# Are Greedy Task Orderings Better Than Random in Continual Linear Regression?

**Matan Tsipory** [* †]      **Ran Levinstein** [* †]      **Itay Evron** [* †]      **Mark Kong** [* ‡]

**Deanna Needell** [‡]          **Daniel Soudry** [†]

## Abstract

We analyze task orderings in continual learning for linear regression, assuming joint realizability of training data. We focus on orderings that greedily maximize dissimilarity between consecutive tasks, a concept briefly explored in prior work but still surrounded by open questions. Using tools from the Kaczmarz method literature, we formalize such orderings and develop geometric and algebraic intuitions around them. Empirically, we demonstrate that greedy orderings converge faster than random ones in terms of the average loss across tasks, both for linear regression with random data and for linear probing on `CIFAR-100` classification tasks. Analytically, in a high-rank regression setting, we prove a loss bound for greedy orderings analogous to that of random ones. However, under general rank, we establish a repetition-dependent separation. Specifically, while prior work showed that for random orderings, with or without replacement, the average loss after $k$ iterations is bounded by $\mathcal{O}(1/\sqrt{k})$—we prove that single-pass greedy orderings may fail catastrophically, whereas those allowing repetition converge at rate $\mathcal{O}(1/\sqrt[3]{k})$. Overall, we reveal nuances within and between greedy and random orderings.

## 1 Introduction

Continual learning is a subfield of machine learning in which a learner is exposed to tasks or datasets sequentially. In such setups, only a single task is fully accessible at any given time, due to, for instance, computational limitations, data retention or privacy constraints, or the temporal nature of the environment. While much of the continual learning research focuses on mitigating forgetting or improving transfer, the role of *task ordering* is not yet fully understood.

Understanding how task order affects learning and what characterizes optimal orderings is important for both theoretical and practical reasons. Such understanding can illuminate failure modes, clarify the interplay between forgetting and transfer, and guide the design of continual environments and algorithms. Furthermore, it can inform active control over task sequences in settings that permit it (*e.g.,* robotic environments), situating the problem at the intersection of continual learning, multitask learning, curriculum learning, and active learning. This line of inquiry raises impactful computational and financial questions in the era of large language models and foundation models:

- *Can task ordering by itself mitigate forgetting, even under vanilla continual training?*
- *What constitutes an "optimal" task ordering?*
- *Is it better to learn when adjacent tasks are similar or dissimilar?*
- *Can greedy strategies systematically outperform random task orderings?*

---

[*] *Equal contribution.*
[†] Technion, Haifa.
[‡] University of California, Los Angeles.

39th Conference on Neural Information Processing Systems (NeurIPS 2025).

One compelling direction in the continual learning literature is the design of task orderings informed by task similarity. This idea appears in several earlier works, with varying degrees of emphasis and differing motivations [e.g., 56, 79, 90, 89, 71, 44, 63, 72]. Most closely related to our work is Bell and Lawrence [14], who were among the first to explicitly and systematically examine such orderings in continual learning. They hypothesized that optimal performance would arise when adjacent tasks are *similar*. Surprisingly, they empirically found the opposite—orderings with *dissimilar* adjacent tasks led to better performance. More recently, Li and Hiratani [62] reached a similar conclusion and further proposed arranging tasks from the least to the most "typical". While these studies are thought-provoking, they are either empirical [71, 14, 72], based on restrictive data assumptions [63, 62], or focused solely on task-incremental settings [79], with some of their findings appearing inconclusive or contradictory. This underscores the need for a more rigorous *theoretical* understanding.

To this end, we formalize "similarity-guided" orderings through *greedy* task selection, leveraging tools from related fields. Building on a projection-based perspective of continual learning [33, 34], we introduce two greedy orderings—Maximum Distance and Maximum Residual—commonly studied in the Kaczmarz [75, 73] and projection methods literatures [2, 41]. Using these orderings, we build a geometric intuition for greediness, as illustrated in Figure 1. We then develop both analytical and empirical insights—*e.g.,* by experimenting on random and CIFAR-100 tasks and proving optimality results for high-rank tasks.

Surprisingly, under general tasks, although without-replacement random orderings are known to converge [35, 10], we show that single-pass greedy orderings may fail catastrophically. We further prove that this drawback is resolved under greedy orderings *with* repetition, for which we establish a dimensionality-independent convergence bound. Finally, we propose a hybrid scheme combining greedy and random orderings, highlighting its empirical and analytical benefits.

We hope that the theoretical foundations—perspectives, tools, and findings—laid out in this paper will inspire future work on task orderings and their potential to mitigate catastrophic forgetting.

**Summary of our contributions.**

1. We formalize similarity-guided orderings in continual linear regression via greedy strategies, drawing on tools and intuitions from the Kaczmarz and projection literature (Section 3).

2. We empirically demonstrate that greedy orderings converge faster than random orderings, both on synthetic regression tasks and on CIFAR-100-based classification tasks (Section 4.1).

3. We prove optimality and convergence guarantees for high-rank tasks (Section 4.2).

4. For general-rank data, we design adversarial task collections in which single-pass greedy orderings provably induce *catastrophic* forgetting, *i.e.,* yield an $\Omega(1)$ loss even as $T \to \infty$ (Section 5.1).

5. In contrast, we prove an $\mathcal{O}(1/\sqrt[3]{k})$ upper bound for greedy orderings *with repetition* (Section 5.2).

6. We combine greedy and random orderings into a hybrid strategy that performs well empirically and inherits the bounds of random orderings, avoiding greedy failure modes (Section 5.3).

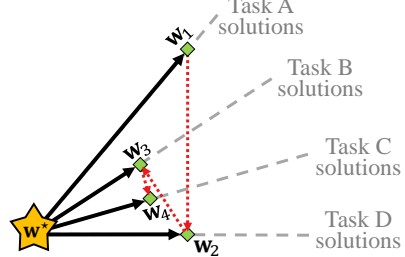

(a) A greedy ordering with *dissimilar* adjacent tasks.

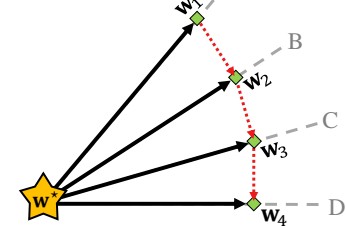

(b) A greedy ordering with *similar* adjacent tasks.

Figure 1: **Intuition.** Consider a collection of jointly-realizable linear regression tasks (*e.g.,* A,B,C,D). Each task has an affine solution space (*e.g.,* where $\mathbf{X_A w = y_A}$), and $\mathbf{w_\star}$ is an "offline" joint solution at the intersection of all tasks. Employing a projection perspective on learning in continual models [33, 34], we see that transitions between *dissimilar* tasks (A→D→B→C) intuitively lead to faster convergence toward the intersection compared to transitions between *similar* tasks (A→B→C→D).

## 2 Setting and Background: Continual linear regression

We focus on continual linear regression, common in theoretical continual learning [e.g., 29, 7, 33, 35, 63, 78, 38, 46]. This setting, though simple, already gives rise to key continual learning phenomena, such as complex interactions between forgetting, task similarity, and overparameterization [see 39].

**Notation.** We reserve bold symbols for matrices and vectors, *e.g.,* $\mathbf{X}, \mathbf{w}$. We use $\|\cdot\|$ to denote the Euclidean norm of vectors and the spectral (L2) norm of matrices. $\mathbf{X}^+$ denotes the Moore–Penrose pseudoinverse of a matrix. Finally, we denote $[n] = 1, \ldots, n$.

Formally, the learner is given access to a *task collection* of $T$ linear regression tasks, *i.e.,* $(\mathbf{X}_1, \mathbf{y}_1), \ldots, (\mathbf{X}_T, \mathbf{y}_T)$ where $\mathbf{X}_m \in \mathbb{R}^{n_m \times d}$, $\mathbf{y}_m \in \mathbb{R}^{n_m}$. We denote the data "radius" by $R \triangleq \max_{m \in [T]} \|\mathbf{X}_m\|$. For $k$ iterations, the learner sequentially learns the tasks according to a *task ordering* $\tau : [k] \to [T]$, which—as this paper shows—can be crucial in continual learning.

---

**Scheme 1** Continual linear regression (to convergence)

---

Initialize $\mathbf{w}_0 = \mathbf{0}_d$
**For each** iteration $t = 1, \ldots, k$:

  $\mathbf{w}_t \leftarrow$ Start from $\mathbf{w}_{t-1}$ and minimize the current task's loss $\mathcal{L}_{\tau(t)}(\mathbf{w}) \triangleq \left\|\mathbf{X}_{\tau(t)}\mathbf{w} - \mathbf{y}_{\tau(t)}\right\|^2$
    with (S)GD to convergence
**Output** $\mathbf{w}_k$

---

We assume throughout the paper that there exist *offline joint solutions* that perfectly solve all $T$ tasks *jointly*. This assumption is common[4] in many theoretical continual learning papers and facilitates the analysis [e.g., 33, 34, 35, 54, 39, 51]. Moreover, it naturally holds in highly overparameterized models and is thus linked to the linear dynamics of deep networks in the neural tangent kernel (NTK) regime [see 49, 23].

**Assumption 2.1** (Joint Linear Realizability of Training Data). Assume the intersection of *all* task solution subspaces is non-empty, *i.e.,* $\mathcal{W}_\star \triangleq \bigcap_{m=1}^T \mathcal{W}_m \triangleq \bigcap_{m=1}^T \left\{ \mathbf{w} \in \mathbb{R}^d \,\middle|\, \mathbf{X}_m\mathbf{w} = \mathbf{y}_m \right\} \neq \varnothing$.

We focus on the joint solution with the minimum norm, often linked to improved generalization.

**Definition 2.2** (Minimum-norm joint solution). Denote specifically $\mathbf{w}_\star \triangleq \underset{\mathbf{w} \in \mathcal{W}_\star}{\arg\min} \|\mathbf{w}\|$.

We follow prominent theoretical work [e.g., 29, 33, 34, 39, 35] and study the model's ability to not "forget" previously seen data, and accumulate expertise on the *training* data (of all tasks). This focus isolates continual dynamics from statistical *generalization* effects that also arise in non-continual, stationary settings.

**Definition 2.3** (Average loss). The (training) loss of an individual task $m \in [T]$ is defined as $\mathcal{L}_m(\mathbf{w}) \triangleq \|\mathbf{X}_m\mathbf{w} - \mathbf{y}_m\|^2$. The loss we analyze is the average across all $T$ tasks, which in our realizable setting takes the following form:

$$\mathcal{L}(\mathbf{w}_k) \triangleq \tfrac{1}{\|\mathbf{w}_\star\|^2 R^2} \cdot \frac{1}{T} \sum_{m=1}^T \mathcal{L}_m(\mathbf{w}_k) = \tfrac{1}{\|\mathbf{w}_\star\|^2 R^2} \cdot \frac{1}{T} \sum_{m=1}^T \|\mathbf{X}_m(\mathbf{w}_k - \mathbf{w}_\star)\|^2 \,,$$

where we also normalize by the generally unavoidable scaling factors $\|\mathbf{w}_\star\|$ and $R \triangleq \max_{m \in [T]} \|\mathbf{X}_m\|$.

*Remark* 2.4 (Forgetting vs. loss). An alternative quantity considered in continual learning [20, 33] is the *forgetting*, defined as the loss *degradation* at iteration $k$ across previously seen tasks *only*, *i.e.,* $\frac{1}{k} \sum_{t=1}^k \left(\mathcal{L}_{\tau(t)}(\mathbf{w}_k) - \mathcal{L}_{\tau(t)}(\mathbf{w}_t)\right)$, or simply as $\frac{1}{k} \sum_{t=1}^k \left\|\mathbf{X}_{\tau(t)}\mathbf{w}_k - \mathbf{y}_{\tau(t)}\right\|^2$ in our realizable setting. Since we mostly focus on single-pass orderings where each task is seen once, the forgetting ultimately coincides with the average loss. Thus, we ease presentation and study the average loss.

---

[4]A different trend in continual learning theory is to assume an underlying linear model, like we do, but allow additive label noise [e.g., 38, 63, 109, 28, 60, 61]. However, this comes at the cost of strong assumptions on the *features*—*e.g.,* commutable covariance matrices or i.i.d. features across tasks. To some extent, the analysis in Section 5.1 of Evron et al. [33] suggests that, under such assumptions, task ordering has limited impact. Thus, it may not be a suitable starting point for studying similarity-guided orderings, in contrast to our assumption.

Another insightful quantity is the distance to $\mathbf{w}_\star$.

**Definition 2.5** (Distance to the joint solution). *After $k$ iterations, the (squared) distance is,*

$$D^2(\mathbf{w}_k) = \tfrac{1}{\|\mathbf{w}_\star\|^2} \cdot \|\mathbf{w}_k - \mathbf{w}_\star\|^2 .$$

This distance upper bounds the loss, as can be shown using simple norm inequalities.

**Proposition 2.6** (Linking the quantities). *After $k$ iterations of Scheme 1 on jointly realizable tasks, the loss is upper bounded by the distance to the joint solution.*

$$\mathcal{L}(\mathbf{w}_k) = \tfrac{1}{\|\mathbf{w}_\star\|^2 R^2} \cdot \frac{1}{T} \sum_{m=1}^{T} \|\mathbf{X}_m (\mathbf{w}_k - \mathbf{w}_\star)\|^2 \leq \tfrac{1}{\|\mathbf{w}_\star\|^2} \cdot \|\mathbf{w}_k - \mathbf{w}_\star\|^2 = D^2(\mathbf{w}_k).$$

In some cases, the distance remains large while the loss vanishes, showing that converging to $\mathbf{w}_\star$ is not mandatory for continual learning [33]. Focusing on the loss paves the way to universal convergence, independent of the problem's complexity, *e.g.,* its condition number [85].

**Geometric interpretation to learning.** In each iteration of Scheme 1, the learner minimizes the squared loss of the current task to convergence.[5] Each iterate $\mathbf{w}_t$ of this scheme above is known [33] to implicitly follow the following closed-form update rule,

$$\mathbf{w}_t = \mathbf{X}_{\tau(t)}^+ \mathbf{y}_{\tau(t)} + \left(\mathbf{I} - \mathbf{X}_{\tau(t)}^+ \mathbf{X}_{\tau(t)}\right) \mathbf{w}_{t-1} . \tag{1}$$

Conveniently, in our realizable setting, this update rule admits an intuitive geometric interpretation.

Evron et al. [33] identified an orthogonal projection operator,

$$\mathbf{P}_m \triangleq \mathbf{I} - \mathbf{X}_m^+ \mathbf{X}_m \in \mathbb{R}^{d \times d},$$

which we use for analysis *only* (Scheme 1 never explicitly computes pseudoinverses or SVDs).
Under the realizability assumption, $\mathbf{y}_{\tau(t)} = \mathbf{X}_{\tau(t)} \mathbf{w}_\star$.
We plug it into Eq. (1) and obtain:

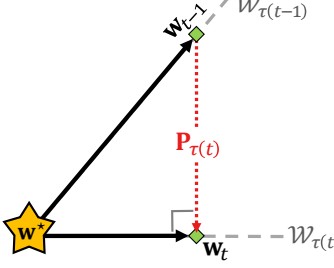

$$\mathbf{w}_t = \mathbf{X}_{\tau(t)}^+ \mathbf{X}_{\tau(t)} \mathbf{w}_\star + \left(\mathbf{I} - \mathbf{X}_{\tau(t)}^+ \mathbf{X}_{\tau(t)}\right) \mathbf{w}_{t-1}$$

$$\mathbf{w}_t - \mathbf{w}_\star = \mathbf{P}_{\tau(t)} \left(\mathbf{w}_{t-1} - \mathbf{w}_\star\right) . \tag{2}$$

Figure 2: **Projection dynamics.**

Geometrically, $\mathbf{w}_{t-1}$ is projected by an affine projection onto the solution space of task $\tau(t)$. In our paper, we adopt this projection-based perspective—proven useful in theoretical work on continual learning [33, 34]—to build intuition about greedy orderings.

## 3   Greedy task orderings: A formal approach and intuition

As discussed in Section 1, the learning order plays a crucial role in the dynamics of many machine learning settings. This phenomenon has also been observed in continual learning, both analytically and empirically. Several works have proposed leveraging "similarity-guided" task orderings—placing dissimilar tasks consecutively. However, the existing literature still lacks the rigor and analytical tools needed to fully understand such orderings. To address this gap, this section draws on connections between continual linear regression and other research areas to formalize greedy task orderings and develop the mathematical tools necessary to study them.

**Geometric intuition.** As illustrated in Figure 2, the projection perspective allows us to decompose $\|\mathbf{w}_t - \mathbf{w}_\star\|^2$ using projection properties and the Pythagorean theorem as:

$$\|\mathbf{w}_t - \mathbf{w}_\star\|^2 = \|\mathbf{w}_{t-1} - \mathbf{w}_\star\|^2 - \|\mathbf{w}_{t-1} - \mathbf{w}_t\|^2$$
$$= \|\mathbf{w}_{t-1} - \mathbf{w}_\star\|^2 - \|(\mathbf{I} - \mathbf{P}_{\tau(t)})(\mathbf{w}_{t-1} - \mathbf{w}_\star)\|^2. \tag{3}$$

Thus, to try and minimize $\|\mathbf{w}_t - \mathbf{w}_\star\|^2$, one could greedily maximize $\|(\mathbf{I} - \mathbf{P}_{\tau(t)})(\mathbf{w}_{t-1} - \mathbf{w}_\star)\|^2$.

---

[5]This simplifies the analysis; other choices exist as well, *e.g.,* a fixed number of steps per task [51, 59].

This has inspired a myriad of studies on Kaczmarz[6] and projection methods [e.g., 2, 75, 16] which employed ordering schemes that greedily maximize $\|\mathbf{w}_{t-1} - \mathbf{w}_t\|^2$, in the following spirit.

**Definition 3.1** (Maximum Distance Ordering). Greedily maximize the distance between iterates, *i.e.,*

$$\tau_{\text{MD}}(t) = \underset{m \in [T] \backslash \tau_{\text{MD}}(1:t-1)}{\operatorname{argmax}} \left\| \mathbf{X}_m^+ \left( \mathbf{X}_m \mathbf{w}_{t-1} - \mathbf{y}_m \right) \right\|^2 = \underset{m \in [T] \backslash \tau_{\text{MD}}(1:t-1)}{\operatorname{argmax}} \left\| \left( \mathbf{I} - \mathbf{P}_m \right) \left( \mathbf{w}_{t-1} - \mathbf{w}_\star \right) \right\|^2$$

at each iteration $t \in [T]$, where $\tau_{\text{MD}} (1:t-1) \triangleq \{\tau_{\text{MD}} (1), \ldots, \tau_{\text{MD}} (t-1)\}$.[7]

Our earlier Figure 1a illustrates the MD ordering and how it leads to faster convergence to $\mathbf{w}_\star$.

**Distance and task similarity.** The distance between iterates $\mathbf{w}_{\tau(t-1)}$ and $\mathbf{w}_{\tau(t)}$ reflects some angle between the affine solution subspaces of their corresponding tasks, and more generally, relates to the principal angles between these subspaces [33]. These angles can be used to quantify task similarity, as illustrated in the setting of Section 4.2 and Figure 1.

An alternative greedy ordering found in the literature is the Maximum Residual ordering [e.g., 2, 40, 75, 106]. This rule is easier to compute in full, or to estimate using a small validation set.

**Definition 3.2** (Maximum Residual Ordering). Greedily select the task exhibiting the greatest error:

$$\tau_{\text{MR}}(t) = \operatorname{argmax}_{m \in [T] \backslash \tau_{\text{MR}}(1:t-1)} \left\| \mathbf{X}_m \mathbf{w}_{t-1} - \mathbf{y}_m \right\|^2, \quad \forall t \in [T] .$$

Notice that the MD and MR orderings are related since $\mathbf{X}_m = \mathbf{X}_m \mathbf{X}_m^+ \mathbf{X}_m = \mathbf{X}_m \left( \mathbf{I} - \mathbf{P}_m \right)$, and,

$$\left\| \mathbf{X}_m \mathbf{w}_{t-1} - \mathbf{y}_m \right\|^2 = \left\| \mathbf{X}_m \left( \mathbf{w}_{t-1} - \mathbf{w}_\star \right) \right\|^2 \leq \left\| \mathbf{X}_m \right\|^2 \left\| \left( \mathbf{I} - \mathbf{P}_m \right) \left( \mathbf{w}_{t-1} - \mathbf{w}_\star \right) \right\|^2 .$$

**Single-pass orderings.** Our paper mostly focuses on "single-pass" greedy orderings, where each task is encountered exactly once. Although disallowing repetitions departs slightly from the motivating literature on projection methods, it is the more common—and arguably more natural—setup in continual learning [see 14, 62]. Even in curriculum or multitask learning, limiting tasks to a single pass can reduce training costs. Nonetheless, in Section 5.2, we examine the effect of repetitions.

**Computational tractability of greedy policies.** As explained above, the benefits of greedy orderings are intuitive. The cost of computing the greedy rules in Definition 3.1 and Eq. (4), of course, introduces a tradeoff between convergence rate and overall computational cost. Before continuing our investigation of these orderings, we briefly address their computational feasibility.

(i) **Estimation:** Greedy rules can often be estimated efficiently in practical scenarios. For example, the maximum residual rule (Definition 3.2) requires the current loss of each available task. This quantity can be approximated using a small subset of samples or via dimensionality reduction techniques [e.g., see 31]. In App. C, we show empirically that the maximum residual rule performs comparably to the maximum distance rule, and that its performance remains unharmed even when approximated using only 1% of the data. In deep networks, computing that rule requires only forward passes and may reduce the number of gradient steps—thereby lowering overall time and memory costs by limiting costly backward passes [47].

(ii) **Heuristics**: Our greedy rules rely on residuals to quantify the similarity between the current task and the remaining ones. This approach is exemplified in Figure 1 and Eq. (5) of Section 4.2, and is related to principal angles between subspaces [see 33]. Alternatively, one could utilize *heuristic* notions of task similarity—*e.g.,* predefined [56] or computed metrics that use Hessians [14], zero-shot performance [62], or task embeddings [1, 71].

(iii) **Structured tasks**: If each step updates relatively few residuals (*e.g.,* in a Kaczmarz setting with sparse columns and rows), only few residuals must be recomputed [75].

---

[6]The Kaczmarz method [52, 32], further explained in App. A, iteratively solves a linear system of equations.

[7]In practice, the MD rule is easy to compute for rank-1 tasks, since it reduces to $\frac{1}{\|\mathbf{x}_m\|^2} \left\| \mathbf{x}_m^\top \mathbf{w}_{t-1} - y_m \right\|^2$. In higher ranks, this rule is harder to compute exactly—but can be estimated, *e.g.,* with a subsample of the task.

# 4 Benefits of greedy orderings

A natural competitor to greedy strategies is the random strategy, uniformly sampling tasks (rows or blocks in the Kaczmarz context) from the task collection $[T]$. That is,

$$\tau_{\mathrm{Unif}}(1), \ldots, \tau_{\mathrm{Unif}}(k) \sim \mathrm{Uniform}\left([T]\right) . \tag{4}$$

As mentioned earlier, greedy strategies have a long-standing history in the Kaczmarz method [75, 76] and its block variants [73, 68, 108, 106, 102] employing either deterministic [75] or probabilistic [11, 12, 107, 95] selection rules. In this context, greedy orderings often achieve better provable bounds on the distance to $\mathbf{w}_\star$ (Definition 2.5), compared to random orderings. In contrast, our focus—and that of related continual learning literature [e.g., 33, 34, 54, 51]—centers on convergence of the loss (Definition 2.3). While the loss is upper bounded by the distance to $\mathbf{w}_\star$ (Proposition 2.6), the existing Kaczmarz rates fall short in two ways: (i) they rely on repeating rows/blocks and do not apply in the single-pass regime central to continual learning; and (ii) even when repetition is allowed, their rates depend on data eigenvalues, potentially making them much slower than our data-independent $\mathcal{O}(1/\sqrt[3]{k})$ guarantee of Theorem 5.3.

In this section, we examine how well the advantages of greedy over random orderings carry over to the loss in continual settings—first empirically, then analytically.

## 4.1 Motivating experiments: Greedy outperforms random ordering

Here, we test different task ordering strategies on synthetic regression tasks and a more complex *classification* setup—using linear probing in a domain-incremental `CIFAR-100` setting.[8]

**Regression tasks: Random data.** The feature matrices $\mathbf{X}_1, \ldots, \mathbf{X}_T$ are drawn from a Gaussian distribution. We compare the two "dissimilarity-maximizing" greedy strategies (MD, MR) to the random ordering (Eq. (4)) and a complementary minimum distance strategy (defined as in Definition 3.1, replacing argmax with argmin). Full details, including more combinations of task count $T$, dimension $d$, and rank $r$, as well as experiments on anisotropic data, are provided in App. B.

**Classification tasks:** `CIFAR-100`. We randomly partition classes into continual binary classification tasks, similarly to Li and Hiratani [62]. We train a linear probe on top of a `ResNet-20` embedder, pretrained on the original `CIFAR-100` multiclass task [45, 55]. We optimize the cross-entropy loss of each task while employing L2 regularization towards the previous parameters [65]. We compare the performance to a 'joint' baseline, trained on all tasks together (not continually). See App. C for full details. There, we conduct further experiments on continual `CIFAR-100` tasks as before, but using embeddings pretrained on (i) `CIFAR-10` and (ii) `CIFAR-100` with only *half* of the samples from each class (in this case, continual learning is performed on tasks formed from the other half). We also examine more computationally efficient greedy orderings, determined from only a fraction of the data—down to 1% (5 samples per class in `CIFAR-100`).

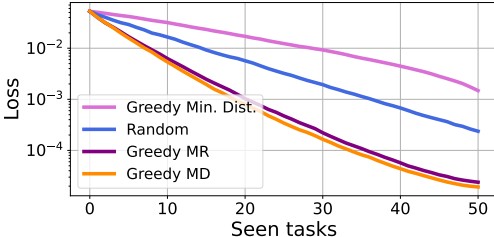 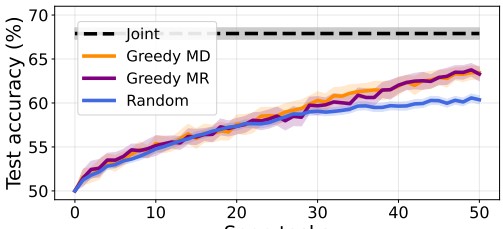

(a) **Regression (random data):** Average loss over $T = 50$ regression tasks of rank $r = 10$ in $d = 100$ dimensions, sampled from an isotropic Gaussian distribution. Details in App. B.

(b) **Classification (`CIFAR-100`):** Average test accuracy over $T = 50$ binary classification tasks, generated by randomly partitioning `CIFAR-100` classes (a domain-incremental setting). Details in App. C.

Figure 3: **Task ordering comparison.** Transitioning between dissimilar tasks consistently outperforms random transitions, with Greedy MR and MD achieving comparable performance.

---

[8]We provide a code snippet for the regression experiments in App. H. The code for the classification experiments is accessible at `https://github.com/matants/greedy_ordering`.

## 4.2 Provable benefits for high-rank, "nearly determined" tasks

To further motivate greedy orderings, we analyze a simple setup where each task's matrix is of nearly full rank, *i.e.,* $\mathrm{rank}(\mathbf{X}_m) = d - 1$, $\forall m \in [T]$. Even in such a setup, it has been shown that arbitrary orderings of $T \to \infty$ may lead to *catastrophic* forgetting and maximal losses [33].

In this setup, each projector can be expressed as a rank-1 operator, *i.e.,* $\mathbf{P}_m = \mathbf{I} - \mathbf{X}_m^+ \mathbf{X}_m = \mathbf{v}_m \mathbf{v}_m^\top$ for a unit vector $\mathbf{v}_m \in \mathbb{R}^d$ that spans the solution space of task $m$. Then, we can rewrite the Maximum Distance (Definition 3.1) rule to explicitly maximize dissimilarity between consecutive tasks, *i.e.,*

$$\tau_{\mathrm{MD}}(t) = \mathrm{argmin}_{m \in [T] \setminus \tau_{\mathrm{MD}}(1:t-1)} \left( \mathbf{v}_m^\top \mathbf{v}_{\tau(t-1)} \right)^2 = \mathrm{argmax}_{m \in [T] \setminus \tau_{\mathrm{MD}}(1:t-1)} \theta_{m,\tau(t-1)}, \quad (5)$$

where we define $\mathbf{v}_{\tau(0)} \triangleq \frac{1}{\|\mathbf{w}_0 - \mathbf{w}_\star\|} (\mathbf{w}_0 - \mathbf{w}_\star)$; see Eq. (6) in App. D.

**Optimality of greedy orderings in terms of distance to $\mathbf{w}_\star$.** Earlier in Eq. (3), we motivated the MD ordering as greedily maximizing the decrease in $\|\mathbf{w}_t - \mathbf{w}_\star\|$. Does this guarantee a minimal distance $\|\mathbf{w}_T - \mathbf{w}_\star\|$ at the end of the sequence? Here, we prove that the MD ordering yields a square-root approximation of the optimal distance at the end of learning.[9]

**Lemma 4.1** (Optimality guarantee when $r = d - 1$). *Let $\mathbf{w}_T^{\tau_{\mathrm{MD}}}$ and $\mathbf{w}_T^{\tau_\star}$ be the iterates after learning $T$ jointly realizable tasks of rank $d - 1$ under the Maximum Distance ordering $\tau_{\mathrm{MD}}$ and an optimal ordering $\tau_\star$ that leads to a minimal distance to the joint solution $\mathbf{w}_\star$. Then, their distances hold,*

$$0 \leq D^2(\mathbf{w}_T^{\tau_\star}) \leq D^2(\mathbf{w}_T^{\tau_{\mathrm{MD}}}) \triangleq \frac{\|\mathbf{w}_T^{\tau_{\mathrm{MD}}} - \mathbf{w}_\star\|^2}{\|\mathbf{w}_\star\|^2} \leq \frac{\|\mathbf{w}_T^{\tau_\star} - \mathbf{w}_\star\|}{\|\mathbf{w}_\star\|} \triangleq D(\mathbf{w}_T^{\tau_\star}) \leq 1.$$

The full proofs for this section are given in App. D.

**What about the loss?** The optimality of the distance does not imply optimality of the average loss, as exemplified in Figure 7 in Section 6. Instead, we now derive an upper bound for the loss.

**Lemma 4.2** (Loss bound when $r = d - 1$). *Under the Maximum Distance greedy ordering over $T$ jointly-realizable tasks of rank $d-1$, the loss of Scheme 1 after $T$ iterations is upper bounded as,*

$$\mathcal{L}(\mathbf{w}_T) = \frac{1}{\|\mathbf{w}_\star\|^2 R^2} \cdot \frac{1}{T} \sum_{m=1}^{T} \|\mathbf{X}_m \mathbf{w}_T - \mathbf{y}_m\|^2 \leq \frac{1}{eT}.$$

This rate matches a recent $\mathcal{O}(\frac{d-r}{T})$ bound for random orderings without replacement [35], whereas the best known rate for such orderings with *general-rank* tasks is $\mathcal{O}(1/\sqrt{T})$ [10]. This raises the question: *for general-rank tasks, can single-pass greedy orderings still compete with random ones, or even outperform them, in worst-case analysis?* Next, we show that they cannot.

## 5 Failure modes and surprises in greedy orderings

Under random orderings, *with or without* replacement, Attia et al. [10] proved a universal, dimensionality-independent rate of $\mathbb{E}_{\tau_{\mathrm{Unif}}} \mathcal{L}(\mathbf{w}_k^{\tau_{\mathrm{Unif}}}) \leq 13/\sqrt{k}$. Surprisingly, we prove a clear separation in our setup: while single-pass greedy orderings can fail *catastrophically*, *i.e., not* decrease with the number of iterations $k = T$, greedy orderings *with* repetition enjoy a bound of $\mathcal{O}(1/\sqrt[3]{k})$.

### 5.1 Greedy orderings can fail where random ones do not

We now present cases where the single-pass greedy ordering forgets *catastrophically*, *i.e.,* suffers an $\Omega(1)$ loss, even after fitting a collection of $T \to \infty$ tasks. Specifically, we present an example in $d = 3$ dimensions where the loss does not diminish, and a construction in $d = T + 1$ dimensions that exploits dimensionality to yield *maximal* forgetting. Full details and proofs are given in App. E.

---

[9]Our optimality result is related to the optimal Hamiltonian path in a predefined similarity graph, as studied in related work [14, 62]. Specifically, in this section's high-rank case, similarity can be *statically* defined as $s_{m,m'} = \cos^2(\theta_{m,m'})$. Then, the greedy MD ordering *approximates* the Hamiltonian path $\tau_\star$ that maximizes $\prod_{t=2}^{T} s_{\tau_\star(t-1),\tau_\star(t)}$ [see 37, 70]. However, under general-rank tasks, our greedy rules (Def. 3.1 and 3.2) are computed *online* at each iteration and depend not only on the previous task $\tau(t-1)$ but also on the previous iterate $\mathbf{w}_{t-1}$. Consequently, these rules do not correspond to a Hamiltonian path on *any* predefined graph.

**Example 5.1** (Adversarial 3d construction). For all $T \in \{4 \cdot 10^i - 1 \mid i = 1, 2, \ldots, 7\}$, there exists a task collection of jointly-realizable tasks in $d = 3$, such that $\mathcal{L}(\mathbf{w}_T^{\tau_{\mathrm{MD}}}), \mathcal{L}(\mathbf{w}_T^{\tau_{\mathrm{MR}}}) > 2.78 \cdot 10^{-5}$.

**Theorem 5.2** (Greedy lower bound). *For any $d \geq 30$, there exists an adversarial task collection with $T = d - 1$ jointly-realizable tasks (of different ranks) such that both greedy orderings (MD, MR) forget catastrophically. That is, the loss at the end of the sequence is, $\mathcal{L}(\mathbf{w}_T^{\tau_{\mathrm{MD}}}), \mathcal{L}(\mathbf{w}_T^{\tau_{\mathrm{MR}}}) \geq \frac{1}{8} - \frac{1}{4d}$.*

We demonstrate the behavior of an adversarial task collection using $T = 999$ tasks in $d = 1000$ dimensions. Our constructed collection "tricks" the greedy orderings: slowly increasing not only the loss on *all* tasks, but also the forgetting of *previous* tasks. The model is thus unable to accumulate knowledge and practically forgets everything it learns.

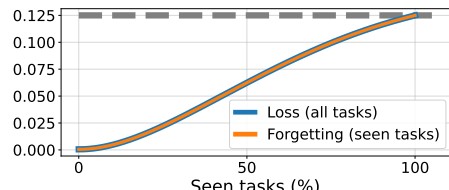

Figure 4: **Learning an adversarial collection.**

## 5.2 Single-pass vs. repetition in greedy orderings

So far, we have focused on *single-pass* greedy orderings, in which each task is learned exactly once. These are conceptually related to without-replacement sampling and (re)shuffling techniques in SGD and the Kaczmarz method, where repetition-free strategies often converge faster than with-replacement sampling, both empirically [17, 76, 96] and in theory [69, 42, 15, 50, 43; but see 84, 26]. We ask: *Does the advantage of orderings without repetition extend to greedy orderings?*

Now, we show that repetition in greedy orderings avoids the failure mode of single-pass ones.

**Theorem 5.3** (Dimensionality-independent bound for greedy orderings with repetition). *Under a Maximum Distance greedy ordering with repetition ($\tau_{\mathrm{MD-R}}$) over $T$ jointly-realizable tasks, the loss of Scheme 1 after $k \geq 2$ iterations is upper bounded as $\mathcal{L}(\mathbf{w}_k^{\tau_{\mathrm{MD-R}}}) = \mathcal{O}(1/\sqrt[3]{k})$.*

App. F provides the proof, a comparison to prior rates (Table 1), and details for the experiment below.

We evaluate the effect of repetition across orderings under random data. As in prior work, random sampling *without* replacement outperforms *with* replacement. In contrast, repetition benefits greedy orderings, likely due to *larger* updates and faster convergence to $\mathbf{w}_\star$. The slowdown in the single-pass case likely reflects the exhaustion of high dissimilarities.

Intuitively, repetition in *random* orderings exposes the learner to less data, while in *greedy* selection it allows considering all tasks at each step.

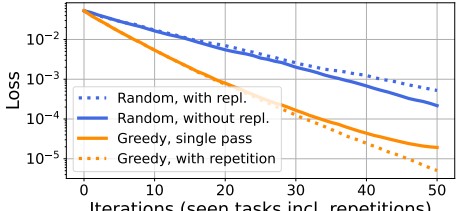

Figure 5: **The effect of repetitions.**

However, these findings do not always hold, as seen in our *classification* example (App. C.5).

## 5.3 Extension: Hybrid task orderings

To leverage both the fast empirical convergence of greedy orderings and the analytical convergence guarantees of random ones, we introduce a "hybrid" strategy: begin with greedy selection and switch to random once the decrements $\|\mathbf{w}_{t-1} - \mathbf{w}_t\|^2$ fall below a threshold. Analytically, using a suitable threshold, we prove in Lemma G.1 that any bound for without-replacement random orderings, *e.g.,* $\mathcal{O}(1/\sqrt{T})$ [10], extends to our hybrid scheme, showing it avoids the failure mode of Section 5.1.

Empirically, the hybrid ordering performs better than random but worse than greedy. This matches our intuition from Eq. (3) and Figure 1a: greedy selection takes larger "steps" (or projections), particularly early on, when most tasks are still available. Once these projections diminish, we switch to the random ordering, which—unlike the greedy approach—cannot be adversarially "tricked" into failure.

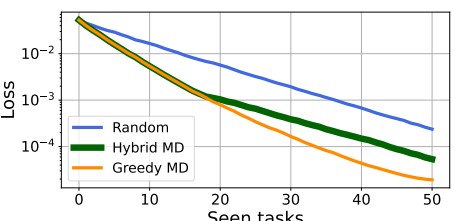

Figure 6: **Hybrid ordering experiment.**

Further details and experiments appear in App. G.

# 6 Discussion and related work

So far, we have studied greedy task orderings, demonstrating empirical and analytical benefits of transitioning between *dissimilar* tasks. Here, we expand on connections and ideas not yet fully covered to better situate our work within the existing literature. In App. A, we discuss further links to Kaczmarz methods, curriculum and active learning, coordinate descent, and example selection in SGD.

**Task orderings in continual learning theory.** Continual learning theory often treats task orderings as arbitrary. However, several analytical works [e.g., 33, 34, 35, 54, 51, 19] show that certain orderings—typically cyclic or random—can mitigate forgetting (matching empirical findings [58]). While some works downplay ordering effects—arguing they are often minor—and defer their study to future work [92], others design continual learning algorithms specifically for *evolving sequences* with very *similar* adjacent tasks [6]. We follow a different line of work that studies how pairwise task similarities, or *dissimilarities*, influence continual schemes.

A particularly relevant work by Bell and Lawrence [14] advocates pairwise task dissimilarity as a guiding principle and was among the first to empirically investigate similarity-guided task orderings. Tasks are represented as vertices in a complete graph, where edge weights correspond to a predefined distance between tasks, and Hamiltonian paths represent full task orderings. They hypothesized that a minimum-weight path (favoring similar tasks in succession) would yield the best continual performance. Yet their experiments on simple neural networks indicated the opposite: maximum-weight paths, placing *dissimilar* tasks adjacently, often led to improved performance. Still, these results were not always statistically significant (see their Figure 5)—motivating us to revisit similarity-guided task orderings from a more analytical perspective.

Li and Hiratani [62] conduct a deeper investigation into similarity-guided task orderings, obtaining more statistically robust empirical results. They likewise find that adjacent tasks should be *dissimilar*, and further explore task "typicality" (discussed below). While deriving results for a linear regression model to support their empirical observations (on neural networks), they rely on a restrictive analytical data model in which features are randomly drawn from a simplified distribution across tasks. In contrast, our analysis accommodates *arbitrary* feature matrices, allowing richer and more realistic forms of task similarity. Like Bell and Lawrence [14], their goal is to characterize optimal orderings in general, whereas we formalize and analyze *greedy* orderings specifically, both as a practical strategy and as a proxy for optimal ones.

Ruvolo and Eaton [88] propose an "information maximization" approach to task ordering, using a diversity-based heuristic related to our maximum residual strategy (Definition 3.2). However, their complex model limits rigorous theoretical analysis of the kind we provide.

Lin et al. [63] examine the role of task similarity and reach conclusions broadly aligned with ours. While influential, their work differs from ours in several ways. First, their analysis relies on a restrictive i.i.d. feature assumption across tasks. They also assume a distinct *teacher* model per task, unlike our setting, where all tasks share a single overparameterized model—as is common in modern deep learning. Consequently, their notion of task similarity relies on teacher similarity, rather than more practical measures such as residuals (as in Definition 3.2) or feature similarity. Although they note ordering effects in their expressions and briefly support them with classification experiments, task ordering is not their primary focus. In contrast, we offer a comprehensive treatment of similarity-guided orderings specifically—providing formal definitions, geometric intuitions, greedy strategies, optimality results, empirical validation, failure modes, and repetition analysis.

**Can similarity-guided task orderings alone mitigate forgetting?** Methods such as replay, regularization, and parameter isolation are widely used to mitigate forgetting in continual learning [53, 59, 83, 21, 87, 27]. However, they depart somewhat from standard ("vanilla") deep learning practices that apply plain gradient methods to the (current) loss. Interestingly, both our work and prior studies show that even without such mechanisms, task ordering *alone* strongly affects forgetting. For instance, simply randomizing the task order—with or without replacement—is known to alleviate forgetting [33, 35, 10, 58]. In contrast, we show that single-pass greedy ordering can exacerbate forgetting (Section 5.1), while allowing task repetitions mitigates this effect (Section 5.2). Moreover, ordering strategies can be combined with other approaches; for example, in our classification experiments we also employ regularization (Section 4.1). This underscores the importance of studying task ordering as a simple, complementary way to mitigate forgetting, while potentially keeping continual learning closer to standard deep learning practice.

**Task typicality at the end of learning.** Li and Hiratani [62] suggest that tasks should be arranged from least to most "typical". While we did not focus on this aspect of orderings, our geometric interpretation can illustrate it.

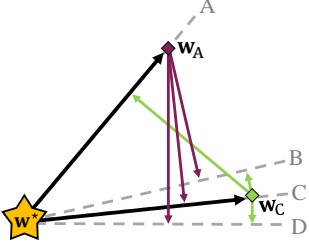

Our motivation was to minimize the distance $\|\mathbf{w}_k - \mathbf{w}_\star\|^2$, which upper bounds the *loss* $\frac{1}{T} \sum_{m=1}^{T} \|\mathbf{X}_m (\mathbf{w}_k - \mathbf{w}_\star)\|^2$. However, this bound can be loose, and minimizing the distance does not guarantee the lowest loss. For example, in Figure 7, although $\|\mathbf{w}_A - \mathbf{w}_\star\|^2 = \|\mathbf{w}_C - \mathbf{w}_\star\|^2$, the point $\mathbf{w}_C$ is a better *ending point* than $\mathbf{w}_A$, inducing a lower loss (the arrows represent the residuals). This happens because task C is more typical—*i.e.,* more similar to other tasks—than task A.

Figure 7: **Task typicality.**

The empirical advantage of greedy ordering may stem from a tendency to postpone typical tasks, perhaps causing its benefits to emerge only in later stages of training (see Figure 3b).

**Regret today or loss tomorrow?** In Section 3, we motivated the use of greedy orderings to minimize the distance to the joint solution $\|\mathbf{w}_k - \mathbf{w}_\star\|^2$, which in turn upper-bounds the average loss over *all* tasks: $\frac{1}{T} \sum_{m=1}^{T} \|\mathbf{X}_m (\mathbf{w}_k - \mathbf{w}_\star)\|^2$. This objective is related, but not identical, to the notion of *regret*, which quantifies the loss along the optimization *path* on *consecutive* tasks, *i.e.,* $\frac{1}{k} \sum_{t=1}^{k} \|\mathbf{X}_{\tau(t)} (\mathbf{w}_{t-1} - \mathbf{w}_\star)\|^2$. From this definition and Figure 1, we observe that regret—though also upper-bounded by the distances $\|\mathbf{w}_{t-1} - \mathbf{w}_\star\|^2$—can often benefit from transitions between *similar* tasks rather than *dissimilar* ones. In other words, to make accurate predictions *during* learning—*e.g.,* in decision-making—transitioning between *similar* tasks may be preferable. Conversely, to minimize average loss *across* tasks—*e.g.,* in curriculum or multitask learning—our findings suggest that transitioning between *dissimilar* tasks is preferable.

**Other continual setups.** The majority of studies support our conclusion that sequential task *dissimilarity* is beneficial [e.g., 88, 14, 71, 81, 33, 63, 66, 91]. Still, the specific continual setup can dramatically influence the behavior of task orderings. We consider a "domain-incremental" setting: learning the same problem across different domains, *i.e.,* $\mathcal{P}(X)$ changes but $\mathcal{P}(Y|X)$ is fixed [99, 57]. Alternatively, "task-incremental" setups involve distinct tasks—possibly with different $\mathcal{P}(Y|X)$—with task identity known at both train *and* test time. There, prior work [79, 72] trained a *separate* model per task and found that *similarity-maximizing* orderings prevail, seemingly contradicting our findings. However, in such scenarios, the focus shifts from *forgetting* to inter-task *transfer*, benefiting from *similar* consecutive tasks (see discussion on regret). Hence, their results complement ours.

Others have studied "class-incremental" learning (CIL), where each task introduces new objects or classes, aiming for strong overall performance (*e.g.,* in split benchmarks [97]). While some CIL papers suggest that consecutive task *similarity* is preferable [44, 67], a closer look reveals that they modify the class composition *within* each task, inducing high *intra-task heterogeneity* [44, 8]. This likely leads to wider minima and stronger "transferability" to other tasks, thus explaining their improved results. Such configurations resemble curriculum more than continual learning.[10] To our knowledge, only Yang and Li [104] report contradictory results, possibly due to their empirical design.[11] Finally, we note that the effects discussed here are related to the *interleaving effect* in educational psychology [77, 86].

**Future work.** One could extend our findings to other settings—such as class- and task-incremental, discussed earlier—and to more complex continual learning methods, such as replay and regularization. Moreover, our linear realizability assumption could be relaxed to accommodate label noise or even extend to nonlinear models, possibly borrowing tools from Kaczmarz literature [13, 106]. It would also be interesting to combine our approach with common wisdom in curriculum learning—*i.e.,* to design orderings that account for both task similarity and difficulty.

Finally, a promising direction for achieving tighter upper bounds in continual linear regression (see Table 1) lies in probabilistic selection rules, inspired by randomized greedy Kaczmarz methods [11, 12, 107, 95], which could combine the strengths of greedy orderings with the robustness of randomness, akin to our proposed hybrid scheme.

---

[10]The learner controls the internal task composition to create "easier" tasks, as in curriculum learning [101].

[11]They construct the first task using a random half of the classes. This strong "pretraining" leads to low initial loss, as the model already learns *half* the classes. This resembles the failure modes discussed in Section 5.1.

## Acknowledgments and Disclosure of Funding

We thank Joseph (Seffi) Naor (Technion) for fruitful discussions. We thank Timothée Lesort (Université de Montréal, MILA-Quebec AI Institute) for fruitful discussions and valuable feedback.

The research of DS was funded by the European Union (ERC, A-B-C-Deep, 101039436). Views and opinions expressed are however those of the author only and do not necessarily reflect those of the European Union or the European Research Council Executive Agency (ERCEA). Neither the European Union nor the granting authority can be held responsible for them. DS also acknowledges the support of the Schmidt Career Advancement Chair in AI.

DN was partially supported by NSF DMS 2408912.

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

# Appendix contents

# A Further related work

Here, we elaborate on additional connections not fully addressed in Section 6.

**Alternative viewpoint: The Kaczmarz method.** Our continual linear regression scheme maps directly to the Kaczmarz method [52, 32], a classical iterative projection algorithm for solving linear systems of equations. In our context, the solved system is, $\mathbf{Xw} = \mathbf{y}$, where

$$\mathbf{X} = \begin{pmatrix} \mathbf{X}_1 \\ \vdots \\ \mathbf{X}_T \end{pmatrix} \in \mathbb{R}^{N \times d}, \quad \mathbf{y} = \begin{pmatrix} \mathbf{y}_1 \\ \vdots \\ \mathbf{y}_T \end{pmatrix} \in \mathbb{R}^N, \quad \text{where } N = \sum_{m=1}^{T} n_m.$$

Evron et al. [33] pointed out that Kaczmarz methods iteratively solve the "block" systems of the form $\mathbf{X}_{\tau(t)}\mathbf{w} = \mathbf{y}_{\tau(t)}$ using an update rule *equivalent* to our continual update in Eq. (1). As a result, the observations and results in our paper extend naturally to the greedy Kaczmarz method. However, whereas Kaczmarz studies typically analyze convergence in terms of the distance to the intersection $\mathbf{w}_\star$, we focus on the *loss*, *i.e.,* the residuals (Definitions 2.5 and 2.3, respectively). For example, in the $r = d - 1$ case of Section 4.2, this distinction allowed proving an upper bound on the *loss* and an "approximation" result on the optimal *distance* (Lemmas 4.2 and 4.1, respectively).

It is worth noting that, via its connection to the Kaczmarz method, our continual linear regression scheme is also related to coordinate descent methods [82]. While prior work in this area shows that greedy selection can outperform random sampling, these results often rely on strong convexity assumptions [74, 36], and typically apply to the Kaczmarz method through a primal-dual lens [33, 105]—again, yielding only convergence to the intersection point $\mathbf{w}_\star$.

**Curriculum learning.** Broadly, curriculum learning enhances training by controlling the order in which data are presented, to accelerate convergence or improve accuracy. The prevailing view is that examples should be ordered from *easy to hard* by their "difficulty" [101, 94]. In contrast, we study *similarity*-guided orderings, aligning with recent findings in continual learning [14, 46].

A key distinction between curriculum and continual learning lies in the unit of ordering: while curriculum learning typically orders individual samples or batches, we focus on orderings of *entire* tasks. Moreover, curriculum learning often takes a single gradient step per sample, whereas continual learning optimizes each task to a low loss before proceeding. Nonetheless, given the maturity of curriculum learning and the interdisciplinary nature of our work, some curriculum studies that operate at the task level are directly relevant [e.g., 79] and were discussed in Section 6.

**Example selection in SGD.** Evron et al. [35] show that learning an *entire* (continual) linear regression task in our Scheme 1 reduces to taking a *single* large gradient step on a modified objective. While they use this reduction to analyze *random* orderings via last-iterate SGD analysis, we leverage it here to draw connections between *greedy* task orderings and greedy example selection in SGD.

Most of the example selection literature considers multi-epoch settings, where each sample is seen multiple times. In such regimes, it is common to *randomly shuffle* the dataset once or at the start of each epoch [e.g., 69, 42], but this is not necessarily *optimal* [80]. For instance, Lu et al. [64] show that greedy permutations—computed at the beginning of each epoch—can yield faster convergence than random ones. However, their analysis requires (1) multiple epochs and (2) very small step sizes, making it inapplicable to our single-pass continual setting.

Das et al. [24] show that a selection rule akin to our maximum residual rule (Definition 3.2) accelerates *early* convergence but may underperform random orderings asymptotically—aligning with our findings on the hybrid approach in Section 5.3. They also analyze an approximate rule, supporting our observations on computational tractability in Section 3. Others select greedily by gradient magnitude instead of loss [103], or "mine" examples at the *mini-batch* level—selecting "hard" samples with a high loss or ones that lead to a significant decrease in loss [93, 100].

**Active Learning.** Active learning aims to reduce labeling cost by querying the most informative samples for labeling, typically from a large unlabeled pool. This setting resembles ours, where the learner may apply a greedy maximum distance rule (Definition 3.1) to select the task or sample expected to induce the greatest model update. For example, a related idea is explored in Cai et al. [18], who propose a greedy maximum distance variant for regression. Since labels are unknown

at selection time in their active learning setup, they approximate the expected model change using a bootstrap method. Empirically, this approach identifies informative examples and consistently improves generalization across datasets.

# B Appendix to Section 4.1: Regression experiments

All figures report averages over 10 runs. In each run, we randomly sample a task collection to evaluate the different ordering strategies. Shaded regions (see App. F.2 and G.3) indicate $\pm 1$ standard error intervals, even when not visually discernible. In App. B.3 we further discuss the statistical significance of our experiments.

**Computational resources.** All regression experiments—including those not shown—were completed within 4 hours on a home PC equipped with an Intel i5-9400F CPU and 16GB of RAM.

## B.1 Isotropic data

Figures 8 and 9 extend the previous experiment on isotropic data (Figure 3a) to varying dimensions $d$, ranks $r$ and task counts $T$. Results confirm consistent patterns: greedy (dissimilarity maximizing) methods outperform random, and MD is better than MR across all settings (sometimes only slightly).

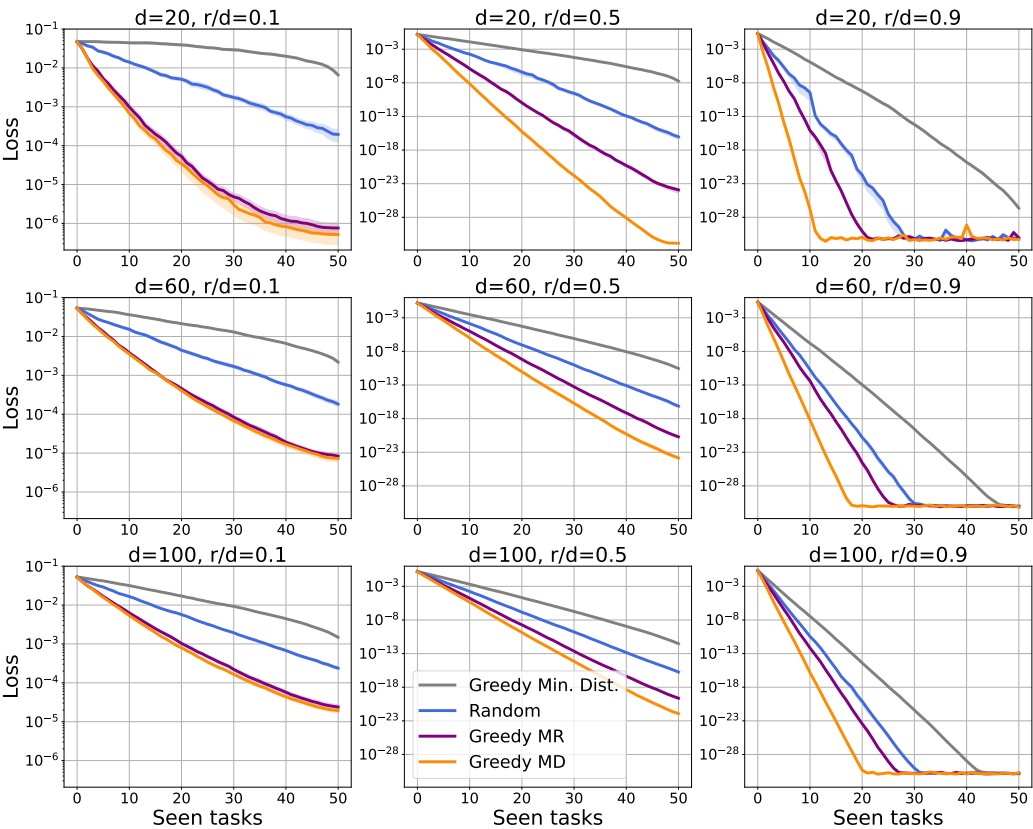

Figure 8: **Comparing orderings for varying dimensions $d$ and ranks $r$ of the data matrices, for isotropic data.** $T = 50$. We observe that, for such isotropic data, the random ordering performance is determined solely by the ratio $r/d$. In contrast, greedy orderings that prioritize *dissimilarity* benefit from a lower dimension when $r/d$ is fixed (to see that, focus on single columns in the grid). We hypothesize that this is because an increased *task "density"* in lower dimensions: when $r/d$ is fixed, increasing $d$ increases $d - r$, expanding the set of possible task projections (see Eq. (2)). As a result, a fixed number of tasks $T$ covers this space more sparsely in higher dimensions. In lower dimensions, the same $T$ tasks yield denser coverage, increasing the likelihood that greedy dissimilarity-based selection identifies tasks with large projections.

In all strategies, higher task rank consistently yields improved performance (focus on single rows). This is because the solution subspaces are of rank $d - r$, so increasing $r$ (with fixed $d$) lowers the subspace rank, increasing the distances between them and resulting in larger projections.

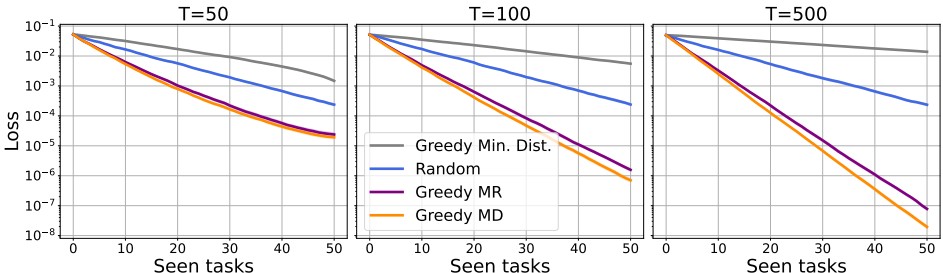

Figure 9: **Comparing orderings for varying task count $T$, for isotropic data.** $d = 100$, $r = 10$. Dissimilarity-based greedy strategies become more effective as the number of tasks increases. This is since in an isotropic setting, where task directions are sampled uniformly, increasing the number of tasks increases the *coverage* of the unit sphere. This results in a higher probability of encountering task pairs with large angular separation between their solution subspaces, which greedy ordering utilizes.

### B.2 Anisotropic data

The following experiments were were performed with anisotropic data, sampled from a Gaussian distribution with exponentially decaying eigenvalues, as detailed in Scheme 2, resulting in high task correlation. This arises because tasks tend to align with the dominant eigen-directions, leading to strong pairwise similarity.

---

**Scheme 2** Generating tasks with high correlation

---

**Require:** Input dimension $d$, task rank $r$, number of tasks $T$, edge eigenvalues $\lambda_1 = 10^{-3}$, $\lambda_d = 10^3$
1: Sample $\mathbf{A} \sim \mathcal{N}(0, 1)^{d \times d}$ and symmetrize: $\mathbf{A}_{\text{sym}} \leftarrow \frac{1}{2}(\mathbf{A} + \mathbf{A}^\top)$
2: Compute SVD: $\mathbf{A}_{\text{sym}} = \mathbf{U}\mathbf{S}\mathbf{U}^\top$
3: Define diagonal spectrum: $\mathbf{\Lambda} \leftarrow \text{diag}\left(\lambda_1 \exp\left(\ln\left(\lambda_d/\lambda_1\right) \frac{i}{d-1}\right)\right)_{i=0}^{d-1}$
4: Construct covariance: $\mathbf{\Sigma} \leftarrow \mathbf{U}\mathbf{\Lambda}\mathbf{U}^\top$
5: **for** $t = 1$ to $T$ **do**
6:     Sample $\mathbf{Z}_t \sim \mathcal{N}(0, 1)^{r \times d}$
7:     Set $\mathbf{X}_t \leftarrow \mathbf{Z}_t \mathbf{\Sigma}^{1/2}$
8: **end for**
9: **Output:** $\{\mathbf{X}_t\}_{t=1}^T$

---

Figures 10 and 11 below reveal some interesting trends compared to the isotropic case.

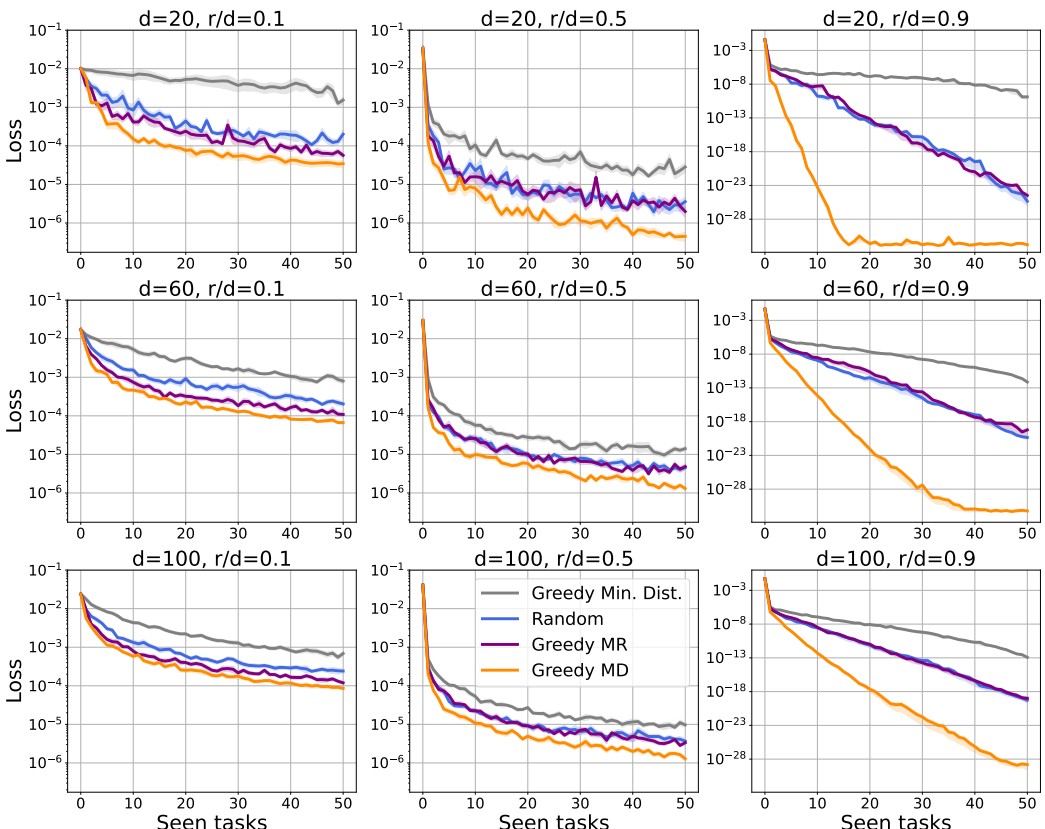

Figure 10: **Comparing orderings for varying dimensions** $d$ **and ranks** $r$ **of the data matrices, for anisotropic data.** $T = 50$. Compared to the isotropic case (Figure 8), we observe slower rates for all strategies. This is easily explained by all pairwise distances between task solution subspaces becoming smaller, due to the higher correlation in the anisotropic case.

Interestingly, as rank increases (focusing on a single row in the grid), the Maximum Residual (MR; Definition 3.2) ordering underperforms and seemingly aligns with the random one. This may stem from the combination of high rank and strong *intra*-task correlation, which leads to *ill-conditioned* data matrices (for each task). In such a case, small perturbations, or steps, in the solution space may cause disproportionately large changes in residuals. As a result, MR is misled into selecting tasks with large residuals that advance the iterate only marginally toward the intersection ($\mathbf{w}_\star$).

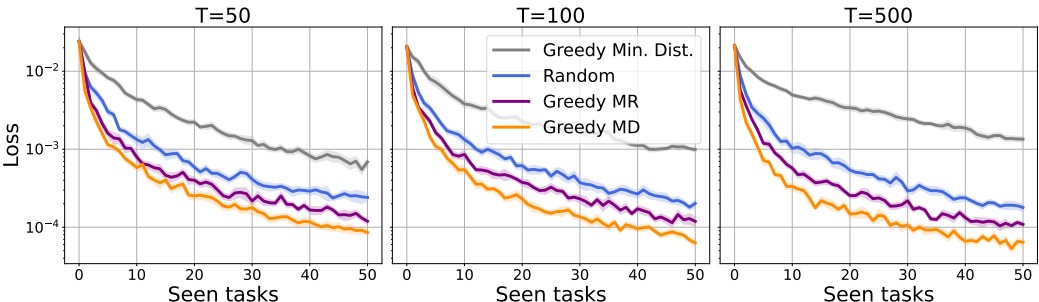

Figure 11: **Comparing orderings for varying task count** $T$**, for anisotropic data.** $d = 100$, $r = 10$. Unlike in the isotropic case (Figure 9), greedy orderings do not significantly benefit from increasing the number of tasks $T$. This is likely since, in the anisotropic case, a large number of tasks must be added to induce the substantial "angles" that greedy orderings can exploit. Put differently, under our anisotropic distribution, the probability that any set of 50 tasks are mutually orthogonal—and thus beneficial to greedy orderings—is extremely small for any reasonable number of tasks $T$.

### B.3 A note on statistical significance

All appendix figures include confidence intervals of $\pm 1$ standard error, although these are often too narrow to be visible. While different task collections introduce slight variations in outcomes, the overall trends are highly consistent. This is illustrated in the following figure, where we replicate the plot from Figure 3a, overlaying individual runs from all 10 repeated experiments. Despite some run-to-run variability, the standard error remains small, reinforcing the robustness of our qualitative conclusions.

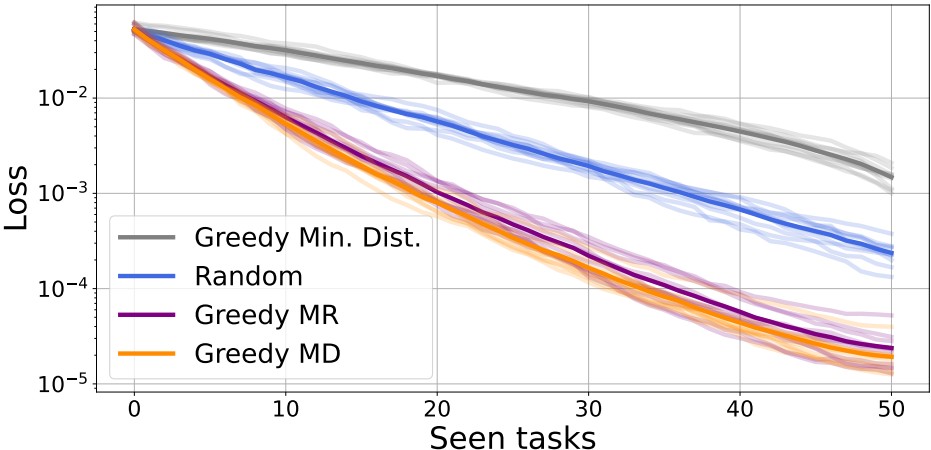

Figure 12: **Showing variations across different experiments.** Same as Figure 3a, we have $T = 50$, $r = 10$, $d = 100$, with random isotropic data. Shaded plots represent each individual experiment. While minor variations exist across experiments, the low standard error confirms the consistency of the results.

# C Appendix to Section 4.1: Classification experiments

## C.1 Code

The code for the classification experiments with `CIFAR-100` is available at `https://github.com/matants/greedy_ordering`.

**Computational resources.** All classification experiments were completed within a month's work on 4 `NVIDIA GeForce GTX 1080 Ti` GPUs.

## C.2 Experiment details

### C.2.1 Model: Linear probing on pretrained `ResNet-20`

Our experiments employ a frozen pretrained `ResNet-20` classifier [45], where the final classification head was removed and replaced with the binary classification head that we train [30, 3].

### C.2.2 Tasks and benchmarks

We employ three benchmarks of domain-incremental `CIFAR-100`-based binary classification:

(A) Using a model pretrained on `CIFAR-100` [55], taken from Chen [22], which achieves 68.83% top-1 classification accuracy on `CIFAR-100` multiclass classification according to Chen [22]. The continual learning tasks are composed of `CIFAR-100` classes randomly split to 50 pairs of binary classification tasks, such that all classes from the same superclass share a label. *This is the setting used in all experiments unless stated otherwise*, including the results presented in Figure 3b.

(B) We partition the 500 training samples of each `CIFAR-100` class to two distinct groups of 250 samples, and use one of the groups to train the `ResNet-20` embedder on the original `CIFAR-100` multiclass task, using the same training recipe as Chen [22] and achieving 61.57% top-1 classification accuracy on the `CIFAR-100` test set after 200 training epochs. The partitioning and training code is included in our provided repository. We then employ a linear probe on top of the resulting model (with the classification head removed), and construct the continual learning tasks using the 250 samples per class that weren't used for training the embedder. The classes that compose each task are the same as in the previous benchmarks.

(C) Using a model pretrained on `CIFAR-10`, taken from Chen [22], with the same `CIFAR-100`-based tasks as (A).

All presented results were composed by experimenting with 25 randomly generated task sets.

### C.2.3 Training

Training was performed with cross-entropy loss on the softmax of the classifier's output with label smoothing of 0.05 [98], with additive L2 regularization towards the previous parameters controlled by the hyperparameter $\lambda$, which is common in continual learning and is necessary to facilitate the projections view, as shown in Evron et al. [34]. In Figure 3b, $\lambda = 5$, and we present an ablation study for $\lambda$ in App. C.4, to address how it affects the performance of different ordering rules.

For each task we used the SGD optimizer with a learning rate of `lr` $= 0.01$ and `ReduceLROnPlateau` on epoch losses, trained for 40 epochs with a batch size of 64. As a baseline, we jointly trained a classifier on all tasks together, without regularization.

### C.2.4 Evaluation

We evaluate the performance of each ordering by calculating the average test (generalization) loss of all tasks after each seen task. Results are presented with 95% confidence intervals, calculated over the different randomly generated task sets, and over the permutations as well for random ordering.

### C.2.5 Ordering computation

For the Random rule (Eq. (4)), we use random sampling *without* replacement from the task set, unless stated otherwise. When presenting results for it, we use 4 random task permutations per task set. The Greedy MR rule (Definition 3.2) requires calculating the loss (without regularization) of all tasks after each task training, and choosing the task with the maximal loss. In App. C.6 we show its performance doesn't degrade when the losses are evaluated on a fraction of the dataset, as small as 1% of the data—5 samples per class in `CIFAR-100`. The Greedy MD rule (Definition 3.1) was calculated by performing full training on each task, as elaborated above (App. C.2.3)—choosing the task that resulted in model parameters farthest from the current model parameters in terms of Euclidean distance. While the MD rule may seem impractical—we show that, in fact, the much simpler Greedy MR rule, that requires a *single* forward pass, achieves identical performance.

### C.3 Out-of-domain feature extractors

To evaluate how our proposed method extends to more general transfer learning settings, we employ multiple benchmarks as elaborated in App. C.2.2.

(A) Figure 13a is the same figure as Figure 3b, shown here for completeness.

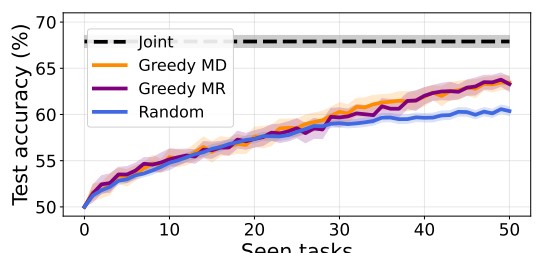

(a) Pretraining: Full `CIFAR-100`; Continual learning: Full `CIFAR-100`.

(B) As shown in Figure 13b, this transfer-learning setting behaves similarly to the case where tasks are drawn from the training data (Figure 3b), though with slightly weaker performance for both the joint baseline and all orderings.

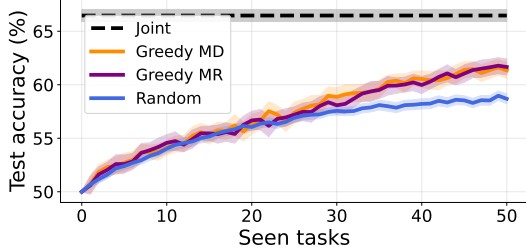

(b) Pretraining: Partitioned `CIFAR-100`; Continual learning: Remaining `CIFAR-100`.

(C) As shown in Figure 13c, dissimilarity-guided orderings still outperform random orderings, though less prominently than in other experiments. We hypothesize that this stems from the model's weak joint interpolation ability (indicated by the dashed curve), which more strongly violates our joint-realizability assumption (Assumption 2.1).

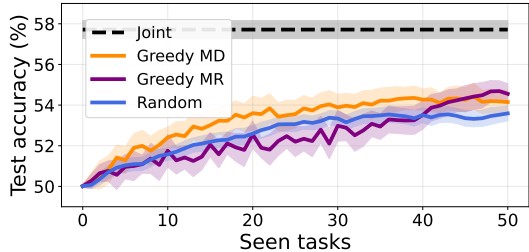

(c) Pretraining: `CIFAR-10`; Continual learning: `CIFAR-100`.

Figure 13: Comparison of orderings under different pretraining setups.

## C.4 Regularization ablation study

Regularization toward previous model parameters is a standard method to mitigate catastrophic forgetting [53, 5, 59], and is crucial for a projections view to emerge in continual classification [34]. Because of its central role, we perform an ablation study on the effect of regularization strength for completeness. As shown in Figure 14, without regularization our continual learning scheme collapses across all orderings, and greedy methods consistently outperform random ordering across all strengths examined. Interestingly, the optimal performance for greedy orderings occurs at smaller regularization strengths ($\lambda$) than for random ordering. This makes sense: greedy methods deliberately select tasks that push parameters further from their current values, so the regularization term has a stronger influence on the loss, requiring smaller $\lambda$ for optimal effect.

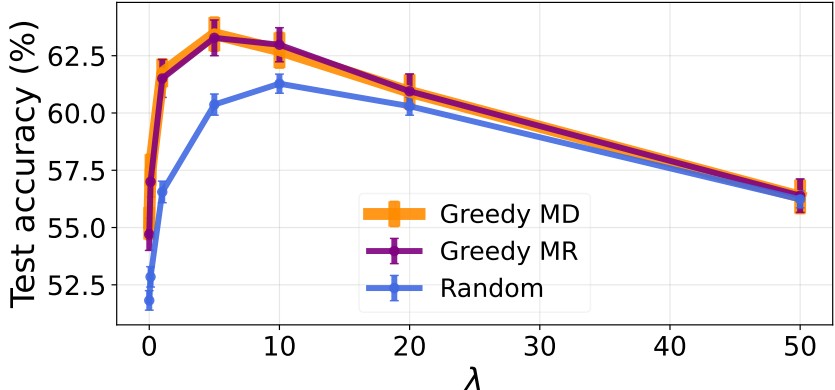

Figure 14: **Regularization strength ablation study.**

## C.5 Allowing task repetition

As discussed in Section 5.2, in continual linear regression, allowing repetitions helps greedy orderings avoid failure modes and guarantees provable convergence. In our classification experiments, however, repetitions do not improve performance: as shown in Figure 15, where each scheme could select from all tasks for 100 iterations (instead of 50), repetitions actually harm the performance of greedy orderings, effectively canceling their advantage over random orderings. We observe a similar phenomenon in linear regression with anisotropic low-rank data, as detailed in App. F.2.

In the classification case, which departs substantially from jointly realizable linear regression, multiple factors could underlie this behavior. It would be interesting to examine how this relates to the connection between greedy ordering and "periphery-to-core" ordering [62], which may break down when repetitions are allowed, or to the distance-to-teacher perspective explored in App. F.2 (Figure 23). As this lies beyond the scope of our paper, we leave it for future work.

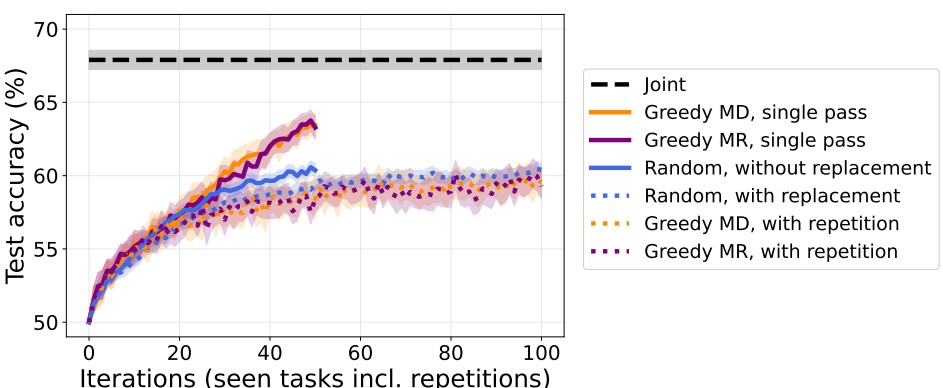

Figure 15: **The effect of task repetition on** `CIFAR-100` **continual classification.**

## C.6 Rule calculation with partial data

To assess practicality, we evaluate the more efficient Greedy MR method—which requires only forward passes—using fractions of the data and compute. Even with just 1% of the data (5 samples per `CIFAR-100` class, i.e., 10 per binary task) to compute each task's loss, Greedy MR maintains its performance and remains stronger than random ordering.

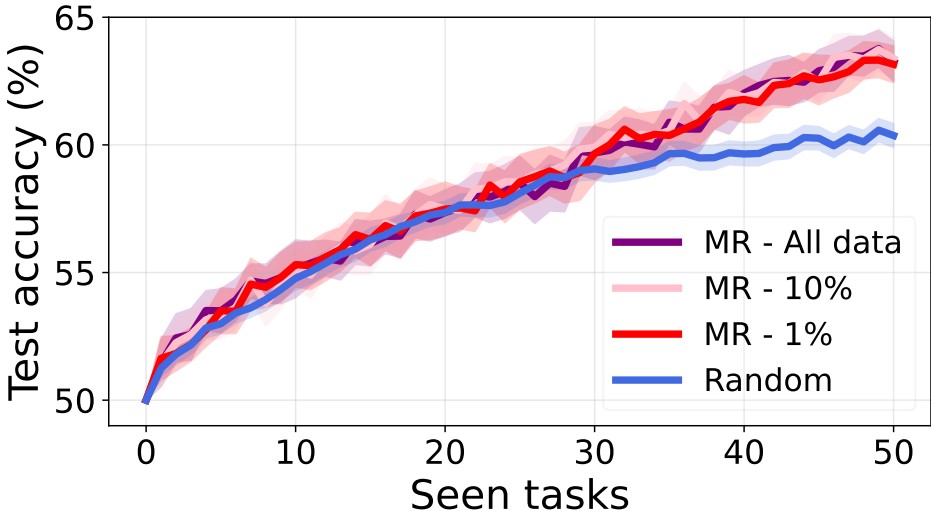

Figure 16: **Greedy MR rule calculation using partial data.**

# D  Appendix to Section 4.2: Proofs for "nearly determined" tasks

## D.1  Optimality guarantee when $r = d - 1$

**Recall Lemma 4.1.** Let $\mathbf{w}_T^{\tau_{\mathrm{MD}}}$ and $\mathbf{w}_T^{\tau_\star}$ be the iterates after learning $T$ jointly realizable tasks of rank $d - 1$ under the Maximum Distance ordering $\tau_{\mathrm{MD}}$ and an optimal ordering $\tau_\star$ that leads to a minimal distance to the joint solution $\mathbf{w}_\star$. Then, their distances hold,

$$0 \le D^2(\mathbf{w}_T^{\tau_\star}) \le D^2(\mathbf{w}_T^{\tau_{\mathrm{MD}}}) \triangleq \frac{\|\mathbf{w}_T^{\tau_{\mathrm{MD}}} - \mathbf{w}_\star\|^2}{\|\mathbf{w}_\star\|^2} \le \frac{\|\mathbf{w}_T^{\tau_\star} - \mathbf{w}_\star\|}{\|\mathbf{w}_\star\|} \triangleq D(\mathbf{w}_T^{\tau_\star}) \le 1 \,.$$

*Proof.* The distance at the end of an ordering $\tau$ is

$$D^2\left(\mathbf{w}_T^\tau\right) \triangleq \frac{\|\mathbf{w}_T^\tau - \mathbf{w}_\star\|^2}{\|\mathbf{w}_\star\|^2} = \frac{1}{\|\mathbf{w}_\star\|^2} \left\| \mathbf{v}_{\tau(T)} \mathbf{v}_{\tau(T)}^\top \cdots \mathbf{v}_{\tau(1)} \mathbf{v}_{\tau(1)}^\top \left(\mathbf{w}_0 - \mathbf{w}^\star\right) \right\|^2$$

$$= \frac{1}{\|\mathbf{w}_\star\|^2} \left(\mathbf{v}_{\tau(1)}^\top \left(\mathbf{w}_0 - \mathbf{w}^\star\right)\right)^2 \cdot \prod_{t=1}^{T-1} \left(\mathbf{v}_{\tau(t)}^\top \mathbf{v}_{\tau(t+1)}\right)^2 \,.$$

Let $\tau = \tau_{\mathrm{MD}}, \tau_\star$ be the greedy MD ordering and an optimal ordering leading to the minimal distance (respectively). Denote for simplicity $c\left(i, j\right) = \begin{cases} \frac{1}{\|\mathbf{w}_\star\|^2} \left(\mathbf{v}_j^\top \left(\mathbf{w}_0 - \mathbf{w}_\star\right)\right)^2 & i = 0, j \in [T] \\ \left(\mathbf{v}_i^\top \mathbf{v}_j\right)^2 & i, j \in [T] \end{cases}$.

Then, we have,

$$D^2\left(\mathbf{w}_T^{\tau_\star}\right) = \frac{1}{\|\mathbf{w}_\star\|^2} \left(\mathbf{v}_{\tau_\star(1)}^\top \left(\mathbf{w}_0 - \mathbf{w}_\star\right)\right)^2 \cdot \prod_{t=1}^{T-1} \left(\mathbf{v}_{\tau_\star(t)}^\top \mathbf{v}_{\tau_\star(t+1)}\right)^2$$

$$= c\left(0, \tau_\star\left(1\right)\right) \prod_{t=1}^{T-1} c\left(\tau_\star\left(t\right), \tau_\star\left(t+1\right)\right)$$

$$= c\left(0, \tau_\star\left(1\right)\right) \prod_{t \in \mathcal{C}} c\left(\tau\left(\tau^{-1}\left(\tau_\star\left(t\right)\right)\right), \tau_\star\left(t+1\right)\right) \cdot \prod_{t \notin \mathcal{C}} c\left(\tau_\star\left(t\right), \tau\left(\tau^{-1}\left(\tau_\star\left(t+1\right)\right)\right)\right),$$

where we define the index set $\mathcal{C} = \left\{t \mid 1 \le t \le T - 1, \ \tau^{-1}\left(\tau_\star\left(t\right)\right) < \tau^{-1}\left(\tau_\star\left(t+1\right)\right)\right\}$.

Employing greediness, we get

$$D^2\left(\mathbf{w}_T^{\tau_\star}\right) \ge c\left(0, \tau\left(1\right)\right) \underbrace{\prod_{t \in \mathcal{C}} c\left(\tau\left(\tau^{-1}\left(\tau_\star\left(t\right)\right)\right), \tau\left(1 + \tau^{-1}\left(\tau_\star\left(t\right)\right)\right)\right)}_{\text{here, } \tau^{-1}(\tau_\star(t)) < T} \cdot$$

$$\cdot \underbrace{\prod_{t \notin \mathcal{C}} c\left(\tau\left(1 + \tau^{-1}\left(\tau_\star\left(t+1\right)\right)\right), \tau\left(\tau^{-1}\left(\tau_\star\left(t+1\right)\right)\right)\right)}_{\text{here, } \tau^{-1}(\tau_\star(t+1)) < T} \,.$$

Then, since $\tau^{-1}\left(\tau_\star\left(\cdot\right)\right)$ "covers" $[T]$ and $c\left(i, j\right) \le 1$, iterating over the entire $1, \ldots, T - 1$ will simply add elements to the product and make it smaller. That is,

$$D^2\left(\mathbf{w}_T^{\tau_\star}\right) \ge c\left(0, \tau\left(1\right)\right) \cdot \prod_{\ell=1}^{T-1} c\left(\tau\left(\ell\right), \tau\left(1 + \ell\right)\right) \cdot \prod_{\ell=1}^{T-1} c\left(\tau\left(1 + \ell\right), \tau\left(\ell\right)\right)$$

$$\ge \left(c\left(0, \tau\left(1\right)\right) \prod_{\ell=1}^{T-1} c\left(\tau\left(\ell\right), \tau\left(1 + \ell\right)\right)\right)^2$$

$$= \left(\frac{1}{\|\mathbf{w}_\star\|^2} \left(\mathbf{v}_{\tau(1)}^\top \left(\mathbf{w}_0 - \mathbf{w}_\star\right)\right)^2 \prod_{t=1}^{T-1} \left(\mathbf{v}_{\tau(t)}^\top \mathbf{v}_{\tau(t+1)}\right)^2\right)^2 = \left(D^2\left(\mathbf{w}_T^\tau\right)\right)^2$$

$$\Rightarrow 1 \ge D\left(\mathbf{w}_T^{\tau_\star}\right) \ge D^2\left(\mathbf{w}_T^{\tau_{\mathrm{MD}}}\right) \ge D^2\left(\mathbf{w}_T^{\tau_\star}\right) \ge 0 \,.$$

$\square$

## D.2 Loss bound when $r = d - 1$

**Recall Lemma 4.2.** Under the Maximum Distance greedy ordering over $T$ jointly-realizable tasks of rank $d-1$, the loss of Scheme 1 after $T$ iterations is upper bounded as,

$$\mathcal{L}(\mathbf{w}_T) = \tfrac{1}{\|\mathbf{w}_\star\|^2 R^2} \cdot \frac{1}{T} \sum_{m=1}^{T} \|\mathbf{X}_m \mathbf{w}_T - \mathbf{y}_m\|^2 \leq \frac{1}{eT} .$$

*Proof.* We aim to bound the average loss using projection matrices,

$$\mathcal{L}_{\tau_{\text{MD}}}(\mathbf{w}_T) = \tfrac{1}{\|\mathbf{w}_\star\|^2 R^2 T} \sum_{m=1}^{T} \|\mathbf{X}_m \mathbf{w}_T - \mathbf{y}_m\|^2 = \tfrac{1}{\|\mathbf{w}_\star\|^2 R^2 T} \sum_{m=1}^{T} \|\mathbf{X}_m (\mathbf{w}_T - \mathbf{w}_\star)\|^2$$

$$= \tfrac{1}{\|\mathbf{w}_\star\|^2 R^2 T} \sum_{m=1}^{T} \|\mathbf{X}_m \mathbf{X}_m^+ \mathbf{X}_m (\mathbf{w}_T - \mathbf{w}_\star)\|^2$$

$$\leq \tfrac{1}{\|\mathbf{w}_\star\|^2 R^2 T} \sum_{m=1}^{T} \|\mathbf{X}_m\|^2 \| (\mathbf{I} - \mathbf{P}_m) (\mathbf{w}_T - \mathbf{w}_\star)\|^2$$

$$[\text{Eq. (2)}] \leq \tfrac{1}{\|\mathbf{w}_\star\|^2 T} \sum_{t=1}^{T} \left\| (\mathbf{I} - \mathbf{P}_{\tau(t)}) \prod_{s=1}^{T} \mathbf{P}_{\tau(s)} (\mathbf{w}_0 - \mathbf{w}_\star) \right\|^2 .$$

Since each task matrix $\mathbf{X}_i$ has rank $d - 1$, each projection $\mathbf{P}_i$ is rank 1 and can be written as $\mathbf{P}_i = \mathbf{v}_i \mathbf{v}_i^\top$ for a unit vector $\mathbf{v}_i$. Substituting this and $\mathbf{v}_{\tau(0)} = \tfrac{1}{\|\mathbf{w}_\star\|} (\mathbf{w}_0 - \mathbf{w}_\star)$, the bound becomes:

$$\mathcal{L}_{\tau_{\text{MD}}}(\mathbf{w}_T) \leq \frac{1}{T} \sum_{t=1}^{T} \left\| (\mathbf{I} - \mathbf{v}_{\tau(t)} \mathbf{v}_{\tau(t)}^\top) \mathbf{v}_{\tau(T)} \mathbf{v}_{\tau(T)}^\top \cdots \mathbf{v}_{\tau(1)} \mathbf{v}_{\tau(1)}^\top \mathbf{v}_{\tau(0)} \right\|^2$$

$$\leq \underbrace{\left(\mathbf{v}_{\tau(1)}^\top \mathbf{v}_{\tau(0)}\right)^2}_{\leq 1} \frac{1}{T} \sum_{t=1}^{T} \left\| (\mathbf{I} - \mathbf{v}_{\tau(t)} \mathbf{v}_{\tau(t)}^\top) \mathbf{v}_{\tau(T)} \right\|^2 \prod_{s=1}^{T-1} \left(\mathbf{v}_{\tau(s+1)}^\top \mathbf{v}_{\tau(s)}\right)^2$$

$$[\text{projection properties}] \leq \left(1 - \frac{1}{T} \sum_{s=1}^{T} \left(\mathbf{v}_{\tau(T)}^\top \mathbf{v}_{\tau(s)}\right)^2\right) \prod_{s=1}^{T-1} \left(\mathbf{v}_{\tau(s+1)}^\top \mathbf{v}_{\tau(s)}\right)^2 .$$

Then, we use algebraic and projection properties to rewrite the greedy ordering as:

$$\tau_{\text{MD}}(t) = \underset{m \in [T] \setminus \tau_{\text{MD}}(1:t-1)}{\arg\max} \| (\mathbf{I} - \mathbf{P}_m) (\mathbf{w}_{t-1} - \mathbf{w}_\star)\|^2$$

$$= \underset{m \in [T] \setminus \tau_{\text{MD}}(1:t-1)}{\arg\max} \left( \|\mathbf{w}_{t-1} - \mathbf{w}_\star\|^2 - \|\mathbf{P}_m (\mathbf{w}_{t-1} - \mathbf{w}_\star)\|^2 \right)$$

$$= \underset{m \in [T] \setminus \tau_{\text{MD}}(1:t-1)}{\arg\min} \|\mathbf{P}_m (\mathbf{w}_{t-1} - \mathbf{w}_\star)\|^2$$

$$= \underset{m \in [T] \setminus \tau_{\text{MD}}(1:t-1)}{\arg\min} \|\mathbf{v}_m \mathbf{v}_m^\top \mathbf{v}_{\tau(t-1)} \mathbf{v}_{\tau(t-1)}^\top (\mathbf{w}_{t-2} - \mathbf{w}_\star)\|^2$$

$$= \underset{m \in [T] \setminus \tau_{\text{MD}}(1:t-1)}{\arg\min} \left(\mathbf{v}_m^\top \mathbf{v}_{\tau(t-1)}\right)^2 . \tag{6}$$

Then, employing greediness as reformulated above and inequality of arithmetic and geometric mean, we obtain:

$$\prod_{s=1}^{T-1} \left(\mathbf{v}_{\tau(s+1)}^\top \mathbf{v}_{\tau(s)}\right)^2 \leq \prod_{s=1}^{T} \left(\mathbf{v}_{\tau(T)}^\top \mathbf{v}_{\tau(s)}\right)^2 \leq \left(\frac{1}{T} \sum_{s=1}^{T} \left(\mathbf{v}_{\tau(T)}^\top \mathbf{v}_{\tau(s)}\right)^2\right)^T .$$

Substituting back into the forgetting, it is now bounded as,

$$\mathcal{L}_{\tau_{\text{MD}}}(\mathbf{w}_T) \leq \left(1 - \frac{1}{T} \sum_{s=1}^{T} \left(\mathbf{v}_{\tau(T)}^\top \mathbf{v}_{\tau(s)}\right)^2\right) \left(\frac{1}{T} \sum_{s=1}^{T} \left(\mathbf{v}_{\tau(T)}^\top \mathbf{v}_{\tau(s)}\right)^2\right)^T \leq \frac{1}{eT} ,$$

where we invoked an algebraic property that $(1 - x)x^T \leq \frac{1}{eT}, \forall x \in [0, 1]$. $\qquad \square$

# E  Appendix to Section 5.1: Lower bound's "adversarial" constructions

## E.1  General dimension construction and proof (Theorem 5.2)

**Recall Theorem 5.2.** For any $d \geq 30$, there exists an adversarial task collection with $T = d - 1$ jointly-realizable tasks of different rank such that both greedy orderings (MD, MR) forget *catastrophically*. That is, the loss at the end of the sequence is, $\mathcal{L}(\mathbf{w}_T^{\mathcal{T}_{\mathrm{MD}}}), \mathcal{L}(\mathbf{w}_T^{\mathcal{T}_{\mathrm{MR}}}) \geq \frac{1}{8} - \frac{1}{4d}$.

**Proof outline.** For a given dimension $d$, we construct a sequence of $d$ iterates $(\mathbf{w}_t)_{t=1}^d$, corresponding to $T = d - 1$ tasks $(\mathbf{X}_t)_{t=2}^d$ of decreasing rank, which are jointly-realizable with $\mathbf{w}_\star = \mathbf{0}$ (*i.e.*, $\forall t \in \{2...T\}$, $\mathbf{y}_t = \mathbf{0}$), and show that:

1. **Bottom line.** Given this specific choice of tasks and matching iterates, the loss (or forgetting) is catastrophic as mentioned in the theorem.

2. The chosen iterates are valid—*i.e.*, they can be obtained from a specific selection rule given the constructed task collection.

3. The chosen ordering adheres to greedy selection rules, both MD and MR, under the chosen tasks. *This part is quite lengthy.*

In the construction, we start the iterates from $t = 1$ and tasks from $t = 2$, contrary to other parts of the paper, for no particular reason other than ease of notation. For this same reason we chose $\mathbf{w}_\star = \mathbf{0}$, and the iterates starting with $\mathbf{w}_1 = \mathbf{e}_1$. The same construction holds for a shifted frame of reference where all iterates (and $\mathbf{w}_\star$) are shifted by $-\mathbf{e}_1$.

### E.1.1  Construction details

We first construct the *iterates* as follows:

$$\mathbf{w}_1 = \mathbf{e}_1 = \left[ 1, \underbrace{0, \ldots, 0}_{d-1 \text{ times}} \right]^\top,$$

$$\forall t \in \{2...d\} : \mathbf{w}_t = \left[ \frac{(\mathbf{w}_{t-1})_1 + \sqrt{(\mathbf{w}_{t-1})_1^2 - 4\beta_t}}{2}, \underbrace{c^{t-2}\frac{1}{\sqrt{d}}, \ldots, c^{t-2}\frac{1}{\sqrt{d}}}_{t-1 \text{ times}}, 0, \ldots, 0 \right]^\top,$$

where $c \triangleq 2^{-1/d}$ and $\beta_t \triangleq \frac{((t-1)c - (t-2))c^{2t-5}}{d}$.

We denote $x_t \triangleq (\mathbf{w}_t)_1$, defined recursively by

$$x_1 = 1, \quad x_t = \frac{x_{t-1} + \sqrt{x_{t-1}^2 - 4\beta_t}}{2}, \forall t \in \{2...d\} . \tag{7}$$

Since $\mathbf{w}_t \neq \mathbf{w}_{t-1}$, we are free to define the unit vector

$$\mathbf{u}_t = \frac{\mathbf{w}_t - \mathbf{w}_{t-1}}{\|\mathbf{w}_t - \mathbf{w}_{t-1}\|} \in \mathrm{span}\left(\mathbf{e}_1, \ldots, \mathbf{e}_t\right) .$$

We now construct the tasks:

$$\mathbf{X}_t = \begin{bmatrix} -\mathbf{u}_t^\top - \\ -\mathbf{e}_{t+1}^\top - \\ \vdots \\ -\mathbf{e}_d^\top - \end{bmatrix} = \begin{bmatrix} -\mathbf{u}_t^\top - \\ \mathbf{I}_{t+1:d} \end{bmatrix} \in \mathbb{R}^{(d-t+1) \times d}, \forall t \in \{2...d\} .$$

Then, it is easy to see that $\mathbf{P}_t \triangleq \mathbf{I}_d - \mathbf{X}_t^+ \mathbf{X}_t = \mathbf{I}_d - \mathbf{I}_{t+1:d} - \mathbf{u}_t \mathbf{u}_t^\top = \underbrace{\mathbf{I}_t}_{\text{rank } t} - \mathbf{u}_t \mathbf{u}_t^\top .$

### E.1.2 Lower bounding the loss

For each task $\mathbf{X}_m$, its individual loss at time $t = d$ is given by:

$$\mathcal{L}_m\left(\mathbf{w}_d\right) \triangleq \|\mathbf{X}_m\mathbf{w}_d\|^2 = \left\|\begin{bmatrix} -\mathbf{u}_m^\top- \\ \mathbf{I}_{m+1:d} \end{bmatrix}\mathbf{w}_d\right\|^2 = \left(\mathbf{u}_m^\top\mathbf{w}_d\right)^2 + \|\mathbf{I}_{m+1:d}\mathbf{w}_d\|^2$$

$$\geq \|\mathbf{I}_{m+1:d}\mathbf{w}_d\|^2 = \sum_{j=m+1}^{d}\left(\mathbf{w}_d\right)_j^2$$

$$[j \geq 2] = (d-m)\frac{c^{2d-4}}{d} = \left(1-\frac{m}{d}\right)c^{2d-4} = \left(1-\frac{m}{d}\right)2^{-(2d-4)/d}$$

$$= \frac{1}{4}\left(1-\frac{m}{d}\right)2^{4/d} \geq \frac{1}{4}\left(1-\frac{m}{d}\right).$$

So the average loss after all iterates, which coincides with the forgetting (see Remark 2.4) is:

$$\mathcal{L}\left(\mathbf{w}_d\right) = \frac{1}{T}\sum_{m\in\{2\ldots d\}}\mathcal{L}_m\left(\mathbf{w}_d\right) = \frac{1}{d-1}\sum_{m=2}^{d}\mathcal{L}_m\left(\mathbf{w}_d\right)$$

$$\geq \frac{1}{4(d-1)}\sum_{m=2}^{d}\left(1-\frac{m}{d}\right) = \frac{1}{4(d-1)}\left(d-1-\frac{\sum_{m=2}^{d}m}{d}\right)$$

$$= \frac{1}{4} - \frac{d+2}{8d} = \frac{1}{8} - \frac{1}{4d}.$$

### E.1.3 Proving that the iterates and tasks exist

In Lemma J.1, we prove that for all $d \geq 30$, $t \in \{2,\ldots,d\}$, we have $x_{t-1}^2 - 4\beta_t \geq 0$, so the square root in the recursive definition of $x_t$ (Eq. (7)) exists.

### E.1.4 Proving that the iterates can be formed from projections of the given tasks

As a sanity check, we notice that $\mathbf{P}_t$ is a real symmetric matrix, and assert its idempotence,

$$\mathbf{P}_t^2 = \left(\mathbf{I}_t - \mathbf{u}_t\mathbf{u}_t^\top\right)^2 = \mathbf{I}_t^2 - \mathbf{u}_t\mathbf{u}_t^\top\mathbf{I}_t - \mathbf{I}_t\mathbf{u}_t\mathbf{u}_t^\top + \mathbf{u}_t\mathbf{u}_t^\top\mathbf{u}_t\mathbf{u}_t^\top$$

$$= \mathbf{I}_t - \mathbf{u}_t\mathbf{u}_t^\top - \mathbf{u}_t\mathbf{u}_t^\top + \mathbf{u}_t\mathbf{u}_t^\top = \mathbf{I}_t - \mathbf{u}_t\mathbf{u}_t^\top = \mathbf{P}_t.$$

First, we show that $\mathbf{w}_t^\top\left(\mathbf{w}_t - \mathbf{w}_{t-1}\right) = 0$, as expected from orthogonality in projections:

$$\mathbf{w}_t^\top\left(\mathbf{w}_t - \mathbf{w}_{t-1}\right) = \sum_{i=1}^{d}\left(\mathbf{w}_t\right)_i^2 - \sum_{i=1}^{d}\left(\mathbf{w}_t\right)_i\left(\mathbf{w}_{t-1}\right)_i = \left(\mathbf{w}_t\right)_1^2 + \sum_{i=2}^{t}\left(\mathbf{w}_t\right)_i^2 - \sum_{i=1}^{t-1}\left(\mathbf{w}_t\right)_i\left(\mathbf{w}_{t-1}\right)_i$$

$$= \left(\mathbf{w}_t\right)_1^2 - \left(\mathbf{w}_t\right)_1\left(\mathbf{w}_{t-1}\right)_1 + \sum_{i=2}^{t}\frac{c^{2t-4}}{d} - \sum_{i=2}^{t-1}\frac{c^{t-2}c^{t-3}}{d}$$

$$= \left(\mathbf{w}_t\right)_1^2 - \left(\mathbf{w}_t\right)_1\left(\mathbf{w}_{t-1}\right)_1 + \frac{(t-1)c^{2t-4} - (t-2)c^{2t-5}}{d}$$

$$= \left(\mathbf{w}_t\right)_1^2 - \left(\mathbf{w}_t\right)_1\left(\mathbf{w}_{t-1}\right)_1 + \underbrace{\frac{((t-1)c - (t-2))c^{2t-5}}{d}}_{=\beta_t}$$

$$= \left(\mathbf{w}_t\right)_1^2 - \left(\mathbf{w}_t\right)_1\left(\mathbf{w}_{t-1}\right)_1 + \beta_t,$$

and it is readily seen that our construction choice of $\left(\mathbf{w}_t\right)_1 = \frac{\left(\mathbf{w}_{t-1}\right)_1 + \sqrt{\left(\mathbf{w}_{t-1}\right)_1^2 - 4\beta_t}}{2}$ implies

$$\mathbf{w}_t^\top\left(\mathbf{w}_t - \mathbf{w}_{t-1}\right) = 0.$$

Finally, we show that the iterates are indeed a sequence the corresponding projections:

$$\mathbf{P}_t \mathbf{w}_{t-1} = \left(\mathbf{I}_t - \mathbf{u}_t \mathbf{u}_t^\top\right) \mathbf{w}_{t-1} = \mathbf{I}_t \mathbf{w}_{t-1} - \mathbf{u}_t \mathbf{u}_t^\top \mathbf{w}_{t-1}$$

$$= \mathbf{w}_{t-1} - \left(\frac{(\mathbf{w}_t - \mathbf{w}_{t-1})^\top}{\|\mathbf{w}_t - \mathbf{w}_{t-1}\|} \mathbf{w}_{t-1}\right) \mathbf{u}_t = \mathbf{w}_{t-1} - \frac{(\mathbf{w}_t - \mathbf{w}_{t-1})^\top \mathbf{w}_{t-1}}{\|\mathbf{w}_t - \mathbf{w}_{t-1}\|^2} (\mathbf{w}_t - \mathbf{w}_{t-1})$$

$$= \mathbf{w}_{t-1} - \frac{(\mathbf{w}_t - \mathbf{w}_{t-1})^\top \mathbf{w}_{t-1} - \overbrace{(\mathbf{w}_t - \mathbf{w}_{t-1})^\top \mathbf{w}_t}^{=0}}{\|\mathbf{w}_t - \mathbf{w}_{t-1}\|^2} (\mathbf{w}_t - \mathbf{w}_{t-1})$$

$$= \mathbf{w}_{t-1} + \frac{(\mathbf{w}_t - \mathbf{w}_{t-1})^\top (\mathbf{w}_t - \mathbf{w}_{t-1})}{\|\mathbf{w}_t - \mathbf{w}_{t-1}\|^2} (\mathbf{w}_t - \mathbf{w}_{t-1})$$

$$= \mathbf{w}_{t-1} + (\mathbf{w}_t - \mathbf{w}_{t-1}) = \mathbf{w}_t \,.$$

### E.1.5   Proving that the iterates adhere to greedy ordering rules

**Maximum Distance (MD).**   We wish to prove that the greedy MD rule agrees with the ordering we chose. That is,

$$\tau_t \triangleq \mathrm{argmax}_{t' \in [T] \setminus \{\tau_2, \ldots, \tau_{t-1}\}} \left\|(\mathbf{I} - \mathbf{P}_{t'}) \mathbf{w}_{t-1}\right\|^2 = t \,.$$

By induction on the validity of the greediness for $\tau_2, \ldots, \tau_{t-1}$, the step is (and the induction base for $t = 2$ is shown exactly the same):

$$\tau_t \triangleq \mathrm{argmax}_{t' \in [T] \setminus \{\tau_2, \ldots, \tau_{t-1}\}} \left\|(\mathbf{I} - \mathbf{P}_{t'}) \mathbf{w}_{t-1}\right\|^2$$

$$[\text{induction assumption}] = \mathrm{argmax}_{t' \in \{t, \ldots, T\}} \left\|(\mathbf{I}_d - \mathbf{P}_{t'}) \mathbf{w}_{t-1}\right\|^2$$

$$= \mathrm{argmax}_{t' \in \{t, \ldots, T\}} \left\|\left(\mathbf{I}_d - \mathbf{I}_{t'} + \mathbf{u}_{t'} \mathbf{u}_{t'}^\top\right) \mathbf{w}_{t-1}\right\|^2$$

$$[t' > t - 1] = \mathrm{argmax}_{t' \in \{t, \ldots, T\}} \left\|\cancel{(\mathbf{I}_d - \mathbf{I}_{t'}) \mathbf{w}_{t-1}} + \mathbf{u}_{t'} \mathbf{u}_{t'}^\top \mathbf{w}_{t-1}\right\|^2$$

$$\left[\|\mathbf{u}_{t'}\|^2 = 1\right] = \mathrm{argmax}_{t' \in \{t, \ldots, T\}} \left(\mathbf{u}_{t'}^\top \mathbf{w}_{t-1}\right)^2$$

$$= \mathrm{argmax}_{t' \in \{t, \ldots, T\}} \left(\frac{(\mathbf{w}_{t'} - \mathbf{w}_{t'-1})^\top}{\|\mathbf{w}_{t'} - \mathbf{w}_{t'-1}\|} \mathbf{w}_{t-1}\right)^2 \,.$$

**Maximum Residual (MR).**   We wish to prove that the greedy MR rule agrees with the ordering we chose. That is,

$$\tau_t \triangleq \mathrm{argmax}_{t' \in [T] \setminus \{\tau_2, \ldots, \tau_{t-1}\}} \left\|\mathbf{X}_{t'} \mathbf{w}_{t-1}\right\|^2 = t \,.$$

By induction on the validity of the greediness for $\tau_2, \ldots, \tau_{t-1}$, the step is (and the induction base for $t = 2$ is shown exactly the same):

$$\tau_t \triangleq \mathrm{argmax}_{t' \in [T] \setminus \{\tau_2, \ldots, \tau_{t-1}\}} \left\|\mathbf{X}_{t'} \mathbf{w}_{t-1}\right\|^2$$

$$[\text{induction assumption}] = \mathrm{argmax}_{t' \in \{t, \ldots, T\}} \left\|\mathbf{X}_{t'} \mathbf{w}_{t-1}\right\|^2 = \mathrm{argmax}_{t' \in \{t, \ldots, T\}} \left\|\begin{bmatrix} -\mathbf{u}_{t'}^\top - \\ \mathbf{I}_{t'+1:d} \end{bmatrix} \mathbf{w}_{t-1}\right\|^2$$

$$[t' > t - 1] = \mathrm{argmax}_{t' \in \{t, \ldots, T\}} \left(\left(\mathbf{u}_{t'}^\top \mathbf{w}_{t-1}\right)^2 + \|\cancel{\mathbf{I}_{t'+1:d} \mathbf{w}_{t-1}}\|^2\right)$$

$$= \mathrm{argmax}_{t' \in \{t, \ldots, T\}} \left(\frac{(\mathbf{w}_{t'} - \mathbf{w}_{t'-1})^\top}{\|\mathbf{w}_{t'} - \mathbf{w}_{t'-1}\|} \mathbf{w}_{t-1}\right)^2 \,.$$

We get that the MR and MD rules coincide in this case.

**How we prove greediness holds: Delta positivity.** We wish to show monotonous decrease (w.r.t. $k \geq t$) of $\left( \frac{\left( (\mathbf{w}_{k-1} - \mathbf{w}_k)^\top \mathbf{w}_{t-1} \right)^2}{\|\mathbf{w}_{k-1} - \mathbf{w}_k\|^2} \right)_k$, which will prove that the iterates we defined are valid under the greedy MD and MR orderings (*i.e.,* adhere to the rules in Def. 3.1 and 3.2).

The difference between consecutive iterates is

$$\mathbf{w}_{k-1} - \mathbf{w}_k = \left[ x_{k-1} - x_k, \underbrace{\frac{c^{k-3}(1-c)}{\sqrt{d}}, \ldots, \frac{c^{k-3}(1-c)}{\sqrt{d}}}_{k-2 \text{ times}}, -\frac{c^{k-2}}{\sqrt{d}}, 0, \ldots, 0 \right].$$

We notice that $\forall k \geq t$ the term $(\mathbf{w}_{k-1} - \mathbf{w}_k)^\top \mathbf{w}_{t-1}$ is **positive** since,

$$(\mathbf{w}_{k-1} - \mathbf{w}_k)^\top \mathbf{w}_{t-1} = \underbrace{(x_{k-1} - x_k)}_{>0, \text{ from I.6}} \underbrace{x_{t-1}}_{>0} + (t-2) \underbrace{\frac{c^{k-3}(1-c)}{\sqrt{d}} \frac{c^{t-3}}{\sqrt{d}}}_{>0} > 0.$$

This means that we can alternatively show monotonous decrease $\forall k \geq t$ for

$$\left( \frac{(\mathbf{w}_{k-1} - \mathbf{w}_k)^\top \mathbf{w}_{t-1}}{\|\mathbf{w}_{k-1} - \mathbf{w}_k\|} \right)_k.$$

To this end, we wish to show that the next quantity is **positive** $\forall t \in \{2, \ldots, d-1\}$ (we are reminded that the first step is at $t=2$ due to our choice, and that at the last step there is only one choice), $\forall k \in \{t, \ldots, d-1\}$:

$$\frac{(\mathbf{w}_{k-1} - \mathbf{w}_k)^\top \mathbf{w}_{t-1}}{\|\mathbf{w}_{k-1} - \mathbf{w}_k\|} - \frac{(\mathbf{w}_k - \mathbf{w}_{k+1})^\top \mathbf{w}_{t-1}}{\|\mathbf{w}_k - \mathbf{w}_{k+1}\|}$$

$$\propto \|\mathbf{w}_k - \mathbf{w}_{k+1}\| (\mathbf{w}_{k-1} - \mathbf{w}_k)^\top \mathbf{w}_{t-1} - \|\mathbf{w}_{k-1} - \mathbf{w}_k\| (\mathbf{w}_k - \mathbf{w}_{k+1})^\top \mathbf{w}_{t-1} \triangleq \Delta_{t,k}.$$

Next, we will show this holds numerically for low dimensions ($d < 25{,}000$), and prove it analytically $\forall d \geq 25{,}000$.

**Showing delta positivity numerically for low dimensions.** We use the following facts to write code that verifies $\Delta_{t,k} > 0 \ \forall d < 25{,}000, \ \forall t \in \{2, \ldots, d-1\}, \ \forall k \in \{t, \ldots, d-1\}$:

$$\|\mathbf{w}_{k-1} - \mathbf{w}_k\| = \sqrt{(x_{k-1} - x_k)^2 + (k-2)\left( \frac{c^{k-3}(1-c)}{\sqrt{d}} \right)^2 + \left( \frac{c^{k-3}}{\sqrt{d}} \right)^2}$$

$$= \sqrt{(x_{k-1} - x_k)^2 + \frac{k-2}{d} c^{2k-6}(1-c)^2 + \frac{1}{d} c^{2k-6}},$$

$$(\mathbf{w}_{k-1} - \mathbf{w}_k)^\top \mathbf{w}_{t-1} = (x_{k-1} - x_k) x_{t-1} + (t-2) \frac{c^{k-3}(1-c)}{\sqrt{d}} \frac{c^{t-3}}{\sqrt{d}}.$$

For each value of dimension $d$, we calculated the sequence $(x)_k$ using its recursive definition, and calculated $\Delta(d) \triangleq \min_{\{t,k \,|\, t \in \{2,\ldots,d-1\}, \, k \in \{t,\ldots,d-1\}\}} \Delta_{t,k}$ using these formulas. As shown in Figure 17, we found $\Delta(d)$ remains positive for all $d \in \{30\ldots47{,}000\}$ (for completeness, any dimension above 25,000 is redundant here). In addition, as will be seen analytically (Eq. (13)), we have that $\Delta(d)$ should correlate with $d^{-\frac{5}{2}}$, and for completeness we show this holds numerically for the lower dimensions as well, by showing $\Delta(d) \cdot d^{\frac{5}{2}}$ is approximately constant.

**Computational resources.** This numerical validation took 4 days to run on a home PC with i5-9400F CPU and 16GB RAM.

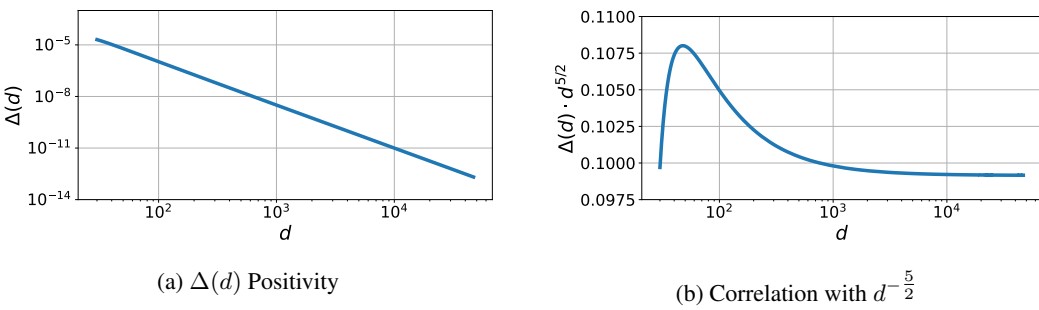

(a) $\Delta(d)$ Positivity

(b) Correlation with $d^{-\frac{5}{2}}$

Figure 17: **Numerical positivity of** $\Delta(d) \triangleq \min_{\{t,k \,|\, t \in \{2,\ldots,d-1\},\, k \in \{t,\ldots,d-1\}\}} \Delta_{t,k}$

**Showing delta positivity analytically for high dimensions.** Due to the length of this part, we defer it to App. I, where we prove that $\forall d \geq 25{,}000,\ \forall t \in \{2,\ldots,d-1\},\ \forall k \in \{t,\ldots,d-1\}$,

$$\Delta_{t,k} \triangleq \|\mathbf{w}_k - \mathbf{w}_{k+1}\| \left(\mathbf{w}_{k-1} - \mathbf{w}_k\right)^\top \mathbf{w}_{t-1} - \|\mathbf{w}_{k-1} - \mathbf{w}_k\| \left(\mathbf{w}_k - \mathbf{w}_{k+1}\right)^\top \mathbf{w}_{t-1} > 0 \,.$$

**Conclusion.** Together with the numerical verification, we have established that $\Delta_{t,k} > 0$ for all $k \geq t$ and all $d \geq 30$. This completes the proof of the iterates' adherence to the greedy ordering rules, and thereby concludes the overall proof of the adversarial construction that yields a lower bound on the loss under single-pass greedy orderings.

## E.2  Adversarial 3d construction (Example 5.1)

**Recall Example 5.1.** For all $T \in \left\{ 4 \cdot 10^i - 1 \mid i = 1, 2, \ldots, 7 \right\}$, there exists a task collection of jointly-realizable tasks in $d = 3$, such that $\mathcal{L}(\mathbf{w}_T^{\mathcal{T}\text{MD}}), \mathcal{L}(\mathbf{w}_T^{\mathcal{T}\text{MR}}) > 2.78 \cdot 10^{-5}$.

### E.2.1  Construction details

For simplicity we employ the joint solution $\mathbf{w}_\star = \mathbf{0}$, which means that for all tasks $m \in [T]$, $\mathbf{y}_m = \mathbf{0}$. For some $K \in \mathbb{N}^+$, we construct $T = 4K - 1$ tasks by defining their *solution subspaces* as follows:

- $K - 1$ copies of $\mathrm{Span}\left(\left(0, \sin\left(\frac{1}{\sqrt{K}}\right), \cos\left(\frac{1}{\sqrt{K}}\right)\right)\right)$,

- $K$ copies of $\mathrm{Span}\left(\left(0, -\sin\left(\frac{1}{\sqrt{K}}\right), \cos\left(\frac{1}{\sqrt{K}}\right)\right)\right)$,

- $K$ copies of $\mathrm{Span}\left(\left(0, \sin\left(\frac{1}{2\sqrt{K}}\right), \cos\left(\frac{1}{2\sqrt{K}}\right)\right), \left(\frac{1}{\sqrt{2}}, 0, \frac{1}{\sqrt{2}}\right)\right)$,

- $K$ copies of $\mathrm{Span}\left(\left(0, -\sin\left(\frac{1}{2\sqrt{K}}\right), \cos\left(\frac{1}{2\sqrt{K}}\right)\right), \left(\frac{1}{\sqrt{2}}, 0, \frac{1}{\sqrt{2}}\right)\right)$.

For a given solution subspace of rank $d - r$, a task feature matrix $\mathbf{X} \in \mathbb{R}^{r \times d}$ of rank $r$ (*i.e.,* with linearly independent rows) is defined such that each row of $\mathbf{X}$ is orthogonal to the solution subspace.

We initialize $\mathbf{w}_0 = \left(0, \sin\left(\frac{1}{\sqrt{K}}\right), \cos\left(\frac{1}{\sqrt{K}}\right)\right)$.

### E.2.2  Lower bound explanation

This is *not* a formal proof, but an explanation of why the construction works. We stick to using greedy MD for the intuition.

While learning tasks consecutively using greedy MD ordering, we start by alternating between projecting onto the 1-D subspaces, since each 2-D subspace contains a line (either $\mathrm{Span}\left(\left(0, \sin\left(\frac{1}{2\sqrt{K}}\right), \cos\left(\frac{1}{2\sqrt{K}}\right)\right)\right)$ or $\mathrm{Span}\left(\left(0, -\sin\left(\frac{1}{2\sqrt{K}}\right), \cos\left(\frac{1}{2\sqrt{K}}\right)\right)\right)$) between them and hence cannot be the farthest away. Once those are used up, we're left with the 2-D subspaces, which we alternate between.

Morally, the angle between the 1-D subspaces is $\mathcal{O}\left(\frac{1}{\sqrt{K}}\right)$, so as $K$ grows large, these first $2K - 1$ steps should bring us to about $(0, 0, \Theta(1))$. The 2-D subspaces intersect in $\mathrm{Span}\left(\left(\frac{1}{\sqrt{2}}, 0, \frac{1}{\sqrt{2}}\right)\right)$, and the angle between them is also $\Theta\left(\frac{1}{\sqrt{K}}\right)$, so they move us a constant fraction of the way toward the closest point on $\mathrm{Span}\left(\left(\frac{1}{\sqrt{2}}, 0, \frac{1}{\sqrt{2}}\right)\right)$, which is $(\Theta(1), 0, \Theta(1))$. Since the first half of the tasks are projections onto 1-D subspaces near $(0, 0, 1)$, the $x$-coordinate contributes $\Theta(1)$ loss due to these tasks.

**Quantifying the asymptotic loss.** In the first $2K - 1$ steps we multiply $\|\mathbf{w}\|$ by $\cos(2/\sqrt{K})$ each time, giving

$$\cos\left(2/\sqrt{K}\right)^{2K-1} = \left(1 - 2/K + O(K^{-2})\right)^{2K-1} = e^{-4} + o(1).$$

The direction is $\left(0, -\sin\left(1/\sqrt{K}\right), \cos\left(1/\sqrt{K}\right)\right)$, so we end at $(0, o(1), e^{-4} + o(1))$.

Next we project onto a 2-D subspace; this is $o(1)$ movement, so $\mathbf{w}$ becomes

$$\mathbf{w} = \left(o(1), o(1), e^{-4} + o(1)\right) = \left(0, 0, e^{-4}\right) + o(1).$$

Let $\mathbf{v}$ be the closest point on $\mathrm{Span}\left(\left(\frac{1}{\sqrt{2}}, 0, \frac{1}{\sqrt{2}}\right)\right)$ to $\mathbf{w} = \left(0, 0, e^{-4}\right) + o(1)$. Then

$$\mathbf{v} = \left(e^{-4}/2, \; 0, \; e^{-4}/2\right) + o(1)\left(1, 0, 1\right).$$

All remaining 2-D subspaces contain the line $L = \mathrm{Span}\left(\left(\frac{1}{\sqrt{2}}, 0, \frac{1}{\sqrt{2}}\right)\right)$, so the projections are onto lines through $\mathbf{v}$ perpendicular to $L$.

The angle between these lines equals the angle between the planes. Let $\mathbf{u}$ be the closest point on $\mathrm{Span}(1,0,1)$ to $\left(0, \sin\left(1/\sqrt{K}\right), \cos\left(1/\sqrt{K}\right)\right)$; then $\mathbf{u} = \left(\frac{1}{2} + o(1)\right)(1,0,1)$. Thus the plane angle is

$$
2\tan^{-1}\left(\frac{\sin(1/(2\sqrt{K}))}{1/\sqrt{2} + o(1)}\right) = (1 + o(1))\frac{\sqrt{2}}{\sqrt{K}}.
$$

Each of the remaining $2K - 1$ projections multiplies $\mathrm{dist}(\mathbf{w}, \mathbf{v})$ by $\cos\left((1 + o(1))\sqrt{2}/\sqrt{K}\right)$, so overall it is scaled by $e^{-2} + o(1)$.

Originally,

$$
\mathrm{dist}\left((0,0,e^{-4}),\ (e^{-4}/2, 0, e^{-4}/2)\right) = \frac{e^{-4}\sqrt{2}}{2} + o(1),
$$

so afterwards the distance is $\frac{e^{-6}\sqrt{2}}{2} + o(1)$.

The direction from $\mathbf{v}$ is parallel to

$$
o(1) + \left(0, 0, e^{-4}\right) - \left(e^{-4}/2, 0, e^{-4}/2\right) = o(1) + \left(-e^{-4}/2, 0, e^{-4}/2\right) = o(1) + \left(-\frac{1}{\sqrt{2}}, 0, \frac{1}{\sqrt{2}}\right),
$$

so the final position of $\mathbf{w}_T$ is

$$
o(1) + (e^{-4}/2, 0, e^{-4}/2) + \left(\frac{e^{-6}\sqrt{2}}{2} + o(1)\right)\left(o(1) + \left(-\frac{1}{\sqrt{2}}, 0, \frac{1}{\sqrt{2}}\right)\right)
$$
$$
= o(1) + \left(e^{-4}/2 - e^{-6}/2,\ 0,\ e^{-4}/2 + e^{-6}/2\right).
$$

Hence, the $x$-coordinate contributes the following approximate loss due to the first half of the tasks:

$$
\mathcal{L}\left(\mathbf{w}_T\right) \approx \frac{1}{2}\left(e^{-4}/2 - e^{-6}/2\right)^2 \approx 3.135 \cdot 10^{-5}. \tag{8}
$$

### E.2.3   Experimental results

Constructing the tasks as explained in App. E.2.1, using QR decomposition to acquire $\mathbf{X}$ for each solution subspace, we observe that the task orderings for greedy MD (Definition 3.1) and greedy MR (Definition 3.2) coincide, as explained in App. E.2.2, for all $T \in \left\{4 \cdot 10^i - 1 \mid i = 1, 2, \ldots, 7\right\}$. As observed in Figure 18a, greedy ordering results in alternating between the first two task groups until these are depleted, then switching to the other two task groups. Loss is is diminishing in a linear rate during learning the first half of the tasks, then increases to the predicted value (Eq. (8)) during learning the second half.

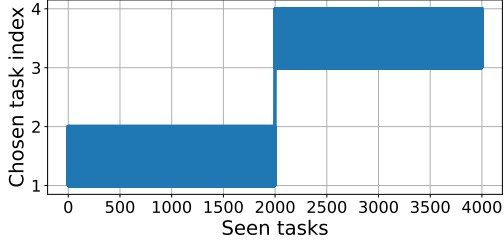

(a) **Chosen task group under greedy ordering.**

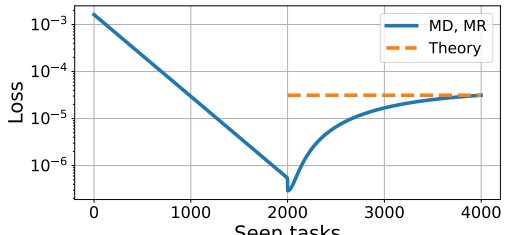

(b) **Average loss during continual learning under greedy ordering.**

Figure 18: **Example of continual learning the 3d adversarial construction with greedy ordering.** Chosen tasks coincide for MD and MR greedy orderings. $K = 1000$, $T = 3999$.

In Figure 19, we show Example 5.1 holds, and the validity of the theoretical value (Eq. (8)).

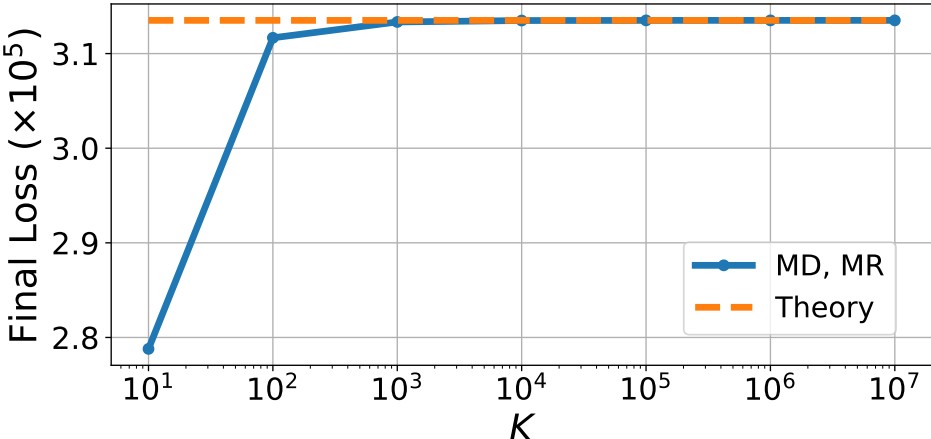

Figure 19: **Final average loss after greedy ordering for the 3d adversarial construction, for different task counts.** $T = 4K + 1$.

### E.2.4 Code for reproducibility

```python
import numpy as np
def orth_complement_rows(W, *, rtol=1e-12):
    """ Given W in R^{(d-r)×d} (full row rank), return X in R^{r×d}
    whose rows form an orthonormal basis of the orthogonal complement
    of the row-space of W. """
    W = np.asarray(W, dtype=float)
    d = W.shape[1]
    # Full QR of W.T  ->  Q is d×d and orthogonal
    # The first rank columns of Q span the rows of W;
    # the remaining columns span their orthogonal complement.
    Q, _ = np.linalg.qr(W.T, mode='complete')        # Q in R^{d×d}
    rank = np.linalg.matrix_rank(W, tol=rtol)
    # Take the last r = d - rank columns of Q, transpose to get rows
    X = Q[:, rank:].T                                 # X in R^{r×d}
    return X

ORDERING = 'MD'  # 'MR' / 'MD'
K = 1000
w1 = np.array([[0, np.sin(1/np.sqrt(K)), np.cos(1/np.sqrt(K))]])
w2 = np.array([[0, -np.sin(1/np.sqrt(K)), np.cos(1/np.sqrt(K))]])
w3 = np.array([[0, np.sin(0.5/np.sqrt(K)), np.cos(0.5/np.sqrt(K))],
    [1/np.sqrt(2), 0, 1/np.sqrt(2)]])
w4 = np.array([[0, -np.sin(0.5/np.sqrt(K)), np.cos(0.5/np.sqrt(K))],
    [1/np.sqrt(2), 0, 1/np.sqrt(2)]])
ws = [w1, w2, w3, w4]
tasks = [orth_complement_rows(w) for w in ws]
projections = [np.eye(3) - np.linalg.pinv(X) @ X for X in tasks]
total_collection_count = [K-1, K, K ,K]
T = np.sum(total_collection_count)
collection_count = total_collection_count.copy()
w = np.array([0, np.sin(1/np.sqrt(K)), np.cos(1/np.sqrt(K))])
residual_per_projection = [np.linalg.norm(X @ w)**2 for X in tasks]
total_loss = (1/T) * np.asarray(residual_per_projection) @
    np.asarray(total_collection_count)
losses = [total_loss]
for i in range(T):
    chosen_task = None
    distance = -np.inf
    new_w, new_w_candidate = None, None
    for task_index in range(4):
        if collection_count[task_index] > 0:
            if ORDERING == 'MD':
                new_w_candidate = projections[task_index] @ w
                new_distance = np.linalg.norm(new_w_candidate - w)**2
            elif ORDERING == 'MR':
                new_distance = np.linalg.norm(tasks[task_index] @ w)**2
            if new_distance > distance:
                chosen_task = task_index
                distance = new_distance
                new_w = new_w_candidate
    if new_w is None:
        new_w = projections[chosen_task] @ w
    w = new_w
    collection_count[chosen_task] -= 1
    residual_per_projection = [np.linalg.norm(X @ w)**2 for X in tasks]
    total_loss = (1/T) * np.asarray(residual_per_projection) @
        np.asarray(total_collection_count)
    losses.append(total_loss)
```

Listing 1: Code for 3d adversarial construction.

# F  Appendix to Section 5.2: Single-pass vs. repetition

## F.1  Appendix to the upper bound for greedy orderings *with* repetition (Theorem 5.3)

**Recall Theorem 5.3.** Under a Maximum Distance greedy ordering *with repetition* ($\tau_{\text{MD-R}}$) over $T$ jointly-realizable tasks, the loss of Scheme 1 after $k \geq 2$ iterations is upper bounded as,

$$\mathcal{L}\left(\mathbf{w}_k^{\tau_{\text{MD-R}}}\right) = \mathcal{O}\left(\frac{1}{\sqrt[3]{k}}\right) .$$

The main propositions leading to the proof of Theorem 5.3 are in App. F.1.2, with auxiliary claims in App. F.1.3. We use geometric analysis to derive an upper bound on how fast iterates can grow, and then bound the iterates relative to the original distance to the teacher $\mathbf{w}_\star$. Finally, we upper-bound the loss by the greedy iterate size when repetitions are allowed.

### F.1.1  Comparison to convergence rates of other task orderings

Table 1: **Loss bounds in continual linear regression over jointly realizable tasks** (based on Table 1 of Evron et al. [35]). The presented bounds are "worst case": upper bounds apply to any collection of $T$ jointly realizable tasks, while lower bounds are achieved by specific constructions. Bounds for random orderings apply to the *expected* loss. We omit scaling terms ($\|\mathbf{w}_\star\|^2 R^2$) and constant multiplicative factors (which are mild).
**Notation:** $k =$ iterations; $d =$ dimensions; $\bar{r}, r_{\max} =$ average/maximum data matrix ranks; $a, b \triangleq \min(a, b)$.

| Bound | Paper / Ordering | Single-pass Greedy | Greedy with Repetition | Random w/o Replacement | Random with Replacement |
|---|---|---|---|---|---|
| **Upper** | Evron et al. [33] | — | — | — | $\dfrac{d - \bar{r}}{k}$ |
| | Evron et al. [35] | — | — | $\dfrac{1}{\sqrt[4]{T}}, \dfrac{d - \bar{r}}{T}$ | $\dfrac{1}{\sqrt[4]{k}}, \dfrac{\sqrt{d - \bar{r}}}{k}, \dfrac{\sqrt{T\bar{r}}}{k}$ |
| | Attia et al. [10] | — | — | $\dfrac{1}{\sqrt{T}}$ | $\dfrac{1}{\sqrt{k}}$ |
| | **Ours** | None | $\dfrac{1}{\sqrt[3]{k}}$ | — | — |
| **Lower** | Evron et al. [33] (∗) | $\dfrac{1}{T}$ | $\dfrac{1}{k}$ | $\dfrac{1}{T}$ | $\dfrac{1}{k}$ |
| | **Ours** | $\Omega(1)$ | — | — | — |

(∗) Although Evron et al. [33] did not state these lower bounds explicitly, their proof of Theorem 10 provides a 2-task construction which, when replicated $\lfloor T/2 \rfloor$ times, produces a $\Theta(1/k)$ bound for both random and greedy orderings with any general $T$.

### F.1.2 Deriving the bound

**Proposition F.1.** *Let $\mathbf{w}_0$, $\mathbf{w}_1 \in \mathbb{R}^d$ and let $\mathbf{P} : \mathbb{R}^d \to \mathbb{R}^d$ be an orthogonal projection, then*

$$\|\mathbf{w}_0 - \mathbf{P}\mathbf{w}_1\|^2 \le \|\mathbf{w}_0 - \mathbf{w}_1\|^2 - \|(\mathbf{I} - \mathbf{P})\,\mathbf{w}_1\|^2 + 2 \|(\mathbf{I} - \mathbf{P})\,\mathbf{w}_1\| \, \|(\mathbf{I} - \mathbf{P})\,\mathbf{w}_0\| \,.$$

*Proof.* Let us denote

$$\mathbf{u} \triangleq (\mathbf{I} - \mathbf{P})\mathbf{w}_0 \,, \quad \mathbf{v} \triangleq (\mathbf{I} - \mathbf{P})\mathbf{w}_1 \,.$$

Then $\mathbf{u}, \mathbf{v} \in \ker(\mathbf{P})$ and are orthogonal to $\mathbf{P}\mathbf{w}_0, \mathbf{P}\mathbf{w}_1 \in \mathrm{im}(\mathbf{P})$, respectively.

Now decompose:

$$\mathbf{w}_0 - \mathbf{w}_1 = (\mathbf{P}\mathbf{w}_0 - \mathbf{P}\mathbf{w}_1) + (\mathbf{u} - \mathbf{v})\,,$$
$$\mathbf{w}_0 - \mathbf{P}\mathbf{w}_1 = (\mathbf{P}\mathbf{w}_0 - \mathbf{P}\mathbf{w}_1) + \mathbf{u}\,.$$

Using orthogonality:

$$\|\mathbf{w}_0 - \mathbf{w}_1\|^2 = \|\mathbf{P}\mathbf{w}_0 - \mathbf{P}\mathbf{w}_1\|^2 + \|\mathbf{u} - \mathbf{v}\|^2 \,, \tag{9}$$
$$\|\mathbf{w}_0 - \mathbf{P}\mathbf{w}_1\|^2 = \|\mathbf{P}\mathbf{w}_0 - \mathbf{P}\mathbf{w}_1\|^2 + \|\mathbf{u}\|^2 \,. \tag{10}$$

Subtracting Eq. (10) from Eq. (9), we get

$$\|\mathbf{w}_0 - \mathbf{w}_1\|^2 - \|\mathbf{w}_0 - \mathbf{P}\mathbf{w}_1\|^2 = \|\mathbf{u} - \mathbf{v}\|^2 - \|\mathbf{u}\|^2 = \|\mathbf{v}\|^2 - 2\mathbf{u}^\top \mathbf{v} \,.$$

So,

$$\|\mathbf{w}_0 - \mathbf{P}\mathbf{w}_1\|^2 = \|\mathbf{w}_0 - \mathbf{w}_1\|^2 - \|\mathbf{v}\|^2 + 2\mathbf{u}^\top \mathbf{v}$$
$$[\text{Cauchy–Schwarz}] \le \|\mathbf{w}_0 - \mathbf{w}_1\|^2 - \|(\mathbf{I} - \mathbf{P})\mathbf{w}_1\|^2 + 2 \|(\mathbf{I} - \mathbf{P})\mathbf{w}_1\| \, \|(\mathbf{I} - \mathbf{P})\mathbf{w}_0\| \,.$$

$\square$

**Corollary F.2.** *Consider the case in Eq. (2), i.e., $(\mathbf{w}_t)_{t=0}^k$ are iterates such that $\mathbf{w}_0 = \mathbf{0}$ and $\forall t \in [k]$, $(\mathbf{w}_t - \mathbf{w}_\star) = \mathbf{P}_{\tau(t)}(\mathbf{w}_{t-1} - \mathbf{w}_\star)$, then $\forall k \ge 2$,*

$$\|\mathbf{w}_0 - \mathbf{w}_k\|^2 \le \|\mathbf{w}_0 - \mathbf{w}_1\|^2 - \sum_{t=2}^k \|\mathbf{w}_t - \mathbf{w}_{t-1}\|^2$$

$$+ 2 \sum_{t=2}^k \|\mathbf{w}_t - \mathbf{w}_{t-1}\| \, \left\|(\mathbf{I} - \mathbf{P}_{\tau(t)})(\mathbf{w}_0 - \mathbf{w}_\star)\right\| \,.$$

*Proof.* We repeatedly apply the above proposition:

$$\|\mathbf{w}_0 - \mathbf{w}_k\|^2 = \|(\mathbf{w}_0 - \mathbf{w}_\star) - (\mathbf{w}_k - \mathbf{w}_\star)\|^2$$
$$= \left\|(\mathbf{w}_0 - \mathbf{w}_\star) - \mathbf{P}_{\tau(k)}(\mathbf{w}_{k-1} - \mathbf{w}_\star)\right\|^2$$
$$\le \underbrace{\|(\mathbf{w}_0 - \mathbf{w}_\star) - (\mathbf{w}_{k-1} - \mathbf{w}_\star)\|^2}_{\text{continue inductively}} - \left\|(\mathbf{I} - \mathbf{P}_{\tau(k)})(\mathbf{w}_{k-1} - \mathbf{w}_\star)\right\|^2$$
$$+ 2 \left\|(\mathbf{I} - \mathbf{P}_{\tau(k)})(\mathbf{w}_{k-1} - \mathbf{w}_\star)\right\| \left\|(\mathbf{I} - \mathbf{P}_{\tau(k)})(\mathbf{w}_0 - \mathbf{w}_\star)\right\|$$
$$\le \|(\mathbf{w}_0 - \mathbf{w}_\star) - (\mathbf{w}_1 - \mathbf{w}_\star)\|^2 - \sum_{t=2}^k \left\|(\mathbf{I} - \mathbf{P}_{\tau(t)})(\mathbf{w}_{t-1} - \mathbf{w}_\star)\right\|^2$$
$$+ 2 \sum_{t=2}^k \left\|(\mathbf{I} - \mathbf{P}_{\tau(t)})(\mathbf{w}_{t-1} - \mathbf{w}_\star)\right\| \left\|(\mathbf{I} - \mathbf{P}_{\tau(t)})(\mathbf{w}_0 - \mathbf{w}_\star)\right\|$$
$$= \|\mathbf{w}_0 - \mathbf{w}_1\|^2 - \sum_{t=2}^k \|\mathbf{w}_t - \mathbf{w}_{t-1}\|^2 + 2 \sum_{t=2}^k \|\mathbf{w}_t - \mathbf{w}_{t-1}\| \left\|(\mathbf{I} - \mathbf{P}_{\tau(t)})(\mathbf{w}_0 - \mathbf{w}_\star)\right\| \,.$$

$\square$

**Corollary F.3.** *If the first step is MD greedy, i.e.,* $\forall m \in [T]$, $\|(\mathbf{I} - \mathbf{P}_m)(\mathbf{w}_0 - \mathbf{w}_\star)\| \leq \|\mathbf{w}_0 - \mathbf{w}_1\|$, *then* $\forall k \geq 2$,

$$\|\mathbf{w}_0 - \mathbf{w}_k\|^2 \leq \|\mathbf{w}_0 - \mathbf{w}_1\|^2 - \sum_{t=2}^{k} \|\mathbf{w}_t - \mathbf{w}_{t-1}\|^2 + 2\|\mathbf{w}_0 - \mathbf{w}_1\| \sum_{t=2}^{k} \|\mathbf{w}_t - \mathbf{w}_{t-1}\| .$$

**Proposition F.4.** *Under greedy MD ordering, either single-pass or with repetition, we have* $\forall k \geq 1$,

$$\|\mathbf{w}_{k+1} - \mathbf{w}_k\| \leq \|\mathbf{w}_0 - \mathbf{w}_1\| \cdot 2e^{4/3} k^{1/3} .$$

*Proof.* Notice that

$$
\begin{aligned}
\|\mathbf{w}_{k+1} - \mathbf{w}_k\| &= \left\|(\mathbf{w}_k - \mathbf{w}_\star) - \mathbf{P}_{\tau(k+1)}(\mathbf{w}_k - \mathbf{w}_\star)\right\| \\
\text{[orth. proj.]} \quad &\leq \left\|(\mathbf{w}_k - \mathbf{w}_\star) - \mathbf{P}_{\tau(k+1)}(\mathbf{w}_0 - \mathbf{w}_\star)\right\| \\
\text{[triangle inequality]} \quad &\leq \left\|(\mathbf{w}_k - \mathbf{w}_\star) - (\mathbf{w}_0 - \mathbf{w}_\star)\right\| + \left\|(\mathbf{w}_0 - \mathbf{w}_\star) - \mathbf{P}_{\tau(k+1)}(\mathbf{w}_0 - \mathbf{w}_\star)\right\| \\
\text{[greediness]} \quad &\leq \|\mathbf{w}_0 - \mathbf{w}_k\| + \|\mathbf{w}_0 - \mathbf{w}_1\| \\
\implies \quad & \|\mathbf{w}_{k+1} - \mathbf{w}_k\| - \|\mathbf{w}_0 - \mathbf{w}_1\| \leq \|\mathbf{w}_0 - \mathbf{w}_k\| ,
\end{aligned}
$$

where we used the fact that the orthogonal projection of a given point on a subspace is the closest point in this subspace to the given point in the Euclidean-norm sense.

If $\|\mathbf{w}_{k+1} - \mathbf{w}_k\| < \|\mathbf{w}_0 - \mathbf{w}_1\|$, the proposition follows immediately. Otherwise $\|\mathbf{w}_{k+1} - \mathbf{w}_k\| \geq \|\mathbf{w}_0 - \mathbf{w}_1\|$, and thus

$$\left(\|\mathbf{w}_{k+1} - \mathbf{w}_k\| - \|\mathbf{w}_0 - \mathbf{w}_1\|\right)^2 \leq \|\mathbf{w}_0 - \mathbf{w}_k\|^2 .$$

Plugging this into Corollary F.3, we get

$$
\begin{aligned}
\left(\|\mathbf{w}_{k+1} - \mathbf{w}_k\| - \|\mathbf{w}_0 - \mathbf{w}_1\|\right)^2 &\leq \|\mathbf{w}_0 - \mathbf{w}_k\|^2 \\
&\leq \|\mathbf{w}_0 - \mathbf{w}_1\|^2 - \sum_{t=2}^{k} \|\mathbf{w}_t - \mathbf{w}_{t-1}\|^2 + 2\|\mathbf{w}_0 - \mathbf{w}_1\| \sum_{t=2}^{k} \|\mathbf{w}_t - \mathbf{w}_{t-1}\|
\end{aligned}
$$

$$\|\mathbf{w}_{k+1} - \mathbf{w}_k\|^2 + \sum_{t=2}^{k} \|\mathbf{w}_t - \mathbf{w}_{t-1}\|^2 \leq 2\|\mathbf{w}_0 - \mathbf{w}_1\| \left(\|\mathbf{w}_{k+1} - \mathbf{w}_k\| + \sum_{t=2}^{k} \|\mathbf{w}_t - \mathbf{w}_{t-1}\|\right)$$

$$\|\mathbf{w}_0 - \mathbf{w}_1\| \geq \frac{\sum_{t=2}^{k+1} \|\mathbf{w}_t - \mathbf{w}_{t-1}\|^2}{2\sum_{t=2}^{k+1} \|\mathbf{w}_t - \mathbf{w}_{t-1}\|} .$$

Using the same derivation, when all steps are greedy, we see $\|\mathbf{w}_2 - \mathbf{w}_1\| \geq \frac{\sum_{t=3}^{k+1} \|\mathbf{w}_t - \mathbf{w}_{t-1}\|^2}{2\sum_{t=3}^{k+1} \|\mathbf{w}_t - \mathbf{w}_{t-1}\|}$, $\ldots$, $\|\mathbf{w}_k - \mathbf{w}_{k-1}\| \geq \frac{\sum_{t=k+1}^{k+1} \|\mathbf{w}_t - \mathbf{w}_{t-1}\|^2}{2\sum_{t=k+1}^{k+1} \|\mathbf{w}_t - \mathbf{w}_{t-1}\|} = \frac{\|\mathbf{w}_{k+1} - \mathbf{w}_k\|^2}{2\|\mathbf{w}_{k+1} - \mathbf{w}_k\|} = \frac{1}{2}\|\mathbf{w}_{k+1} - \mathbf{w}_k\|$, acquiring a lower bound $\forall t \in [k]$: $\|\mathbf{w}_t - \mathbf{w}_{t-1}\| \geq \frac{\sum_{j=t+1}^{k+1} \|\mathbf{w}_j - \mathbf{w}_{j-1}\|^2}{2\sum_{j=t+1}^{k+1} \|\mathbf{w}_j - \mathbf{w}_{j-1}\|}$. Hence, we define the sequence $(C_t)_{t=1}^{k+1}$ by the backward recurrence $C_{k+1} = \|\mathbf{w}_{k+1} - \mathbf{w}_k\|$ and $\forall t \in [k]$, $C_t = \frac{\sum_{j=t+1}^{k+1} C_j^2}{2\sum_{j=t+1}^{k+1} C_j}$. By applying Claim F.7, using the sequences $a_i \triangleq \|\mathbf{w}_{k+1-i} - \mathbf{w}_{k-i}\|$, $b_i \triangleq C_{k+1-i}$ for all $i \in \{0, \ldots, k\}$, we observe that $\|\mathbf{w}_0 - \mathbf{w}_1\| \geq C_1$.

We investigate the sequence $(C_t)$ in Corollary F.10, where we prove that if $C_{k+1} = \|\mathbf{w}_{k+1} - \mathbf{w}_k\| > 0$, this sequence maintains $\forall k \geq 1$,

$$C_1 \geq C_{k+1} \cdot \frac{1}{2e^{4/3}} k^{-1/3} ,$$

(and indeed if $\|\mathbf{w}_{k+1} - \mathbf{w}_k\| = 0$, the proposition follows immediately), and thus we get that

$$\|\mathbf{w}_0 - \mathbf{w}_1\| \geq C_1 \geq \|\mathbf{w}_{k+1} - \mathbf{w}_k\| \frac{1}{2e^{4/3}} k^{-1/3}$$

$$\implies \|\mathbf{w}_{k+1} - \mathbf{w}_k\| \leq \|\mathbf{w}_0 - \mathbf{w}_1\| \cdot 2e^{4/3} k^{1/3} .$$

$\square$

**Lemma F.5.** *Under greedy MD ordering, either single pass or with repetition, we have $\forall k \geq 1$,*

$$\|\mathbf{w}_{k-1} - \mathbf{w}_k\|^2 \leq \frac{4e^{8/3}}{3} \frac{\|\mathbf{w}_0 - \mathbf{w}_\star\|^2}{k^{1/3}}.$$

*Proof.* Applying the above proposition for a starting index $t - 1$ (instead of 0), we see that $\forall k > t$,

$$\|\mathbf{w}_{k-1} - \mathbf{w}_k\| \leq \|\mathbf{w}_{t-1} - \mathbf{w}_t\| \cdot 2e^{4/3} (k-t)^{1/3}$$

$$\implies \frac{\|\mathbf{w}_{k-1} - \mathbf{w}_k\|^2}{4e^{8/3} (k-t)^{2/3}} \leq \|\mathbf{w}_{t-1} - \mathbf{w}_t\|^2.$$

From the Pythagorean theorem we have

$$\|\mathbf{w}_k - \mathbf{w}_\star\|^2 = \|\mathbf{w}_{k-1} - \mathbf{w}_\star\|^2 - \|\mathbf{w}_{k-1} - \mathbf{w}_k\|^2 = \|\mathbf{w}_0 - \mathbf{w}_\star\|^2 - \sum_{t=1}^{k} \|\mathbf{w}_{t-1} - \mathbf{w}_t\|^2.$$

Plugging in the above proposition, we get

$$0 \leq \|\mathbf{w}_k - \mathbf{w}_\star\|^2 = \|\mathbf{w}_0 - \mathbf{w}_\star\|^2 - \sum_{t=1}^{k} \|\mathbf{w}_{t-1} - \mathbf{w}_t\|^2$$

$$\leq \|\mathbf{w}_0 - \mathbf{w}_\star\|^2 - \|\mathbf{w}_{k-1} - \mathbf{w}_k\|^2 - \sum_{t=1}^{k-1} \frac{\|\mathbf{w}_{k-1} - \mathbf{w}_k\|^2}{4e^{8/3} (k-t)^{2/3}}$$

$$= \|\mathbf{w}_0 - \mathbf{w}_\star\|^2 - \|\mathbf{w}_{k-1} - \mathbf{w}_k\|^2 \left( 1 + \frac{1}{4e^{8/3}} \sum_{i=1}^{k-1} \frac{1}{i^{2/3}} \right)$$

$$\left[ \sum_{i=1}^{k-1} \frac{1}{i^{2/3}} \geq 3 \left( k^{1/3} - 1 \right) \right] \leq \|\mathbf{w}_0 - \mathbf{w}_\star\|^2 - \|\mathbf{w}_{k-1} - \mathbf{w}_k\|^2 \left( 1 + \frac{3 \left( k^{1/3} - 1 \right)}{4e^{8/3}} \right)$$

$$\implies \|\mathbf{w}_{k-1} - \mathbf{w}_k\|^2 \leq \frac{\|\mathbf{w}_0 - \mathbf{w}_\star\|^2}{\left( 1 + \frac{3 \left( k^{1/3} - 1 \right)}{4e^{8/3}} \right)} = \frac{4e^{8/3}}{3} \frac{\|\mathbf{w}_0 - \mathbf{w}_\star\|^2}{k^{1/3} + \frac{4e^{8/3}}{3} - 1}$$

$$\leq \frac{4e^{8/3}}{3} \frac{\|\mathbf{w}_0 - \mathbf{w}_\star\|^2}{k^{1/3}},$$

where we used $\sum_{i=1}^{k-1} \frac{1}{i^{2/3}} \geq \int_1^k \frac{1}{x^{2/3}} dx = 3 \left( k^{1/3} - 1 \right)$. $\qquad\square$

**We are now ready to prove Theorem 5.3:**

*Proof of Theorem 5.3.* Under greedy MD ordering with repetitions we have:

$$\mathcal{L}(\mathbf{w}_k) = \frac{1}{\|\mathbf{w}_\star\|^2 R^2 T} \sum_{m=1}^{T} \|\mathbf{X}_m \mathbf{w}_k - \mathbf{y}_m\|^2 = \frac{1}{\|\mathbf{w}_\star\|^2 R^2 T} \sum_{m=1}^{T} \|\mathbf{X}_m (\mathbf{w}_k - \mathbf{w}_\star)\|^2$$

$$\leq \frac{1}{\|\mathbf{w}_\star\|^2 R^2 T} \sum_{m=1}^{T} \|\mathbf{X}_m\|^2 \|(\mathbf{I} - \mathbf{P}_m)(\mathbf{w}_k - \mathbf{w}_\star)\|^2$$

$$\leq \frac{1}{\|\mathbf{w}_\star\|^2 T} \sum_{m=1}^{T} \|(\mathbf{I} - \mathbf{P}_m)(\mathbf{w}_k - \mathbf{w}_\star)\|^2$$

$$[\text{greedy+repetitions}] \leq \frac{1}{\|\mathbf{w}_\star\|^2} \|\mathbf{w}_k - \mathbf{w}_{k+1}\|^2$$

$$[\text{above}, \mathbf{w}_0 = \mathbf{0}] \leq \frac{1}{\|\mathbf{w}_\star\|^2} \frac{4e^{8/3}}{3} \frac{\|\mathbf{w}_\star\|^2}{(k+1)^{1/3}} = \frac{4e^{8/3}}{3} \frac{1}{(k+1)^{1/3}}$$

$$= \mathcal{O}\left( k^{-1/3} \right).$$

$$\qquad\square$$

### F.1.3 Auxiliary claims

**Claim F.6.** For $s, r > 0$ and $\alpha \geq \frac{1}{2}$ define

$$f_{s,r}(\alpha) \triangleq r \frac{s + \alpha^2 r}{s + \alpha r}.$$

(a) For all $s, r > 0$, we have $\dfrac{\partial f_{s,r}(\alpha)}{\partial \alpha} \geq 0$ on $\alpha \in \left[\frac{1}{2}, \infty\right)$; hence $f$ is non-decreasing in $\alpha$.

(b) For all $\alpha \geq \frac{1}{2}$ and $s > 0$, we have $\dfrac{\partial f_{s,r}(\alpha)}{\partial r} > 0$; hence $f_{s,r}(\alpha)$ is strictly increasing in $r$.

(c) For all $\alpha \in \left[\frac{1}{2}, 1\right]$ and $r > 0$, we have $\dfrac{\partial f_{s,r}(\alpha)}{\partial s} \geq 0$, hence $f_{s,r}(\alpha)$ is non-decreasing in $s$ on that $\alpha$-range.

*Proof.* Below, we prove each statement separately, in order.

(a) Derivative with respect to $\alpha$:

$$\frac{\partial f_{s,r}}{\partial \alpha} = r \frac{(2\alpha r)(s + \alpha r) - (s + \alpha^2 r)(r)}{(s + \alpha r)^2}.$$

Expanding the numerator:

$$(2\alpha r)(s + \alpha r) = 2\alpha r s + 2\alpha^2 r^2,$$
$$(s + \alpha^2 r)r = rs + \alpha^2 r^2,$$
$$\text{difference} = (2\alpha - 1)rs + \alpha^2 r^2.$$

Thus

$$\frac{\partial f_{s,r}}{\partial \alpha} = \frac{r^2\left[(2\alpha - 1)s + \alpha^2 r\right]}{(s + \alpha r)^2} \geq 0.$$

(b) Derivative with respect to $r$:

$$\frac{\partial f_{s,r}}{\partial r} = \frac{(s + 2\alpha^2 r)(s + \alpha r) - \alpha(rs + \alpha^2 r^2)}{(s + \alpha r)^2}.$$

Expand the numerator term-by-term:

$$(s + 2\alpha^2 r)(s + \alpha r) = s^2 + s\alpha r + 2\alpha^2 rs + 2\alpha^3 r^2,$$

$$\alpha(rs + \alpha^2 r^2) = \alpha rs + \alpha^3 r^2,$$

$$\text{difference} = s^2 + 2\alpha^2 rs + \alpha^3 r^2 > 0.$$

Hence $\partial f_{s,r} / \partial r > 0$.

(c) Derivative with respect to $s$:

$$\frac{\partial f_{s,r}}{\partial s} = r \frac{(s + \alpha r) - (s + \alpha^2 r)}{(s + \alpha r)^2} = \frac{r^2 \alpha(1 - \alpha)}{(s + \alpha r)^2}.$$

For $\alpha \in \left[\frac{1}{2}, 1\right]$ the factor $\alpha(1 - \alpha) \geq 0$, yielding a non-negative derivative.

$\square$

**Claim F.7.** Let $n \geq 1$ and let $(a_i)_{i=0}^n$ be a sequence of strictly positive numbers such that

$$\forall i \geq 1, \qquad a_i \geq \frac{\sum_{j=0}^{i-1} a_j^2}{2 \sum_{j=0}^{i-1} a_j}.$$

Define a second sequence $(b_i)_{i=0}^n$ recursively by

$$b_0 = a_0 > 0, \qquad b_i = \frac{\sum_{j=0}^{i-1} b_j^2}{2 \sum_{j=0}^{i-1} b_j} \quad (i \geq 1).$$

Then $a_n \geq b_n$.

*Proof.* For either sequence $x \in \{a, b\}$ and each $k \geq 1$ set

$$S_k^{(x)} \triangleq \sum_{j=0}^k x_j, \qquad Q_k^{(x)} \triangleq \sum_{j=0}^k x_j^2, \qquad R_k^{(x)} \triangleq \frac{Q_k^{(x)}}{S_k^{(x)}} > 0.$$

For $k \geq 1$ we can express the defining relations as

$$a_k \geq \frac{R_{k-1}^{(a)}}{2}, \qquad b_k = \frac{R_{k-1}^{(b)}}{2},$$

and, using $Q_{k-1}^{(x)} = R_{k-1}^{(x)} S_{k-1}^{(x)}$,

$$R_k^{(x)} = \frac{Q_k^{(x)}}{S_k^{(x)}} = \frac{x_k^2 + Q_{k-1}^{(x)}}{x_k + S_{k-1}^{(x)}} = \frac{x_k^2 + R_{k-1}^{(x)} S_{k-1}^{(x)}}{x_k + S_{k-1}^{(x)}} = R_{k-1}^{(x)} \frac{S_{k-1}^{(x)} + \left(\frac{x_k}{R_{k-1}^{(x)}}\right)^2 R_{k-1}^{(x)}}{S_{k-1}^{(x)} + \left(\frac{x_k}{R_{k-1}^{(x)}}\right) R_{k-1}^{(x)}}$$

$$= f_{S_{k-1}^{(x)}, R_{k-1}^{(x)}}\left(\frac{x_k}{R_{k-1}^{(x)}}\right), \tag{$*$}$$

where $f_{s,r}(\alpha)$ is defined as in Claim F.6.

We now prove by induction that $\forall k \in \{0, \ldots, n\}$,

$$a_k \geq b_k, \qquad S_k^{(a)} \geq S_k^{(b)}, \qquad R_k^{(a)} \geq R_k^{(b)}.$$

**Base case $k = 0$.** All equalities $a_0 = b_0$, $R_0^{(a)} = R_0^{(b)} = a_0$, $S_0^{(a)} = S_0^{(b)} = a_0$ hold.

**Induction step $k \to k+1$.** Assume the inequalities hold for $k$.

(i)   *Comparing $a_{k+1}$ and $b_{k+1}$.* By the recursion and the inductive hypothesis,

$$a_{k+1} \geq \frac{R_k^{(a)}}{2} \geq \frac{R_k^{(b)}}{2} = b_{k+1}.$$

(ii)  *Comparing $S_{k+1}^{(a)}$ and $S_{k+1}^{(b)}$.* From the above and the inductive hypothesis,

$$S_{k+1}^{(a)} = S_k^{(a)} + a_{k+1} \geq S_k^{(b)} + b_{k+1} = S_{k+1}^{(b)}.$$

(iii) *Comparing $R_{k+1}^{(a)}$ and $R_{k+1}^{(b)}$.* From $(*)$ we have

$$R_{k+1}^{(b)} = f_{S_k^{(b)}, R_k^{(b)}}\left(\frac{1}{2}\right), \qquad R_{k+1}^{(a)} = f_{S_k^{(a)}, R_k^{(a)}}\left(\frac{a_{k+1}}{R_k^{(a)}}\right),$$

where $\frac{a_{k+1}}{R_k^{(a)}} \geq \frac{1}{2}$. By the inductive hypothesis $S_k^{(a)} \geq S_k^{(b)}$ and $R_k^{(a)} \geq R_k^{(b)}$, and by Claim F.6 the function $f_{s,r}(\alpha)$ is *(a)* increasing in $\alpha$, *(b)* increasing in $r$ for all $\alpha \geq \frac{1}{2}$, and *(c)* increasing in $s$ when $\alpha = \frac{1}{2}$. Therefore

$$R_{k+1}^{(a)} = f_{S_k^{(a)}, R_k^{(a)}}\left(\frac{a_{k+1}}{R_k^{(a)}}\right) \geq f_{S_k^{(a)}, R_k^{(a)}}\left(\frac{1}{2}\right) \geq f_{S_k^{(b)}, R_k^{(b)}}\left(\frac{1}{2}\right) = R_{k+1}^{(b)}.$$

Hence the inequalities hold for $k+1$, concluding the proof by induction. Taking $k = n$ yields $a_n \geq b_n$, proving the claim. $\qquad\square$

**Claim F.8.** $\forall x > -1$, $\log(1+x) \geq \frac{x}{1+x}$ .

*Proof.* For $x > -1$, define $f(x) = \ln(1+x) - \frac{x}{1+x}$, then

$$f'(x) = \frac{1}{1+x} - \frac{1+x-x}{(1+x)^2} = \frac{x}{(1+x)^2} .$$

Since $f'(x) < 0$ for $x < 0$ and $f'(x) > 0$ for $x > 0$, the point $x = 0$, where $f(0) = 0$, is the global minimum of $f$. Hence $\ln(1+x) \geq \frac{x}{1+x}$ for all $x > -1$. $\qquad\square$

**Claim F.9.** Let $\lambda > \mu > 0$, $\kappa > 0$. Define the sequences $(A_n)_{n=0}^\infty$, $(B_n)_{n=0}^\infty$, $\left(\tilde{C}_n\right)_{n=0}^\infty$ by $A_0 > 0$, $B_0 > 0$, $\tilde{C}_0 = \kappa \frac{B_0}{A_0}$, and for $n \geq 1$,

$$A_n \triangleq A_{n-1} + \lambda \frac{B_{n-1}}{A_{n-1}}, \qquad B_n \triangleq B_{n-1} + \mu \left(\frac{B_{n-1}}{A_{n-1}}\right)^2, \qquad \tilde{C}_n \triangleq \kappa \frac{B_n}{A_n},$$

then $\forall n \geq 1$, $\tilde{C}_n \geq \tilde{C}_0 \cdot \gamma n^{-\frac{\lambda-\mu}{2\lambda-\mu}}$ , where $\gamma \triangleq \exp\left(-\frac{2\lambda(\lambda-\mu)}{\mu(2\lambda-\mu)}\right)$.

*Proof.* It is readily seen that $\forall n \geq 0$, $A_n > 0$ and $B_n > 0$, immediately by induction. Define the helper sequence $\forall n \geq 0$, $f_n = \frac{A_n^2}{B_n} > 0$, then $A_n = \sqrt{B_n f_n}$ and $\forall n \geq 1$,

$$f_n = \frac{\left(A_{n-1} + \lambda \frac{B_{n-1}}{A_{n-1}}\right)^2}{B_{n-1} + \mu \left(\frac{B_{n-1}}{A_{n-1}}\right)^2} = \frac{\left(\sqrt{B_{n-1}f_{n-1}} + \lambda \frac{B_{n-1}}{\sqrt{B_{n-1}f_{n-1}}}\right)^2}{B_{n-1} + \mu \left(\frac{B_{n-1}}{\sqrt{B_{n-1}f_{n-1}}}\right)^2}$$

$$= \frac{B_{n-1}\left(\sqrt{f_{n-1}} + \frac{\lambda}{\sqrt{f_{n-1}}}\right)^2}{B_{n-1}\left(1 + \frac{\mu}{f_{n-1}}\right)}$$

$$[B_{n-1} > 0] = \frac{\frac{1}{f_{n-1}}(f_{n-1} + \lambda)^2}{1 + \frac{\mu}{f_{n-1}}} = \frac{f_{n-1}^2 + 2\lambda f_{n-1} + \lambda^2}{f_{n-1} + \mu}$$

$$= f_{n-1} + 2\lambda - \mu + \frac{(\lambda - \mu)^2}{f_{n-1} + \mu}$$

$$\geq f_{n-1} + 2\lambda - \mu ,$$

then $f_n \geq f_{n-1} + 2\lambda - \mu \geq f_0 + n(2\lambda - \mu) = \frac{A_0^2}{B_0} + n(2\lambda - \mu)$ .

Now, observe $\forall n \geq 1$,

$$\tilde{C}_n = \kappa \frac{B_n}{A_n} = \kappa \frac{B_{n-1} + \mu \left(\frac{B_{n-1}}{A_{n-1}}\right)^2}{A_{n-1} + \lambda \frac{B_{n-1}}{A_{n-1}}} = \kappa \frac{B_{n-1}}{A_{n-1}} \frac{1 + \mu \frac{B_{n-1}}{A_{n-1}^2}}{1 + \lambda \frac{B_{n-1}}{A_{n-1}^2}} = \tilde{C}_{n-1} \frac{1 + \frac{\mu}{f_{n-1}}}{1 + \frac{\lambda}{f_{n-1}}}$$

$$= \tilde{C}_0 \prod_{i=0}^{n-1} \frac{f_i + \mu + \lambda - \lambda}{f_i + \lambda} = \tilde{C}_0 \prod_{i=0}^{n-1}\left(1 - \frac{\lambda - \mu}{f_i + \lambda}\right) ,$$

and note that $\forall i \geq 0$, $0 < \frac{\lambda - \mu}{f_i + \lambda} < 1$. Taking the log we get,

$$\log \tilde{C}_n = \log \tilde{C}_0 + \sum_{i=0}^{n-1} \log\left(1 - \frac{\lambda - \mu}{f_i + \lambda}\right)$$

$$[\text{Claim F.8}] \geq \log \tilde{C}_0 + \sum_{i=0}^{n-1} \frac{-\frac{\lambda - \mu}{f_i + \lambda}}{1 - \frac{\lambda - \mu}{f_i + \lambda}} = \log \tilde{C}_0 - \sum_{i=0}^{n-1} \frac{\lambda - \mu}{f_i + \mu}$$

$$[\lambda > \mu] \geq \log \tilde{C}_0 - \sum_{i=0}^{n-1} \frac{\lambda - \mu}{f_0 + i\,(2\lambda - \mu) + \mu} \geq \log \tilde{C}_0 - \sum_{i=0}^{n-1} \frac{\lambda - \mu}{i\,(2\lambda - \mu) + \mu}$$

$$[\text{assuming } n \geq 2] = \log \tilde{C}_0 - \frac{\lambda - \mu}{\mu} - \sum_{i=1}^{n-1} \frac{\lambda - \mu}{i\,(2\lambda - \mu) + \mu} \geq \log \tilde{C}_0 - \frac{\lambda - \mu}{\mu} - \sum_{i=1}^{n-1} \frac{\lambda - \mu}{i\,(2\lambda - \mu)}$$

$$\left[\sum_{i=1}^{m} \frac{1}{i} \leq 1 + \log m\right] \geq \log \tilde{C}_0 - \frac{\lambda - \mu}{\mu} - \frac{\lambda - \mu}{2\lambda - \mu}\,(1 + \log(n-1))$$

$$\geq \log \tilde{C}_0 - \frac{\lambda - \mu}{\mu} - \frac{\lambda - \mu}{2\lambda - \mu} - \frac{\lambda - \mu}{2\lambda - \mu} \log n$$

$$\implies \tilde{C}_n \geq \tilde{C}_0 \cdot n^{-\frac{\lambda - \mu}{2\lambda - \mu}} \exp\left(-\frac{2\lambda\,(\lambda - \mu)}{\mu\,(2\lambda - \mu)}\right).$$

When $n = 1$, we have $\log \tilde{C}_1 \geq \log \tilde{C}_0 - \frac{\lambda - \mu}{\mu} > \log \tilde{C}_0 - \frac{\lambda - \mu}{\mu} - \frac{\lambda - \mu}{2\lambda - \mu}$, hence $\tilde{C}_1 \geq \tilde{C}_0 \exp\left(-\frac{2\lambda(\lambda - \mu)}{\mu(2\lambda - \mu)}\right)$, concluding the proof for all $n \geq 1$. $\qquad\square$

**Corollary F.10.** *For* $(C_t)_{t=1}^{k+1}$ *defined by the backwards recurrence* $C_{k+1} > 0$ *and* $\forall t \in [k]$, $C_t = \frac{\sum_{j=t+1}^{k+1} C_j^2}{2 \sum_{j=t+1}^{k+1} C_j}$, *we have that* $\forall k \geq 1$,

$$C_1 \geq C_{k+1} \cdot \frac{1}{2e^{4/3}} k^{-1/3}.$$

*Proof.* By defining $\forall n \in \{0, \dots, k\}$ : $A_n = \sum_{j=k+1-n}^{k+1} C_j$, $B_n = \sum_{j=k+1-n}^{k+1} C_j^2$, we note that $A_0 = C_{k+1} > 0$, $B_0 = C_{k+1}^2 > 0$, and for all $1 \leq n \leq k$,

$$A_n = A_{n-1} + C_{k+1-n} = A_{n-1} + \frac{\sum_{j=k+1-n+1}^{k+1} C_j^2}{2 \sum_{j=k+1-n+1}^{k+1} C_j} = A_{n-1} + \frac{1}{2}\frac{B_{n-1}}{A_{n-1}},$$

$$B_n = B_{n-1} + C_{k+1-n}^2 = B_{n-1} + \left(\frac{1}{2}\frac{B_{n-1}}{A_{n-1}}\right)^2 = B_{n-1} + \frac{1}{4}\left(\frac{B_{n-1}}{A_{n-1}}\right)^2.$$

Defining the sequence $\tilde{C}_n \triangleq \kappa \frac{B_n}{A_n}$, with $\kappa = \frac{1}{2}$, we have from the above claim $\forall n \geq 1$,

$$\tilde{C}_n \geq \tilde{C}_0 \cdot \gamma n^{-\frac{\lambda - \mu}{2\lambda - \mu}},$$

where $\lambda = \frac{1}{2}$, $\mu = \frac{1}{4}$, $\gamma = \exp\left(-\frac{2\lambda(\lambda - \mu)}{\mu(2\lambda - \mu)}\right) = e^{-4/3}$, and $\tilde{C}_0 = \frac{1}{2}\frac{B_0}{A_0} = \frac{1}{2}C_{k+1}$. Thus, we have,

$$\tilde{C}_n \geq C_{k+1} \cdot \frac{e^{-4/3}}{2} n^{-1/3}, \quad \forall n \geq 1.$$

Finally, observe that $\tilde{C}_n \triangleq \frac{1}{2}\frac{B_n}{A_n} = \frac{\sum_{j=k+1-n}^{k+1} C_j^2}{2 \sum_{j=k+1-n}^{k+1} C_j} = C_{k-n}$, for all $0 \leq n \leq k-1$, and plugging in $n = k - 1 \geq 1$, when $k \geq 2$, we get

$$C_1 \geq C_{k+1} \cdot \frac{e^{-4/3}}{2} (k-1)^{-1/3} \geq C_{k+1} \cdot \frac{e^{-4/3}}{2} k^{-1/3}.$$

Specifically, when $k = 1$ we get $C_1 = \frac{C_2}{2} = C_{k+1} \cdot \frac{1}{2} \geq C_{k+1} \cdot \frac{e^{-4/3}}{2} k^{-1/3}$, concluding the proof for all $k \geq 1$. $\qquad\square$

## F.2 Regression experiments on single-pass vs. repetition

Here, we extend the experiment on the effect of repetition (Figure 5) to additional data regimes. Figure 5 was produced using the same data as Figure 3a, *i.e.*, $d = 100$, $r = 10$, $T = 50$. Throughout this section, the Maximum Distance ordering (Definition 3.1) is denoted by "Greedy" for brevity.

**Isotropic data.** We find that the conclusions of Section 5.2 extend to more regimes: repetitions are beneficial in greedy ordering while replacement harms random ordering.

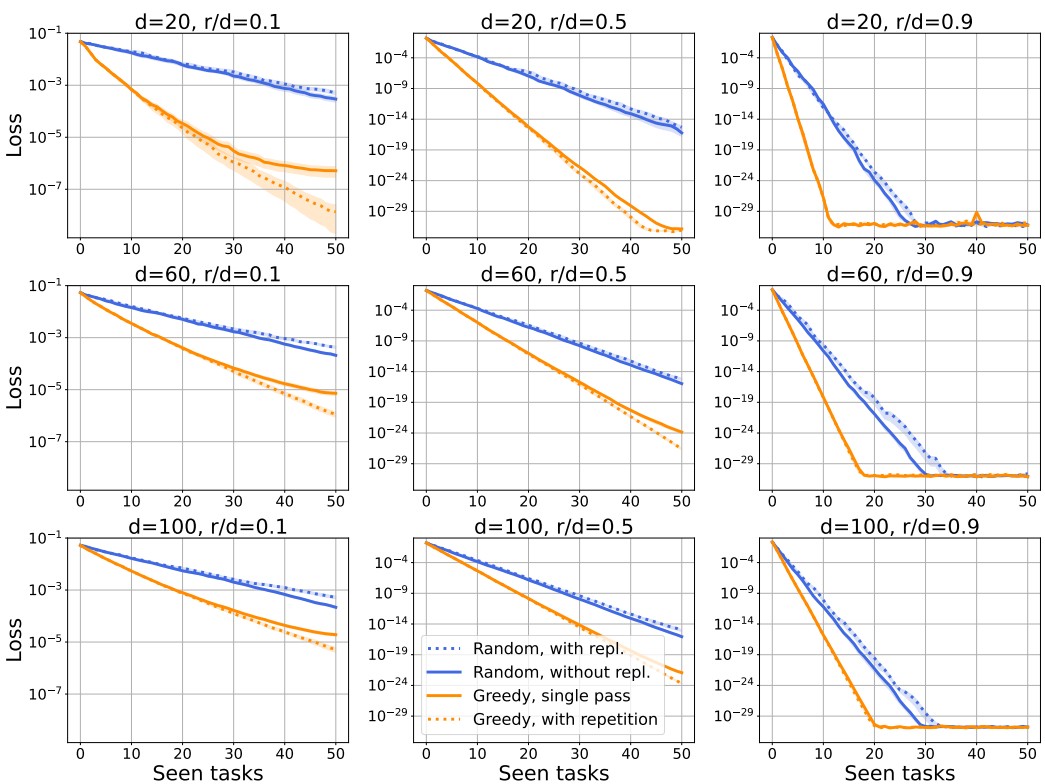

Figure 20: **The effect of repetitions for varying dimensions $d$ and ranks $r$ of the data matrices, for isotropic data.** $T = 50$. Random orderings without-replacement consistently outperform their with-replacement counterparts. In contrast, greedy orderings with repetition outperform the single-pass variant. As explained in Section 5.2, repetition in greedy orderings is beneficial because it enables larger steps (and converging faster to the joint solution $\mathbf{w}_\star$).

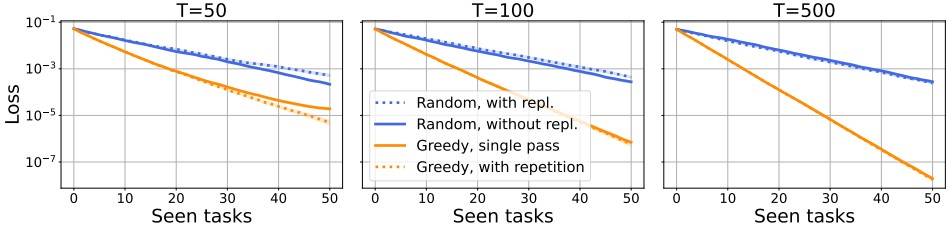

Figure 21: **The effect of repetitions for varying task count $T$, for isotropic data.** $d = 100$, $r = 10$. As task count increases, the differences between with and without repetition diminish. Notice, however, that in all subplots we only learn the first 50 tasks. It is readily observed in the left subplot that the effect of repetition becomes pronounced in the latter parts of the task sequences. As can be expected, repetition offers less benefit when many diverse, unexplored tasks remain.

**Anisotropic data.** Next, we observe that the effect of repetitions diminishes for correlated data.

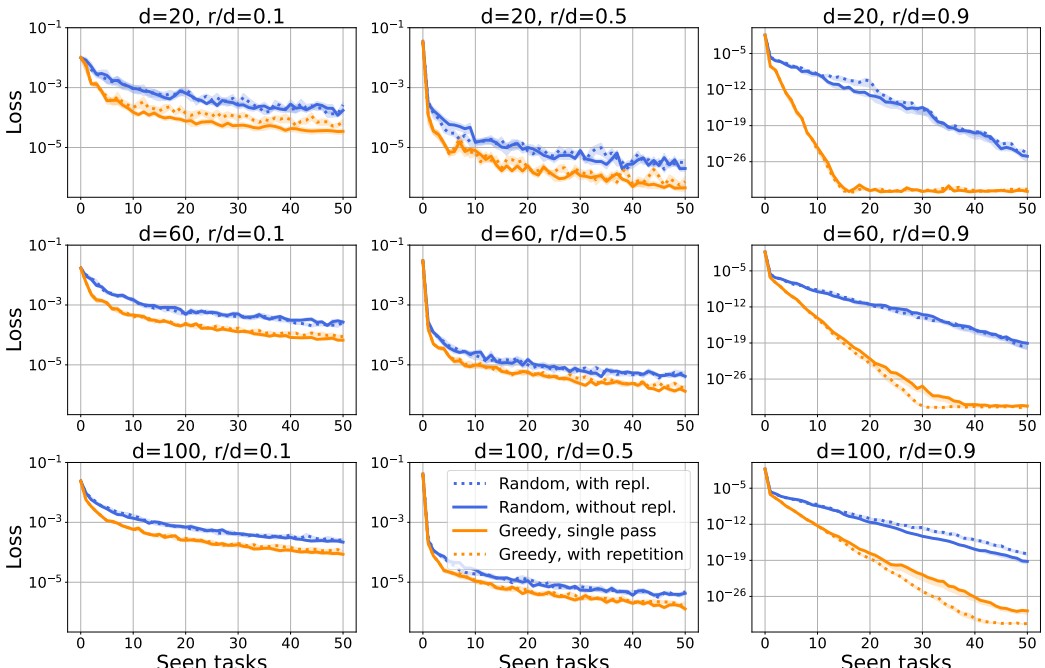

Figure 22: **The effect of repetitions for varying dimensions** $d$ **and ranks** $r$ **of the data matrices, for anisotropic data.** $T = 50$. Previously in App. B (Figure 10), we explained that the performance of all ordering strategies deteriorates when the pairwise distances between task solution subspaces are small. This effect is even more pronounced for low-rank tasks (left columns), where the complementary high-rank solution subspaces overlap substantially. We also observe that in those low-rank regimes, different orderings exhibit more similar performance, and consequently, repetitions become less impactful. While we cannot fully explain the small performance degradation observed when allowing repetitions in greedy ordering for low-dimension, low-rank settings (top-left subfigure), it may be related to the slower convergence to the joint solution $\mathbf{w}_\star$, nullifying the effect of the loss upper bound induced by the distance to $\mathbf{w}_\star$ (Proposition 2.6; see also Figure 23 below).

Below, we observe another aspect related to the "diminished" effect of repetitions in this setting.

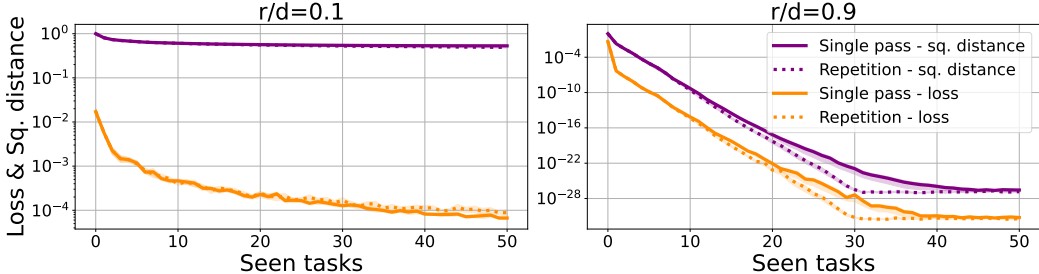

Figure 23: **The distance to** $\mathbf{w}_\star$ **is a loose upper bound on the loss for high similarity tasks.** Greedy ordering with $d = 60$, $T = 50$. While repetitions do lead to a slightly faster decrease in the squared distance to $\mathbf{w}_\star$, this decrease remains slow when tasks are highly similar (as in the low-rank setting on the left). Consequently, the upper bound of Proposition 2.6 becomes looser, as the loss itself decreases more rapidly. This discrepancy makes it difficult to draw firm conclusions about the convergence of the loss, including the exact impact of repetitions. A similar gap between the loss and the distance to the joint solution $\mathbf{w}_\star$ in highly similar tasks was also noted by Evron et al. [33, Section 5.1 therein].

**Remark.** We omit the figure for the corresponding experiment with varying number of tasks $T$, as it offers no additional insights beyond those shown in Figure 21.

# G Appendix to Section 5.3: Hybrid task ordering

## G.1 Hybrid ordering scheme

Motivated by the success of greedy Kaczmarz and importance sampling methods [75, 4], as well as recent convergence bounds for random orderings in continual learning [35, 10], we introduce a "hybrid" strategy in Section 5.3. Hybrid schemes have also been explored in the contexts of Kaczmarz methods [75, 25], coordinate descent [36], and multiplicative Schwarz methods [40].

In this approach, tasks are selected greedily as long as the decrements $\|\mathbf{w}_{t-1} - \mathbf{w}_t\|^2$ (see Eq. (3)) remain above a threshold; afterward, selection switches to random sampling. The proposed hybrid method can be used with either the greedy Maximum Distance rule (Definition 3.1), as in Scheme 3, or the greedy Maximum Residual rule (Definition 3.2) as in Scheme 4.

---

**Scheme 3** MD hybrid ordering ($\tau_{\text{H-MD}}$)

---

**Input:** $\beta_{\text{MD}} \in \left[0, \|\mathbf{w}_0 - \mathbf{w}_\star\|^2\right]$

**For each** iteration $t = 1, \dots, T$:         # Use greedy selection as long as the threshold is met

   $m' \leftarrow \operatorname{argmax}_{m \in [T] \setminus \tau_{\text{H-MD}}(1:t-1)} \|(\mathbf{I} - \mathbf{P}_m)(\mathbf{w}_{t-1} - \mathbf{w}_\star)\|^2$       # Compute greedy selection

   **If**   $\|(\mathbf{I} - \mathbf{P}_{m'})(\mathbf{w}_{t-1} - \mathbf{w}_\star)\|^2 \geq \beta_{\text{MD}}$   **Then**   $\tau_{\text{H-MD}}(t) \leftarrow m'$   **Else**   **Break**

$\tau_{\text{H-MD}}(t : T) \sim \operatorname{Unif}\left([T] \setminus \tau_{\text{H-MD}}(1 : t-1)\right)$   # Choose remaining tasks randomly w/o replacement

---

**Scheme 4** MR hybrid ordering ($\tau_{\text{H-MR}}$)

---

**Input:** $\beta_{\text{MR}} \in \left[0, R^2 \|\mathbf{w}_0 - \mathbf{w}_\star\|^2\right]$         # Reminder: $R \triangleq \max_{m \in [T]} \|\mathbf{X}_m\|$

**For each** iteration $t = 1, \dots, T$:         # Use greedy selection as long as the threshold is met

   $m' \leftarrow \operatorname{argmax}_{m \in [T] \setminus \tau_{\text{H-MR}}(1:t-1)} \|\mathbf{X}_m \mathbf{w}_{t-1} - \mathbf{y}_m\|^2$       # Compute greedy selection

   **If**   $\|\mathbf{X}_{m'} \mathbf{w}_{t-1} - \mathbf{y}_{m'}\|^2 \geq \beta_{\text{MR}}$   **Then**   $\tau_{\text{H-MR}}(t) \leftarrow m'$   **Else**   **Break**

$\tau_{\text{H-MR}}(t : T) \sim \operatorname{Unif}\left([T] \setminus \tau_{\text{H-MR}}(1 : t-1)\right)$   # Choose remaining tasks randomly w/o replacement

---

While our analysis sets the threshold $\beta$ using $\|\mathbf{w}_0 - \mathbf{w}_\star\|$ and $R$, the hybrid methods remain useful, *e.g.,* with a heuristic $\beta$.

Analytically, using a suitable threshold $\beta$, any upper bound for without-replacement random orderings, *e.g.,* an $\mathcal{O}\left(1/\sqrt{k}\right)$ bound [10], can extend to our hybrid schemes, showing again that they avoid the failure mode of Section 5.1, as shown in the following Lemma G.1. Moreover, we show that the upper bound that we derive for hybrid orderings continues to improve as long as the stopping criterion is not triggered. Put more simply, it is beneficial to follow the greedy ordering as long as the resulting iterates are "large enough" (however, the actual stopping time will depend on the data).

## G.2 Hybrid ordering upper bound

**Lemma G.1** (Hybrid ordering bound). *Consider any known upper bound for the expected normalized loss (Definition 2.3) in random ordering without replacement over $T$ jointly-realizable tasks, of the form $\mathbb{E}_{\tau_{\mathrm{Unif}}}\left[\mathcal{L}(\mathbf{w}_T^{\tau_{\mathrm{Unif}}})\right] \leq \frac{C}{T^\alpha}$ with $C > 0$ and $0 < \alpha \leq 1$, such that $\frac{C}{T^\alpha} \leq \frac{1}{2-\alpha}$. Then, defining $\tilde{\beta}_{\min} \triangleq \frac{T^\alpha - C(1-\alpha)}{CT}$, the following holds:*
*When $\beta_{MD} \geq \|\mathbf{w}_0 - \mathbf{w}_\star\|^2 \tilde{\beta}_{\min}$ (or $\beta_{MR} \geq R^2 \|\mathbf{w}_0 - \mathbf{w}_\star\|^2 \tilde{\beta}_{\min}$), the loss under Scheme 3 (or Scheme 4) is upper bounded as $\mathbb{E}_{\tau_H}\left[\mathcal{L}(\mathbf{w}_T^{\tau_H})\right] \leq \frac{C}{T^\alpha}$. Furthermore, choosing $\beta_{MD} = \|\mathbf{w}_0 - \mathbf{w}_\star\|^2 \tilde{\beta}_{\min}$ (or $\beta_{MR} = R^2 \|\mathbf{w}_0 - \mathbf{w}_\star\|^2 \tilde{\beta}_{\min}$), i.e., postponing the stopping time as much as our (data-dependent) condition allows, leads to the tightest upper bound (in our derivations).*

*Proof.* For MD and MR hybrid orderings, we denote $\beta_{\mathrm{MD}} = \tilde{\beta} \|\mathbf{w}_0 - \mathbf{w}_\star\|^2$ and $\beta_{\mathrm{MR}} = \tilde{\beta} R^2 \|\mathbf{w}_0 - \mathbf{w}_\star\|^2$, respectively. Note the following holds for all $m \in [T]$, $\mathbf{w} \in \mathbb{R}^d$:

$$\|\mathbf{X}_m \mathbf{w} - \mathbf{y}_m\|^2 = \|\mathbf{X}_m (\mathbf{w} - \mathbf{w}_\star)\|^2 = \left\|\mathbf{X}_m \mathbf{X}_m^+ \mathbf{X}_m (\mathbf{w} - \mathbf{w}_\star)\right\|^2 \leq R^2 \|(\mathbf{I} - \mathbf{P}_m)(\mathbf{w} - \mathbf{w}_\star)\|^2 .$$

So, when $\|\mathbf{X}_m \mathbf{w}_{t-1} - \mathbf{y}_m\|^2 \geq \tilde{\beta} R^2 \|\mathbf{w}_0 - \mathbf{w}_\star\|^2$, immediately $\|(\mathbf{I} - \mathbf{P}_m)(\mathbf{w}_{t-1} - \mathbf{w}_\star)\|^2 \geq \tilde{\beta} \|\mathbf{w}_0 - \mathbf{w}_\star\|^2$, i.e., if the condition for continuing with greedy MR steps in Scheme 4 holds, then $\|(\mathbf{I} - \mathbf{P}_m)(\mathbf{w}_{t-1} - \mathbf{w}_\star)\|^2 \geq \tilde{\beta} \|\mathbf{w}_0 - \mathbf{w}_\star\|^2$ (this holds by definition for Scheme 3).

The last step $t$ for which $\max_{m \in [T] \backslash \tau(1:t-1)} \|(\mathbf{I} - \mathbf{P}_m)(\mathbf{w}_{t-1} - \mathbf{w}_\star)\|^2 \geq \tilde{\beta} \|\mathbf{w}_0 - \mathbf{w}_\star\|^2$ consecutively holds is some $t = s$, where $0 \leq s \leq T$. The following holds:

$$\|\mathbf{w}_s - \mathbf{w}_\star\|^2 = \|\mathbf{w}_0 - \mathbf{w}_\star\|^2 - \sum_{t=1}^{s} \|\mathbf{w}_t - \mathbf{w}_{t-1}\|^2 \leq \|\mathbf{w}_0 - \mathbf{w}_\star\|^2 \left(1 - \tilde{\beta}s\right) .$$

We are reminded of the definition for the (normalized) loss for a solution vector $\mathbf{w}$ with a task collection $\mathcal{T}$, starting from some starting point $\mathbf{w}_0$ and having a minimum norm joint solution $\mathbf{w}_\star$:

$$\mathcal{L}^{(\mathcal{T}, \mathbf{w}_0)}[\mathbf{w}] \triangleq \frac{1}{\|\mathbf{w}_0 - \mathbf{w}_\star\|^2 R^2} \frac{1}{|\mathcal{T}|} \sum_{m \in \mathcal{T}} \|\mathbf{X}_m (\mathbf{w} - \mathbf{w}_\star)\|^2 .$$

Running the hybrid scheme on the task collection $[T]$ yields the following expected loss:

$$\mathbb{E}_\tau \mathcal{L}^{([T], \mathbf{w}_0)}[\mathbf{w}_T] = \frac{1}{\|\mathbf{w_0} - \mathbf{w}_\star\|^2 R^2} \frac{1}{T} \sum_{m=1}^{T} \mathbb{E}\left[\|\mathbf{X}_m (\mathbf{w}_T - \mathbf{w}_\star)\|^2\right]$$

$$= \frac{1}{\|\mathbf{w}_0 - \mathbf{w}_\star\|^2 R^2} \frac{1}{T} \left[\sum_{t=1}^{s} \mathbb{E}\left[\|\mathbf{X}_{\tau(t)} (\mathbf{w}_T - \mathbf{w}_\star)\|^2\right] + \sum_{m \in [T] \backslash \tau(1:s)} \mathbb{E}\left[\|\mathbf{X}_m (\mathbf{w}_T - \mathbf{w}_\star)\|^2\right]\right]$$

$$\leq \frac{1}{\|\mathbf{w}_0 - \mathbf{w}_\star\|^2 R^2} \frac{1}{T} \left[R^2 \sum_{t=1}^{s} \mathbb{E}\left[\|\mathbf{w}_T - \mathbf{w}_\star\|^2\right] + \sum_{m \in [T] \backslash \tau(1:s)} \mathbb{E}\left[\|\mathbf{X}_m (\mathbf{w}_T - \mathbf{w}_\star)\|^2\right]\right]$$

$$\overset{(1)}{\leq} \frac{1}{\|\mathbf{w}_0 - \mathbf{w}_\star\|^2 R^2} \frac{1}{T} \left[R^2 \sum_{t=1}^{s} \mathbb{E}\left[\|\mathbf{w}_s - \mathbf{w}_\star\|^2\right] + \sum_{m \in [T] \backslash \tau(1:s)} \mathbb{E}\left[\|\mathbf{X}_m (\mathbf{w}_T - \mathbf{w}_\star)\|^2\right]\right]$$

$$\overset{(2)}{=} \frac{1}{\|\mathbf{w}_0 - \mathbf{w}_\star\|^2 R^2} \frac{1}{T} \left[R^2 s \|\mathbf{w}_s - \mathbf{w}_\star\|^2 + \sum_{m \in [T] \backslash \tau(1:s)} \mathbb{E}\left[\|\mathbf{X}_m (\mathbf{w}_T - \mathbf{w}_\star)\|^2\right]\right]$$

$$= \frac{\|\mathbf{w}_s - \mathbf{w}_\star\|^2}{T \|\mathbf{w}_0 - \mathbf{w}_\star\|^2} \left(s + (T-s) \left(\frac{1}{\|\mathbf{w}_s - \mathbf{w}_\star\|^2 R^2} \frac{1}{T-s} \sum_{m \in [T] \backslash \tau(1:s)} \mathbb{E}\left[\|\mathbf{X}_m (\mathbf{w}_T - \mathbf{w}_\star)\|^2\right]\right)\right)$$

$$= \frac{\|\mathbf{w}_s - \mathbf{w}_\star\|^2}{T \|\mathbf{w}_0 - \mathbf{w}_\star\|^2} \left(s + (T-s) \mathbb{E}_\tau \mathcal{L}^{([T] \backslash \tau(1:s), \mathbf{w}_s)}[\mathbf{w}_T]\right)$$

$$\leq \frac{1 - \tilde{\beta}s}{T} \left(s + (T-s) \mathbb{E}_\tau \mathcal{L}^{([T] \backslash \tau(1:s), \mathbf{w}_s)}[\mathbf{w}_T]\right) ,$$

where (1) is since $s \leq T$, and (2) is since $\mathbf{w}_s$ is deterministic. This means we can plug in any upper bound for the expected normalized loss of the random ordering, for the collection of $T - s$ tasks $[T] \setminus \tau(1:s)$ with the starting point $\mathbf{w}_s$, replacing dependence on $T$ with $T - s$. If we have an upper bound for the expected normalized loss of random ordering of $f(T)$ tasks, which is a positive and decreasing function of $T$, we obtain the following upper bound for hybrid ordering:

$$\mathbb{E}_\tau \mathcal{L}^{([T], \mathbf{w}_0)}[\mathbf{w}_T] \leq \frac{1 - \tilde{\beta}s}{T} (s + (T - s) f(T - s)) . \tag{11}$$

As a sanity check, setting $s = 0$ removes the greedy iterates, and the bound reduces to that of random ordering.

For the rest of the proof, we work with bounds of the following form:

$$f(T) = \begin{cases} \frac{C}{T^\alpha} & \frac{C}{T^\alpha} \leq 1 \\ 1 & \text{else} \end{cases} , \tag{12}$$

such that $f(T) \leq 1, \forall C, \alpha, T$. This only means that we ignore the cases where the bound on random orderings is entirely vacuous, *i.e.,* it is larger than 1.

We want a condition on $\tilde{\beta}$ for which continuing with greedy iterates as long as $\|(\mathbf{I} - \mathbf{P}_m)(\mathbf{w}_{t-1} - \mathbf{w}_\star)\|^2 \geq \tilde{\beta} \|\mathbf{w}_0 - \mathbf{w}_\star\|^2$, necessarily improves the bound. This means we want the bound to decrease with $s$. Thus, we demand $\forall s \in [T] : \frac{\mathrm{d}}{\mathrm{d}s} \left( \frac{1 - \tilde{\beta}s}{T} (s + (T - s) f(T - s)) \right) \leq 0$:

$$\frac{\mathrm{d}}{\mathrm{d}s} \left( \frac{1 - \tilde{\beta}s}{T} (s + (T - s) f(T - s)) \right)$$

$$= \frac{1}{T} \left( -\tilde{\beta}(s + (T - s) f(T - s)) + \left(1 - \tilde{\beta}s\right)(1 + (-f(T - s) - (T - s) f'(T - s))) \right)$$

$$= \frac{1}{T} \left( -\tilde{\beta}s - \tilde{\beta}Tf(T - s) + \tilde{\beta}sf(T - s) + 1 - f(T - s) - (T - s) f'(T - s) - \tilde{\beta}s \right.$$

$$\left. + \tilde{\beta}sf(T - s) + \tilde{\beta}s(T - s) f'(T - s) \right)$$

$$= \frac{1}{T} \left( 1 - 2\tilde{\beta}s - \left(1 + \tilde{\beta}T - 2\tilde{\beta}s\right) f(T - s) - \left(1 - \tilde{\beta}s\right)(T - s) f'(T - s) \right) .$$

When demanding this expression to be $\leq 0$, we get:

$$\tilde{\beta}(-2s - (T - 2s) f(T - s) + s(T - s) f'(T - s)) \leq -1 + f(T - s) + (T - s) f'(T - s)$$

$$\tilde{\beta} \geq \frac{1 - f(T - s) - (T - s) f'(T - s)}{Tf(T - s) + 2s(1 - f(T - s)) - s(T - s) f'(T - s)} .$$

As a sanity check, note that since $f(T - s) \leq 1$ and $f'(T - s) \leq 0$, the numerator is non-negative and the denominator is positive.

Continuing:

$$\tilde{\beta} \geq \frac{1 - f(T - s) - (T - s) f'(T - s)}{Tf(T - s) + 2s(1 - f(T - s)) - s(T - s) f'(T - s)}$$

$$= \frac{1 - f(T - s) - (T - s) f'(T - s)}{s(1 - f(T - s) - (T - s) f'(T - s)) - s + Tf(T - s) + s - sf(T - s)}$$

$$= \left( s + \frac{(T - s) f(T - s)}{1 - f(T - s) - (T - s) f'(T - s)} \right)^{-1}$$

$$\tilde{\beta}^{-1} \leq s + \frac{(T - s) f(T - s)}{1 - f(T - s) - (T - s) f'(T - s)} .$$

We demand this holds $\forall s \in [T]$. We assume, for now, that $\frac{C}{(T-s)^\alpha} \leq 1$. We will later examine the other case. Thus, plugging in $f(T - s) = \frac{C}{(T-s)^\alpha}$:

$$\tilde{\beta}^{-1} \leq s + \frac{(T-s)\frac{C}{(T-s)^\alpha}}{1 - \frac{C}{(T-s)^\alpha} + (T-s)\frac{\alpha C}{(T-s)^{\alpha+1}}} = s + \frac{C(T-s)^{1-\alpha}}{1 - \frac{C}{(T-s)^\alpha} + \frac{\alpha C}{(T-s)^\alpha}}$$

$$= s + \frac{C(T-s)}{(T-s)^\alpha \left(1 - \frac{C(1-\alpha)}{(T-s)^\alpha}\right)} = s + \frac{C(T-s)}{(T-s)^\alpha - C(1-\alpha)} \triangleq g(s).$$

We are reminded that we assumed $\frac{C}{(T-s)^\alpha} \leq 1$, thus $(T-s)^\alpha \geq C > C(1-\alpha)$, so the denominator here is positive. Denote $\tilde{\beta}^{-1} \leq g(s) \triangleq s + \frac{C(T-s)}{(T-s)^\alpha - C(1-\alpha)}$. In order to find an upper bound on $\tilde{\beta}^{-1}$ which holds for all $s$, we look for the minimum of $g(s)$. Differentiating, we get:

$$\frac{\mathrm{d}g(s)}{\mathrm{d}s} = 1 + \frac{-C((T-s)^\alpha - C(1-\alpha)) - C(T-s)\left(-\alpha(T-s)^{\alpha-1}\right)}{((T-s)^\alpha - C(1-\alpha))^2}$$

$$= 1 - C\frac{-\alpha(T-s)^\alpha + (T-s)^\alpha - C(1-\alpha)}{((T-s)^\alpha - C(1-\alpha))^2}$$

$$= 1 + \alpha C\frac{(T-s)^\alpha}{((T-s)^\alpha - C(1-\alpha))^2} - C\frac{1}{(T-s)^\alpha - C(1-\alpha)}$$

$$\geq 1 + \alpha C\frac{(T-s)^\alpha - C(1-\alpha)}{((T-s)^\alpha - C(1-\alpha))^2} - C\frac{1}{(T-s)^\alpha - C(1-\alpha)}$$

$$= 1 - \frac{C(1-\alpha)}{(T-s)^\alpha - C(1-\alpha)} = \frac{(T-s)^\alpha - 2C(1-\alpha)}{(T-s)^\alpha - C(1-\alpha)}.$$

Hence $g'(s) \geq 0$ for $s$ small enough such that $(T-s)^\alpha \geq 2C(1-\alpha)$, and $g'(s) < 0$ for larger values, up to the maximum value under the current assumption of $s = T - C^{1/\alpha}$. Hence, the minimum of $g(s)$ in $[0, T - C^{1/\alpha}]$ will be one of the boundary points:

$$g\left(T - C^{1/\alpha}\right) = T - C^{1/\alpha} + \frac{C \cdot C^{1/\alpha}}{C - C(1-\alpha)} = T - C^{1/\alpha} + \frac{C^{1/\alpha}}{\alpha}$$

$$= T + C^{1/\alpha}\left(\alpha^{-1} - 1\right) \geq T$$

$$g(0) = \frac{CT}{T^\alpha - C(1-\alpha)}$$

$$g\left(T - C^{1/\alpha}\right) - g(0) \geq T\left(1 - \frac{C}{T^\alpha - C(1-\alpha)}\right) = T\left(\frac{T^\alpha - C(2-\alpha)}{T^\alpha - C(1-\alpha)}\right).$$

This can only be negative when $T < (C(2-\alpha))^{1/\alpha}$, i.e., $f(T) = \frac{C}{T^\alpha} > \frac{1}{2-\alpha}$, which is a case not covered in this Lemma, since we assumed $\frac{C}{T^\alpha} \leq \frac{1}{2-\alpha}$ (and the bound is quite useless if it is larger than $\frac{1}{2}$ anyway). Thus, it is guaranteed that the lowest upper bound for $\tilde{\beta}^{-1}$ is for $s = 0$, and we get:

$$\tilde{\beta}^{-1} \leq \frac{CT}{T^\alpha - C(1-\alpha)}$$

$$\tilde{\beta} \geq \tilde{\beta}_{\min} = \frac{T^\alpha - C(1-\alpha)}{CT}.$$

Under this choice of $\tilde{\beta}$, the upper bound of Eq. (11) is monotonically decreasing with $s$ as long as $\frac{C}{(T-s)^\alpha} \leq 1$.

We now address the case of $s$ large enough such that $\frac{C}{(T-s)^\alpha} > 1$. In this case, our upper bound from Eq. (11) becomes: $\mathbb{E}_\tau \mathcal{L}^{([T], \mathbf{w}_0)}[\mathbf{w}_T] \leq \frac{1-\tilde{\beta}s}{T}(s + (T-s) \cdot 1) = 1 - \tilde{\beta}s$, which is also monotonically decreasing with $s$. From continuity of the bound at $s = T - C^{1/\alpha}$ (it does not matter that this might not be an integer), we get that the bound is monotonically decreasing with $s$ for all $s > 0$ when $\tilde{\beta} \geq \tilde{\beta}_{\min} = \frac{T^\alpha - C(1-\alpha)}{CT}$, and thus beats the bound for random ordering, achieved at $s = 0$. $\qquad\square$

## G.3 Hybrid ordering experiments

Figure 6 was acquired using the same data as Figure 3a, and using the dimension and rank-dependent upper bound of $2\,(d-r)/k$ from Evron et al. [35] to set $\beta$, since the universal bound of $14/k^{1/4}$ requires more than 50 iterations to be effective. The hybrid method results with intermediate performance between random and greedy. The figures demonstrate that the hybrid approach combines trends we have seen earlier (App. B) for random and greedy MD, in terms of the effect of dimension, rank, task count and task correlation on the performance.

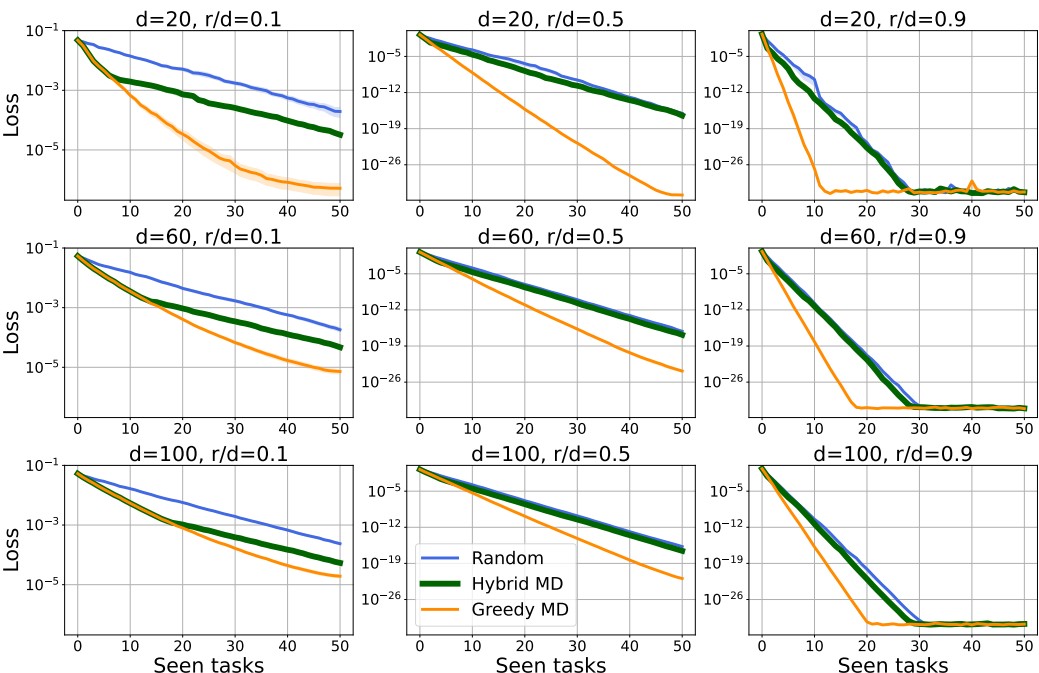

Figure 24: **Hybrid performance for varying dimensions $d$ and ranks $r$ of the data matrices, for isotropic data.** $T = 50$. In high-rank and/or low-dimensional settings, the rank-dependent upper bound employed by the hybrid strategy in this case is lower, prompting an earlier transition from the greedy to the random phase. Interestingly, the performance of the random phase within the hybrid method is slightly inferior to that of fully random ordering—possibly because the initial greedy steps deplete the set of "extreme" tasks that would otherwise drive greater progress.

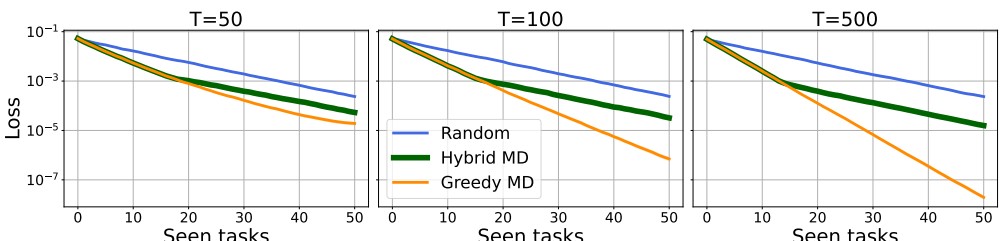

Figure 25: **Hybrid performance for varying task count $T$, for isotropic data.** $d = 100$, $r = 10$. We see similar trends. Note that the previously observed slight drop in performance of the random iterates following the greedy phase is less pronounced with higher task counts, possibly since more extreme tasks remain available for selection.

**Anisotropic data.** Similar trends were observed under anisotropic data, and we therefore omit the corresponding figures for brevity.

# H  Appendix to Section 4.1: Code snippet for regression experiments

The regression experiments are intentionally simple, and for completeness and reproducibility we provide a short code snippet. Running it generates a basic linear regression experiment on isotropic data, comparing random and greedy orderings (Eq. (4), Def. 3.1 and 3.2).

```python
# Minimal Block Kaczmarz experiment with a runnable demo + simple plot
import numpy as np
import matplotlib.pyplot as plt
from numpy.linalg import pinv
from typing import List, Tuple, Optional

# -------- Core utilities --------

def generate_data(r: int, d: int, T: int, seed: Optional[int] = None) ->
    Tuple[np.ndarray, np.ndarray, np.ndarray, List[tuple], List[np.ndarray],
    List[np.ndarray]]:
    """
    Returns:
        X: (T*r, d) matrix; b: (T*r,) labels; w_true: (d,) teacher.
        blocks: list of (X_t, b_t) with X_t in R^{r×d}, b_t in R^{r}.
        pinv_blocks: list of pseudoinverses X_t^+.
        md_proj: list of X_t^+ X_t (used by MD-based selection).
    """
    if seed is not None:
        np.random.seed(seed)

    X_blocks = [np.random.randn(r, d) for _ in range(T)]
    max_rad = max(np.linalg.norm(B, 2) for B in X_blocks)
    X_blocks = [B / max_rad for B in X_blocks]

    w_true = np.random.randn(d)
    w_true /= np.linalg.norm(w_true)

    X = np.vstack(X_blocks)
    b = X @ w_true

    X_blocks = [X[i*r:(i+1)*r, :] for i in range(T)]
    b_blocks = [b[i*r:(i+1)*r] for i in range(T)]
    pinv_blocks = [pinv(Xt) for Xt in X_blocks]
    md_proj = [pinv_blocks[t] @ X_blocks[t] for t in range(T)]
    blocks = list(zip(X_blocks, b_blocks))
    return X, b, w_true, blocks, pinv_blocks, md_proj
```

Listing 2: Data generation.

```python
def _pick_uniform(T: int, used: Optional[List[int]], allow_repetition: bool) ->
↪  Optional[int]:
    if allow_repetition:
        return int(np.random.randint(0, T))
    pool = list(set(range(T)) - set(used or []))
    return int(np.random.choice(pool)) if pool else None

def _pick_greedy_mr(blocks: List[tuple], w: np.ndarray, used:
↪  Optional[List[int]], allow_repetition: bool) -> Optional[int]:
    best, best_val = None, -np.inf
    for i, (Xt, bt) in enumerate(blocks):
        if (not allow_repetition) and used and i in used:
            continue
        val = np.linalg.norm(Xt @ w - bt) ** 2
        if val >= best_val:
            best, best_val = i, val
    return best

def _pick_greedy_md(md_proj: List[np.ndarray], w: np.ndarray, w_true:
↪  np.ndarray, used: Optional[List[int]], allow_repetition: bool) ->
↪  Optional[int]:
    best, best_val = None, -np.inf
    for i, P in enumerate(md_proj):
        if (not allow_repetition) and used and i in used:
            continue
        val = np.linalg.norm(P @ (w - w_true)) ** 2
        if val >= best_val:
            best, best_val = i, val
    return best
```

Listing 3: Ordering strategies.

```
1   def block_kaczmarz(
2       blocks: List[tuple],
3       pinv_blocks: List[np.ndarray],
4       md_proj: List[np.ndarray],
5       w_true: np.ndarray,
6       T: int,
7       d: int,
8       max_iters: int,
9       selection: str = "uniform",    # one of {"uniform","greedy_mr","greedy_md"}
10      allow_repetition: bool = False,
11  ) -> Tuple[np.ndarray, List[float], List[float]]:
12      """Runs Block Kaczmarz and returns (w, out_losses, dist_to_teacher) per
        ↪  iteration."""
13      w = np.zeros(d)
14      out_losses: List[float] = []
15      dist: List[float] = []
16      used: List[int] = []
17
18      for _ in range(max_iters):
19          if selection == "uniform":
20              t = _pick_uniform(T, used, allow_repetition)
21          elif selection == "greedy_mr":
22              t = _pick_greedy_mr(blocks, w, used, allow_repetition)
23          elif selection == "greedy_md":
24              t = _pick_greedy_md(md_proj, w, w_true, used, allow_repetition)
25          else:
26              raise ValueError("selection must be one of
                ↪  {'uniform','greedy_mr','greedy_md'}")
27
28          if t is None:  # no available block under no-replacement
29              out_losses.append(np.nan)
30              dist.append(np.linalg.norm(w - w_true) ** 2)
31              continue
32
33          Xt, bt = blocks[t]
34          w += pinv_blocks[t] @ (bt - Xt @ w)  # single block least squares step
35
36          if not allow_repetition:
37              used.append(t)
38
39          # metrics
40          loss = 0.0
41          for Xs, bs in blocks:
42              r = Xs @ w - bs
43              loss += np.linalg.norm(r) ** 2
44          out_losses.append(loss / T)
45          dist.append(np.linalg.norm(w - w_true) ** 2)
46
47      return w, out_losses, dist
```

Listing 4: Experiment loop.

```python
# -------- One reproducible experiment + simple figure --------

# Problem size / schedule
d = 100
r = max(1, int(0.1 * d))    # rows per block (r/d = 0.1)
T = 50                      # number of blocks
max_iters = T               # single pass without replacement
seed = 42                   # reproducible

# Data and precomputations
_, _, w_true, blocks, pinv_blocks, md_proj = generate_data(r, d, T, seed=seed)

# Compare three orderings (no replacement for apples-to-apples single pass)
strategies = [
    ("Random",     {"selection": "uniform",   "allow_repetition": False}),
    ("Greedy MR",  {"selection": "greedy_mr", "allow_repetition": False}),
    ("Greedy MD",  {"selection": "greedy_md", "allow_repetition": False}),
]

results = {}
for name, opts in strategies:
    _, out_losses, dist = block_kaczmarz(
        blocks=blocks,
        pinv_blocks=pinv_blocks,
        md_proj=md_proj,
        w_true=w_true,
        T=T,
        d=d,
        max_iters=max_iters,
        **opts
    )
    results[name] = (np.asarray(out_losses), np.asarray(dist))

# Print final metrics
print("Final loss and distance-to-teacher:")
for name in strategies:
    key = name[0]
    L, D = results[key]
    # pick the last non-nan value
    last_idx = np.where(~np.isnan(L))[0][-1]
    print(f"  {key:10s}  loss={L[last_idx]:.4e}   dist={D[last_idx]:.4e}")

# Simple comparison figure
plt.figure(figsize=(6, 4))
xs = np.arange(1, max_iters + 1)
for name, (L, _) in results.items():
    plt.plot(xs, L[:max_iters], label=name)
plt.xlabel("Iterations (seen tasks)")
plt.ylabel("Average loss")
plt.yscale("log")
plt.grid(True, alpha=0.3)
plt.legend()
plt.tight_layout()
plt.show()
```

Listing 5: Simple ordering comparison experiment.

# I Lower bound technical appendix: Delta positivity proof

This section supplements App. E.1, which we recommend reviewing in advance. Here, we prove that $\forall d \geq 25{,}000$, $\forall t \in \{2, \ldots, d-1\}$, $\forall k \in \{t, \ldots, d-1\}$,

$$\Delta_{t,k} \triangleq \|\mathbf{w}_k - \mathbf{w}_{k+1}\| (\mathbf{w}_{k-1} - \mathbf{w}_k)^\top \mathbf{w}_{t-1} - \|\mathbf{w}_{k-1} - \mathbf{w}_k\| (\mathbf{w}_k - \mathbf{w}_{k+1})^\top \mathbf{w}_{t-1} > 0\,.$$

In some places in our proofs, we will need a closed-form approximation of the first coordinates $x_k \triangleq (\mathbf{w}_k)_1$ which we obtain recursively. Such an approximation was suggested in Hucht [48]:

$$\tilde{x}_k = \sqrt{1 - \frac{1}{\ln 4} + 4^{-\frac{k}{d}} \left( \frac{1}{\ln 4} - \frac{k}{d} \right)}\,.$$

This will be formalized and proven in App. J. In addition this gives us a lower bound $x_k \geq 0.45$, $\forall k \in [d]$ when $d \geq 25{,}000$ (Corollary J.3).

## I.1 Proof outline

The proof is straightforward: we decompose $\Delta_{t,k}$ to smaller parts, and attempt to lower bound each of these parts. We then combine all of these lower bounds to achieve an overall lower bound on $\Delta_{t,k}$ and find a sufficient condition on $d$ for which this lower bound is positive. This condition, revealed in Eq. (13), is already satisfied when $d \geq 25{,}000$, concluding the proof. We begin by bounding some intermediate quantities that appear later in the derivation, and starting in App. I.3.6 we decompose and lower bound $\Delta_{t,k}$.

## I.2 Auxiliary: Algebraic inequalities

**Claim I.1.** $\forall d \in \mathbb{N}$ and $1 \leq n \leq d$, it holds that $1 - c^n \triangleq 1 - 2^{-n/d} \in \left[ \frac{n \ln(2)}{d} - \frac{n^2 \ln^2(2)}{2d^2}, \frac{n \ln(2)}{d} \right]$.
Particularly, this shows $1 - c \in \left[ \frac{\ln(2)}{d} - \frac{\ln^2(2)}{2d^2}, \frac{\ln(2)}{d} \right]$.

*Proof.* To show the upper bound, we define $\alpha = n/d \in (0, 1]$ and $f(\alpha) = 1 - 2^{-\alpha} - \alpha \ln(2)$, and notice that $f$ is *decreasing* in $(0, 1]$ since

$$f'(\alpha) = \left( 2^{-\alpha} - 1 \right) \ln(2) \propto 2^{-\alpha} - 1 < 0, \quad \forall \alpha \in (0, 1]\,.$$

Then, this means $f(\alpha) = 1 - 2^{-n/d} - \frac{n \ln(2)}{d} \leq \lim_{\alpha \to 0^+} f(\alpha) = 0$ as required.

Conversely, we get the lower bound by showing that the function $g\left( \alpha = \frac{n}{d} \right) = 1 - 2^{-n/d} - \left( \frac{n \ln(2)}{d} - \frac{n^2 \ln^2(2)}{2d^2} \right)$ is *increasing* in $(0, 1]$,

$$g(\alpha) = 1 - 2^{-\alpha} - \left( \alpha \ln(2) - \frac{\alpha^2 \ln^2(2)}{2} \right), \quad \lim_{\alpha \to 0^+} g(0) = 1 - 2^{-0} - 0 = 0\,,$$

$$g'(\alpha) = \ln(2) \left( 2^{-\alpha} + \alpha \ln(2) - 1 \right) \propto 2^{-\alpha} + \alpha \ln(2) - 1 = -f(\alpha) + 1$$

$$\geq -\lim_{\alpha \to 0^+} f(\alpha) + 1 = -\left( 1 - 2^{-0} - 0 \right) + 1 = 1 > 0\,.$$

$\square$

**Claim I.2.** For $\forall d, n, m \in \mathbb{N}$ and $k \in [d]$, we have $c^{nk-m} \geq 2^{-n}$.

*Proof.* Notice that $c^z = 2^{-z/d}$ is *decreasing* with $z$. Plugging in $z = nk - m \leq nd$, we get $c^z \geq c^{nd} = 2^{-n}$. $\square$

**Claim I.3.** $\forall k \in [1, d]$ it holds that $1 - (1 - c)(k - 1) = ((k - 1)c - (k - 2)) \in [0, 1]$.

*Proof.* It is clear that $(1 - c)(k - 1) \triangleq (1 - 2^{-1/d})(k - 1) \geq 0$. Then, we can simply show that from Claim I.1:

$$\underbrace{(1 - c)}_{\geq 0}(k - 1) \leq (1 - c)(d - 1) \leq \frac{\ln(2)}{d}(d - 1) < \ln(2) < 1.$$

$\square$

**Claim I.4.** $\forall k \in [1, d]$ it holds that $kc - (k - 1) > 0$.

*Proof.* From Claim I.1 we have $1 - c \leq \frac{\ln 2}{d} \Rightarrow c \geq 1 - \frac{\ln 2}{d}$. Plugging that in we get:

$$kc - (k - 1) \geq k\left(1 - \frac{\ln 2}{d}\right) - k + 1 = 1 - \ln 2\frac{k}{d} \geq 1 - \ln 2 > 0.$$

$\square$

**Claim I.5.** $\forall k \in [d]$ it holds that $\beta_k \in \left[\frac{0.3c^{2k-5}}{d}, \frac{c^{2k-5}}{d}\right]$.

*Proof.*

$$\beta_k = \frac{((k - 1)c - (k - 2))c^{2k-5}}{d} = \left(1 - \underbrace{(1 - c)(k - 1)}_{\in[0, \ln(2)] \subset [0, 0.7]}\right) \cdot \frac{c^{2k-5}}{d} \in \left[\frac{0.3c^{2k-5}}{d}, \frac{c^{2k-5}}{d}\right].$$

$\square$

**Claim I.6.** $x_k$ is decreasing and $\forall k \in [d], \ x_k \leq 1$.

*Proof.* Decreasing follows immediately from positivity of $\beta_k$ (see Claim I.5) and the construction, and since $x_1 = 1$ we get $\forall k \in [d], \ x_k \leq 1$. $\square$

**Claim I.7.** $\forall k \in [2, d]$ it holds that $\beta_k \leq \frac{1}{cd}$.

*Proof.* Since $c < 1$, we have $\beta_k \leq \frac{c^{2k-5}}{d} \leq \frac{c^{2 \cdot 2 - 5}}{d} = \frac{1}{cd}$. $\square$

**Claim I.8.** $\forall a > 0, b \in \mathbb{R} \setminus \{0\}$ such that $a + b \geq 0$, it holds that $\sqrt{a + b} < \sqrt{a} + \frac{b}{2\sqrt{a}}$.

*Proof.* $0 < b^2 \iff 4a(a + b) < 4a^2 + 4ab + b^2 = (2a + b)^2 \iff \sqrt{a + b} < \sqrt{a} + \frac{b}{2\sqrt{a}}$. $\square$

**Claim I.9.** $\forall d \geq 1 : \ 2^{1/d} \geq 1 + \frac{\ln 2}{d}$.

*Proof.* Using Taylor's expansion: $2^{1/d} = e^{\frac{\ln 2}{d}} = 1 + \frac{\ln 2}{d} + \sum_{i=2}^{\infty} \frac{1}{i!}\left(\frac{\ln 2}{d}\right)^i \geq 1 + \frac{\ln 2}{d}$. $\square$

**Claim I.10.** If $|x_k - \tilde{x}_k| \leq \epsilon$ and $x_k \geq 0$, $\left|x_k^2 - \tilde{x}_k^2\right| \leq 2x_k\epsilon_d + \epsilon_d^2$.

*Proof.* Defining $r = \tilde{x}_k - x_k$, we have,

$$\left|x_k^2 - \tilde{x}_k^2\right| = \left|x_k^2 - (x_k + r)^2\right| = \left|2x_k r - r^2\right| \leq |2x_k r| + r^2 = 2|x_k(\tilde{x}_k - x_k)| + (\tilde{x}_k - x_k)^2$$
$$\leq 2x_k\epsilon + \epsilon^2.$$

$\square$

### I.3 Proof body

#### I.3.1 Analyzing $x_{k-1} - x_k$, $(x_{k-1} - x_k)^2$, $\frac{x_k}{x_{k-1}}$

**Proposition I.11.** *For any $k \geq 2$, it holds that,*

$$x_{k-1} - x_k \triangleq f_{x_{k-1}}(\beta_k) \in \left[ \frac{\beta_k}{x_{k-1}} + \frac{\beta_k^2}{x_{k-1}^3} + \frac{2\beta_k^3}{x_{k-1}^5}, \frac{\beta_k}{x_{k-1}} + \frac{\beta_k^2}{x_{k-1}^3} + \frac{2\beta_k^3}{\left(x_{k-1}^2 - 4\beta_k\right)^{5/2}} \right]$$

$$[\text{when } d \geq 25{,}000] \subseteq \left[ \frac{\beta_k}{x_{k-1}} + \frac{\beta_k^2}{x_{k-1}^3}, \frac{\beta_k}{x_{k-1}} + \frac{\beta_k^2}{x_{k-1}^3} + \frac{113c^{6k-15}}{d^3} \right].$$

*Proof.* By construction, we have

$$x_{k-1} - x_k = \frac{x_{k-1} - \sqrt{x_{k-1}^2 - 4\beta_k}}{2}.$$

Define $f_z(x) = \frac{1}{2}\left(z - \sqrt{z^2 - 4x}\right)$ for $z \in [0.45, 1]$ (see Claim I.6, Corollary J.3) and $z^2 \gg x > 0$.
Expand with Taylor:

$$f_z(0) = f_z(0)$$

$$f_z^{(1)}(x) = -\frac{1}{4}\frac{-4}{\sqrt{z^2 - 4x}} = \frac{1}{\sqrt{z^2 - 4x}} \qquad\qquad \implies f_z^{(1)}(0) = \frac{1}{z},$$

$$f_z^{(2)}(x) = 2\left(z^2 - 4x\right)^{-3/2} \qquad\qquad \implies f_z^{(2)}(0) = \frac{2}{z^3},$$

$$f_z^{(3)}(x) = 2\frac{3}{2}\cdot 4\left(z^2 - 4x\right)^{-5/2} = 12\left(z^2 - 4x\right)^{-5/2} \qquad \implies f_z^{(3)}(0) = \frac{12}{z^5},$$

and notice that generally $\forall z^2 \gg x > 0$ we have $f_z^{(n)}(x) > 0$.

Then, by Lagrange's form of the remainder, the error of the quadratic approximation (around $x = 0$) is given by

$$f_z(x) = \frac{f(0)}{0!}x^0 + \frac{f^{(1)}(0)}{1!}x^1 + \frac{f^{(2)}(0)}{2!}x^2 + R_2(x) = \frac{x}{z} + \frac{x^2}{z^3} + R_2(x),$$

where

$$R_2(x) = \frac{f^{(3)}(x_0)}{3!}(x-0)^3 = \frac{12\left(z^2 - 4x_0\right)^{-5/2}}{6}x^3 \in \left[ \frac{2x^3}{z^5}, \frac{2x^3}{\left(z^2 - 4x\right)^{5/2}} \right].$$

since $x_0 \in [0, x]$.

We get that

$$x_{k-1} - x_k = f_{x_{k-1}}(\beta_k) \in \left[ \frac{\beta_k}{x_{k-1}} + \frac{\beta_k^2}{x_{k-1}^3} + \frac{2\beta_k^3}{x_{k-1}^5}, \frac{\beta_k}{x_{k-1}} + \frac{\beta_k^2}{x_{k-1}^3} + \frac{2\beta_k^3}{\left(x_{k-1}^2 - 4\beta_k\right)^{5/2}} \right].$$

Finally, since $\beta_k \leq \frac{c^{2k-5}}{d}$ and $x_{k-1} \in [0.45, 1]$ we have

$$\frac{2\beta_k^3}{\left(x_{k-1}^2 - 4\beta_k\right)^{5/2}} \leq \frac{2\left(\frac{c^{2k-5}}{d}\right)^3}{\left(0.45^2 - 4\frac{c^{2k-5}}{d}\right)^{5/2}} \leq \frac{2c^{6k-15}}{d^3\left(0.45^2 - 4\frac{1}{cd}\right)^{5/2}}$$

$$\leq \frac{2c^{6k-15}}{d^{1/2}\left(0.2d - 4\cdot 2^{1/d}\right)^{5/2}}$$

$$\left[d \geq 10{,}000 \Rightarrow 4\cdot 2^{1/d} \leq 0.00041d\right] \leq \frac{2c^{6k-15}}{d^{1/2}\left(0.2d - 0.00041d\right)^{5/2}} \leq \frac{113c^{6k-15}}{d^3}.$$

$\square$

**Proposition I.12.** *For any $k \geq 2$ (and $d \geq 25{,}000$), it holds that,*

$$\frac{x_k}{x_{k-1}} \in \left(1 - \frac{\beta_k}{x_{k-1}^2} - \left(1 + \frac{10}{d}\right)\frac{\beta_k^2}{x_{k-1}^4}, \ 1 - \frac{\beta_k}{x_{k-1}^2}\right).$$

*Proof.* We employ the bounds we found for $x_{k-1} - x_k$:

$$\frac{x_k}{x_{k-1}} = \frac{1}{x_{k-1}}\left(x_k - x_{k-1} + x_{k-1}\right) = 1 - \frac{1}{x_{k-1}}\left(x_{k-1} - x_k\right)$$

$$\in \left[1 - \frac{\beta_k}{x_{k-1}^2} - \frac{\beta_k^2}{x_{k-1}^4} - \frac{2\beta_k^3}{x_{k-1}\left(x_{k-1}^2 - 4\beta_k\right)^{5/2}}, \ 1 - \frac{\beta_k}{x_{k-1}^2} - \frac{\beta_k^2}{x_{k-1}^4} - \frac{2\beta_k^3}{x_{k-1}^6}\right]$$

$$\subset \left[1 - \frac{\beta_k}{x_{k-1}^2} - \frac{\beta_k^2}{x_{k-1}^4} - \frac{2\beta_k^3}{x_{k-1}\left(x_{k-1}^2 - 4\beta_k\right)^{5/2}}, \ 1 - \frac{\beta_k}{x_{k-1}^2}\right].$$

Notice that from the bounds on $\beta_k, x_{k-1}$, we have:

$$\frac{2\beta_k^3}{x_{k-1}\left(x_{k-1}^2 - 4\beta_k\right)^{5/2}} = \frac{2\beta_k^2 \beta_k x_{k-1}^3}{x_{k-1}^4\left(x_{k-1}^2 - 4\beta_k\right)^{5/2}} \leq \frac{2\beta_k^2 \frac{c^{2k-5}}{d} 1^3}{x_{k-1}^4\left(0.45^2 - 4\frac{c^{2k-5}}{d}\right)^{5/2}}$$

$$\leq \frac{2\beta_k^2 \frac{1}{d}}{x_{k-1}^4\left(0.45^2 - 4\frac{1}{cd}\right)^{5/2}} = \frac{2\beta_k^2}{d \cdot 0.45^2 x_{k-1}^4\left(1 - \frac{4}{0.45^2}\frac{1}{2^{-1/d}d}\right)^{5/2}}$$

$$[d \geq 10{,}000] \leq \frac{2\beta_k^3}{d \cdot 0.45^2 x_{k-1}^4\left(1 - \frac{4}{0.45^2}\frac{1}{2^{-1/10000}\cdot 10000}\right)^{5/2}} \leq \frac{\beta_k^3}{x_{k-1}^4}\cdot\frac{10}{d}.$$

Since $d \geq 10{,}000$, we obtain $\frac{2\beta_k^3}{x_{k-1}\left(x_{k-1}^2 - 4\beta_k\right)^{5/2}} \leq \frac{10}{d}\frac{\beta_k^2}{x_{k-1}^4}$. Overall, we get

$$\frac{x_k}{x_{k-1}} \in \left(1 - \frac{\beta_k}{x_{k-1}^2} - \left(1 + \frac{10}{d}\right)\frac{\beta_k^2}{x_{k-1}^4}, \ 1 - \frac{\beta_k}{x_{k-1}^2}\right).$$

$\square$

**Proposition I.13.** *For $k \geq 2$ (and $d \geq 25{,}000$), it holds that,*

$$(x_{k-1} - x_k)^2 = \left( \frac{x_{k-1} - \sqrt{x_{k-1}^2 - 4\beta_k}}{2} \right)^2 \in \left[ \frac{\beta_k^2}{x_{k-1}^2}, \frac{\beta_k^2}{x_{k-1}^2} + \frac{113c^{6k-15}}{d^3} \right] .$$

*Proof.* We exploit the Taylor expansion of the following function (for $z^2 \gg x > 0$),

$$f(x) = \left( \frac{z - \sqrt{z^2 - 4x}}{2} \right)^2 ,$$

$$f^{(1)}(x) = \frac{z}{\sqrt{z^2 - 4x}} - 1, \ f^{(2)}(x) = \frac{2z}{(z^2 - 4x)^{3/2}}, \ f^{(3)}(x) = \frac{12z}{(z^2 - 4x)^{5/2}} .$$

Then, by Lagrange's form of the remainder, the error of the quadratic approximation (around $x = 0$) is given by

$$f(x) = \frac{f(0)}{0!} x^0 + \frac{f^{(1)}(0)}{1!} x^1 + \frac{f^{(2)}(0)}{2!} x^2 + R_2(x)$$

$$= 0 + 0 \cdot x^1 + \frac{2}{2z^2} x^2 + R_2(x) = \frac{x^2}{z^2} + R_2(x) ,$$

where

$$R_2(x) = \frac{f^{(3)}(x_0)}{3!} (x - 0)^3 = \frac{12z}{6(z^2 - 4x_0)^{5/2}} x^3 = \frac{2z}{(z^2 - 4x_0)^{5/2}} x^3 \in \left[ \frac{2x^3}{z^4}, \frac{2z \cdot x^3}{(z^2 - 4x)^{5/2}} \right] ,$$

since $x_0 \in [0, x]$.

Then, setting $z = x_{k-1} \in [0.45, 1]$, we can now conclude that,

$$(x_{k-1} - x_k)^2 = \left( \frac{x_{k-1} - \sqrt{x_{k-1}^2 - 4\beta_k}}{2} \right)^2 \triangleq f(\beta_k)$$

$$\in \left[ \frac{\beta_k^2}{x_{k-1}^2} + \frac{2\beta_k^3}{x_{k-1}^4}, \frac{\beta_k^2}{x_{k-1}^2} + \frac{2\beta_k^3 x_{k-1}}{(x_{k-1}^2 - 4\beta_k)^{5/2}} \right] .$$

We simplify the lower bound as $(x_{k-1} - x_k)^2 \geq \frac{\beta_k^2}{x_{k-1}^2} + \frac{2\beta_k^3}{x_{k-1}^4} \geq \frac{\beta_k^2}{x_{k-1}^2}$.

Finally, for the upper bound, since $\beta_k \leq \frac{c^{2k-5}}{d}$ and $x_{k-1} \in [0.45, 1]$ we have

$$(x_{k-1} - x_k)^2 \leq \frac{\beta_k^2}{x_{k-1}^2} + \frac{2\beta_k^3 x_{k-1}}{(x_{k-1}^2 - 4\beta_k)^{5/2}} \leq \frac{\beta_k^2}{x_{k-1}^2} + \frac{2c^{6k-15} x_{k-1}}{d^3 \left( x_{k-1}^2 - 4\frac{c^{2k-5}}{d} \right)^{5/2}}$$

$$\leq \frac{\beta_k^2}{x_{k-1}^2} + \frac{2 \cdot c^{6k-15}}{d^3 \left( 0.45^2 - 4\frac{1}{cd} \right)^{5/2}} \leq \frac{\beta_k^2}{x_{k-1}^2} + \frac{2c^{6k-15}}{d^{1/2} \left( 0.2d - 4 \cdot 2^{1/d} \right)^{5/2}}$$

$$\left[ \xRightarrow[4 \cdot 2^{1/d} \leq 0.00041d]{d \geq 10{,}000} \right] \leq \frac{\beta_k^2}{x_{k-1}^2} + \frac{2c^{6k-15}}{d^{1/2} \left( 0.2d - 0.00041d \right)^{5/2}} \leq \frac{\beta_k^2}{x_{k-1}^2} + \frac{113c^{6k-15}}{d^3} .$$

$\square$

### I.3.2 Expanding the inner product $\left(\mathbf{w}_{k-1} - \mathbf{w}_k\right)^\top \mathbf{w}_{t-1}$

**Proposition I.14.** *Let* $t \in [d]$ *and* $t < k \leq d$ *(and* $d \geq 25{,}000$*). Then,*

$$\left(\mathbf{w}_{k-1} - \mathbf{w}_k\right)^\top \mathbf{w}_{t-1} \in \left[\frac{x_{t-1}}{x_{k-1}}\beta_k + \frac{x_{t-1}}{x_{k-1}^3}\beta_k^2 + \frac{t-2}{d}c^{k+t-6}\left(1-c\right),\right.$$

$$\left.\frac{x_{t-1}}{x_{k-1}}\beta_k + \frac{x_{t-1}}{x_{k-1}^3}\beta_k^2 + \frac{t-2}{d}c^{k+t-6}\left(1-c\right) + \frac{113c^{6k-15}}{d^3}\right] .$$

*Proof.* We use the expanded form of the inner product, that is,

$$0 < \left(\mathbf{w}_{k-1} - \mathbf{w}_k\right)^\top \mathbf{w}_{t-1} = \left(x_{k-1} - x_k\right)x_{t-1} + \left(t-2\right)\frac{c^{k-3}\left(1-c\right)}{\sqrt{d}}\frac{c^{t-3}}{\sqrt{d}}$$

$$= \left(x_{k-1} - x_k\right)x_{t-1} + \frac{t-2}{d}c^{k+t-6}\left(1-c\right) .$$

Since we already showed $x_{k-1} - x_k \in \left[\frac{\beta_k}{x_{k-1}} + \frac{\beta_k^2}{x_{k-1}^3}, \frac{\beta_k}{x_{k-1}} + \frac{\beta_k^2}{x_{k-1}^3} + \frac{113c^{6k-15}}{d^3}\right]$, we now have,

$$\left(\mathbf{w}_{k-1} - \mathbf{w}_k\right)^\top \mathbf{w}_{t-1} \geq \frac{x_{t-1}}{x_{k-1}}\beta_k + \frac{x_{t-1}}{x_{k-1}^3}\beta_k^2 + \frac{t-2}{d}c^{k+t-6}\left(1-c\right) ,$$

and,

$$\left(\mathbf{w}_{k-1} - \mathbf{w}_k\right)^\top \mathbf{w}_{t-1} \leq x_{t-1}\left(\frac{\beta_k}{x_{k-1}} + \frac{\beta_k^2}{x_{k-1}^3} + \frac{113c^{6k-15}}{d^3}\right) + \frac{t-2}{d}c^{k+t-6}\left(1-c\right)$$

$$\left[x_{t-1} \leq 1\right] \leq \frac{x_{t-1}}{x_{k-1}}\beta_k + \frac{x_{t-1}}{x_{k-1}^3}\beta_k^2 + \frac{t-2}{d}c^{k+t-6}\left(1-c\right) + \frac{113c^{6k-15}}{d^3} .$$

$\square$

### I.3.3 Bounding $h\left(k\right) \triangleq \sqrt{\frac{\beta_k^2}{x_{k-1}^2} + \frac{k-2}{d}c^{2k-6}\left(1-c\right)^2 + \frac{1}{d}c^{2k-6}}$

**Proposition I.15.** *For any* $k \geq 2$ *(when* $d \geq 25{,}000$*),* $h\left(k\right) \in \left[\frac{c^{k-3}}{\sqrt{d}}, \frac{c^{k-3}}{\sqrt{d}} + \frac{5.42}{d^{3/2}}\right]$.

*Proof.* The lower bound is easy to obtain:

$$h\left(k\right) = \sqrt{\frac{\beta_k^2}{x_{k-1}^2} + \frac{k-2}{d}c^{2k-6}\left(1-c\right)^2 + \frac{1}{d}c^{2k-6}} > \sqrt{\frac{1}{d}c^{2k-6}} \geq \frac{c^{k-3}}{\sqrt{d}} .$$

To get the upper bound, we employ the inequality $\left(1-c\right) \triangleq 1 - 2^{-1/d} \leq \frac{\ln(2)}{d}$, and get,

$$h\left(k\right) = \sqrt{\frac{\beta_k^2}{x_{k-1}^2} + \frac{k-2}{d}c^{2k-6}\left(1-c\right)^2 + \frac{c^{2k-6}}{d}}$$

$$\leq \sqrt{\frac{\beta_k^2}{\left(0.45\right)^2} + \left(1-c\right)^2 + \frac{c^{2k-6}}{d}} \leq \sqrt{\frac{\beta_k^2}{\left(0.45\right)^2} + \frac{\ln^2\left(2\right)}{d^2} + \frac{c^{2k-6}}{d}}$$

$$\left[\text{I.7}\right] \leq \sqrt{\frac{1}{\left(0.45\right)^2 c^2 d^2} + \frac{\ln^2\left(2\right)}{d^2} + \frac{c^{2k-6}}{d}}$$

$$\left[\begin{array}{c}d \geq 10{,}000 \\ \Rightarrow c^2 \geq 0.99986\end{array}\right] \leq \sqrt{\frac{c^{2k-6}}{d} + \frac{5.42}{d^2}}$$

$$\left[\text{I.8}\right] < \sqrt{\frac{c^{2k-6}}{d}} + \frac{1}{2\sqrt{\frac{c^{2k-6}}{d}}} \cdot \frac{5.42}{d^2} = \frac{c^{k-3}}{\sqrt{d}} + \frac{1}{2c^{k-3}} \cdot \frac{5.42}{d^{3/2}}$$

$$\left[c^{k-m} \geq 2^{-1}\right] \leq \frac{c^{k-3}}{\sqrt{d}} + \frac{5.42}{d^{3/2}} .$$

$\square$

## I.3.4 Bounding $\frac{h(k+1)}{h(k)}$

**Proposition I.16.** *For any $k \geq 2$ (when $d \geq 500$),*

$$\frac{h(k+1)}{h(k)} \in \left[ c - \frac{5.5c^{2k-5}}{x_{k-1}^2 d^2}, \; c + \frac{2.44}{x_k^4 c^{2k-3} d^2} \right].$$

*Proof.* We start by expanding the expression in a way that will be useful for both the upper and the lower bounds,

$$\frac{h(k+1)}{h(k)} = \sqrt{\frac{\frac{\beta_{k+1}^2}{x_k^2} + \frac{k-1}{d}c^{2k-4}(1-c)^2 + \frac{1}{d}c^{2k-4}}{\frac{\beta_k^2}{x_{k-1}^2} + \frac{k-2}{d}c^{2k-6}(1-c)^2 + \frac{1}{d}c^{2k-6}}}$$

$$= \sqrt{\frac{\frac{\beta_{k+1}^2}{x_k^2} + \frac{k-2}{d}c^{2k-4}(1-c)^2 + \frac{1}{d}c^{2k-4}}{\frac{\beta_k^2}{x_{k-1}^2} + \frac{k-2}{d}c^{2k-6}(1-c)^2 + \frac{1}{d}c^{2k-6}} + \frac{\frac{1}{d}c^{2k-4}(1-c)^2}{\frac{\beta_k^2}{x_{k-1}^2} + \frac{k-2}{d}c^{2k-6}(1-c)^2 + \frac{1}{d}c^{2k-6}}}$$

$$= \sqrt{c^2 + \frac{\frac{\beta_{k+1}^2}{x_k^2} - \frac{c^2\beta_k^2}{x_{k-1}^2}}{\frac{\beta_k^2}{x_{k-1}^2} + \frac{k-2}{d}c^{2k-6}(1-c)^2 + \frac{1}{d}c^{2k-6}} + \frac{\frac{1}{d}c^{2k-4}(1-c)^2}{\frac{\beta_k^2}{x_{k-1}^2} + \frac{k-2}{d}c^{2k-6}(1-c)^2 + \frac{1}{d}c^{2k-6}}}.$$

**For the upper bound.** We show that,

$$\frac{h(k+1)}{h(k)}$$

$$= \sqrt{c^2 + \frac{\frac{\beta_{k+1}^2}{x_k^2} - \frac{c^2\beta_k^2}{x_{k-1}^2}}{\frac{\beta_k^2}{x_{k-1}^2} + \frac{k-2}{d}c^{2k-6}(1-c)^2 + \frac{1}{d}c^{2k-6}} + \frac{\frac{1}{d}c^{2k-4}(1-c)^2}{\frac{\beta_k^2}{x_{k-1}^2} + \frac{k-2}{d}c^{2k-6}(1-c)^2 + \frac{1}{d}c^{2k-6}}}$$

$$\overset{(1)}{\leq} \sqrt{c^2 + \boldsymbol{\beta_k^2}\frac{\frac{1}{x_k^2} - \frac{c^2}{x_{k-1}^2}}{\frac{\beta_k^2}{x_{k-1}^2} + \frac{k-2}{d}c^{2k-6}(1-c)^2 + \frac{1}{d}c^{2k-6}} + \frac{\frac{1}{d}(1-c)^2}{\frac{\beta_k^2}{x_{k-1}^2} + \frac{k-2}{d}c^{2k-6}(1-c)^2 + \frac{1}{d}c^{2k-6}}}$$

$$\overset{(2)}{\leq} \sqrt{c^2 + \frac{\boldsymbol{1}}{\boldsymbol{c^2 d^2}}\frac{\frac{1}{x_k^2} - \frac{c^2}{x_{k-1}^2}}{\frac{\beta_k^2}{x_{k-1}^2} + \frac{k-2}{d}c^{2k-6}(1-c)^2 + \frac{1}{d}c^{2k-6}} + \frac{\frac{1}{d}\left(\frac{\boldsymbol{\ln(2)}}{\boldsymbol{d}}\right)^2}{\frac{\beta_k^2}{x_{k-1}^2} + \frac{k-2}{d}c^{2k-6}(1-c)^2 + \frac{1}{d}c^{2k-6}}}$$

$$\leq \sqrt{c^2 + \frac{1}{c^2 d^2}\frac{\frac{1}{x_k^2} - \frac{c^2}{x_{k-1}^2}}{\frac{1}{d}c^{2k-6}} + \frac{\frac{\ln^2(2)}{d^3}}{\frac{1}{d}c^{2k-6}}} = \sqrt{c^2 + \frac{x_{k-1}^2 - c^2 x_k^2}{x_{k-1}^2 x_k^2 c^{2k-4}d} + \frac{\ln^2(2)}{c^{2k-6}d^2}}$$

$$\overset{(3)}{\leq} \sqrt{c^2 + \frac{x_{k-1}^2 - c^2 x_k^2}{x_k^4 c^{2k-4}d} + \frac{\ln^2(2)}{c^{2k-6}d^2}} = \sqrt{c^2 + \frac{x_{k-1}^2 - c^2 x_{k-1}^2}{x_k^4 c^{2k-4}d} + \frac{c^2 x_{k-1}^2 - c^2 x_k^2}{x_k^4 c^{2k-4}d} + \frac{\ln^2(2)}{c^{2k-6}d^2}}$$

$$\overset{(4)}{\leq} \sqrt{c^2 + (1-c^2)\frac{1}{x_k^4 c^{2k-4}d} + c^2\frac{x_{k-1}^2 - x_k^2}{x_k^4 c^{2k-4}d} + \frac{\ln^2(2)}{c^{2k-6}d^2}},$$

where (1) is since $\beta_{k+1} < \beta_k$, $c < 1$; (2) is since $\beta_k < \frac{1}{cd}$, $1 - c \leq \frac{\ln 2}{d}$; (3) is since $x_k \leq x_{k-1}$; and (4) is since $x_{k-1} \leq 1$. To upper bound $x_{k-1}^2 - x_k^2$ we use the recursive formula of $x_k$, showing

that

$$x_{k-1}^2 - x_k^2 = x_{k-1}^2 - \frac{x_{k-1}^2 + 2x_{k-1}\sqrt{x_{k-1}^2 - 4\beta_k} + x_{k-1}^2 - 4\beta_k}{4}$$

$$= \frac{x_{k-1}^2}{2} - \frac{x_{k-1}^2\sqrt{1 - 4\frac{\beta_k}{x_{k-1}^2}} - 2\beta_k}{2}$$

$$\left[1 - z \le \sqrt{1-z}\right] \le \frac{x_{k-1}^2}{2} - \frac{x_{k-1}^2\left(1 - 4\frac{\beta_k}{x_{k-1}^2}\right) - 2\beta_k}{2} = \frac{4\beta_k + 2\beta_k}{2} = 3\beta_k \,.$$

Back to our expression,

$$\frac{h(k+1)}{h(k)} \le \sqrt{c^2 + (1-c^2)\frac{1}{x_k^4 c^{2k-4} d} + \frac{3\beta_k}{x_k^4 c^{2k-6} d} + \frac{\ln^2(2)}{c^{2k-6} d^2}}$$

$$\left[\beta_k \le \frac{1}{cd}\right] \le \sqrt{c^2 + (1-c^2)\frac{1}{x_k^4 c^{2k-4} d} + \frac{3}{x_k^4 c^{2k-5} d^2} + \frac{\ln^2(2)}{c^{2k-6} d^2}}$$

$$\left[1 - c^2 \le \frac{\ln(4)}{d}\right] \le \sqrt{c^2 + \frac{\ln(4)}{x_k^4 c^{2k-4} d^2} + \frac{3}{x_k^4 c^{2k-5} d^2} + \frac{\ln^2(2)}{c^{2k-6} d^2}}$$

$$[c < 1] \le \sqrt{c^2 + \frac{\ln(4)}{x_k^4 c^{2k-4} d^2} + \frac{3}{x_k^4 \boldsymbol{c^{2k-4}} d^2} + \frac{\ln^2(2)}{\boldsymbol{c^{2k-4}} d^2}} \le \sqrt{c^2 + \frac{4.39 + 0.49 x_k^4}{x_k^4 c^{2k-6} d^2}}$$

$$[x_k \le 1] \le \sqrt{c^2 + \frac{4.88}{x_k^4 c^{2k-4} d^2}} = c\sqrt{1 + \frac{4.88}{x_k^4 c^{2k-2} d^2}} \le c + \frac{2.44}{x_k^4 c^{2k-3} d^2} \,,$$

where in the last inequality we used the fact that $\forall z > 0$, $\sqrt{1+z} \le 1 + \frac{z}{2}$ (since $\left(1 + \frac{z}{2}\right)^2 = 1 + z + \frac{z^2}{4} \ge 1 + z = \left(\sqrt{1+z}\right)^2$).

**For the lower bound.** We show that,

$$\frac{h(k+1)}{h(k)}$$

$$= \sqrt{c^2 + \frac{\frac{\beta_{k+1}^2}{x_k^2} - \frac{c^2\beta_k^2}{x_{k-1}^2}}{\frac{\beta_k^2}{x_{k-1}^2} + \frac{k-2}{d}c^{2k-6}(1-c)^2 + \frac{1}{d}c^{2k-6}} + \frac{\frac{1}{d}c^{2k-4}(1-c)^2}{\frac{\beta_k^2}{x_{k-1}^2} + \frac{k-2}{d}c^{2k-6}(1-c)^2 + \frac{1}{d}c^{2k-6}}}$$

$$\ge \sqrt{c^2 + \frac{\frac{\beta_{k+1}^2}{x_k^2} - \frac{c^2\beta_k^2}{x_{k-1}^2}}{\frac{\beta_k^2}{x_{k-1}^2} + \frac{k-2}{d}c^{2k-6}(1-c)^2 + \frac{1}{d}c^{2k-6}}}$$

$$\ge c\sqrt{1 + \frac{\frac{\beta_{k+1}^2}{x_k^2} - \frac{\beta_k^2}{x_{k-1}^2}}{\frac{\beta_k^2}{x_{k-1}^2} + \frac{k-2}{d}c^{2k-6}(1-c)^2 + \frac{1}{d}c^{2k-6}}}$$

$$\overset{(1)}{\ge} c\sqrt{1 + \frac{1}{x_{k-1}^2}\frac{\beta_{k+1}^2 - \beta_k^2}{\frac{\beta_k^2}{x_{k-1}^2} + \frac{k-2}{d}c^{2k-6}(1-c)^2 + \frac{1}{d}c^{2k-6}}} \,,$$

where (1) is since $x_k \le x_{k-1}$. Since $(\beta_k)_k$ is positive and decreasing, $\beta_{k+1}^2 - \beta_k^2 < 0$, and so we can simplify the expression using the fact that $\sqrt{1-z} \ge 1 - z$, $\forall z \in (0,1)$:

$$\frac{h(k+1)}{h(k)} \ge c\sqrt{1 - \frac{|\beta_k^2 - \beta_{k+1}^2|}{\frac{1}{d}c^{2k-6}x_{k-1}^2}} \ge c - \frac{|\beta_k^2 - \beta_{k+1}^2|}{\frac{1}{d}c^{2k-5}x_{k-1}^2} \,.$$

Focusing on $\frac{\left|\beta_{k+1}^2 - \beta_k^2\right|}{\frac{1}{d}c^{2k-5}}$, and since $1 - (1-c)(k-1) = ((k-1)c - (k-2))$,

$$\frac{\left|\beta_{k+1}^2 - \beta_k^2\right|}{\frac{1}{d}c^{2k-5}} = \frac{\beta_k^2 - \beta_{k+1}^2}{\frac{1}{d}c^{2k-5}} = \frac{\left(\frac{((k-1)c-(k-2))c^{2k-5}}{d}\right)^2 - \left(\frac{(kc-(k-1))c^{2k-3}}{d}\right)^2}{\frac{1}{d}c^{2k-5}}$$

$$= \frac{c^{2k-5}}{d}\left((1-(1-c)(k-1))^2 - (1-(1-c)k)^2 c^4\right)$$

$$= \frac{c^{2k-5}}{d}\left((1-c^4) - 2k\underbrace{(1-c)(1-c^4)}_{\geq 0} + k^2(1-c)^2(1-c^4) + \right.$$

$$\left. + 2(1-c) + \underbrace{(1-c)^2}_{\geq 0}(-2k+1)\right)$$

$$\leq \frac{c^{2k-5}}{d}\left((1-c^4) + k^2(1-c)^2(1-c^4) + 2(1-c) + (1-c)^2\right).$$

Using the previously derived bounds of $1 - c \in \left[\frac{\ln(2)}{d} - \frac{\ln^2(2)}{2d^2}, \frac{\ln(2)}{d}\right]$, we can get,

$$\frac{\left|\beta_{k+1}^2 - \beta_k^2\right|}{\frac{1}{d}c^{2k-5}} \leq \frac{c^{2k-5}}{d}\left((1-c^4) + k^2\frac{\ln^2(2)}{d^2}(1-c^4) + 2\frac{\ln(2)}{d} + \frac{\ln^2(2)}{d^2}\right)$$

$$[d \geq 500 \geq 1000\ln^2(2)] \leq \frac{c^{2k-5}}{d}\left((1-c^4) + k^2\frac{\ln^2(2)}{d^2}(1-c^4) + 2\frac{\ln(2)}{d} + \frac{0.001}{d}\right)$$

$$[k \leq d] \leq \frac{c^{2k-5}}{d}\left((1-c^4) + \ln^2(2)(1-c^4) + \frac{\ln(4) + 0.001}{d}\right)$$

$$\leq \frac{c^{2k-5}}{d}\left((1+\ln^2(2))(1-c^4) + \frac{\ln(4) + 0.001}{d}\right).$$

Notice that we can use the previously derived bound of $1 - c^n \leq \frac{n\ln(2)}{d}$, thus obtaining

$$\frac{\left|\beta_{k+1}^2 - \beta_k^2\right|}{\frac{1}{d}c^{2k-5}} \leq \frac{c^{2k-5}}{d}\left((1+\ln^2(2))\frac{4\ln(2)}{d} + \frac{\ln(4) + 0.001}{d}\right) \leq 5.5\frac{c^{2k-5}}{d^2}.$$

Finally, we get,

$$\frac{h(k+1)}{h(k)} \geq c - \frac{\left|\beta_k^2 - \beta_{k+1}^2\right|}{\frac{1}{d}c^{2k-5}x_{k-1}^2} \geq c - \frac{5.5c^{2k-5}}{x_{k-1}^2 d^2}.$$

$\square$

### I.3.5 Expanding the norm

**Proposition I.17.** *For any $k \geq 2$, $\|\mathbf{w}_{k-1} - \mathbf{w}_k\| \in \left[ h(k), h(k) + \frac{56.5 c^{6k-15}}{c^{k-3} d^{5/2}} \right]$, where $h(k) \triangleq$*
$$\sqrt{\frac{\beta_k^2}{x_{k-1}^2} + \frac{k-2}{d} c^{2k-6} (1-c)^2 + \frac{1}{d} c^{2k-6}}.$$

*Proof.* By construction we have

$$\|\mathbf{w}_{k-1} - \mathbf{w}_k\| = \sqrt{(x_{k-1} - x_k)^2 + (k-2) \left( \frac{c^{k-3}(1-c)}{\sqrt{d}} \right)^2 + \left( \frac{c^{k-3}}{\sqrt{d}} \right)^2}$$

$$= \sqrt{(x_{k-1} - x_k)^2 + \frac{k-2}{d} c^{2k-6} (1-c)^2 + \frac{1}{d} c^{2k-6}}.$$

Before, we proved that $(x_{k-1} - x_k)^2 \in \left[ \frac{\beta_k^2}{x_{k-1}^2}, \frac{\beta_k^2}{x_{k-1}^2} + \frac{113 c^{6k-15}}{d^3} \right]$. Now, we show the resulting bounds for $\|\mathbf{w}_{k-1} - \mathbf{w}_k\|$ which employ that bound.

The lower bound is immediate, since

$$\|\mathbf{w}_{k-1} - \mathbf{w}_k\| = \sqrt{(x_{k-1} - x_k)^2 + \frac{k-2}{d} c^{2k-6} (1-c)^2 + \frac{1}{d} c^{2k-6}}$$

$$\geq \sqrt{\frac{\beta_k^2}{x_{k-1}^2} + \frac{k-2}{d} c^{2k-6} (1-c)^2 + \frac{1}{d} c^{2k-6}} \triangleq h(k).$$

The upper bound requires an additional algebraic inequality of $\forall a, b > 0 : \sqrt{a+b} < \sqrt{a} + \frac{b}{2\sqrt{a}}$ and the inequality of $h(k) \geq \frac{1}{\sqrt{d}} c^{k-3}$, i.e.,

$$\|\mathbf{w}_{k-1} - \mathbf{w}_k\| \leq \sqrt{\frac{\beta_k^2}{x_{k-1}^2} + \frac{k-2}{d} c^{2k-6} (1-c)^2 + \frac{1}{d} c^{2k-6} + \frac{113 c^{6k-15}}{d^3}}$$

$$= \sqrt{h^2(k) + \frac{113 c^{6k-15}}{d^3}} \leq h(k) + \frac{113 c^{6k-15}}{2h(k) d^3} \leq h(k) + \frac{56.5 c^{6k-15}}{c^{k-3} d^{5/2}}.$$

$\square$

### I.3.6 Combining the expansions

**Proposition I.18.** *When $d \geq 25{,}000$,*

$$\|\mathbf{w}_k - \mathbf{w}_{k+1}\| \left(\mathbf{w}_{k-1} - \mathbf{w}_k\right)^\top \mathbf{w}_{t-1} - \|\mathbf{w}_{k-1} - \mathbf{w}_k\| \left(\mathbf{w}_k - \mathbf{w}_{k+1}\right)^\top \mathbf{w}_{t-1}$$
$$\geq A_1\left(k\right) + A_2\left(k\right) + A_3\left(k\right)$$
$$- \frac{1}{d^{7/2}} \left(\frac{96 c^{6k-15}}{x_k} + 113 c^{7k-18}\right) - \frac{1}{d^{9/2}} \left(\frac{56.5 c^{9k-18}}{x_k^3} + 614 c^{6k-30}\right),$$

*where* $A_1\left(k\right) \triangleq x_{t-1} \left(\frac{h(k+1)}{x_{k-1}} \beta_k - \frac{h(k)}{x_k} \beta_{k+1}\right)$, $A_2\left(k\right) \triangleq x_{t-1} \left(\frac{h(k+1)}{x_{k-1}^3} \beta_k^2 - \frac{h(k)}{x_k^3} \beta_{k+1}^2\right)$, $A_3\left(k\right) \triangleq \frac{t-2}{d}\left(1-c\right) c^{k+t-6}\left(h\left(k+1\right) - ch\left(k\right)\right)$.

*Proof.* Keeping in mind that we wish to bound

$$\|\mathbf{w}_k - \mathbf{w}_{k+1}\| \left(\mathbf{w}_{k-1} - \mathbf{w}_k\right)^\top \mathbf{w}_{t-1} - \|\mathbf{w}_{k-1} - \mathbf{w}_k\| \left(\mathbf{w}_k - \mathbf{w}_{k+1}\right)^\top \mathbf{w}_{t-1},$$

we start lower bounding the right expression. Using the bounds for $\left(\mathbf{w}_{k-1} - \mathbf{w}_k\right)^\top \mathbf{w}_{t-1}$ and $\|\mathbf{w}_{k-1} - \mathbf{w}_k\|$ we derived above, we get,

$$- \|\mathbf{w}_{k-1} - \mathbf{w}_k\| \left(\mathbf{w}_k - \mathbf{w}_{k+1}\right)^\top \mathbf{w}_{t-1}$$
$$\geq - \|\mathbf{w}_{k-1} - \mathbf{w}_k\| \left(\frac{x_{t-1}}{x_k} \beta_{k+1} + \frac{x_{t-1}}{x_k^3} \beta_{k+1}^2 + \frac{\left(t-2\right) c^{k+t-5}\left(1-c\right)}{d} + \frac{113 c^{6k-15}}{d^3}\right)$$
$$\geq - \|\mathbf{w}_{k-1} - \mathbf{w}_k\| \left(\frac{x_{t-1}}{x_k} \beta_{k+1} + \frac{x_{t-1}}{x_k^3} \beta_{k+1}^2 + \frac{\left(t-2\right) c^{k+t-5}\left(1-c\right)}{d}\right)$$
$$\quad - \|\mathbf{w}_{k-1} - \mathbf{w}_k\| \frac{113 c^{6k-15}}{d^3}$$
$$\geq \underbrace{- \left(h\left(k\right) + \frac{56.5 c^{6k-15}}{c^{k-3} d^{5/2}}\right) \left(\frac{x_{t-1}}{x_k} \beta_{k+1} + \frac{x_{t-1}}{x_k^3} \beta_{k+1}^2 + \frac{t-2}{d} c^{k+t-5}\left(1-c\right)\right)}_{\triangleq a(k)}$$
$$\underbrace{- \|\mathbf{w}_{k-1} - \mathbf{w}_k\| \frac{113 c^{6k-15}}{d^3}}_{\triangleq b(k)}.$$

The right function is easily bounded as,

$$b\left(k\right) = - \|\mathbf{w}_{k-1} - \mathbf{w}_k\| \frac{113 c^{6k-15}}{d^3}$$
$$\geq - \left(h\left(k\right) + \frac{56.5 c^{6k-15}}{c^{k-3} d^{5/2}}\right) \frac{113 c^{6k-15}}{d^3}$$
$$= - \frac{113 c^{6k-15}}{d^3} h\left(k\right) - \frac{6384.5 c^{12k-30}}{d^{11/2} c^{k-3}}$$
$$\geq - \frac{113 c^{6k-15}}{d^3} \left(\frac{\sqrt{c^{2k-6}}}{\sqrt{d}} + \frac{5.42}{d^{3/2}}\right) - \frac{6384.5 c^{12k-30}}{d^{11/2} c^{k-3}}$$
$$= - \frac{113 c^{6k-15} c^{k-3}}{d^{7/2}} - \frac{612.46 c^{6k-15}}{d^{9/2}} - \frac{6384.5 c^{12k-30}}{d^{11/2} c^{k-3}}$$
$$\left[c^{k-3} \geq c^d = 0.5\right] \geq - \frac{113 c^{7k-18}}{d^{7/2}} - \frac{612.46 c^{6k-15}}{d^{9/2}} - \frac{12769 c^{12k-30}}{d^{11/2}}.$$

The left function is further decomposed as,

$$a\left(k\right) = \underbrace{-h\left(k\right)\left(\frac{x_{t-1}}{x_k}\beta_{k+1} + \frac{x_{t-1}}{x_k^3}\beta_{k+1}^2 + \frac{t-2}{d}c^{k+t-5}\left(1-c\right)\right)}_{\triangleq a_1(k)}$$

$$\underbrace{-\frac{56.5c^{5k-12}}{d^{5/2}}\left(\frac{x_{t-1}}{x_k}\beta_{k+1} + \frac{x_{t-1}}{x_k^3}\beta_{k+1}^2 + \frac{t-2}{d}c^{k+t-5}\left(1-c\right)\right)}_{\triangleq a_2(k)}.$$

Then,

$$a_2\left(k\right) = -\frac{56.5c^{5k-12}}{d^{5/2}}\left(\frac{x_{t-1}}{x_k}\beta_{k+1} + \frac{x_{t-1}}{x_k^3}\beta_{k+1}^2 + \frac{t-2}{d}c^{k+t-5}\left(1-c\right)\right)$$

$$\left[x_{t-1} \leq 1, \frac{t-2}{d} < 1\right] \geq -\frac{56.5c^{5k-12}}{d^{5/2}}\left(\frac{1}{x_k}\beta_{k+1} + \frac{1}{x_k^3}\beta_{k+1}^2 + c^{k+t-5}\left(1-c\right)\right)$$

$$\left[\beta_{k+1} \leq \frac{c^{2k-3}}{d}\right] \geq -\frac{56.5c^{5k-12}}{d^{5/2}}\left(\frac{c^{2k-3}}{x_kd} + \frac{c^{4k-6}}{x_k^3d^2} + c^{k+t-5}\left(1-c\right)\right)$$

$$\geq -\frac{56.5c^{5k-12}}{d^{5/2}}\left(\frac{c^{2k-3}}{x_kd} + \frac{c^{4k-6}}{x_k^3d^2} + \frac{\ln\left(2\right)}{d}c^{k+t-5}\right)$$

$$\left[x_k \leq 1\right] \geq -\frac{56.5c^{5k-12}}{d^{5/2}}\left(\frac{c^{2k-3} + \ln\left(2\right)c^{k+t-5}}{x_kd} + \frac{c^{4k-6}}{x_k^3d^2}\right)$$

$$= -\frac{56.5c^{6k-15}}{d^{5/2}\cdot x_kd}\left(c^k + \ln\left(2\right)c^{t-2} + \frac{c^{3k-3}}{x_k^2d}\right)$$

$$\left[c < 1\right] \geq -\frac{56.5c^{6k-15}}{d^{7/2}\cdot x_k}\left(1 + \ln\left(2\right) + \frac{c^{3k-3}}{x_k^2d}\right)$$

$$\geq -\frac{96c^{6k-15}}{d^{7/2}\cdot x_k} - \frac{56.5c^{9k-18}}{d^{9/2}\cdot x_k^3}.$$

Overall we got,

$$\|\mathbf{w}_k - \mathbf{w}_{k+1}\| (\mathbf{w}_{k-1} - \mathbf{w}_k)^\top \mathbf{w}_{t-1} - \|\mathbf{w}_{k-1} - \mathbf{w}_k\| (\mathbf{w}_k - \mathbf{w}_{k+1})^\top \mathbf{w}_{t-1},$$

$$\geq \|\mathbf{w}_k - \mathbf{w}_{k+1}\| (\mathbf{w}_{k-1} - \mathbf{w}_k)^\top \mathbf{w}_{t-1} + a_1(k) + a_2(k) + b(k)$$

$$\geq \|\mathbf{w}_k - \mathbf{w}_{k+1}\| (\mathbf{w}_{k-1} - \mathbf{w}_k)^\top \mathbf{w}_{t-1} + a_1(k)$$
$$- \frac{96c^{6k-15}}{d^{7/2} \cdot x_k} - \frac{56.5c^{9k-18}}{d^{9/2} \cdot x_k^3} - \frac{113c^{7k-18}}{d^{7/2}} - \frac{612.46c^{6k-15}}{d^{9/2}} - \frac{12769c^{12k-30}}{d^{11/2}}$$

$$\geq \|\mathbf{w}_k - \mathbf{w}_{k+1}\| (\mathbf{w}_{k-1} - \mathbf{w}_k)^\top \mathbf{w}_{t-1} + a_1(k)$$
$$- \frac{96c^{6k-15}}{d^{7/2} \cdot x_k} - \frac{56.5c^{9k-18}}{d^{9/2} \cdot x_k^3} - \frac{113c^{7k-18}}{d^{7/2}} - \frac{612.46c^{6k-30}}{d^{9/2}} - \frac{12769c^{6k-30}}{d^{11/2}}$$

$$\overset{(1)}{\geq} \|\mathbf{w}_k - \mathbf{w}_{k+1}\| (\mathbf{w}_{k-1} - \mathbf{w}_k)^\top \mathbf{w}_{t-1}$$
$$+ a_1(k) - \frac{96c^{6k-15}}{d^{7/2} \cdot x_k} - \frac{56.5c^{9k-18}}{d^{9/2} \cdot x_k^3} - \frac{113c^{7k-18}}{d^{7/2}} - \frac{614c^{6k-30}}{d^{9/2}}$$

$$\geq \|\mathbf{w}_k - \mathbf{w}_{k+1}\| (\mathbf{w}_{k-1} - \mathbf{w}_k)^\top \mathbf{w}_{t-1}$$
$$+ a_1(k) - \frac{1}{d^{7/2}}\left(\frac{96c^{6k-15}}{x_k} + 113c^{7k-18}\right) - \frac{1}{d^{9/2}}\left(\frac{56.5c^{9k-18}}{x_k^3} + 614c^{6k-30}\right),$$

where (1) is since $d \geq 10{,}000$. Focusing on the left terms, we get the **overall** expression, which we need to show is **positive**. We again use previously-derived inequalities, to show,

$$\|\mathbf{w}_k - \mathbf{w}_{k+1}\| (\mathbf{w}_{k-1} - \mathbf{w}_k)^\top \mathbf{w}_{t-1} + a_1(k)$$

$$= \|\mathbf{w}_k - \mathbf{w}_{k+1}\| (\mathbf{w}_{k-1} - \mathbf{w}_k)^\top \mathbf{w}_{t-1}$$
$$- h(k)\left(\frac{x_{t-1}}{x_k}\beta_{k+1} + \frac{x_{t-1}}{x_k^3}\beta_{k+1}^2 + \frac{t-2}{d}c^{k+t-5}(1-c)\right)$$

$$\geq h(k+1)\left(\frac{x_{t-1}}{x_{k-1}}\beta_k + \frac{x_{t-1}}{x_{k-1}^3}\beta_k^2 + \frac{t-2}{d}c^{k+t-6}(1-c)\right)$$
$$- h(k)\left(\frac{x_{t-1}}{x_k}\beta_{k+1} + \frac{x_{t-1}}{x_k^3}\beta_{k+1}^2 + \frac{t-2}{d}c^{k+t-5}(1-c)\right)$$

$$= \underbrace{x_{t-1}\left(\frac{h(k+1)}{x_{k-1}}\beta_k - \frac{h(k)}{x_k}\beta_{k+1}\right)}_{\triangleq A_1(k)} + \underbrace{x_{t-1}\left(\frac{h(k+1)}{x_{k-1}^3}\beta_k^2 - \frac{h(k)}{x_k^3}\beta_{k+1}^2\right)}_{\triangleq A_2(k)}$$

$$+ \underbrace{\frac{t-2}{d}(1-c)c^{k+t-6}(h(k+1) - ch(k))}_{\triangleq A_3(k)},$$

which we will bound separately below. $\qquad\square$

### I.3.7 The second term, $A_2(k)$, is insignificant $\mathcal{O}\left(\frac{1}{d^{7/2}}\right)$

**Proposition I.19.** *When $d \geq 25{,}000$,*

$$A_2(k) = x_{t-1}\left(\frac{h(k+1)}{x_{k-1}^3}\beta_k^2 - \frac{h(k)}{x_k^3}\beta_{k+1}^2\right) \geq -\frac{14.88 x_{t-1} c^{3k-12}}{x_k^3 d^{7/2}} \,.$$

*Proof.* We start from,

$$A_2(k) = x_{t-1}\left(\frac{h(k+1)}{x_{k-1}^3}\beta_k^2 - \frac{h(k)}{x_k^3}\beta_{k+1}^2\right) = \frac{x_{t-1}}{x_k^3}h(k)\beta_{k+1}^2\underbrace{\left(\frac{h(k+1)}{h(k)}\frac{x_k^3}{x_{k-1}^3}\frac{\beta_k^2}{\beta_{k+1}^2}-1\right)}_{\triangleq a(k)}.$$

Dissecting the terms in $a(k)$,

$$\frac{\beta_k}{\beta_{k+1}} = \frac{\frac{((k-1)c-(k-2))c^{2k-5}}{d}}{\frac{(kc-(k-1))c^{2k-3}}{d}} = \frac{1}{c^2} + \underbrace{\frac{1-c}{(kc-(k-1))c^2}}_{>0,\ \text{from I.4}} \geq \frac{1}{c^2}\,,$$

$$\frac{\beta_k^2}{\beta_{k+1}^2} \geq \frac{1}{c^4}\,.$$

We already showed that $\frac{x_k}{x_{k-1}} \in \left(1-\frac{\beta_k}{x_{k-1}^2}-\left(1+\frac{10}{d}\right)\frac{\beta_k^2}{x_{k-1}^4},\ 1-\frac{\beta_k}{x_{k-1}^2}\right)$, and we simplify it further:

$$\frac{x_k}{x_{k-1}} \geq 1 - \frac{\beta_k}{x_{k-1}^2} - \left(1+\frac{10}{d}\right)\frac{\beta_k^2}{x_{k-1}^4} \overset{\beta_k \leq \frac{1}{d}}{\geq} 1 - \frac{\beta_k}{x_{k-1}^2} - \left(1+\frac{10}{d}\right)\frac{\beta_k}{x_{k-1}^4 d}$$

$$[x_{k-1} \geq 0.45] \geq 1 - \beta_k\left(\frac{1}{(0.45)^2} + \frac{\left(1+\frac{10}{d}\right)}{(0.45)^4 d}\right) \overset{d \geq 10{,}000}{\geq} 1 - 4.95\beta_k \,.$$

Now, using the algebraic inequality $\forall z \in (0,1),\ (1-z)^3 = 1-3z+3z^2-z^3 > 1-3z$, we get,

$$\frac{x_k^3}{x_{k-1}^3} > 1 - 14.85\beta_k \,.$$

Moreover, recall that we already showed that $\frac{h(k+1)}{h(k)} \geq c - \frac{5.5 c^{2k-5}}{x_{k-1}^2 d^2}$. Now, focusing on $a(k)$,

$$\begin{aligned}
a(k) &= \frac{h(k+1)}{h(k)} \cdot \frac{x_k^3}{x_{k-1}^3}\frac{\beta_k^2}{\beta_{k+1}^2} - 1 \\
&\geq \left(c - \frac{5.5 c^{2k-5}}{x_{k-1}^2 d^2}\right)(1-14.85\beta_k)\frac{1}{c^4} - 1 \\
&= \left(\frac{1}{c^3} - \frac{5.5 c^{2k-5}}{x_{k-1}^2 d^2}\right)(1-14.85\beta_k) - 1 \\
&\geq \left(1 - \frac{5.5 c^{2k-5}}{x_{k-1}^2 d^2}\right)(1-14.85\beta_k) - 1 \\
&= -14.85\beta_k - \frac{5.5 c^{2k-5}}{x_{k-1}^2 d^2} + \frac{81.675 c^{2k-9}}{x_{k-1}^2 d^2}\beta_k \\
&\geq -14.85\beta_k - \frac{5.5 c^{2k-9}}{x_{k-1}^2 d^2} \\
\left[\beta_k \leq \frac{c^{2k-5}}{d}\right] &\geq -\frac{14.85 c^{2k-5}}{d} - \frac{5.5 c^{2k-9}}{x_{k-1}^2 d^2} \geq -\frac{14.85 c^{2k-5}}{d} - \frac{5.5 c^{2k-9}}{(0.45)^2 d^2} \\
&\geq -\frac{14.85 c^{2k-5}}{d} - \frac{27.17 c^{2k-9}}{d^2} \geq -\frac{14.85 c^{2k-9}}{d} - \frac{27.17 c^{2k-9}}{d^2} \\
[d \geq 10{,}000] &\geq -\frac{14.86 c^{2k-9}}{d} \,.
\end{aligned}$$

And finally,

$$\frac{1}{x_{t-1}}A_2(k) = \frac{1}{x_k^3}h(k)\beta_{k+1}^2 \cdot a(k) \geq -\frac{1}{x_k^3}h(k)\beta_{k+1}^2 \cdot \frac{14.86c^{2k-9}}{d}$$

$$\left[\begin{matrix}\beta_{k+1}\leq\frac{1}{d},\\ h(k)\leq\frac{c^{k-3}}{\sqrt{d}}+\frac{5.42}{d^{3/2}}\end{matrix}\right] \geq -\frac{1}{x_k^3}\frac{\frac{c^{k-3}}{\sqrt{d}}+\frac{5.42}{d^{3/2}}}{d^2}\frac{14.86c^{2k-9}}{d} = -\frac{1}{x_k^3}\frac{c^{3k-12}}{d^{5/2}}\frac{14.86}{d} - \frac{1}{x_k^3}\frac{5.42}{d^{7/2}}\frac{14.86c^{2k-9}}{d}$$

$$[c<1] \geq -\frac{14.86c^{3k-12}}{x_k^3 d^{7/2}} - \frac{80.55c^{2k-12}}{x_k^3 d^{9/2}}$$

$$\left[\begin{matrix}d\geq 10{,}000,\\ c^{k-3}\geq c^d=0.5\end{matrix}\right] \geq -\frac{14.86c^{3k-12}}{x_k^3 d^{7/2}} - \frac{0.0081\cdot c^{3k-12}}{x_k^3 d^{7/2}c} \geq -\frac{\left(14.86+\frac{0.0081}{0.5}\right)c^{3k-12}}{x_k^3 d^{7/2}}$$

$$A_2(k) \geq -\frac{14.88x_{t-1}c^{3k-12}}{x_k^3 d^{7/2}},$$

thus concluding this part. $\qquad\square$

### I.3.8   The third term, $A_3(k)$, is insignificant $\mathcal{O}\left(\frac{1}{d^{7/2}}\right)$

**Proposition I.20.** *When $d \geq 25{,}000$,*

$$|A_3(k)| = \left|\frac{t-2}{d}(1-c)c^{k+t-6}(h(k+1)-ch(k))\right| \leq \frac{6.77c^{k-6}}{x_k^4}\left(\frac{c^{k-3}}{d^{7/2}}+\frac{5.42}{d^{9/2}}\right).$$

*Proof.* Notice that,

$$|A_3(k)| = \left|\frac{t-2}{d}(1-c)c^{k+t-6}(h(k+1)-ch(k))\right|$$

$$= \frac{t-2}{d}(1-c)c^{k+t-6}|h(k+1)-ch(k)| \leq (1-c)c^{k+t-6}|h(k+1)-ch(k)|$$

$$\leq \ln(2)c^{k+t-6}\frac{h(k)}{d}\left|\frac{h(k+1)}{h(k)}-c\right| \leq \ln(2)c^{k+t-6}\left(\frac{c^{k-3}}{d^{3/2}}+\frac{5.42}{d^{5/2}}\right)\left|\frac{h(k+1)}{h(k)}-c\right|,$$

where we used the facts that $1-c \leq \frac{\ln 2}{d}$ and $h(k) \leq \frac{c^{k-3}}{\sqrt{d}}+\frac{5.42}{d^{3/2}}$.

Using $\frac{h(k+1)}{h(k)} \in \left[c-\frac{5.5c^{2k-5}}{x_{k-1}^2 d^2},\ c+\frac{2.44}{x_k^4 c^{2k-3}d^2}\right]$, we finally get,

$$|A_3(k)| \leq \ln(2)c^{k+t-6}\left(\frac{c^{k-3}}{d^{3/2}}+\frac{5.42}{d^{5/2}}\right)\left|\frac{h(k+1)}{h(k)}-c\right|$$

$$\leq \ln(2)c^{k+t-6}\left(\frac{c^{k-3}}{d^{3/2}}+\frac{5.42}{d^{5/2}}\right)\max\left(\frac{5.5c^{2k-5}}{x_{k-1}^2 d^2},\frac{2.44}{x_k^4 c^{2k-3}d^2}\right)$$

$$[x_k<x_{k-1}] \leq \ln(2)c^{k+t-6}\left(\frac{c^{k-3}}{d^{7/2}}+\frac{5.42}{d^{9/2}}\right)\max\left(\frac{5.5c^{2k-5}}{x_k^4},\frac{2.44}{x_k^4 c^{2k-3}}\right)$$

$$[c<1] \leq \frac{\ln(2)c^{k+t-6}}{x_k^4}\left(\frac{c^{k-3}}{d^{7/2}}+\frac{5.42}{d^{9/2}}\right)\max\left(5.5c^{2k-5},\frac{2.44}{c^{2k}}\right)$$

$$[c^{nk-m}\geq 2^{-n},\ k\geq 2,\ c<1] \leq \frac{\ln(2)c^{k+t-6}}{x_k^4}\left(\frac{1}{cd^{7/2}}+\frac{5.42}{d^{9/2}}\right)\max\left(\frac{5.5}{c},9.76\right)$$

$$\left[c\geq 2^{-1/10000}\geq 0.9999\right] \leq \frac{\ln(2)c^{k+t-6}}{x_k^4}\left(\frac{1}{0.9999d^{7/2}}+\frac{5.42}{d^{9/2}}\right)\max\left(\frac{5.5}{0.9999},9.76\right)$$

$$\leq \frac{6.77c^{k+t-6}}{x_k^4}\left(\frac{1}{d^{7/2}}+\frac{5.42}{d^{9/2}}\right).$$

$\qquad\square$

### I.3.9 Back to the first term, $A_1(k)$

**Proposition I.21.** *When* $d \geq 25{,}000$,

$$A_1(k) = x_{t-1}\left(\frac{h(k+1)}{x_{k-1}}\beta_k - \frac{h(k)}{x_k}\beta_{k+1}\right) \geq x_{t-1}\left(\frac{0.0429c^{3k-6}}{x_k^3 d^{5/2}} - \frac{173.07c^{3k-9}}{x_k^2 d^{7/2}}\right).$$

*Proof.* We have

$$A_1(k) = x_{t-1}\left(\frac{h(k+1)}{x_{k-1}}\beta_k - \frac{h(k)}{x_k}\beta_{k+1}\right) = \underbrace{\frac{x_{t-1}}{x_k}h(k)\beta_{k+1}}_{=\Theta(d^{-3/2})}\underbrace{\left(\frac{x_k}{x_{k-1}}\frac{h(k+1)}{h(k)}\frac{\beta_k}{\beta_{k+1}} - 1\right)}_{\triangleq a(k)}.$$

We are going to use the previously-derived lower bounds of $\frac{x_k}{x_{k-1}} \geq 1 - \frac{\beta_k}{x_{k-1}^2} - \left(1 + \frac{10}{d}\right)\frac{\beta_k^2}{x_{k-1}^4}$ and $\frac{h(k+1)}{h(k)} \geq c - \frac{5.5c^{2k-5}}{x_{k-1}^2 d^2}$. To lower bound $\frac{\beta_k}{\beta_{k+1}} = \frac{1}{c^2} + \frac{1-c}{(1-k(1-c))c^2}$, we need a slightly stronger bound than before. Specifically, notice that for any $z \in (0,1)$, $\frac{z}{1-z} \geq z$. Then, since $1 - c \in \left[\frac{\ln(2)}{d} - \frac{\ln^2(2)}{2d^2}, \frac{\ln(2)}{d}\right] \implies k(1-c) \in \left[\frac{k}{d}\ln(2) - \frac{k}{2d^2}\ln^2(2), \frac{k}{d}\ln(2)\right] \subseteq (0,1)$, and

$$\frac{1-c}{(1-k(1-c))c^2} = \frac{1}{c^2 k}\frac{k(1-c)}{(1-k(1-c))} \geq \frac{1}{c^2 k}k(1-c) = \frac{1-c}{c^2}.$$

We now get,

$$\frac{\beta_k}{\beta_{k+1}} \geq \frac{1}{c^2} + \frac{1-c}{c^2} = \frac{2-c}{c^2}.$$

We are now ready to lower bound $a(k)$ as,

$$\begin{aligned}
a(k) &= \frac{x_k}{x_{k-1}}\frac{h(k+1)}{h(k)}\frac{\beta_k}{\beta_{k+1}} - 1 \\
&\geq \left(1 - \frac{\beta_k}{x_{k-1}^2} - \left(1 + \frac{10}{d}\right)\frac{\beta_k^2}{x_{k-1}^4}\right)\left(c - \frac{5.5c^{2k-5}}{x_{k-1}^2 d^2}\right)\left(\frac{2-c}{c^2}\right) - 1 \\
&= \left(1 - \frac{\beta_k}{x_{k-1}^2} - \left(1 + \frac{10}{d}\right)\frac{\beta_k^2}{x_{k-1}^4}\right)\left(1 - \frac{5.5c^{2k-6}}{x_{k-1}^2 d^2}\right)\left(\frac{2-c}{c}\right) - 1.
\end{aligned}$$

Using Claim I.9: $\frac{2-c}{c} = \frac{2}{c} - 1 = 2 \cdot 2^{1/d} - 1 \geq 2\left(1 + \frac{\ln(2)}{d}\right) - 1 = 1 + \frac{\ln(4)}{d}$, we get:

$$\begin{aligned}
a(k) &\geq \left(1 - \frac{\beta_k}{x_{k-1}^2} - \left(1 + \frac{10}{d}\right)\frac{\beta_k^2}{x_{k-1}^4}\right)\left(1 - \frac{5.5c^{2k-6}}{x_{k-1}^2 d^2}\right)\left(1 + \frac{\ln(4)}{d}\right) - 1 \\
&= \underbrace{\frac{\ln(4)}{d} - \frac{\beta_k}{x_{k-1}^2}}_{\mathcal{O}(d^{-1})} - \underbrace{\left(\frac{5.5c^{2k-6}}{x_{k-1}^2 d^2} + \frac{\left(1 + \frac{10}{d}\right)\beta_k^2}{x_{k-1}^4} + \frac{\ln(4)\beta_k}{x_{k-1}^2 d}\right)}_{\Theta(d^{-2})} \\
&\quad + \underbrace{\frac{5.5c^{2k-6}\beta_k}{x_{k-1}^4 d^2} - \frac{5.5\ln(4)c^{2k-6}}{x_{k-1}^2 d^3} - \frac{\left(1 + \frac{10}{d}\right)\ln(4)\beta_k^2}{x_{k-1}^4 d}}_{\mathcal{O}(d^{-3})} \\
&\quad + \underbrace{\frac{5.5\left(1 + \frac{10}{d}\right)c^{2k-6}\beta_k^2}{x_{k-1}^6 d^2} + \frac{5.5\ln(4)c^{2k-6}}{x_{k-1}^2 d^3}\left(\frac{\beta_k}{x_{k-1}^2} + \frac{\left(1 + \frac{10}{d}\right)\beta_k^2}{x_{k-1}^4}\right)}_{\mathcal{O}(d^{-4})}.
\end{aligned}$$

Lower bounding negligible positive terms by $0$, we get,

$$a(k) \geq \underbrace{\frac{\ln(4)}{d} - \frac{\beta_k}{x_{k-1}^2}}_{\mathcal{O}(d^{-1})} - \underbrace{\left(\frac{5.5c^{2k-6}}{x_{k-1}^2 d^2} + \frac{\left(1 + \frac{10}{d}\right)\beta_k^2}{x_{k-1}^4} + \frac{\ln(4)\beta_k}{x_{k-1}^2 d}\right)}_{\Theta(d^{-2})}$$

$$- \underbrace{\left(\frac{5.5\ln(4)c^{2k-6}}{x_{k-1}^2 d^3} + \frac{\left(1 + \frac{10}{d}\right)\ln(4)\beta_k^2}{x_{k-1}^4 d}\right)}_{\Theta(d^{-3})}.$$

We will now simplify the least significant terms above further. We start from an *upper* bound to the $\Theta(d^{-2})$ term (since its sign is negative in the expression above),

$$\frac{5.5c^{2k-6}}{x_{k-1}^2 d^2} + \frac{\left(1 + \frac{10}{d}\right)\beta_k^2}{x_{k-1}^4} + \frac{\ln(4)\beta_k}{x_{k-1}^2 d} \leq \frac{5.5c^{2k-6}}{x_{k-1}^2 d^2} + \frac{1.001\beta_k^2}{x_{k-1}^2 x_{k-1}^2} + \frac{\ln(4)\beta_k}{x_{k-1}^2 d}$$

$$\left[\beta_k \leq \frac{c^{2k-5}}{d} \leq \frac{1}{cd}, \ x_{k-1} \geq 0.45\right] \leq \frac{5.5c^{2k-6}}{x_{k-1}^2 d^2} + \frac{1.001c^{4k-10}}{x_{k-1}^2 d^2 0.45^2} + \frac{\ln(4)c^{2k-5}}{x_{k-1}^2 d^2}$$

$$\leq \frac{c^{2k-6}}{x_{k-1}^2 d^2}\left(5.5 + 4.95c^{2k-4} + \ln 4 \cdot c\right)$$

$$[k \geq 2, \ c \leq 1] \leq \frac{c^{2k-6}}{x_{k-1}^2 d^2}\left(5.5 + 4.95 + \ln 4\right) \leq \frac{11.84c^{2k-6}}{x_{k-1}^2 d^2}.$$

Similarly, for the $\Theta(d^{-3})$ term, we again employ the upper bound $\beta_k \leq \frac{c^{2k-5}}{d} \leq \frac{1}{cd}$, and obtain,

$$\frac{5.5\ln(4)c^{2k-6}}{x_{k-1}^2 d^3} + \frac{\left(1 + \frac{10}{d}\right)\ln(4)\beta_k^2}{x_{k-1}^4 d} \leq \frac{5.5\ln(4)c^{2k-6}}{x_{k-1}^2 d^3} + \frac{1.001\ln(4)c^{2k-5}}{x_{k-1}^4 d^3 c^3}$$

$$\leq \frac{c^{2k-6}}{x_{k-1}^4 d^3}\left(5.5\ln(4) + 1.001\ln(4)c^{-2}\right) \leq \frac{c^{2k-8}}{x_{k-1}^4 d^3}\left(5.5\ln(4) + 1.001\ln(4)\right) \leq \frac{9.02c^{2k-8}}{x_{k-1}^4 d^3}.$$

And so, we get the following lower bound,

$$a(k) \geq \frac{\ln(4)}{d} - \frac{\beta_k}{x_{k-1}^2} - \left(\frac{11.84c^{2k-6}}{x_{k-1}^2 d^2} + \frac{9.02c^{2k-8}}{x_{k-1}^4 d^3}\right)$$

$$[x_k \leq x_{k-1}] \geq \frac{\ln(4)}{d} - \frac{\beta_k}{x_k^2} - \frac{c^{2k-6}}{x_k^2 d^2}\left(11.84 + \frac{9.02c^{-2}}{x_k^2 d}\right).$$

Back to the overall term we are trying to lower bound,

$$\frac{1}{x_{t-1}}A_1(k) = \frac{h(k+1)}{x_{k-1}}\beta_k - \frac{h(k)}{x_k}\beta_{k+1} = \underbrace{\frac{1}{x_k}h(k)\beta_{k+1}}_{=\Theta\left(d^{-3/2}\right)}\underbrace{\left(\frac{x_k}{x_{k-1}}\frac{h(k+1)}{h(k)}\frac{\beta_k}{\beta_{k+1}} - 1\right)}_{\triangleq a(k)}$$

$$\geq \frac{1}{x_k}h(k)\beta_{k+1}\left(\frac{\ln(4)}{d} - \frac{\beta_k}{x_k^2} - \frac{c^{2k-6}}{x_k^2 d^2}\left(11.84 + \frac{9.02c^{-2}}{x_k^2 d}\right)\right)$$

$$= \frac{1}{x_k}h(k)\beta_{k+1}\left(\frac{\ln(4)}{d} - \frac{\beta_k}{x_k^2}\right) - \frac{1}{x_k}h(k)\beta_{k+1}\frac{c^{2k-6}}{x_k^2 d^2}\left(11.84 + \frac{9.02c^{-2}}{x_k^2 d}\right)$$

$$\overset{(1)}{\geq} \frac{1}{x_k}\frac{c^{k-3}}{\sqrt{d}}\beta_{k+1}\left(\frac{\ln(4)}{d} - \frac{\beta_k}{x_k^2}\right) - \frac{1}{x_k}\frac{\frac{c^{k-3}}{\sqrt{d}} + \frac{5.42}{d^{3/2}}}{d}\frac{c^{2k-6}}{x_k^2 d^2}\left(11.84 + \frac{9.02c^{-2}}{x_k^2 d}\right)$$

$$\overset{(2)}{\geq} \frac{1}{x_k}\frac{c^{k-3}}{\sqrt{d}}\beta_{k+1}\left(\frac{\ln(4)}{d} - \frac{\beta_k}{x_k^2}\right) - \frac{c^{3k-9}}{x_k^3 d^{7/2}}\left(11.84 + \frac{9.02 \cdot 2^{2/10000}}{0.45^2 d}\right)$$

$$- \frac{5.42c^{2k-6}}{x_k^3 d^{9/2}}\left(11.84 + \frac{9.02 \cdot 2^{2/10000}}{0.45^2 d}\right),$$

where (1) is since $h(k) \in \left[\frac{c^{k-3}}{\sqrt{d}}, \frac{c^{k-3}}{\sqrt{d}} + \frac{5.42}{d^{3/2}}\right]$, $\beta_{k+1} \leq \frac{1}{d}$; (2) is since $d \geq 10{,}000$ and $x_k \geq 0.45$. Furthermore,

$$
\frac{1}{x_{t-1}} A_1(k)
$$
$$
\geq \frac{1}{x_k} \frac{c^{k-3}}{\sqrt{d}} \beta_{k+1} \left(\frac{\ln(4)}{d} - \frac{\beta_k}{x_k^2}\right) - \frac{11.84 c^{3k-9}}{x_k^3 d^{7/2}} - \frac{44.55 c^{3k-9}}{x_k^3 d^{9/2}} - \frac{64.18 c^{2k-6}}{x_k^3 d^{9/2}} - \frac{241.46 c^{2k-6}}{x_k^3 d^{11/2}}
$$
$$
\geq \frac{1}{x_k} \frac{c^{k-3}}{\sqrt{d}} \beta_{k+1} \left(\frac{\ln(4)}{d} - \frac{\beta_k}{x_k^2}\right) - \frac{11.84 c^{3k-9}}{x_k^3 d^{7/2}} - \frac{44.55 c^{2k-9}}{x_k^3 d^{9/2}} - \frac{64.18 c^{2k-9}}{x_k^3 d^{9/2}} - \frac{241.46 c^{2k-9}}{x_k^3 d^{11/2}}
$$
$$
\overset{(3)}{\geq} \frac{1}{x_k} \frac{c^{k-3}}{\sqrt{d}} \beta_{k+1} \left(\frac{\ln(4)}{d} - \frac{\beta_k}{x_k^2}\right) - \frac{11.84 c^{3k-9}}{x_k^3 d^{7/2}} - \frac{109 c^{2k-9}}{x_k^3 d^{9/2}},
$$

where (3) is since $d \geq 10{,}000$. Overall, we get,

$$
A_1(k) \geq \frac{x_{t-1}}{x_k} \frac{c^{k-3}}{\sqrt{d}} \beta_{k+1} \left(\frac{\ln(4)}{d} - \frac{\beta_k}{x_k^2}\right) - \frac{11.84 x_{t-1} c^{3k-9}}{x_k^3 d^{7/2}} - \frac{109 x_{t-1} c^{2k-9}}{x_k^3 d^{9/2}}.
$$

It remains to get a lower bound for $\beta_{k+1}\left(\frac{\ln(4)}{d} - \frac{\beta_k}{x_k^2}\right)$.

First, we show

$$
b(k) \triangleq \frac{\ln(4)}{d} - \frac{\beta_k}{x_k^2} = \frac{\ln(4)}{d} - \frac{(1 - (1-c)(k-1)) c^{2k-5}}{x_k^2 d}
$$
$$
= \frac{\ln(4)}{d} - \frac{c^{2k-5}}{x_k^2 d} + \frac{1}{x_k^2 c^3} \cdot (1-c)\left(\frac{k-1}{d}\right) 4^{-\frac{k-1}{d}}
$$
$$
\geq \frac{\ln(4)}{d} - \frac{c^{2k-5}}{x_k^2 d} + \frac{1}{x_k^2 c^3} \cdot \frac{1-c}{4}\left(\frac{k-1}{d}\right),
$$

where we used an algebraic property that $4^{-z} \geq \frac{1}{4}, \forall z \in [0,1]$. Continuing,

$$
b(k) \geq \frac{\ln(4)}{d} - \frac{c^{2k-5}}{x_k^2 d} + \frac{1}{4 x_k^2 c^3} \cdot \left(\frac{\ln(2)}{d} - \frac{\ln^2(2)}{2d^2}\right)\left(\frac{k-1}{d}\right)
$$
$$
= \frac{\ln(4)}{d} - \frac{c^{2k-5}}{x_k^2 d} + \frac{\ln(2)}{4 x_k^2 c^3 d}\left(\frac{k-1}{d}\right) - \frac{\ln^2(2)}{8 x_k^2 c^3 d^2}\left(\frac{k-1}{d}\right)
$$
$$
[c \leq 1] \geq \frac{\ln(4)}{d} - \frac{c^{2k-5}}{x_k^2 d} + \frac{\ln(2)}{4 x_k^2 d}\left(\frac{k-1}{d}\right) - \frac{\ln^2(2)}{8 x_k^2 c^3 d^2}\left(\frac{k-1}{d}\right)
$$
$$
\geq \frac{\ln(4)}{d} - \frac{c^{2k-5}}{x_k^2 d} + \frac{\ln(2)}{4 x_k^2 d}\left(\frac{k-1}{d}\right) - \frac{\ln^2(2)}{8 (0.45)^2 c^3 d^2} \cdot 1
$$
$$
\geq \frac{\ln(4)}{d} - \frac{c^{2k-5}}{x_k^2 d} + \frac{\ln(2)}{4 x_k^2 d} \cdot \frac{k-1}{d} - \frac{0.3}{c^3 d^2} \geq \frac{\ln(4)}{d} + \frac{\ln(2)\left(\frac{k-1}{d}\right) - 4 c^{2k-5}}{4 x_k^2 d} - \frac{0.3}{c^3 d^2}.
$$

Below, we are going to use the closed-form approximation of $x_k$, for which we have established $|x_k - \tilde{x}_k| \leq \frac{170.4}{d} = \epsilon$ (Lemma J.2), and also note that $\left|x_k^2 - \tilde{x}_k^2\right| \leq 2 x_k \epsilon + \epsilon^2$ (Claim I.10). Also, recall that, $\tilde{x}_k = \sqrt{1 - \frac{1}{\ln 4} + 4^{-\frac{k}{d}}\left(\frac{1}{\ln 4} - \frac{k}{d}\right)} = \sqrt{1 - \frac{1}{\ln 4} + c^{2k}\left(\frac{1}{\ln 4} - \frac{k}{d}\right)}$. We now use these

relations to further refine the lower bound on $b(k)$:

$$b(k) + \frac{0.3}{c^3 d^2} \geq \frac{\ln(4)}{d} + \frac{\ln(2)\left(\frac{k-1}{d}\right) - 4c^{2k-5}}{4x_k^2 d} = \frac{4x_k^2 \ln(4) + \ln(2)\left(\frac{k-1}{d}\right) - 4c^{2k-5}}{4x_k^2 d}$$

$$\geq \frac{4\tilde{x}_k^2 \ln(4) + \ln(2)\left(\frac{k-1}{d}\right) - 4c^{2k-5}}{4x_k^2 d} - \frac{4\ln(4)}{4x_k^2 d}\left|x_k^2 - \tilde{x}_k^2\right|$$

$$\geq \frac{4\tilde{x}_k^2 \ln(4) + \ln(2)\left(\frac{k-1}{d}\right) - 4c^{2k-5}}{4x_k^2 d} - \frac{\ln(4)}{x_k^2 d}\left(2x_k\epsilon + \epsilon^2\right)$$

$$= \frac{4\left(\ln(4) - 1 + c^{2k}\left(1 - \frac{k}{d}\ln(4)\right)\right) + \ln(2)\left(\frac{k-1}{d}\right) - 4c^{2k-5}}{4x_k^2 d} - \frac{\epsilon^2}{x_k^2 d} - \frac{\ln(16)\epsilon}{x_k d}$$

$$\geq \frac{1.545 + \ln(2)\left(\frac{k-1}{d}\right) + 4c^{2k}\left(1 - \frac{k}{d}\ln(4)\right) - 4c^{2k-5}}{4x_k^2 d} - \frac{\epsilon^2}{x_k^2 d} - \frac{\ln(16)\epsilon}{x_k d}$$

$$= \frac{1.545 + \ln(2)\left(\frac{k-1}{d}\right) - 4c^{2k}\ln(4)\frac{k}{d}}{4x_k^2 d} - \frac{c^{2k-5}\left(1 - c^5\right)}{x_k^2 d} - \frac{\epsilon^2}{x_k^2 d} - \frac{\ln(16)\epsilon}{x_k d}$$

$$[\mathrm{I.1}] \geq \frac{1.545 + \ln(2)\left(\frac{k-1}{d}\right) - 4c^{2k}\ln(4)\frac{k}{d}}{4x_k^2 d} - \frac{5\ln(2)c^{2k-5}}{x_k^2 d^2} - \frac{\epsilon^2}{x_k^2 d} - \frac{\ln(16)\epsilon}{x_k d} \; .$$

Focusing on the left nominator,

$$1.545 + \ln(2)\left(\frac{k-1}{d}\right) - 4c^{2k}\ln(4)\frac{k}{d} = 1.545 + \underbrace{\ln(2)}_{>0}\left(\frac{1}{d}(k-1) - 8c^{2k}\frac{k}{d}\right)$$

$$= 1.545 + \ln(2)\left(\left(1 - 8c^{2k}\right)\frac{k}{d} - \frac{1}{d}\right)1.545 - \ln(2)\left(\left(8c^{2k} - 1\right)\frac{k}{d} + \frac{1}{d}\right)$$

$$= 1.545 - \ln(2)\left(\left(8 \cdot 4^{-\frac{k}{d}} - 1\right)\frac{k}{d} + \frac{1}{d}\right) \; .$$

To upper bound $g(x) = 8x \cdot 4^{-x}$ (inside $x \in [0,1]$), we show that

$$0 \stackrel{!}{=} g'(x) = 8 \cdot 4^{-x} - 8x\ln(4)4^{-x} = 8 \cdot 4^{-x}\left(1 - x\ln(4)\right) \; ,$$

solved by $x = \frac{1}{\ln(4)}$, which falls inside $x \in [0,1]$, meaning it is a global optimum.

The second derivative is

$$g''(x) = \left(8 \cdot 4^{-x} - 8\ln(4) \cdot 4^{-x}x\right)' = -4^{2-x}\ln(2) - 8\ln(4) \cdot 4^{-x}\left(1 - x\ln(4)\right)$$

$$g''\left(\frac{1}{\ln(4)}\right) = -4^{2-\frac{1}{\ln(4)}}\ln(2) = -\frac{16\ln(2)}{e} < 0 \; ,$$

meaning that the $x = \frac{1}{\ln(4)}$ is the global **maximum.** Also note: $\left(4^{\frac{1}{\ln 4}}\right)^{\ln 4} = 4 \Rightarrow 4^{\frac{1}{\ln 4}} = e.$

So overall, we get,

$$1.545 + \ln(2)\left(\frac{k-1}{d}\right) - 4c^{2k}\ln(4)\frac{k}{d} \geq 1.545 - \ln(2)\left(\left(8 \cdot 4^{-\frac{k}{d}} - 1\right)\frac{k}{d} + \frac{1}{d}\right)$$

$$\geq 1.545 - \ln(2)\left(\left(8 \cdot 4^{-\frac{1}{\ln 4}} - 1\right)\frac{1}{\ln 4} + \frac{1}{d}\right) = 1.545 - \frac{\ln(2)}{d} - \frac{\ln(2)}{2\ln(2)}\left(\frac{8}{e} - 1\right)$$

$$\geq 1.545 - 0.972 - \frac{0.7}{d} \geq 0.573 - \frac{0.7}{d} \; .$$

Finally,

$$b(k) + \frac{0.3}{c^3 d^2} + \frac{\epsilon^2}{x_k^2 d} + \frac{\ln(16)\epsilon}{x_k d} + \frac{5\ln(2)c^{2k-5}}{x_k^2 d^2} \geq \frac{1.545 + \ln(2)\left(\frac{k-1}{d}\right) - 4c^{2k}\ln(4)\frac{k}{d}}{4x_k^2 d}$$

$$\geq \frac{0.573 - \frac{0.7}{d}}{4x_k^2 d} \geq \frac{\mathbf{0.14325}}{\mathbf{x_k^2 d}} - \frac{0.175}{x_k^2 d^2} \; .$$

Going back to $\beta_{k+1}\left(\frac{\ln(4)}{d} - \frac{\beta_k}{x_k^2}\right) = \beta_{k+1}b\left(k\right)$, we have

$$\beta_{k+1}b\left(k\right) \geq \beta_{k+1}\left(\frac{0.14325}{x_k^2 d} - \left(\frac{0.3}{c^3 d^2} + \frac{\epsilon^2}{x_k^2 d} + \frac{\epsilon \ln(16)}{x_k d} + \frac{5\ln(2)c^{2k-5}}{x_k^2 d^2} + \frac{0.175}{x_k^2 d^2}\right)\right)$$

$$[c < 1,\ k \geq 2] \geq \beta_{k+1}\left(\frac{0.14325}{x_k^2 d} - \left(\frac{0.3}{c^3 d^2} + \frac{\epsilon^2}{x_k^2 d} + \frac{\epsilon \ln(16)}{x_k d} + \frac{5\ln(2)}{cx_k^2 d^2} + \frac{0.175}{x_k^2 d^2}\right)\right)$$

$$= \beta_{k+1}\left(\frac{0.14325}{x_k^2 d} - \frac{1}{x_k}\left(\frac{0.3x_k}{c^3 d^2} + \frac{\epsilon^2}{x_k d} + \frac{\epsilon \ln(16)}{d} + \frac{5\ln(2)}{cx_k d^2} + \frac{0.175}{x_k d^2}\right)\right)$$

$$[0.45 \leq x_k \leq 1] \geq \beta_{k+1}\left(\frac{0.14325}{x_k^2 d} - \frac{1}{x_k}\left(\frac{0.3}{c^3 d^2} + \frac{\epsilon \ln(16)}{d} + \frac{5\ln(2)}{c \cdot 0.45 d^2} + \frac{0.175}{0.45 d^2} + \frac{\epsilon^2}{0.45 d}\right)\right)$$

$$\geq \beta_{k+1}\left(\frac{0.14325}{x_k^2 d} - \frac{1}{x_k}\left(\frac{0.3}{c^3 d^2} + \frac{2.78\epsilon}{d} + \frac{7.71}{cd^2} + \frac{0.39}{d^2} + \frac{2.3\epsilon^2}{d}\right)\right).$$

Since $c = 2^{-1/d} \geq 0.9999,\ \forall d \geq 10{,}000$, and plugging in $\epsilon = \frac{170.4}{d}$:

$$\beta_{k+1}\left(\frac{\ln(4)}{d} - \frac{\beta_k}{x_k^2}\right)$$

$$\geq \beta_{k+1}\left(\frac{0.14325}{x_k^2 d} - \frac{1}{x_k}\left(\frac{0.3}{0.9999^3 d^2} + \frac{2.78 \cdot 170.4}{d^2} + \frac{7.71}{0.9999 d^2} + \frac{0.39}{d^2} + \frac{2.3 \cdot 170.4^2}{d^3}\right)\right)$$

$$\geq \beta_{k+1}\left(\frac{0.14325}{x_k^2 d} - \frac{1}{x_k}\left(\frac{482.2}{d^2} + \frac{66783.17}{d^3}\right)\right)$$

$$\overset{(1)}{\geq} \beta_{k+1}\left(\frac{0.14325}{x_k^2 d} - \frac{1}{x_k}\left(\frac{488.88}{d^2}\right)\right) \geq \frac{\beta_{k+1}}{x_k d}\left(\frac{0.14325}{x_k} - \frac{488.88}{d}\right),$$

where (1) is since $d \geq 10{,}000$. The inside of the parenthesis is **positive** $\forall d \geq \left\lceil\frac{488.88}{0.14325}\right\rceil = 3413$, so we can bound the expression by lower bounding $\frac{\beta_{k+1}}{x_k^2}$.

$$\beta_{k+1}\left(\frac{\ln(4)}{d} - \frac{\beta_k}{x_k^2}\right) \geq \frac{\beta_{k+1}}{x_k d}\left(\frac{0.14325}{x_k} - \frac{488.88}{d}\right) \geq \frac{0.3c^{2k-3}}{d^2}\left(\frac{0.14325}{x_k^2} - \frac{488.88}{x_k d}\right)$$

$$\geq c^{2k-3}\left(\frac{0.0429}{x_k^2 d^2} - \frac{146.7}{x_k d^3}\right).$$

And then,

$$A_1\left(x\right) \geq x_{t-1}\left(\frac{1}{x_k}\frac{c^{k-3}}{\sqrt{d}}\beta_{k+1}\left(\frac{\ln(4)}{d} - \frac{\beta_k}{x_k^2}\right) - \frac{11.84c^{3k-9}}{x_k^3 d^{7/2}} - \frac{109c^{2k-9}}{x_k^3 d^{9/2}}\right)$$

$$\geq x_{t-1}\left(\frac{1}{x_k}\frac{c^{k-3}}{\sqrt{d}}c^{2k-3}\left(\frac{0.0429}{x_k^2 d^2} - \frac{146.7}{x_k d^3}\right) - \frac{11.84c^{3k-9}}{x_k^3 d^{7/2}} - \frac{109c^{2k-9}}{x_k^3 d^{9/2}}\right)$$

$$= x_{t-1}\left(\frac{0.0429c^{3k-6}}{x_k^3 d^{5/2}} - \frac{146.7c^{3k-6}}{x_k^2 d^{7/2}} - \frac{11.84c^{3k-9}}{x_k^3 d^{7/2}} - \frac{109c^{2k-9}}{x_k^3 d^{9/2}}\right)$$

$$[c < 1] \geq x_{t-1}\left(\frac{0.0429c^{3k-6}}{x_k^3 d^{5/2}} - \frac{146.7c^{3k-9}}{x_k^2 d^{7/2}} - \frac{11.84c^{3k-9}}{x_k^3 d^{7/2}} - \frac{109c^{3k-9}}{c^k x_k^3 d^{9/2}}\right)$$

$$\overset{(1)}{\geq} x_{t-1}\left(\frac{0.0429c^{3k-6}}{x_k^3 d^{5/2}} - \frac{146.7c^{3k-9}}{x_k^2 d^{7/2}} - \frac{11.84c^{3k-9}}{0.45x_k^2 d^{7/2}} - \frac{109c^{3k-9}}{0.5 \cdot 0.45 x_k^2 d^{9/2}}\right)$$

$$\geq x_{t-1}\left(\frac{0.0429c^{3k-6}}{x_k^3 d^{5/2}} - \frac{173.02c^{3k-9}}{x_k^2 d^{7/2}} - \frac{484.45c^{3k-9}}{x_k^2 d^{9/2}}\right)$$

$$[d \geq 10{,}000] \geq x_{t-1}\left(\frac{0.0429c^{3k-6}}{x_k^3 d^{5/2}} - \frac{173.07c^{3k-9}}{x_k^2 d^{7/2}}\right),$$

where (1) is since $x_k \geq 0.45,\ c^k \geq c^d = 0.5$.

$\square$

## I.4 Conclusion

We are reminded that we want to show positivity of $\Delta_{t,k} \triangleq \|\mathbf{w}_k - \mathbf{w}_{k+1}\| (\mathbf{w}_{k-1} - \mathbf{w}_k)^\top \mathbf{w}_{t-1} - \|\mathbf{w}_{k-1} - \mathbf{w}_k\| (\mathbf{w}_k - \mathbf{w}_{k+1})^\top \mathbf{w}_{t-1}$. Applying Proposition I.18, we show $\forall k \geq t$:

$$
\begin{aligned}
\Delta_{t,k} &= \|\mathbf{w}_k - \mathbf{w}_{k+1}\| (\mathbf{w}_{k-1} - \mathbf{w}_k)^\top \mathbf{w}_{t-1} - \|\mathbf{w}_{k-1} - \mathbf{w}_k\| (\mathbf{w}_k - \mathbf{w}_{k+1})^\top \mathbf{w}_{t-1} \\
&\geq A_1(k) + A_2(k) + A_3(k) \\
&\quad - \frac{1}{d^{7/2}} \left( \frac{96c^{6k-15}}{x_k} + 113c^{7k-18} \right) - \frac{1}{d^{9/2}} \left( \frac{56.5c^{9k-18}}{x_k^3} + 614c^{6k-30} \right) \\
&\overset{(1)}{\geq} A_1(k) + A_2(k) + A_3(k) \\
&\quad - \frac{1}{d^{7/2}} \left( \frac{96c^{6k-15}}{x_k} + 113c^{7k-18} \right) - \frac{1}{d^{9/2}} \left( \frac{56.5c^{7k-18}}{x_k^3} + 614c^{7k-18}c^{-k-12} \right) \\
&\overset{(2)}{\geq} A_1(k) + A_2(k) + A_3(k) \\
&\quad - \frac{1}{d^{7/2}} \left( \frac{96c^{6k-15}}{x_k} + 113c^{7k-18} \right) - \frac{c^{7k-18}}{d^{9/2}} \left( \frac{56.5}{x_k^3} + 614 \cdot 1.00007^{12} \cdot 2 \right) \\
&\overset{(3)}{\geq} A_1(k) + A_2(k) + A_3(k) \\
&\quad - \frac{1}{d^{7/2}} \left( \frac{96c^{6k-15}}{0.45} + 113c^{7k-18} \right) - \frac{c^{7k-18}}{d^{9/2}} \left( \frac{56.5}{0.45^3} + 1238.36 \right) \\
&\geq A_1(k) + A_2(k) + A_3(k) - \frac{1}{d^{7/2}} \left( 213.34c^{6k-15} + 113c^{7k-18} \right) - \frac{1858.39c^{7k-18}}{d^{9/2}} \\
&\overset{(4)}{\geq} A_1(k) + A_2(k) + A_3(k) - \frac{1}{d^{7/2}} \left( 213.34c^{6k-15} + 113.19c^{7k-18} \right),
\end{aligned}
$$

where (1) is since $c < 1$, $k \geq 2$; (2) is since $2^{1/10000} \leq 1.00007$, $c^{-k} \leq c^{-d} = 2$; (3) is since $x_{k-1} \geq 0.45$; and (4) is since $d \geq 10{,}000$. Plugging in the results of Propositions I.19, I.20 and I.21, we derive

$$
\begin{aligned}
\Delta_{t,k} &= \|\mathbf{w}_k - \mathbf{w}_{k+1}\| (\mathbf{w}_{k-1} - \mathbf{w}_k)^\top \mathbf{w}_{t-1} - \|\mathbf{w}_{k-1} - \mathbf{w}_k\| (\mathbf{w}_k - \mathbf{w}_{k+1})^\top \mathbf{w}_{t-1} \\
&\geq \underbrace{x_{t-1} \left( \frac{0.0429c^{3k-6}}{x_k^3 d^{5/2}} - \frac{173.07c^{3k-9}}{x_k^2 d^{7/2}} \right)}_{A_1(k)} \underbrace{- \frac{14.88x_{t-1}c^{3k-12}}{x_k^3 d^{7/2}}}_{A_2(k)} \underbrace{- \frac{6.77c^{k+t-6}}{x_k^4} \left( \frac{1}{d^{7/2}} + \frac{5.42}{d^{9/2}} \right)}_{A_3(k)} \\
&\quad - \frac{1}{d^{7/2}} \left( 213.34c^{6k-15} + 113.19c^{7k-18} \right) \\
&= \frac{0.0429x_{t-1}c^{3k-6}}{x_k^3 d^{5/2}} - \frac{173.07x_{t-1}c^{3k-9}}{x_k^2 d^{7/2}} - \frac{14.88x_{t-1}c^{3k-12}}{x_k^3 d^{7/2}} - \frac{6.77c^{k+t-6}}{x_k^4} \left( \frac{1}{d^{7/2}} + \frac{5.42}{d^{9/2}} \right) \\
&\quad - \frac{1}{d^{7/2}} \left( 213.34c^{6k-15} + 113.19c^{7k-18} \right) \\
&\overset{(1)}{\geq} \frac{0.0429c^{3k-6}}{x_k^2 d^{5/2}} - \frac{173.07x_{t-1}c^{3k-9}}{x_k^2 d^{7/2}} - \frac{14.88x_{t-1}c^{3k-12}}{x_k^3 d^{7/2}} - \frac{6.77c^{k+t-6}}{x_k^4} \left( \frac{1}{d^{7/2}} + \frac{5.42}{d^{9/2}} \right) \\
&\quad - \frac{1}{d^{7/2}} \left( 213.34c^{6k-15} + 113.19c^{7k-18} \right) \\
&\overset{(2)}{\geq} \frac{0.0429c^{3k-6}}{x_k^2 d^{5/2}} - \frac{173.07c^{3k-9}}{x_k^2 d^{7/2}} - \frac{14.88c^{3k-12}}{x_k^3 d^{7/2}} - \frac{6.77c^{k+t-6}}{x_k^4} \left( \frac{1}{d^{7/2}} + \frac{5.42}{d^{9/2}} \right) \\
&\quad - \frac{1}{d^{7/2}} \left( 213.34c^{6k-15} + 113.19c^{7k-18} \right) \\
&\overset{(3)}{\geq} \frac{0.0429c^{3k-6}}{x_k^2 d^{5/2}} - \frac{173.07c^{3k-9}}{x_k^2 d^{7/2}} - \frac{14.88c^{3k-12}}{x_k^3 d^{7/2}} - \frac{6.78c^{k+t-6}}{x_k^4 d^{7/2}} \\
&\quad - \frac{1}{d^{7/2}} \left( 213.34c^{6k-15} + 113.19c^{7k-18} \right)
\end{aligned}
$$

$$\overset{(4)}{\geq} \frac{0.0429c^{3k-6}}{x_k^2 d^{5/2}} - \frac{173.07c^{3k-9}}{x_k^2 d^{7/2}} - \frac{14.88c^{3k-12}}{x_k^3 d^{7/2}} - \frac{6.78c^{k+t-6}}{x_k^4 d^{7/2}}$$

$$- \frac{213.34c^{6k-15} + 113.19c^{7k-18}}{x_k^2 d^{7/2}}$$

$$= \frac{1}{x_k^2 d^{5/2}} \left( 0.0429c^{3k-6} - \frac{1}{d} \left( 173.07c^{3k-9} \right. \right.$$

$$\left. \left. + \frac{14.88c^{3k-12}}{x_k} + \frac{6.78c^{k+t-6}}{x_k^2} + 213.34c^{6k-15} + 113.19c^{7k-18} \right) \right)$$

$$\overset{(5)}{\geq} \frac{1}{x_k^2 d^{5/2}} \left( 0.0429c^{3k-6} - \frac{1}{d} \left( 173.07c^{3k-9} \right. \right.$$

$$\left. \left. + \frac{14.88c^{3k-12}}{0.45} + \frac{6.78c^{k+t-6}}{0.45^2} + 213.34c^{6k-15} + 113.19c^{7k-18} \right) \right)$$

$$\geq \frac{1}{x_k^2 d^{5/2}} \left( 0.0429c^{3k-6} - \frac{1}{d} \left( 173.07c^{3k-9} \right. \right.$$

$$\left. \left. + 33.07c^{3k-12} + 33.49c^{k+t-6} + 213.34c^{6k-15} + 113.19c^{7k-18} \right) \right)$$

$$= \frac{c^{3k-6}}{x_k^2 d^{5/2}} \left( 0.0429 - \frac{1}{d} \left( 173.07c^{-3} \right. \right.$$

$$\left. \left. + 33.07c^{-6} + 33.49c^{-2k+t} + 213.34c^{3k-9} + 113.19c^{4k-12} \right) \right)$$

$$\overset{(6)}{\geq} \frac{c^{3k-6}}{x_k^2 d^{5/2}} \left( 0.0429 - \frac{1}{d} \left( 173.07c^{-3} + 33.07c^{-6} + 33.49c^{-2k} + 213.34c^{-3} + 113.19c^{-4} \right) \right)$$

$$\overset{(7)}{\geq} \frac{c^{3k-6}}{x_k^2 d^{5/2}} \left( 0.0429 - \frac{1}{d} \left( 173.07c^{-3} + 33.07c^{-6} + 33.49 \cdot 4 + 213.34c^{-3} + 113.19c^{-4} \right) \right)$$

$$\overset{(8)}{\geq} \frac{c^{3k-6}}{x_k^2 d^{5/2}} \left( 0.0429 - \frac{1}{d} \left( 173.07 \cdot 0.9999^{-3} \right. \right.$$

$$\left. \left. + 33.07 \cdot 0.9999^{-6} + 33.49 \cdot 4 + 213.34 \cdot 0.9999^{-3} + 113.19 \cdot 0.9999^{-4} \right) \right) ,$$

where (1) is since $x_{t-1} > x_k$; (2) is since $x_{t-1} \leq 1$; (3) is since $d \geq 10{,}000$; (4) is since $x_k \leq 1$; (5) is since $x_k \geq 0.45$; (6) is since $c < 1$, $k \geq 2$; (7) is since $c^{-2k} = 4^{k/d} \leq 4$; and (8) is since $d \geq 10{,}000 \Rightarrow c \geq 0.9999$.

Finally, we conclude that,

$$\Delta_{t,k} \geq \frac{c^{3k-6}}{x_k^2 d^{5/2}} \left( 0.0429 - \frac{666.82}{d} \right) . \tag{13}$$

Hence, a sufficient condition for $\frac{(\mathbf{w}_{k-1}-\mathbf{w}_k)^\top \mathbf{w}_{t-1}}{\|\mathbf{w}_{k-1}-\mathbf{w}_k\|} - \frac{(\mathbf{w}_k-\mathbf{w}_{k+1})^\top \mathbf{w}_{t-1}}{\|\mathbf{w}_k-\mathbf{w}_{k+1}\|}$ to be positive and monotonicity to hold, is that $d \geq \lceil \frac{666.82}{0.0429} \rceil = 15{,}544$. Since this is smaller than 25,000, this concludes our proof of positivity of $\Delta_{t,k}$.

# J  Lower bound technical appendix: Properties of the recursive construction

This section complements App. E.1 by confirming that the recursive construction introduced there is well-defined; readers are encouraged to review it first for context.

Specifically, we prove that the recurrence defining the sequence $(x_k)$ is well-posed, in the sense that the square root is always taken over a nonnegative quantity:

**Lemma J.1** (Existence of the recursive sequence). *Given the sequence* $(x)_k$ *recursively defined by* $x_1 = 1$, $x_k = \frac{x_{k-1}+\sqrt{x_{k-1}^2-4\beta_k}}{2}$, $\forall k \in \{2, \dots, d\}$ *where* $c \triangleq 2^{-1/d}$ *and* $\beta_k \triangleq \frac{((k-1)c-(k-2))c^{2k-5}}{d}$, *we have* $\forall d \geq 30$, $\forall k \in \{2, \dots, d\}$ *that*

$$x_{k-1}^2 - 4\beta_k \geq 0 \,.$$

In addition, we prove the following lemma:

**Lemma J.2** (Approximation by closed-form reference). *Given the sequence* $(x)_k$ *recursively defined by* $x_1 = 1$, $x_k = \frac{x_{k-1}+\sqrt{x_{k-1}^2-4\beta_k}}{2}$, $\forall k \in \{2, \dots, d\}$ *where* $c \triangleq 2^{-1/d}$ *and* $\beta_k \triangleq \frac{((k-1)c-(k-2))c^{2k-5}}{d}$, *and the sequence* $\tilde{x}_k = \sqrt{1 - \frac{1}{\ln 4} + 4^{-\frac{k}{d}}\left(\frac{1}{\ln 4} - \frac{k}{d}\right)}$, *we have* $\forall d \geq 30$, $\forall k \in [d]$:

$$|x_k - \tilde{x}_k| \leq \frac{170.4}{d} \,.$$

Before proving this lemma, we note the following will immediately hold:

**Corollary J.3** (Lower bound on $x_k$). $\forall d \geq 25{,}000$, $\forall k \in [d]$: $x_k \geq 0.45$.

*Proof.* $x_k$ is decreasing (Claim I.5), so $\forall k \in [d]$:

$$x_k \geq x_d \geq \tilde{x}_d - \frac{170.4}{d} = \sqrt{1 - \frac{1}{\ln 4} + 4^{-1}\left(\frac{1}{\ln 4} - 1\right)} - \frac{170.4}{d}$$

$$[d \geq 25{,}000] \geq 0.45 \,.$$

$\qquad\qquad\qquad\qquad\qquad\qquad\qquad\qquad\qquad\qquad\qquad\qquad\qquad\qquad\qquad\qquad\qquad\qquad\qquad\square$

This bound is extensively used in the proof of App. I.

## J.1  Proof outline

First, we show the above holds numerically for $30 \leq d < 100{,}000$, as can be seen in Figure 26. We then prove analytically for $d \geq 100{,}000$, by constructing an ODE for which the sequence $(x)_k$ serves as an Euler trajectory. We then bound the distance between the solution to this ODE and a known function $\tilde{x}(\tau)$. Combining this bound with Euler's method global truncation error bound, we obtain a bound for the distance between $(x)_k$ and $(\tilde{x})_k$. We then use this bound to show the existence of the sequence $(x)_k$ for all $k \in \{2, \dots, d\}$.

**Computational resources**  The numerical validation took 6 hours to run on a home PC with i5-9400F CPU and 16GB RAM.

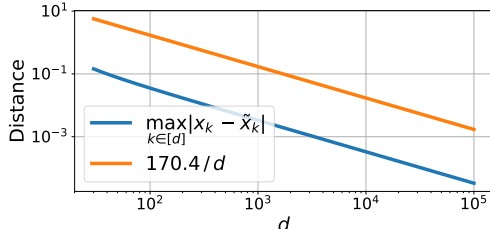

(a) $|x_k - \tilde{x}_k| \leq \frac{170.4}{d}$. This is a loose, analytically derived upper bound.

(b) Actual upper bound is $< \frac{4.5}{d}$.

Figure 26: **Numerical proof of Lemma J.2 for d<100,000.** Using the recursive definition of $x_k$, we calculated the sequence for each value of $d$, $\forall k \in [d]$, and compared with $\tilde{x}_k$.

### J.1.1 Euler's method construction and bottom line

Here, we leverage the global truncation error of Euler's method to establish the bound. The auxiliary propositions supporting this result are proved in the following section. Define

$$f\left(\tau, x\right) = d\frac{\sqrt{x^2 - 4\beta\left(\tau + \frac{1}{d}\right)} - x}{2} \,,$$

$$\beta\left(\tau\right) = \frac{\left(\left(d\tau - 1\right)2^{-1/d} - \left(d\tau - 2\right)\right)2^{(5 - 2d\tau)/d}}{d} \,.$$

Then, using step size of $h = \frac{1}{d}$ in Euler's method we have the iterates

$$x_{k+1} = x_k + h \cdot f\left(\tau_k, x_k\right) \,,$$
$$\tau_{k+1} = \tau_k + h \,,$$

and thus

$$x_{k+1} = x_k + \frac{\sqrt{x_k^2 - 4\beta\left(\frac{k+1}{d}\right)} - x_k}{2} = \frac{x_k + \sqrt{x_k^2 - 4\beta_{k+1}}}{2} \,,$$

which are exactly the iterates we want to solve for.

These are the Euler's iterates for the differential equation

$$x'\left(\tau\right) = f\left(\tau, x\left(\tau\right)\right) \,, \quad x\left(0\right) = 1 \,. \tag{14}$$

While it's hard to find an exact solution to this equation, we proved that for $d \geq 100{,}000$:

$$\left|x\left(\tau\right) - \tilde{x}\left(\tau\right)\right| \leq \frac{38.9822}{d}$$

in Proposition J.19, where we define the *function* suggested in Hucht [48],

$$\tilde{x}\left(\tau\right) = \sqrt{1 - \frac{1}{\ln 4} + 4^{-\tau}\left(\frac{1}{\ln 4} - \tau\right)} \,,$$

such that $\tilde{x}_k = \tilde{x}\left(\frac{k}{d}\right)$. Next, we bound the iterates using the global truncation error of Euler's method, obtaining

$$\left|x_k - x\left(\frac{k}{d}\right)\right| \leq \frac{131.3685}{d} \,.$$

Combining this with the previous result, Proposition J.26 yields

$$\left|x_k - \tilde{x}_k\right| \leq \frac{131.3685}{d} + \frac{38.9822}{d} \leq \frac{170.4}{d} \,.$$

Finally, we use this bound to show the iterates exists $\forall k \in [d]$.

### J.2 Full proof

#### J.2.1 Auxiliary propositions

We begin with preliminary claims and move on to the propositions used in the previous section.

**Claim J.4.** $f(d) \triangleq -d\left(1 - 2^{-1/d}\right)$ is decreasing $\forall d \geq 1$.

*Proof.* $f'(d) = -\left(1 - 2^{-1/d}\right) - d\left((-1) \cdot \frac{1}{d^2} \ln 2 \cdot 2^{-1/d}\right) = -\left(1 - 2^{-1/d}\right) + \frac{\ln 2}{d} 2^{-1/d} = -1 - \left(1 - \frac{\ln 2}{d}\right) 2^{-1/d} < 0.$ $\square$

**Claim J.5.** $\forall d \geq 1 : d\left(2^{1/d} - 1\right) \geq \ln 2$.

*Proof.* Using Taylor's expansion:

$$2^{1/d} = e^{\frac{\ln 2}{d}} = 1 + \frac{\ln 2}{d} + \sum_{i=2}^{\infty} \frac{1}{i!} \left(\frac{\ln 2}{d}\right)^i$$

$$\Rightarrow d\left(2^{1/d} - 1\right) = \ln 2 + \sum_{i=2}^{\infty} \frac{1}{i!} \frac{(\ln 2)^i}{d^{i-1}} \geq \ln 2.$$

$\square$

**Claim J.6.** $-d\left(1 - 2^{-1/d}\right) \geq -\ln 2$ (alternatively: $2^{-1/d} \geq 1 - \frac{\ln 2}{d}$).

*Proof.* From Claim J.4 we know that $-d\left(1 - 2^{-1/d}\right)$ is decreasing with d, so we have,

$$-d\left(1 - 2^{-1/d}\right) \geq \lim_{d \to \infty} -d\left(1 - 2^{-1/d}\right) = \lim_{h \to 0^+} \frac{2^{-h} - 1}{h} = \lim_{h \to 0^+} \frac{2^{-h} - 2^0}{h}.$$

We recognize this as the definition of the derivative of $2^{-x}$ for $x = 0^+$, so we have:

$$\lim_{d \to \infty} -d\left(1 - 2^{-1/d}\right) = \frac{d\left(2^{-x}\right)}{dx}\left(x = 0^+\right) = -\ln 2 \cdot 2^0 = -\ln 2.$$

$\square$

**Claim J.7.** $\beta(\tau) > 0$, decreasing and convex for $\tau < \frac{1}{\ln 2}$.

*Proof.* Reminder that $\beta(\tau) = \frac{\left((d\tau - 1)2^{-1/d} - (d\tau - 2)\right)2^{(5-2d\tau)/d}}{d}$, and $d \geq 1$.

Denote $\beta(\tau) = \frac{1}{d} f(\tau) g(\tau)$, where $f(\tau) = (d\tau - 1) 2^{-1/d} - (d\tau - 2)$ and $g(\tau) = 2^{\frac{5-2d\tau}{d}}$.

We have $\forall \tau, \ g(\tau) > 0$.

Note that from Claim J.6 $1 - \frac{\ln 2}{d} \leq 2^{-1/d} \leq 1$, so:

$$f(\tau) = 2^{-1/d} d\tau - 2^{-1/d} - d\tau + 2 \geq \left(1 - \frac{\ln 2}{d}\right) d\tau - 1 - d\tau + 2 = -\tau \ln 2 + 1,$$

so $f(\tau) > 0$ for $\tau < \frac{1}{\ln 2}$. Thus $\beta(\tau) > 0$ for $\tau < \frac{1}{\ln 2}$.

Now we note that $f'(\tau) = d\left(2^{-1/d} - 1\right) < 0$, $\forall d \geq 1$, $\forall \tau$ and $g'(\tau) = -2 \ln 2 \cdot 2^{\frac{5-2d\tau}{d}} < 0$, $\forall d$, $\forall \tau$

So:

$$\beta'(\tau) = \frac{1}{d}\left(f'(\tau) g(\tau) + g'(\tau) f(\tau)\right) < 0,$$

as long as $g(\tau) > 0$ and $f(\tau) \geq 0$ - which we get for $\tau < \frac{1}{\ln 2}$.

Now note $f''(\tau) = 0$ and $g''(\tau) = 4\ln^2 2 \cdot 2^{\frac{5-2d\tau}{d}} > 0$, so:

$$\beta''(\tau) = \frac{1}{d}\left(f''(\tau)g(\tau) + f'(\tau)g'(\tau) + g''(\tau)f(\tau) + f'(\tau)g'(\tau)\right)$$
$$= \frac{1}{d}\left(2f'(\tau)g'(\tau) + g''(\tau)f(\tau)\right) > 0,$$

as long as $f(\tau) \geq 0$ - which we get for $\tau < \frac{1}{\ln 2}$. $\qquad\square$

**Claim J.8.** $x(\tau)$ defined by the ODE in Eq. (14) satisfies the ODE $x(0) = 1$, $x'(\tau) = d\frac{\sqrt{x^2 - g(\tau)} - x}{2}$ with $g(\tau) = 4\beta\left(\tau + \frac{1}{d}\right)$. We also have:

$$\forall d \geq 3, \quad \max_{s \in [0,1]} g(s) = g(0) = 4\frac{2^{3/d}}{d}.$$

*Proof.* Substituting $x'(\tau) = d\frac{\sqrt{x^2 - 4\beta\left(\tau + \frac{1}{d}\right)} - x}{2}$ in $x'(\tau) = d\frac{\sqrt{x^2 - g(\tau)} - x}{2}$ we get $g(\tau) = 4\beta\left(\tau + \frac{1}{d}\right)$.

For $\tau \in [0,1]$ and $d \geq 3$, $\tau + \frac{1}{d} \leq \frac{1}{\ln 2}$. We get from Claim J.7 that $\beta$ is decreasing, so:

$$\max_{s \in [0,1]} g(s) = g(0) = 4\beta\left(\frac{1}{d}\right) = 4\frac{\left((1-1)2^{-1/d} - (1-2)\right)2^{(5-2)/d}}{d} = 4\frac{2^{3/d}}{d}.$$
$\qquad\square$

*Remark* J.9. The solution to the ODE $x(\tau)x'(\tau) = -f(x)$, $x(0) = 1$ is

$$x(\tau) = \sqrt{1 - 2\int_0^\tau f(s)\,\mathrm{d}s}.$$

**Claim J.10.** $\forall 0 \leq a \leq \mu \leq 1 : 1 - \frac{1-\sqrt{1-\mu}}{\mu}a \leq \sqrt{1-a} \leq 1 - \frac{a}{2}$.

*Proof.* The right side inequality is trivial: $\left(1 - \frac{a}{2}\right)^2 = 1 - a + \frac{a^2}{4} \geq 1 - a = \left(\sqrt{1-a}\right)^2$.

For the left side: denote $f(a) = \sqrt{1-a}$. $f$ is concave: $f'(a) = -\frac{1}{2\sqrt{1-a}}$, $f''(a) = \frac{\frac{-2}{2\sqrt{1-a}}}{4(1-a)} \leq 0$. So we have $\forall 0 \leq a \leq \mu \leq 1$:

$$\sqrt{1-a} = f(a) \geq \frac{(\mu - a)f(0) + af(\mu)}{\mu}$$
$$= 1 - \frac{a}{\mu} + \frac{a}{\mu}\sqrt{1-\mu} = 1 - \frac{1-\sqrt{1-\mu}}{\mu}a.$$
$\qquad\square$

*Remark* J.11. The solution to the ODE in Eq. (14) is only defined when the quantity under the square root remains nonnegative, *i.e.,*

$$x^2(\tau) \geq 4\beta\left(\tau + \frac{1}{d}\right).$$

Given that $x(0) = 1$ and that $4\beta\left(\frac{1}{d}\right) = 4\frac{2^{3/d}}{d} < 1$ for all $d \geq 6$, *i.e.,* $x^2(0) > 4\beta\left(\frac{1}{d}\right)$ *strictly*, from continuity we have that $x(\tau) \geq \sqrt{4\beta\left(\tau + \frac{1}{d}\right)} > 0$ for all $\tau \in [0, \zeta]$, for some $\zeta \in (0,1]$. Going forward we focus on $\tau \in [0, \zeta]$ when stating facts about $x(\tau)$, and eventually show that $\zeta = 1$ in Corollary J.23.

**Proposition J.12.** *Assuming* $\forall \tau \in [0, \zeta] : 0 \leq \frac{g(\tau)}{x^2} \leq 1$ *and* $x > 0$*, the solution of* $x(0) = 1$*,* $x'(\tau) = d\frac{\sqrt{x^2 - g(\tau)} - x}{2}$ *holds*

$$x(\tau) \in \left[ \sqrt{1 - d\frac{1 - \sqrt{1 - \mu}}{\mu} \int_0^\tau g(s)\,\mathrm{d}s}, \; \sqrt{1 - \frac{d}{2} \int_0^\tau g(s)\,\mathrm{d}s} \right] ,$$

*for* $\mu \triangleq \max\limits_{s \in [0, \zeta]} \dfrac{g(s)}{x(s)^2}$.

*Proof.* Let $0 \leq \mu \leq 1$ such that $\forall \tau \in [0, \zeta] : 0 \leq \frac{g(\tau)}{x^2} \leq \mu \leq 1$. From Claim J.10 we have:

$$1 - \frac{1 - \sqrt{1 - \mu}}{\mu} \frac{g(\tau)}{x^2} \leq \sqrt{1 - \frac{g(\tau)}{x^2}} \leq 1 - \frac{g(\tau)}{2x^2}$$

$$\left( 1 - \frac{1 - \sqrt{1 - \mu}}{\mu} \frac{g(\tau)}{x^2} \right) x \leq x\sqrt{1 - \frac{g(\tau)}{x^2}} \leq \left( 1 - \frac{g(\tau)}{2x^2} \right) x$$

$$x - \frac{1 - \sqrt{1 - \mu}}{\mu} \frac{g(\tau)}{x} \leq \sqrt{x^2 - g(\tau)} \leq x - \frac{g(\tau)}{2x}$$

$$- \frac{1 - \sqrt{1 - \mu}}{\mu} \frac{g(\tau)}{x} \leq \sqrt{x^2 - g(\tau)} - x \leq - \frac{g(\tau)}{2x}$$

$$- d\frac{1 - \sqrt{1 - \mu}}{2\mu} \frac{g(\tau)}{x} \leq d\frac{\sqrt{x^2 - g(\tau)} - x}{2} \leq - \frac{dg(\tau)}{4x}$$

$$\frac{\left( \sqrt{1 - \mu} - 1 \right) dg(\tau)}{2\mu x(\tau)} \leq x'(\tau) \leq \frac{-dg(\tau)}{4x(\tau)}$$

$$\frac{\left( \sqrt{1 - \mu} - 1 \right) dg(\tau)}{2\mu} \leq x(\tau) x'(\tau) \leq \frac{-dg(\tau)}{4} ,$$

where the last transition is valid since $x > 0$. From Remark J.9, we know that the solution to the ODE $x(\tau) x'(\tau) = -f(x)$, $x(0) = 1$ is $x(\tau) = \sqrt{1 - 2 \int_0^\tau f(s)\,\mathrm{d}s}$. Put differently this means $x(\tau) = \sqrt{1 + 2 \int_0^\tau x(s) x'(s)\,\mathrm{d}s}$ (when $x(0) = 1$). We aim to plug this into the inequalities, so we now achieve the required form:

$$\int_0^\tau \frac{\left( \sqrt{1 - \mu} - 1 \right) dg(s)}{2\mu}\,\mathrm{d}s \leq \int_0^\tau x(s) x'(s)\,\mathrm{d}s \leq \int_0^\tau \frac{-dg(s)}{4}\,\mathrm{d}s$$

$$1 - 2 \int_0^\tau \frac{d}{2} \frac{\left( 1 - \sqrt{1 - \mu} \right)}{\mu} g(s)\,\mathrm{d}s \leq 1 + 2 \int_0^\tau x(s) x'(s)\,\mathrm{d}s \leq 1 - 2 \int_0^\tau \frac{d}{4} g(s)\,\mathrm{d}s$$

$$1 - d\frac{1 - \sqrt{1 - \mu}}{\mu} \int_0^\tau g(s)\,\mathrm{d}s \leq 1 + 2 \int_0^\tau x(s) x'(s)\,\mathrm{d}s \leq 1 - \frac{d}{2} \int_0^\tau g(s)\,\mathrm{d}s .$$

Denoting $A \triangleq d \int_0^\tau g(s)\,\mathrm{d}s \geq 0$, note that for the LHS,

$$1 - \frac{1 - \sqrt{1 - \mu}}{\mu} A = \frac{1}{\mu} \left( \mu - A + A\sqrt{1 - \mu} \right) \geq \frac{1}{\mu} (\mu - A + A(1 - \mu)) = 0 .$$

Hence it is legal to take a square root:

$$\sqrt{1 - d\frac{1 - \sqrt{1 - \mu}}{\mu} \int_0^\tau g(s)\,\mathrm{d}s} \leq \sqrt{1 + 2 \int_0^\tau x(s) x'(s)\,\mathrm{d}s} \leq \sqrt{1 - \frac{d}{2} \int_0^\tau g(s)\,\mathrm{d}s} .$$

Finally, plugging in $x(\tau) = \sqrt{1 + 2 \int_0^\tau x(s) x'(s)\,\mathrm{d}s}$, we get:

$$\sqrt{1 - d\frac{1 - \sqrt{1 - \mu}}{\mu} \int_0^\tau g(s)\,\mathrm{d}s} \leq x(\tau) \leq \sqrt{1 - \frac{d}{2} \int_0^\tau g(s)\,\mathrm{d}s} .$$

$\square$

**Proposition J.13.** *For $d \geq 100{,}000$ and $\tau \in [0, 1]$, $0 \leq d \int_0^\tau g(s)\,\mathrm{d}s \leq 1.5821$.*

*Proof.* For $\tau \in [0, 1]$ and $d \geq 3$, $\tau + \frac{1}{d} < \frac{1}{\ln 2}$. We get from Claim J.7 that $\beta$ is positive and thus $d \int_0^\tau g(s)\,\mathrm{d}s = 4d \int_0^\tau \beta\left(s + \frac{1}{d}\right)\mathrm{d}s \geq 0$. For the right side inequality, we have:

$$d \int_0^\tau g(s)\,\mathrm{d}s = 4d \int_0^\tau \beta\left(s + \frac{1}{d}\right)\mathrm{d}s$$

$$[\beta \geq 0] \leq 4d \int_0^1 \beta\left(s + \frac{1}{d}\right)\mathrm{d}s$$

$$= 4d \int_0^1 \left(\frac{\left((ds + 1 - 1)\, 2^{-1/d} - (ds + 1 - 2)\right) 2^{(5 - 2ds - 2)/d}}{d}\right)\mathrm{d}s$$

$$= 4 \cdot 2^{3/d}\left[\int_0^1 2^{-2s}\mathrm{d}s - d\left(1 - 2^{-1/d}\right)\int_0^1 s 2^{-2s}\mathrm{d}s\right]$$

$$= 4 \cdot 2^{3/d}\left[\left[-\frac{2^{-2s}}{\ln 4}\right]\Big|_0^1 - d\left(1 - 2^{-1/d}\right)\left[-\frac{2^{-2s}(s \ln 4 + 1)}{\ln^2 4}\right]\Big|_0^1\right]$$

$$= 4 \cdot 2^{2/d}\left[2^{1/d}\frac{3}{4\ln 4} - d\left(2^{1/d} - 1\right)\left[\frac{4 - (\ln 4 + 1)}{4\ln^2 4}\right]\right].$$

From Claim J.5 we know that $d\left(2^{1/d} - 1\right) \geq \ln 2$, so:

$$d \int_0^\tau g(s)\,\mathrm{d}s \leq 4d \int_0^1 \beta\left(s + \frac{1}{d}\right)\mathrm{d}s \leq 4 \cdot 2^{2/d}\left[2^{1/d}\frac{3}{4\ln 4} - \ln 2\left[\frac{3 - \ln 4}{4\ln^2 4}\right]\right]$$

$$[d \geq 100{,}000] \leq 4 \cdot 2^{2/100000}\left[2^{1/100000}\frac{3}{4\ln 4} - \ln 2\left[\frac{3 - \ln 4}{4\ln^2 4}\right]\right] \leq 1.5821.$$

$\square$

**Claim J.14.** $\frac{1 - \sqrt{1-x}}{x} - \frac{1}{2} \leq \frac{x}{2}$ *for $x \in (0, 1]$.*

*Proof.* Note that

$$\frac{1 - \sqrt{1-x}}{x} = \frac{1 - \sqrt{1-x}}{x}\frac{1 + \sqrt{1-x}}{1 + \sqrt{1-x}} = \frac{x}{x\left(1 + \sqrt{1-x}\right)},$$

so we define $a(x) \triangleq \frac{1}{1 + \sqrt{1-x}}$.

This function is monotonically increasing, continuous and convex for $x \in [0, 1]$:

$$a'(x) = \frac{\frac{1}{2\sqrt{1-x}}}{\left(1 + \sqrt{1-x}\right)^2},$$

$$a''(x) = \frac{-1 \cdot \left(2\frac{-1}{2\sqrt{1-x}}\left(1 + \sqrt{1-x}\right)^2 + 2\left(1 + \sqrt{1-x}\right)\frac{-1}{2\sqrt{1-x}} \cdot 2\sqrt{1-x}\right)}{4(1-x)\left(1 + \sqrt{1-x}\right)^4}$$

$$= \frac{\frac{1}{\sqrt{1-x}}\left(1 + \sqrt{1-x}\right)^2 + 2\left(1 + \sqrt{1-x}\right)}{4(1-x)\left(1 + \sqrt{1-x}\right)^4} \geq 0,$$

and thus from convexity we have for $x \in (0, 1]$:

$$\frac{1 - \sqrt{1-x}}{x} = a(x) \leq (1-x)\,a(0) + x a(1) = (1-x)\frac{1}{2} + x = \frac{1}{2} + \frac{1}{2}x$$

$$\implies 2\left(\frac{1 - \sqrt{1-x}}{x} - \frac{1}{2}\right) \leq x.$$

$\square$

**Proposition J.15.** *For $d \geq 100{,}000$, $\mu \triangleq \max_{s \in [0,\zeta]} \frac{g(s)}{x(s)^2} \leq \frac{19.158}{d}$.*

*Proof.* From Remark J.11 we note that $x(\tau)$ is positive, and since $\beta\left(\tau + \frac{1}{d}\right) > 0$, we know $x(\tau)$ is decreasing in $[0, \zeta]$ (Claim J.7), and thus the minimum of $x(\tau)^2$ is $x(\zeta)^2$, and $x(\zeta)^2 \leq x(0)^2 = 1$. Applying the upper bound of $g(s)$ from Claim J.8, we get

$$\mu \triangleq \max_{s \in [0,\zeta]} \frac{g(s)}{x(s)^2} \leq \frac{\max_{s \in [0,\zeta]} g(s)}{\min_{s \in [0,\zeta]} x(s)^2} = \frac{4\frac{2^{3/d}}{d}}{x(\zeta)^2} \, .$$

From Proposition J.12 we know that $\sqrt{1 - d\frac{1 - \sqrt{1-\mu}}{\mu} \int_0^\zeta g(s)\, \mathrm{d}s} \leq x(\zeta)$.

Squaring both (positive) sides, substituting and denoting $A = d \int_0^\zeta g(s)\, \mathrm{d}s$, we get

$$1 - \frac{1 - \sqrt{1-\mu}}{\mu} A \leq x(\zeta)^2 \, .$$

Note that $A \geq 0$ (from Proposition J.13), and that $f(\mu) \triangleq \frac{1 - \sqrt{1-\mu}}{\mu}$ is increasing, since using Claim J.10, we have $f'(\mu) = \frac{2 - \mu - 2\sqrt{1-\mu}}{2\mu^2 \sqrt{1-\mu}} \geq \frac{2 - \mu - 2(1 - \mu/2)}{2\mu^2 \sqrt{1-\mu}} = 0$. Combining these and $\mu \leq \frac{4\frac{2^{3/d}}{d}}{x(\zeta)^2}$, we get that,

$$1 - \frac{1 - \sqrt{1 - \frac{4\frac{2^{3/d}}{d}}{x(\zeta)^2}}}{4\frac{2^{3/d}}{d}/x(\zeta)^2} A \leq 1 - \frac{1 - \sqrt{1-\mu}}{\mu} A \leq x(\zeta)^2$$

$$\frac{4\frac{2^{3/d}}{d}}{x(\zeta)^2} - A + A\sqrt{1 - \frac{4\frac{2^{3/d}}{d}}{x(\zeta)^2}} \leq 4\frac{2^{3/d}}{d}$$

$$A\sqrt{1 - \frac{4\frac{2^{3/d}}{d}}{x(\zeta)^2}} \leq -4\frac{2^{3/d}}{d}\left(\frac{1}{x(\zeta)^2} - 1\right) + A \, .$$

For simplicity denote $z = \frac{1}{x(\zeta)^2}$, $r = 4\frac{2^{3/d}}{d}$. Recall that $z \geq 1$, and we are looking for an upper bound for it, so we can have a lower bound for $x(\zeta)^2$. We have:

$$A^2 (1 - rz) \leq (-r(z-1) + A)^2 = r^2 (z-1)^2 - 2Ar(z-1) + A^2$$
$$-A^2 rz \leq r^2 z^2 - 2r^2 z + r^2 - 2Arz + 2Ar$$
$$0 \leq rz^2 + \left(A^2 - 2A - 2r\right)z + 2A + r \, .$$

Finding the roots, $z_{1,2} = \dfrac{2r + A(2 - A) \pm \sqrt{(2r + A(2 - A))^2 - 4r(2A + r)}}{2r}$.

Since we are looking for an upper bound, we care about the smaller root:

$$z \leq \frac{2r + A(2 - A) - \sqrt{4r^2 + 4rA(2 - A) + A^2 (2 - A)^2 - 8rA - 4r^2}}{2r}$$

$$= \frac{2r + A(2 - A) - \sqrt{-4rA^2 + A^2 (2 - A)^2}}{2r} = 1 + \frac{A(2 - A)}{2r}\left(1 - \sqrt{1 - \frac{4r}{(2 - A)^2}}\right) \, .$$

For $d \geq 100{,}000$, $\frac{4r}{(2-A)^2} = \frac{4\cdot 4 \frac{2^{3/d}}{d}}{(2-A)^2} \leq \frac{4\cdot 4\cdot \frac{2^{3/100000}}{100000}}{(2-1.5821)^2} \leq 10^{-3} < 1$ , (we used Proposition J.13), so we can apply Claim J.14:

$$z \leq 1 + \frac{A(2-A)}{2r}\left(\frac{4r}{2(2-A)^2}\left(\frac{4r}{(2-A)^2}+1\right)\right) = 1 + \frac{A}{2-A} + \frac{4Ar}{(2-A)^3}$$

$$\left[\begin{array}{c} d \geq 100{,}000 \\ \Longrightarrow r \leq 4.1\cdot 10^{-5} \end{array}\right] \leq 1 + \frac{A}{2-A} + 4\cdot 4.1\cdot 10^{-5}\frac{A}{(2-A)^3}$$

$$\left[A \leq 1.5821,\ \text{J.13}\right] \leq 1 + \frac{1.5821}{0.4179} + 4\cdot 4.1\cdot 10^{-5}\cdot \frac{1.5821}{0.4179^3} \leq 4.7894 \,.$$

Then, for $d \geq 10^5$: $\dfrac{1}{x(\zeta)^2} \leq 4.7894 \implies \mu \leq 4.7894\cdot 4\dfrac{2^{3/d}}{d} \leq \dfrac{19.1576\cdot 2^{3/10^5}}{d} \leq \dfrac{19.158}{d}.$ $\quad\square$

**Proposition J.16.** *For $d \geq 100{,}000$, we have $\forall \tau \in [0,\zeta]$, $\left|x(\tau) - \sqrt{1 - \frac{d}{2}\int_0^\tau g(s)\,\mathrm{d}s}\right| \leq \frac{33.1539}{d}$.*

*Proof.* Denote $A = d\int_0^\tau g(s)\,\mathrm{d}s$. We know that

$$\sqrt{1 - \frac{\left(1-\sqrt{1-\mu}\right)}{\mu}A} \leq x(\tau) \leq \sqrt{1 - \frac{1}{2}A}$$

$$\left|x(\tau) - \sqrt{1 - \frac{1}{2}A}\right| \leq \sqrt{1 - \frac{1}{2}A} - \sqrt{1 - \frac{\left(1-\sqrt{1-\mu}\right)}{\mu}A}$$

$$\leq \frac{1 - \frac{1}{2}A - 1 + \frac{\left(1-\sqrt{1-\mu}\right)}{\mu}A}{\sqrt{1-\frac{1}{2}A} + \sqrt{1 - \frac{\left(1-\sqrt{1-\mu}\right)}{\mu}A}}$$

$$\leq \frac{\left(\frac{\left(1-\sqrt{1-\mu}\right)}{\mu} - \frac{1}{2}\right)A}{\sqrt{1-\frac{1}{2}A} + \sqrt{1 - \frac{\left(1-\sqrt{1-\mu}\right)}{\mu}A}}$$

$$\leq \frac{A}{\sqrt{1-\frac{1}{2}A}}\left(\frac{\left(1-\sqrt{1-\mu}\right)}{\mu} - \frac{1}{2}\right).$$

From Proposition J.13 we have for $d \geq 100{,}000$ that $A \leq 1.5821$, then $\frac{A}{\sqrt{1-\frac{1}{2}A}} \leq \frac{1.5821}{\sqrt{1-\frac{1.5821}{2}}} \leq 3.4611$.

We further know from Claim J.14 that $\frac{1-\sqrt{1-\mu}}{\mu} - \frac{1}{2} \leq \frac{\mu}{2}$ for $\mu \in (0,1]$.

Combining these and applying Proposition J.15, we get:

$$\left|x(\tau) - \sqrt{1 - \frac{1}{2}A}\right| \leq 3.4611\frac{\mu}{2} \leq \frac{3.4611\cdot 19.158}{2d} \leq \frac{33.1539}{d} \,.$$

$\square$

**Claim J.17.** $\forall x \in [0,1]:\ 2^x \leq 1 + x.$

*Proof.* $2^x$ is convex, so we get in $[0,1]$:

$$2^x \leq (1-x)2^0 + x2^1 = 1 - x + 2x = 1 + x \,.$$

$\square$

**Proposition J.18.** *For $d \geq 100,000$, We have $\left| \tilde{x}\left(\tau\right) - \sqrt{1 - \frac{d}{2}\int_0^\tau g\left(s\right)\mathrm{d}s} \right| \leq \frac{5.8283}{d}$.*

*Proof.* We are reminded of the following:

$$\tilde{x}\left(\tau\right) = \sqrt{1 - \frac{1}{\ln 4} + 4^{-\tau}\left(\frac{1}{\ln 4} - \tau\right)},$$

$$\beta\left(\tau\right) = \frac{\left(\left(d\tau - 1\right)2^{-1/d} - \left(d\tau - 2\right)\right)2^{(5-2d\tau)/d}}{d},$$

$$g\left(\tau\right) = 4\beta\left(\tau + \frac{1}{d}\right) = 4\frac{\left(d\tau 2^{-1/d} - \left(d\tau - 1\right)\right)2^{(3-2d\tau)/d}}{d}.$$

Define $A\left(\tau\right) \triangleq \frac{d}{2}\int_0^\tau g\left(s\right)\mathrm{d}s$, $B\left(\tau\right) = \frac{1}{\ln 4} - 4^{-\tau}\left(\frac{1}{\ln 4} - \tau\right)$.

We have $A\left(0\right) = B\left(0\right) = 0$, and $A\left(\tau\right) - B\left(\tau\right)$ is non negative and increasing for $\tau \in [0, 1]$:

$$\frac{\mathrm{d}\left(A\left(\tau\right) - B\left(\tau\right)\right)}{\mathrm{d}\tau} = \frac{d}{2}g\left(\tau\right) - 4^{-\tau}\left(2 - \tau\ln 4\right)$$

$$= \frac{d}{2}4\frac{\left(d\tau 2^{-1/d} - \left(d\tau - 1\right)\right)2^{(3-2d\tau)/d}}{d} - 4^{-\tau}\left(2 - \tau\ln 4\right)$$

$$= 4^{-\tau}\left(2\left(2^{3/d} - 1\right) + \tau\left(\ln 4 - 2\cdot 2^{3/d}d\left(1 - 2^{-1/d}\right)\right)\right).$$

If we assume $\ln 4 \geq 2\cdot 2^{3/d}d\left(1 - 2^{-1/d}\right)$, then the derivative is in fact positive and we are done. If we assume the opposite we have:

$$\frac{\mathrm{d}\left(A\left(\tau\right) - B\left(\tau\right)\right)}{\mathrm{d}\tau} = 4^{-\tau}\left(2\left(2^{3/d} - 1\right) - \tau\left(2\cdot 2^{3/d}d\left(1 - 2^{-1/d}\right) - \ln 4\right)\right)$$

$$[\tau \in [0, 1]] \geq 4^{-\tau}\left(2\left(2^{3/d} - 1\right) - \left(2\cdot 2^{3/d}d\left(1 - 2^{-1/d}\right) - \ln 4\right)\right)$$

$$= 4^{-\tau}\left(2\cdot 2^{3/d}\left(1 - d\left(1 - 2^{-1/d}\right)\right) - 2 + \ln 4\right)$$

$$[\text{J.6}] \geq 4^{-\tau}\left(2\cdot 2^{3/d}\left(1 - \ln 2\right) - 2 + \ln 4\right)$$

$$\geq 4^{-\tau}\left(2\left(1 - \ln 2\right) - 2 + \ln 4\right) = 4^{-\tau}\left(2 - 2\ln 2 - 2 + \ln 4\right) = 0.$$

This means that

$$0 \leq A\left(\tau\right) - B\left(\tau\right) \leq A\left(1\right) - B\left(1\right).$$

In Proposition J.13 we saw that:

$$d\int_0^1 g\left(s\right)\mathrm{d}s \leq 4\cdot 2^{2/d}\left[2^{1/d}\frac{3}{4\ln 4} - \ln 2\left[\frac{3 - \ln 4}{4\ln^2 4}\right]\right]$$

$$\implies A\left(1\right) \leq 2\cdot 2^{2/d}\left[2^{1/d}\frac{3}{4\ln 4} - \ln 2\left[\frac{3 - \ln 4}{4\ln^2 4}\right]\right]$$

$$[\text{J.17}, d \geq 2, \ln 4 = 2\ln 2] \leq 2\left(1 + \frac{2}{d}\right)\left[\frac{3\left(1 + \frac{1}{d}\right)}{8\ln 2} - \ln 2\left[\frac{3 - \ln 4}{16\ln^2 2}\right]\right]$$

$$= 2\left(1 + \frac{2}{d}\right)\left[\frac{6 + \frac{6}{d}}{16\ln 2} - \frac{3 - \ln 4}{16\ln 2}\right] = \left(1 + \frac{2}{d}\right)\left[\frac{3 + 2\ln 2 + \frac{6}{d}}{8\ln 2}\right].$$

Subtracting $B(1)$ we get:

$$A(1) - B(1) \leq \left(1 + \frac{2}{d}\right)\left[\frac{3 + 2\ln 2 + \frac{6}{d}}{8\ln 2}\right] - \left(\frac{1}{\ln 4} - \frac{1}{4}\left(\frac{1}{\ln 4} - 1\right)\right)$$

$$= \left(1 + \frac{2}{d}\right)\left[\frac{3 + 2\ln 2 + \frac{6}{d}}{8\ln 2}\right] - \frac{3 + 2\ln 2}{8\ln 2}$$

$$= \frac{6}{d \cdot 8\ln 2} + \frac{2}{d}\left[\frac{3 + 2\ln 2 + \frac{6}{d}}{8\ln 2}\right]$$

$$[d \geq 100{,}000] \leq \frac{2.6641}{d},$$

leading to

$$0 \leq A(\tau) - B(\tau) \leq \frac{2.6641}{d}.$$

Now note that,

$$\tilde{x}(\tau) - \sqrt{1 - \frac{d}{2}\int_0^\tau g(s)\,\mathrm{d}s} = \sqrt{1 - B(\tau)} - \sqrt{1 - A(\tau)} = \frac{1 - B(\tau) - (1 - A(\tau))}{\sqrt{1 - B(\tau)} + \sqrt{1 - A(\tau)}}$$

$$= \frac{A(\tau) - B(\tau)}{\sqrt{1 - A(\tau)} + \sqrt{1 - B(\tau)}}.$$

Since $0 \leq A(\tau) - B(\tau)$, we have $\tilde{x}(\tau) \geq \sqrt{1 - \frac{d}{2}\int_0^\tau g(s)\,\mathrm{d}s}$. Lower bounding the denominator:

$$\sqrt{1 - A(\tau)} + \sqrt{1 - B(\tau)} \geq \sqrt{1 - B(\tau)} = \sqrt{1 - \frac{1}{\ln 4} + 4^{-\tau}\left(\frac{1}{\ln 4} - \tau\right)}$$

$$\geq \sqrt{1 - \frac{1}{\ln 4} + 4^{-1}\left(\frac{1}{\ln 4} - 1\right)} \geq 0.4571,$$

and so,

$$0 \leq \tilde{x}(\tau) - \sqrt{1 - \frac{d}{2}\int_0^\tau g(s)\,ds} \leq \frac{2.6641}{0.4571d} \leq \frac{5.8283}{d}.$$

$\square$

**Proposition J.19.** *For $d \geq 100{,}000$, we have $\forall \tau \in [0, \zeta]$, $|\tilde{x}(\tau) - x(\tau)| \leq \frac{38.9822}{d}$.*

*Proof.* From Proposition J.18 and Proposition J.16:

$$|\tilde{x}(\tau) - x(\tau)| = \left|\tilde{x}(\tau) - \sqrt{1 - \frac{d}{2}\int_0^\tau g(s)\,ds} + \sqrt{1 - \frac{d}{2}\int_0^\tau g(s)\,ds} - x(\tau)\right|$$

$$\leq \left|\tilde{x}(\tau) - \sqrt{1 - \frac{d}{2}\int_0^\tau g(s)\,ds}\right| + \left|\sqrt{1 - \frac{d}{2}\int_0^\tau g(s)\,ds} - x(\tau)\right|$$

$$\leq \frac{5.8283}{d} + \frac{33.1539}{d} = \frac{38.9822}{d}.$$

$\square$

**Corollary J.20.** *For $d \geq 100{,}000$, we have $\forall \tau \in [0, \zeta]$, $x(\tau) \geq 0.4567$.*

*Proof.* Note that $\tilde{x}(\tau) = \sqrt{1 - \frac{1}{\ln 4} + 4^{-\tau}\left(\frac{1}{\ln 4} - \tau\right)}$ is decreasing for $\tau \in [0, 1]$:

$$\tilde{x}'(\tau) = \frac{-4^{-\tau}(2 - \tau\ln 4)}{2\sqrt{1 - \frac{1}{\ln 4} + 4^{-\tau}\left(\frac{1}{\ln 4} - \tau\right)}} \leq 0.$$

Note that it is lowest at $\tau = 1$. Combined with $|x(\tau) - \tilde{x}(\tau)| \le \frac{38.9822}{d}$, we get:

$$x \ge \sqrt{1 - \frac{1}{\ln 4} + \frac{1}{4}\left(\frac{1}{\ln 4} - 1\right)} - \frac{38.9822}{d} \ge 0.4571 - \frac{38.9822}{d}$$

$$[d \ge 100{,}000] \ge 0.4571 - \frac{38.9822}{100{,}000} \ge 0.4567.$$

$\square$

**Claim J.21.** For $a \ge b \ge 0$, $\sqrt{a-b} \ge \sqrt{a} - \sqrt{b}$.

*Proof.*

$$\sqrt{a-b} = \sqrt{\left(\sqrt{a} - \sqrt{b}\right)\left(\sqrt{a} + \sqrt{b}\right)} \ge \sqrt{\left(\sqrt{a} - \sqrt{b}\right)^2} = \sqrt{a} - \sqrt{b}.$$

$\square$

**Proposition J.22** (Existence of the solution to the ODE). *For $d \ge 100{,}000$, we have $x^2(\tau) \ge 4\beta\left(\tau + \frac{1}{d}\right), \forall \tau \in [0,1]$.*

*Proof.* Note the following for $\tau = \zeta$:

$$x^2(\zeta) \ge 0.4567^2 > 0.2 > 4\frac{2}{d} > 4\beta\left(\zeta + \frac{1}{d}\right),$$

for $d \ge 100{,}000$, from Claim J.8 and Corollary J.20. From continuity and the strict inequality, there exists $\delta > 0$ such that $\forall \tau \in [\zeta, \zeta + \delta]$, $x^2(\tau) \ge 4\beta\left(\tau + \frac{1}{d}\right)$. Observing the definition of the ODE Eq. (14), we have the following for all $\tau \in [\zeta, \zeta + \delta]$:

$$x'(\tau) = d\frac{\sqrt{x^2 - 4\beta\left(\tau + \frac{1}{d}\right)} - x}{2}$$

$$[\text{J.21}] \ge d\frac{x - 2\sqrt{\beta\left(\tau + \frac{1}{d}\right)} - x}{2} = -d\sqrt{\beta\left(\tau + \frac{1}{d}\right)} \ge -d\sqrt{\frac{2}{d}} = -\sqrt{2d}$$

$$\implies x(\tau) \ge x(\zeta) - \sqrt{2d}(\tau - \zeta) \ge 0.4567 - \sqrt{2d}(\tau - \zeta).$$

We can show that $\delta \ge \frac{0.3}{\sqrt{d}}$, by showing the solution must exist at $\tau = \zeta + \frac{0.3}{\sqrt{d}}$:

$$x\left(\zeta + \tilde{\delta}\right) \ge 0.4567 - \sqrt{2d}\tilde{\delta} \ge 2\sqrt{\frac{2}{d}} > \sqrt{4\beta\left(\zeta + \tilde{\delta} + \frac{1}{d}\right)}$$

$$\iff \tilde{\delta} \le \frac{1}{\sqrt{2d}}\left(0.4567 - 2\sqrt{\frac{2}{d}}\right) \quad\quad \Longleftarrow \quad\quad \tilde{\delta} = \frac{0.3}{\sqrt{d}}.$$

Hence, we have that $x^2(\tau) \ge 4\beta\left(\tau + \frac{1}{d}\right)$ for $\tau \in \left[0, \zeta + \frac{0.3}{\sqrt{d}}\right]$, and thus we can apply all previous claims replacing $\zeta$ with $\zeta + \frac{0.3}{\sqrt{d}}$, and specifically,

$$x^2(\zeta + \frac{0.3}{\sqrt{d}}) \ge 0.4567^2 > 0.2 > 4\frac{2}{d} > 4\beta\left(\zeta + \frac{0.3}{\sqrt{d}} + \frac{1}{d}\right),$$

Which allows repeating all the previous steps without alteration. After repeating $\left\lceil\frac{\sqrt{d}}{0.3}\right\rceil$ times, we get that $x^2(\tau) \ge 4\beta\left(\tau + \frac{1}{d}\right)$ for $\tau \in [0,1]$, concluding the proof. $\square$

**Corollary J.23.** *All previous propositions, and specifically Proposition J.19 and Corollary J.20, apply $\forall \tau \in [0,1]$ (indicating $\zeta = 1$).*

**Proposition J.24.** *For $d \geq 100{,}000$, $L \triangleq \max_{x,\tau \in [0,1]} \left| \frac{\mathrm{d}}{\mathrm{d}x} f(\tau, x) \right| \leq 4.7955$.*

*Proof.* $L$, the Lipschitz constant of $f$, is given by

$$L \triangleq \max_{x,\tau \in [0,1]} \left| \frac{\mathrm{d}}{\mathrm{d}x} f(\tau, x) \right| .$$

We have:

$$\frac{\mathrm{d}}{\mathrm{d}x} f(\tau, x) = \frac{\mathrm{d}}{\mathrm{d}x} \left[ d \frac{\sqrt{x^2 - 4\beta \left(\tau + \frac{1}{d}\right)} - x}{2} \right] = \frac{d}{2} \left( \frac{x}{\sqrt{x^2 - 4\beta \left(\tau + \frac{1}{d}\right)}} - 1 \right) .$$

Assume that $x \geq x_{\min}$, $\tau \in [0,1]$. from $d \geq 3$, $\tau + \frac{1}{d} \leq \frac{1}{\ln 2}$. From Claim J.7, we get $\beta \left(\tau + \frac{1}{d}\right) \geq 0$. This means that $\frac{\mathrm{d}}{\mathrm{d}x} f(\tau, x) \geq 0$. So

$$L = \max_{x,\tau \in [0,1]} \frac{\mathrm{d}}{\mathrm{d}x} f(\tau, x) .$$

For any fixed $x$, the maximum $\beta \left(\tau + \frac{1}{d}\right)$ will maximize $L$. From Claim J.7, we know that $\beta$ is decreasing with $\tau$, so to maximize $L$, $\tau = 0$. To maximize $\frac{d}{2} \left( \frac{x}{\sqrt{x^2 - 4\beta \left(\frac{1}{d}\right)}} - 1 \right)$, note that this function is decreasing with respect to $x$:

$$\frac{\mathrm{d}}{\mathrm{d}x} \left[ \frac{d}{2} \left( \frac{x}{\sqrt{x^2 - 4\beta \left(\frac{1}{d}\right)}} - 1 \right) \right] = \frac{d}{2} \left( \frac{\sqrt{x^2 - 4\beta \left(\frac{1}{d}\right)} - x \frac{x}{\sqrt{x^2 - 4\beta \left(\tau + \frac{1}{d}\right)}}}{x^2 - 4\beta \left(\frac{1}{d}\right)} \right)$$

$$= \frac{d}{2} \left( \frac{x^2 - 4\beta \left(\frac{1}{d}\right) - x^2}{\left(x^2 - 4\beta \left(\frac{1}{d}\right)\right)^{\frac{3}{2}}} \right) = \frac{d}{2} \left( \frac{-4\beta \left(\frac{1}{d}\right)}{\left(x^2 - 4\beta \left(\frac{1}{d}\right)\right)^{\frac{3}{2}}} \right) \leq 0 ,$$

hence the optimal $x$ is $x_{\min}$. We get that

$$L = \frac{d}{2} \left( \frac{x_{\min}}{\sqrt{x_{\min}^2 - 4\beta \left(\frac{1}{d}\right)}} - 1 \right) .$$

Now,

$$4\beta \left(\frac{1}{d}\right) = 4 \frac{\left((1-1) 2^{-1/d} - (1-2)\right) 2^{(5-2)/d}}{d} = 4 \frac{2^{3/d}}{d} ,$$

and applying Corollary J.20 we get:

$$L \leq \frac{d}{2} \left( \frac{0.4567}{\sqrt{0.4567^2 - 4 \frac{2^{3/d}}{d}}} - 1 \right)$$

$$[d \geq 100{,}000] \leq \frac{d}{2} \left( \frac{0.4567}{\sqrt{0.4567^2 - 4 \frac{2^{3/100000}}{d}}} - 1 \right) \leq \frac{d}{2} \left( \frac{0.4567}{\sqrt{0.4567^2 - \frac{4.0001}{d}}} - 1 \right)$$

$$\leq \frac{d}{2} \left( \frac{1}{\sqrt{1 - \frac{19.1783}{d}}} - 1 \right) = \frac{d}{2} \left( \frac{1}{\sqrt{1 - \frac{19.1783}{d}}} - 1 \right) \frac{\frac{1}{\sqrt{1 - \frac{19.1783}{d}}} + 1}{\frac{1}{\sqrt{1 - \frac{19.1783}{d}}} + 1}$$

$$= \frac{d}{2} \left( \frac{\frac{1}{1 - \frac{19.1783}{d}} - 1}{\frac{1}{\sqrt{1 - \frac{19.1783}{d}}} + 1} \right) \leq \frac{d}{2} \left( \frac{\frac{19.1783}{d}}{1 + 1} \right) = \frac{19.1783}{4} \frac{1}{1 - \frac{19.1783}{d}}$$

$$\leq \frac{19.1783}{4} \frac{1}{1 - \frac{19.1783}{100000}} \leq 4.7955 .$$

$\square$

**Proposition J.25.** *From $d \geq 100,000$, $M \triangleq \max_{\tau \in [0,1]} \left| \frac{d^2}{d\tau^2} x(\tau) \right| \leq 10.5027$*

*Proof.* $M$ is defined as an upper bound on the second derivative (absolute value) of $x(\tau)$ in the relevant interval:

$$M \triangleq \max_{\tau \in [0,1]} \left| \frac{d^2}{d\tau^2} x(\tau) \right| = \max_{\tau \in [0,1]} \left| \frac{d}{d\tau} f(\tau, x(\tau)) \right|.$$

We have:

$$\frac{d}{d\tau} f(\tau, x(\tau)) = \frac{\partial}{\partial \tau} \left[ d \frac{\sqrt{x^2 - 4\beta(\tau + \frac{1}{d})} - x}{2} \right] + x'(\tau) \frac{\partial}{\partial x} \left[ d \frac{\sqrt{x^2 - 4\beta(\tau + \frac{1}{d})} - x}{2} \right]$$

$$= \frac{\partial}{\partial \tau} \left[ d \frac{\sqrt{x^2 - 4\beta(\tau + \frac{1}{d})} - x}{2} \right] + f(\tau, x(\tau)) \frac{\partial}{\partial x} \left[ d \frac{\sqrt{x^2 - 4\beta(\tau + \frac{1}{d})} - x}{2} \right].$$

For the first term:

$$\frac{\partial}{\partial \tau} \left[ d \frac{\sqrt{x^2 - 4\beta(\tau + \frac{1}{d})} - x}{2} \right] = \frac{d}{2} \left( \frac{-4}{2\sqrt{x^2 - 4\beta(\tau + \frac{1}{d})}} \right) \frac{\partial}{\partial \tau} \beta \left( \tau + \frac{1}{d} \right)$$

$$= - \frac{d}{\sqrt{x^2 - 4\beta(\tau + \frac{1}{d})}} \frac{\partial}{\partial \tau} \beta \left( \tau + \frac{1}{d} \right).$$

From Claim J.7 we know $\beta$ is positive and decreasing, so $\frac{d}{\sqrt{x^2 - 4\beta(\tau + \frac{1}{d})}}$ is maximized at $\tau = 0$; In addition $\frac{\partial}{\partial \tau} \beta(\tau + \frac{1}{d}) \leq 0$, and thus the entire expression is non negative. We know $\beta$ is convex, so the absolute value of the negative $\frac{\partial}{\partial \tau} \beta(\tau + \frac{1}{d})$ is also maximized at $\tau = 0$. All in all, the entire expression is maximized at $\tau = 0$, and is bounded by:

$$0 \leq - \frac{d}{\sqrt{x^2 - 4\beta(\frac{1}{d})}} \frac{\partial}{\partial \tau} \beta \left( \tau + \frac{1}{d} \right) \Big|_{\tau=0}$$

$$= - \frac{d}{\sqrt{x^2 - 4\beta(\frac{1}{d})}} \frac{1}{d} \left( d \left( 2^{-1/d} - 1 \right) 2^{\frac{5 - 2d(1/d)}{d}} \right.$$

$$\left. -2 \ln 2 \cdot 2^{\frac{5 - 2d(1/d)}{d}} \left( (d(1/d) - 1) 2^{-1/d} - (d(1/d) - 2) \right) \right)$$

$$= \frac{2^{3/d}}{\sqrt{x^2 - 4\beta(\frac{1}{d})}} \left( 2 \ln 2 + d \left( 1 - 2^{-1/d} \right) \right)$$

$$[J.6] \leq \frac{2^{3/d}}{\sqrt{x^2 - 4 \frac{2^{3/d}}{d}}} (2 \ln 2 + \ln 2)$$

$$\leq \frac{2^{3/100000}}{\sqrt{x^2 - 4 \frac{2^{3/100000}}{100000}}} (3 \ln 2)$$

$$\leq \frac{2.08}{\sqrt{x^2 - 4.1 \cdot 10^{-5}}}$$

$$[J.20] \leq \frac{2.08}{\sqrt{0.4567^2 - 4.1 \cdot 10^{-5}}} \leq 4.555.$$

Moving to the second term, we have from Proposition J.24,

$$0 \leq \frac{\partial}{\partial x} \left[ d \frac{\sqrt{x^2 - 4\beta(\tau + \frac{1}{d})} - x}{2} \right] \leq L = 4.7955.$$

Now we need to bound $f(\tau, x) = d\dfrac{\sqrt{x^2 - 4\beta\left(\tau + \frac{1}{d}\right)} - x}{2}$, which is always negative. From Claim J.7 we know $\beta$ is positive and decreasing, so its maximum, which minimizes this and thus maximizes its absolute value, is received at $\tau = 0$. We also know that $f(0, x)$ increases with $x$ (see the beginning of the proof for Proposition J.24), so its absolute value decreases with $x$. Utilizing these facts we get:

$$0 \geq d\frac{\sqrt{x^2 - 4\beta\left(\tau + \frac{1}{d}\right)} - x}{2} \geq d\frac{\sqrt{x^2 - 4\beta\left(\frac{1}{d}\right)} - x}{2}$$

$$[d \geq 100{,}000 \Rightarrow x \geq 0.4567] \geq d\frac{\sqrt{0.4567^2 - 4\frac{2^{3/d}}{d}} - 0.4567}{2}$$

$$\geq d\frac{\sqrt{0.4567^2 - 4\frac{2^{3/100{,}000}}{d}} - 0.4567}{2}$$

$$\geq \frac{0.4567}{2}\frac{\sqrt{1 - \frac{19.1782}{d}} - 1}{\frac{1}{d}}$$

$$= -\frac{0.4567}{2} \cdot 19.1782\frac{1 - \sqrt{1 - \frac{19.1782}{d}}}{\frac{19.1782}{d}}$$

$$[\text{J.14}] \geq -\frac{0.4567}{2} \cdot 19.1782\left(\frac{1}{2} + \frac{19.1782}{2d}\right)$$

$$[d \geq 100{,}000] \geq -\frac{0.4567}{2} \cdot 19.1782\left(\frac{1}{2} + \frac{19.1782}{2 \cdot 100000}\right) \geq -2.1901\,.$$

To summarize, we have $-2.1901 \leq f(\tau, x) \leq 0$, and thus

$$\frac{\mathrm{d}}{\mathrm{d}\tau}f(\tau, x(\tau)) \in [0, 4.555] + [-2.1901, 0] \cdot [0, 4.7955] \subseteq [-10.5027, 4.555]\,,$$

and in absolute value,

$$M \leq 10.5027\,.$$

$\qquad\square$

**Proposition J.26.** *For $d \geq 100{,}000$, $k \in \{2, \ldots, d\}$ for which the sequence $(x)_k$ exists, i.e., $x_{j-1}^2 - 4\beta\left(\frac{j}{d}\right) \geq 0$ for all $2 \leq j \leq k$,*

$$|x_k - \tilde{x}_k| \leq \frac{170.4}{d}\,,$$

*where $\tilde{x}_k = \sqrt{1 - \frac{1}{\ln 4} + 4^{-\frac{k}{d}}\left(\frac{1}{\ln 4} - \frac{k}{d}\right)}$.*

*Proof.* As noted in App. J.1.1, the sequence $(x)_k$ are Euler's iterates for the ODE in Eq. (14). Using the global truncation error of Euler's method [9, chapter 6.2] we get:

$$\left|x_k - x\left(\frac{k}{d}\right)\right| \leq \frac{hM}{2L}\left(\exp\left(L\left(\frac{k}{d} - 0\right)\right) - 1\right) \leq \frac{M}{2Ld}\left(e^L - 1\right),$$

where

$$L = \max_{x, \tau \in [0,1]}\left|\frac{\mathrm{d}}{\mathrm{d}x}f(\tau, x)\right| \text{ (where } \tau \text{ is treated as a constant)},$$

$$M = \max_{\tau \in [0,1]}\left|\frac{\mathrm{d}^2}{\mathrm{d}\tau^2}x(\tau)\right| = \left|\frac{\mathrm{d}}{\mathrm{d}\tau}f(\tau, x(\tau))\right|\,.$$

For $d \geq 100{,}000$ we have $L \leq 4.7955$ from Proposition J.24, and Proposition J.25 gives $M \leq 10.5027$.

So in total

$$\left| x_k - x\left(\frac{k}{d}\right) \right| \leq \frac{10.5027}{2 \cdot 4.7955d} \cdot \left(e^{4.7955} - 1\right) \leq \frac{131.3685}{d} .$$

Combining this with Proposition J.19, we get

$$
\begin{aligned}
\left| x_k - \tilde{x}_k \right| = \left| x_k - \tilde{x}\left(\frac{k}{d}\right) \right| &= \left| x_k - x\left(\frac{k}{d}\right) + x\left(\frac{k}{d}\right) - \tilde{x}\left(\frac{k}{d}\right) \right| \\
&\leq \left| x_k - x\left(\frac{k}{d}\right) \right| + \left| x\left(\frac{k}{d}\right) - \tilde{x}\left(\frac{k}{d}\right) \right| \\
&\leq \frac{131.3685}{d} + \frac{38.9822}{d} \leq \frac{170.4}{d} .
\end{aligned}
$$

$\square$

**Corollary J.27.** *For $d \geq 100{,}000$, $k \in \{2, \ldots, d\}$ for which the sequence $(x)_k$ exists, i.e., $x_{j-1}^2 - 4\beta\left(\frac{i}{d}\right) \geq 0$ for all $2 \leq j \leq k$: $x_k \geq 0.45$.*

*Proof.* $x_k$ is decreasing (Claim I.5), so $\forall k \in [d]$:

$$x_k \geq x_d \geq \tilde{x}_d - \frac{170.4}{d} = \sqrt{1 - \frac{1}{\ln 4} + 4^{-1}\left(\frac{1}{\ln 4} - 1\right)} - \frac{170.4}{d}$$

$$[d \geq 100{,}000] \geq 0.45 .$$

$\square$

**Proposition J.28** (Existence of the sequence)**.** *For $d \geq 100{,}000$, the sequence $(x)_k$ exists for all $k \in [d]$, i.e., $x_{j-1}^2 - 4\beta_j \geq 0$, for all $2 \leq j \leq d$.*

The proof is similar to that of Proposition J.22, but simpler. We first note that

$$x_1^2 = 1 > 4\frac{2}{d} > 4\beta\left(\frac{2}{d}\right) = 4\beta_2 ,$$

which means $x_2$ exists. Now assuming for some $k \in \{2, \ldots, d\}$, $x_k$ exists, then from Corollary J.27, $x_k \geq 0.45$, and we have

$$x_k^2 \geq 0.45^2 > 0.2 > 4\frac{2}{d} > 4\beta\left(\frac{k+1}{d}\right) = 4\beta_{k+1} ,$$

which means that $x_{k+1}$ exists, and by induction the proof is done.

