# OpenReview forum: "Are Greedy Task Orderings Better Than Random in Continual Linear Regression?"
_NeurIPS.cc/2025/Conference — NeurIPS 2025 poster_

### Official Review · Reviewer_Lb47 · 2025-07-01

**Clarity:** 2
**Significance:** 2
**Originality:** 2
**Rating:** 3
**Confidence:** 3

**Summary:**

The paper investigates the impact of task ordering strategies in continual learning for linear regression. It focuses on greedy task orderings that maximize dissimilarity between consecutive tasks. Using tools from the Kaczmarz method literature, the authors formalize these orderings and develop geometric and algebraic intuitions around them. The paper also provides theoretical bounds on the loss for greedy orderings in a simplified setting and constructs an adversarial task sequence to demonstrate potential failure modes.

**Questions:**

See weaknesses.

**Ethical Concerns:**

["NO or VERY MINOR ethics concerns only"]

**Final Justification:**

I have read the response and found the applicability of the theory to a more realistic setting is doubtful. But it makes some contribution theoretically, so I raise my score from 2 to 3.

**Quality:**

2

**Strengths And Weaknesses:**

Strengths

1. The paper provides a rigorous theoretical analysis of greedy task orderings in continual linear regression. It formalizes the concept of similarity-guided orderings using tools from the Kaczmarz method literature, offering new insights into the dynamics of task ordering in continual learning.

2. The paper introduces a hybrid task ordering strategy that combines greedy and random orderings is innovative. This approach leverages the strengths of both strategies. The hybrid strategy inherits the bounds of random orderings, making it a robust alternative.

3. The paper draws meaningful connections between continual learning, Kaczmarz methods, and other related areas such as projection methods and importance sampling.

Weakness

1. The study focuses solely on the linear regression setting, which is relatively simple and may not fully capture the complexities of recent advances in continual learning.

2. The empirical results are based on synthetic data and are very small-sized, which may not fully reflect the characteristics and challenges of real-world datasets. Conducting experiments on real-world data would provide a more comprehensive evaluation of the proposed methods and their practical utility.

3. Though the author mentioned this in the checklist, it is better to provide the sample code for better judgment of the implementation complexity.

---

> ### Author Rebuttal · Authors · 2025-07-30
>
> We thank the reviewer for their feedback. We are encouraged that they appreciated the contribution of our work to the understanding of the effect of task ordering, as well as the novelty of the suggested hybrid method and the discussion of meaningful connections to other fields.
>
> Before answering in detail, we note that the reviewer mostly pointed out as weaknesses the simplicity of our analytical setting (which is already discussed in the paper; e.g., lines 73-74) and the lack of experiments on “real-world” data. However, we analyzed a setting that is **widely used** in theoretical CL studies [e.g., 5, 21, 24, 26, 28, 29, 36, 48, 60; cited in our lines 72-74, 82-86] in order to better explain and characterize phenomena that were **already** observed empirically in more complicated models and datasets (as cited in our line 33).
> Moreover, even this simple setting allowed uncovering novel failure modes not previously observed—that would have been almost impossible to uncover in more complex models—and identifying a separation between greedy and random orderings in the effects of task repetition.
>
> Thus, we respectfully find the attached rating extremely harsh, not accurately reflecting the place of our paper and contributions in the **theoretical** continual learning literature.
>
> ---
>
> Below, we would like to comment on the concerns raised by the reviewer in more detail:
> 1. **Simplicity of the continual linear regression setting.**
> We agree with the reviewer that our (and frankly, any other) simple setting “may not fully capture the complexities of recent advances in CL”.
> However, theoretical progress is made gradually and always starts from the simplest analyzable models (as done in the papers we cited above). Our simple model already exhibits phenomena related to similarity-guided task orderings—including ones observed empirically in recent papers [e.g., 10, 54, 63, 50, 67, *1 below]—thus providing a good opportunity to better understand such phenomena analytically. See our paper’s lines 31-41 for a more detailed discussion.
> Is the reviewer aware of other theoretical papers that rigorously analyze these task ordering aspects in more complicated models?
> 2. **Experiments use synthetic data.**
> The goal of our experiments was **not** to show the benefits of similarity-based orderings, as these were *already* demonstrated in various settings (mentioned in lines 32-33), including vision tasks with neural networks [10, 48, 54, 63], NLP tasks with word2vec embeddings [50], tabular data with linear and logistic regression [67], and in incremental fine-tuning of LLMs [*1 below].
> Instead, the goal of our experiments was to exemplify specific aspects of greedy orderings in ways that support our analysis, facilitate discussions, and develop intuition into our analytical findings which are the *main contributions* of our work. For example, we used our simple experiments to clearly discuss repetition in greedy orderings (Section 5.2, Appendix E.1) and the effects of the rank and dimensionality (Appendix B).
> Of course, more experiments can always be interesting, but we believe this could be done in future work.
>
> 3. **Providing a sample of the code.** We accept the reviewer’s suggestion and in the updated manuscript we will improve the reproducibility by sharing a full code sample of the straightforward implementation of our experiments.
>
> ---
> **Additional references:**
> [*1] Andrew Bai, Chih-Kuan Yeh, Cho-Jui Hsieh, and Ankur Taly. 2025. An Efficient Rehearsal Scheme for Catastrophic Forgetting Mitigation during Multi-stage Fine-tuning. In Findings of the Association for Computational Linguistics: NAACL 2025.

---

> ### Comment · Reviewer_Lb47 · 2025-08-05
>
> Thanks the authors for the reply. There have been lots of works done in the linear regression setting, but not all are restricted in this setting, for e.g. [1]. As you mentioned, if the phenomena that were already observed empirically in more complicated models and datasets, what will the theory provide some new interesting insights?
>
>
>
> [1] Towards a General Framework for Continual Learning with Pre-training

---

> > ### Author Response · Authors · 2025-08-06
> >
> > We thank the reviewer for the follow up.
> >
> > We are confused—how does the cited paper [\*1] answer our question *("Is the reviewer aware of other theoretical papers that rigorously analyze these task ordering aspects in more complicated models?")*. Specifically, [\*1] does **not** address task ordering and offers no suitable tools for analyzing orderings or deriving convergence guarantees.
> >
> > In response to the reviewer’s further question: we note that our work not only explains earlier empirical findings but also uncovers new ones—for instance, we discover a novel failure mode of single-pass (dis)similarity-guided orderings. A detailed list of our practical insights can be found in responses to other reviewers, e.g., E1hb.
> >
> > We would be happy to engage with the reviewer in discussion regarding any remaining concerns.
> >
> > ---
> >
> > [\*1] Wang, Liyuan, et al. "Towards a general framework for continual learning with pre-training." arXiv preprint arXiv:2310.13888 (2023).

---

### Official Review · Reviewer_E1hb · 2025-07-02

**Clarity:** 3
**Significance:** 2
**Originality:** 3
**Rating:** 4
**Confidence:** 4

**Summary:**

The paper tackles the continual linear regression problem. The main algorithm analyzed is a Kaczmarz-type method that chooses the next task with a greedy rule. The main theorems are as follows. Lemma 4.1 shows that the squared distance of the greedy method to ground-truth is smaller than or equal to the (unsquared) distance of an "optimal" method. Lemma 4.2 proves an upper bound $O(1/T)$ on the losses of the algorithm where $T$ is the number of tasks. These two results assume the data matrix has rank $d-1$, where $d$ is the data dimension. Theorem 5.1 shows that in high dimensions ($d\geq 30$) the loss of the greedy rule is lower bounded by a nonzero constant (for example, 1/16). Proposition 5.2 instead provides an upper bound of the loss that is of order $O(1/log k)$, where $k$ is the number of iterations

**Questions:**

- At line 988 of Theorem 5.1, I saw $u_t$ is defined based on $w_t$ and $X_t$ is constructed using $u_t$. But the learner only produces $w_t$ after seeing $X_t$. Does it mean that the adversarial construct assume knowledge about all iterates? If so, the values of Theorem 5.1 would be greatly reduced.

- While I am not very familiar with literature, I imagine there would be some convergence bound on greedy Kaczmarz or greedy Gauss Seidel method. I furthermore imagine that these convergence rates are linear, depending on eigenvalues of data. How would the bounds in the paper compare with those results? A very detailed comparison is needed to understand the contribution of the paper.

- Line 241: Once these projections diminish, we switch to the random ordering, which—unlike the greedy approach—cannot be adversarially “tricked” into failure ... why the random ordering won't fail? I think the randomized rule would also admit some lower bound, as shown in earlier work.
- The main title "Task Similarity Matters" is a little bit confusing to me. It seems that task similarity doesn't matter for randomized rules. Could the authors better explain the rationale behind the title? On the other hand, in fact, in my opinion, the connection to "task similarity" is very weak. I believe the notation of task similarity is not related to algorithms. The greedy rule instead is an algorithm that chooses a task based on the current iterate vector, not on task similarity. Could the authors better justify why task similarity is a suitable angle towards appreciating the paper?

**Ethical Concerns:**

["NO or VERY MINOR ethics concerns only"]

**Final Justification:**

The rebuttals addressed several of my concerns, and promised to include a stronger bound in the revision.

I have read the comments from other reviewers and also the author's reponses.

My original concern was that the paper didn't analyze a practically implementable algorithm. This was partially addressed in the rebuttal.

Based on these, I increased my score from 3 to 4, and increased the score on "Significance" from 1 to 2. I believe the paper represents a solid contribution to the theory of greedy rules. Personally, I love to see and encourage more theoretical works like this, as continual learning theory is scarce compared to empirical works. Moreover, the paper provides new results from the Kaczmarz or  Gauss-Seidel viewpoint.

That being said, I am not recommending acceptance with a higher score, as its practical significance is still not very clear to me. For example, computing the maximum distance rule is much more expensive compared to randomized rules. In fact, this is a shared concern among reviewers.


Finally, regarding the concerns of Reviewer Lb47:
- *... may not fully capture the complexities of recent advances in continual learning.*
- *...  may not fully reflect the characteristics and challenges of real-world datasets.*

I respectfully disagree with that, as it is a general criticism without specifics. To be rigorous, I am not sure what "the complexities of recent advances in continual learning" are, and what "the characteristics and challenges of real-world datasets" are. Do the reviewer have a precise definition on these?

**Limitations:**

Personally, I think the main limitation is that the paper works with simplified settings and a hard-to-implement algorithm. Even though the analysis of greedy rules is more difficult and the paper seems to make solid theoretical contributions, this does not justify the study of an algorithm that can not be used in practice.

**Paper Formatting Concerns:**

I have no concerns on paper formatting.

**Quality:**

3

**Strengths And Weaknesses:**

Strengths:
- the paper reads smooth and is very clear, with figures and visualization properly placed to explain the ideas and results.
- the paper contains a solid array of technical results on continual regression with a greedy rule.


Weaknesses:

A major weakness in my humble opinion is on the significance and relevance of the study. To begin with, continual linear regression is an idealized problem, and the paper furthermore analyzes a hard-to-implement greedy rule that need to know the ground-truth vector (Definition 3.1). My first question therefore is: how would the authors justify the practical value and impact of the studied problem and the analyzed algorithm?

A potential justification might be that the theorems proved could provide practical guidance for continual learning. However, this point is not very clear to the reviewer. Could the authors explain: what practical insights do we obtain from the theorems?

- Proposition 5.2 doesn't provide a good bound. In analysis people only ignore the log term. The bound $O(1/log k)$ of Proposition 5.2 is like a bound $O(1)$. The authors are encouraged to illustrate the difficulty of obtaining a better bound that is dimension independent.

---

> ### Author Rebuttal · Authors · 2025-07-31
>
> We thank the reviewer for their review. We are pleased that they found the paper clear and deriving meaningful results.
>
> Below, we answer the knowledgeable questions by the reviewer, using them to clarify several important aspects of our work, particularly regarding its practicality and significance. We hope these clarifications enable the reviewer to reassess our paper.
>
>
> In short, here is how we resolve/alleviate the ‘weaknesses’ raised by the reviewer:
> * The **practicality** of the greedy selection rule → we rephrase a formula, clarifying that no knowledge of the ground-truth solution is necessary.
> * The **simplicity** of our linear regression setup → we explain that this setup suffices to derive meaningful (and even surprising) results that illuminate recent empirical findings.
> * The **strength** of the $\mathcal{O}(1/\log k )$ bound in the greedy with-repetition case → we improve it to $\mathcal{O}\left(1/\sqrt[3]{k}\right)$.
>
> ---
>
> ### Now, we go into detail on the raised ‘weaknesses’:
> 1. **Significance and relevance:**
>     - **Is the greedy rule “hard-to-implement”? (no)**
> We clarify that both of the greedy rules we analyze do *not* require any prior knowledge of the offline (‘ground-truth’) solution.
> The reviewer seemed especially worried about the Maximum Distance rule (Definition 3.1); however, its objective can be rewritten as  $ \left\Vert ( \mathbf{I} - \mathbf{P}_m ) ( \mathbf{w}\_{t-1} - \mathbf{w}\_{\star} ) \right\Vert^{2} = \left\Vert \mathbf{X}_m^{+} ( \mathbf{X}_m \mathbf{w}\_{t-1} - \mathbf{y}_m ) \right\Vert^{2} $,
> better illustrating that no knowledge of $\mathbf{w}\_{\star}$ is required. Thus, it *is* possible to compute the MD rule exactly, making it not purely theoretical.
> One might claim that explicitly computing $\mathbf{X}^{+}\_{m}$ is also prohibitive (but not impossible!); however, it can be estimated quite easily with small subsets of the training data from different tasks—again, without knowledge of the offline solution $\mathbf{w}\_{\star}$ (see also lines 150-155).
> We thank the reviewer for pointing out this misunderstanding which we will resolve with a clearer explanation in the revised version.
>     - **What practical insights could be obtained from the analysis on the ‘idealized’ continual linear regression setting?**
> The main goal of our paper (discussed in lines 31-41) was to deepen the theoretical understanding of practical benefits of similarity-guided orderings—which were already demonstrated in prior empirical work [e.g., 10, 34, 43, 47, 48, 54, 55, 61, 68, 69].
> Our elegant linear regression framework not only offers a theoretical **foundation** for recent empirical findings in the field, but also yields **insights** that can guide future practice. For example:
>
>
>         - **Foundation:** A rigorous analysis showing provable advantages of (dis)similarity-guided task orderings, justifying recent empirical findings in CL [e.g., 10, 47], in a more permissive analytical framework compared to recent prior efforts [47, 48].
>
>
>         - **Practical insight:** We uncover a surprising *failure* mode of single-pass greedy orderings (Theorem 5.1).
>
>         - **Practical insight:** We compare greedy and random orderings from multiple perspectives, highlighting differences in their empirical convergence behavior. Contrary to conventional wisdom on random orderings, we demonstrate that in *greedy* orderings, task repetition often leads to faster convergence, both theoretically and empirically.
>
>         - **Practical insight:** Leveraging the empirical benefits of greedy orderings while avoiding their possible pitfalls, we suggest a hybrid method—novel to the CL community—that provably converges.
>
>       We will better emphasize these contributions in the revised version.
>
>
> 2. **The $\mathcal{O}(1/\log k)$ upper bound for greedy orderings with repetition is weak.**
> Thank you for this insight. After working hard on improving this, we derived **a stronger (dimensionality-independent) bound of $\mathcal{O}\left(1/\sqrt[3]{k}\right)$**. This emphasizes the effect of task repetition in these greedy orderings and strengthens our theoretical contributions.
> We will include this in the revised manuscript.
>
>
>
> ### Addressing the raised **questions**:
> 1. **Does the adversarial construction assume prior knowledge of the chosen iterates?**
> The *learner* does *not* assume knowledge about future iterates, but simply greedily maximizes over unseen (prespecified) tasks at each iteration (as in Definition 3.1, 3.2).
> To show *how* a task collection that leads to failure can be *constructed*, we “reverse engineer” it as follows:
> We define vectors $(\mathbf{w}_t)$ by specifying their coordinates, then define tasks $\left(\mathbf{X}_t, \mathbf{y}_t\right)$ in terms of the $(\mathbf{w}_t)$. With these tasks and an explicitly-defined initialization vector, we show that the greedy task selection naturally yields $(\mathbf{w}_t)$ as the iterates.
> Consequently, Theorem 5.1 retains its validity and significance.
>
> 2. **Comparison to convergence bounds from Kaczmarz method literature.**
> Continual learning seeks to minimize the average task loss (Definition 2.3), whereas the greedy Kaczmarz literature minimizes the distance to the joint solution (Definition 2.5). Proposition 2.6 bridges the two: any bound on the distance immediately upper‑bounds the average loss.
> Existing linear convergence rates for greedy Kaczmarz—e.g., Nutini et al. [57] and other works cited in lines 168‑175—fall short in two ways: (i) they rely on repeated rows/blocks and therefore do not apply in the single-pass regime central to continual learning; (ii) when repetitions are allowed, their rates depend on data eigenvalues, so they can be much slower than our (improved) data‑independent $\mathcal{O}\left(1/\sqrt[3]{k}\right)$ guarantee—especially for nearly collinear tasks, as illustrated in Evron et al. [24, Thm 8 & Lem 9].
> To make this contrast concrete, we will add a small‑angle experiment to the supplementary material. In addition, we will extend the discussion in lines 168-175 accordingly.
>
>
> 3. **Doesn’t random ordering also fail?**
> When we mentioned that *“the greedy approach can be tricked into failure”* (Section 5.1), we meant a complete, “catastrophic” failure that yields $\Omega(1)$ forgetting after learning (even infinitely) many tasks.
> The reviewer is right in suggesting that random orderings also exhibit worst-case behaviors in certain cases, but these are much more mild, admitting an $\mathcal{O}\left(1/\sqrt[4]{k}\right)$ bound, as shown in prior work [26]—which we do not consider as “failure” modes.
> Following the reviewer’s question, we will make this clearer in the next version.
>
>
> 4. **How is task similarity connected to greedily choosing the next task based on the current iterate?**
> This is indeed a subtle issue to which we have given considerable thought, and we have explicitly discussed it in the paper.
> As we explain in lines 133-136, the distance between iterates (affecting both greedy rules) relates to the principal angles between the solution subspaces of tasks.
> In turn, these angles measure task similarity, as we illustrate intuitively in Figure 1 and further exemplify in Section 4.2 (see lines 195-196).
> Furthermore, we discuss heuristic similarity measures (lines 156-160), and the usage of greedy rules as an idealized theoretical “proxy” for optimal and similarity-guided task orderings (lines 164-166).
> Finally, the reviewer’s intuition that “task similarity doesn’t matter for randomized rule” is largely correct (or at least, it matters less). Such random orderings are mostly invariant to many of the more “complex” interactions between tasks, which our paper aims to better understand.
> The fact that the greedy (dis)similarity-guided orderings *beat* random ones in our experiments and in many other empirical works (including those that rely on heuristic notions of similarity), implies that, indeed, “task similarity matters”.
>
>
> 5. As a side note, in their summary, the reviewer mentioned that our adversarial lower bound applies from a dimension $d \geq 30$, calling this a “high” dimension.
> Consequently, we improved our result, and in the next version we will include another adversarial construction in *3 dimensions only* that shows that high dimensionality isn’t necessary for demonstrating “catastrophic” (non-diminishing) forgetting.
>
>
> We again thank the reviewer for their feedback. We hope we clarified the significance of our work, and we would be happy to engage further with the reviewer to address any additional questions.

---

> ### Author Response · Authors · 2025-08-04
>
> We thank the reviewer for engaging in discussion with us and for raising the rating following our clarifications. We are pleased that the reviewer now recognizes a solid theoretical contribution in our paper, one that is currently scarce in the continual learning community.
>
> While we do not wish to overstep or place any pressure on the reviewer (if we may—we are happy with the new rating), we believe that the reviewer’s comment on the computational cost of the Maximum Distance rule raises an important point that merits brief discussion, if only for the benefit of other readers.
>
> Indeed, the MD rule is more computationally expensive than random selection, as it naively requires computing (individual) pseudoinverses. However, we view it as a formal theoretical proxy for more efficient, prespecified (offline) or online metrics designed to quantify either “update magnitudes” or task similarity. Such affordable proxies are common across many areas of machine learning and include: (i) loss magnitudes, as in our Maximum Residual rule; (ii) gradient and/or Hessian magnitudes; (iii) heuristic measures of task similarity; and so on. We also kindly refer the reviewers to lines 146–166 of our paper (*“Computational tractability of greedy policies”*).
>
> A better theoretical understanding of the MD rule could not only shed light on the recent empirical success of similarity-guided task orderings, but also serve as a key step toward developing more computationally efficient methods for designing successful task orderings.

---

### Official Review · Reviewer_rjvw · 2025-07-03

**Clarity:** 2
**Significance:** 3
**Originality:** 3
**Rating:** 4
**Confidence:** 4

**Summary:**

The paper considers the theoretical performance of continual linear regression when the order of several jointly realizable tasks changes.
The paper proves the optimality of greedy ordering according to the maximum distance (MD) in the limited setting when the rank of data in each task is $d-1$.
The paper also shows an adversarial counterexample on general data in which the greedy ordering leads to catastrophic forgetting.
The paper further discusses learning with repetition and the hybrid task ordering.

**Questions:**

Q1: How can you compute MD when the task is not rank-1? We do not have access to the $w_*$.

Q2: In Scheme 2, similarly, how can you compute Line 3 a 4 without knowing $w_*$?

**Ethical Concerns:**

["NO or VERY MINOR ethics concerns only"]

**Final Justification:**

Thank you for your detailed explanation. I think these arguments make sense, and they kind of address my concern. Generally speaking, I would be happy to see and to encourage CL theory works. I am raising the score from 3 to 4.

That said, it will be quite some work to integrate these arguments into the paper. Regardless of the final result, please ensure to carefully incorporate them into the intro and discussion part of your paper.

**Limitations:**

N/A on societal impact

**Quality:**

3

**Strengths And Weaknesses:**

While I believe the work is sound, the result is novel, and the writing is clear, my worry is that it seems to me for now that the result is a little disconnected from continual learning. I am also not aware of how the insights on task ordering can benefit the area of continuous learning and Kazcmarz methods. Please feel free to educate me on this; if there is substantial evidence that the result can provide some insights into practice, I will be happy to raise the score.

The authors mentioned in the footnote on page 3 that they only consider a noiseless optimization-based setting in analyzing the effect of ordering. Furthermore, they are considering the continual linear regression algorithm without any other regularization, replay, etc. for improving the CL performance. On practical relevance, this algorithm is known to suffer from forgetting and not used for performance in practice; on theoretical significance to CL, [Evron, 2022] already discussed this algorithm and some effect of task ordering, and the additional message of this paper seems incremental. Also, I am not exactly sure for the theoretical significance of such ordering in the Kazcmarz methods field, plus that direction is not claimed as the main contribution of the paper.

---

> ### Author Rebuttal · Authors · 2025-07-31
>
> We thank the reviewer for their review. We are pleased that they found our paper novel, clearly written, and theoretically sound.
>
> ### The reviewer’s feedback entails a lot of ideas, carefully addressed below:
>
> 1. **Benefit to the continual learning and Kaczmarz fields.**
> We are happy that the reviewer is open to reconsider their score, especially on the significance of the paper. Let us mention some specific benefits of our work.
>
>     - **In continual learning (CL):**
>     Our main goal (discussed in lines 31-41) was to deepen the theoretical understanding of the empirical success of similarity-guided orderings [e.g., see 10, 34, 43, 47, 48, 54, 55, 61, 68, 69].
>     Our analytical framework not only offers a theoretical **foundation** for recent empirical findings, but also yields **insights** that can guide future practice. For example:
>
>        - **Foundation:** A rigorous analysis showing provable advantages of (dis)similarity-guided task orderings, justifying recent empirical findings in CL [e.g., 10, 47], in a more permissive analytical framework compared to recent prior efforts [47, 48]. Moreover, throughout our paper and specifically in Section 6, we thoroughly discuss and compare related work from several different fields of research (Kaczmarz, SGD, sample mining, etc.), thus establishing bridges that can benefit both theory and practice.
>
>         - **Theoretical progress:** New upper bounds for the loss include Lemma 4.2 and Proposition 5.2—which will be improved in the revised version to a meaningful dimensionality-independent $\mathcal{O}(1/\\sqrt[3]{k})$ bound for the loss of greedy orderings with repetitions.
>
>         - **Practical insight:** We uncover a surprising *failure* mode of single-pass greedy orderings (Theorem 5.1).
>
>         - **Practical insight:** We compare greedy and random orderings from multiple perspectives, highlighting differences in their empirical behavior. Contrary to conventional wisdom on random orderings, we demonstrate that in *greedy* orderings, task repetition often leads to faster convergence, both theoretically and empirically.
>
>         - **Practical insight:** Leveraging the empirical benefits of greedy orderings while avoiding their possible pitfalls, we suggest a hybrid method (Section 5.3)—novel to the CL community—that provably converges.
>
>
>     - **In the Kaczmarz literature**, there is much ongoing work on how to practically accelerate convergence using better sampling schemes (of rows or blocks; equivalent to tasks in the CL context). Evidently, greedy methods are of high interest in the Kaczmarz community [e.g., 57, 58, 52, 56, 79, 82, 84, 7, 8, 74, 83; cited in our lines 168-175].
>     Our work promotes the understanding of such methods by:
>     (a) analyzing the **loss behavior** rather than the convergence to the intersection (as more common in the Kaczmarz field);
>     (b) focusing on **single-pass** orderings rather than orderings with repetitions (which are more common in current Kaczmarz literature, despite sometimes behaving worse [58]);
>     (c) showing non-trivial **failure modes** of these methods.
>
> 2. **Practicality of the studied scheme: CL without replay, regularization, etc.**
> As the reviewer mentioned, we study a plain CL scheme, without actively trying to prevent forgetting with replay, regularization, and so on.
> However, we do not think it limits the significance of our work, for several reasons:
>     - **The greediness itself can mitigate forgetting, even in plain schemes!**
> A nice take-home message from our work is that even without active methods like replay or regularization—greedy orderings make the model forget less and converge faster (in some cases with proven convergence rates). This illuminates the effects of the task ordering specifically, rather than those of other algorithmic factors.
>     - **Inherent analytical limitations.**
> Existing analytical tools are more mature for plain CL schemes than for regularization- and replay-based methods, which have received comparatively less theoretical attention.
> This makes it possible to conduct a more elegant and fundamental analysis of task ordering effects, without the need to account for additional factors such as buffer size or regularization strength.
>
> 3. **A note on the setting being “noiseless”.** We point out that we only require the *training* data to be realizable by a linear model. This naturally happens given sufficient overparameterization, e.g., in neural networks that, famously, can perfectly fit data, even with random labels [*1, *2]. Importantly, this makes our analysis relevant to linearized approximations of such networks (e.g., using the Neural Tangent Kernel; see lines 82-86 in our paper).
>
> 4. **Difference from Evron et al. (2022).**
> We respectfully disagree that our work is “incremental” in comparison to Evron et al. [15].
> In fact, other than the initial setup (jointly realizable continual linear regression), the intersection is small. Of course, we build upon some of their analytical tools (giving them full credit where we do so), but the focus, findings, and insights of the two papers are very different.
> In particular: (a) they never mention reordering tasks to mitigate forgetting; (b) they discuss effects of task similarity given two tasks *only*; (c) they propose no task selection rule like our similarity-guided greedy rules; (d) they do not investigate ordering optimality;
> (e) their convergence results (somewhat restrictively) require task repetition, while we study the benefits of more practical greedy orderings that require no repetition; and so on.
>
>
> ### Regarding the reviewer’s **questions**:
> 1. **Can the greedy MD rule be calculated without prior knowledge of the offline solution $\mathbf{w}_{\star}$?**
> Yes, both greedy rules analyzed do *not* require any prior knowledge of the offline solution.
> The greedy MD rule’s objective (Definition 3.1) can be rewritten as
> $ \Vert ( \mathbf{I} - \mathbf{P}_m ) ( \mathbf{w}\_{t-1} - \mathbf{w}\_{\star} ) \Vert^{2} = \Vert \mathbf{X}_m^{+} ( \mathbf{X}_m \mathbf{w}\_{t-1} - \mathbf{y}_m ) \Vert^{2} $,
> better illustrating that no knowledge of $\mathbf{w}\_{\star}$ is required. Thus, it *is* possible to compute the MD rule exactly, making it not purely theoretical.
> One might claim that explicitly computing $\mathbf{X}^{+}\_{m}$ is also prohibitive (but not impossible!); however, it can be estimated quite easily with small subsets of the training data from different tasks—again, without knowledge of the offline solution $\mathbf{w}\_{\star}$ (see also lines 150-155).
> We thank the reviewer for pointing out this misunderstanding which we will resolve with a clearer explanation in the revised version.
>
>
>
> 2. **Similarly, how is the hybrid method evaluated without prior knowledge?**
> Hopefully, the above clarifies how the Maximum Distance rule is computed in practice and that it requires no knowledge of the solution $\mathbf{w}\_{\star}$. The same applies to lines 3-4 in the hybrid Scheme 2.
>
>
> We hope we have addressed the reviewer’s questions and concerns, and we would be happy to engage further should any additional questions arise.
>
> ---
> **Additional references:**
> [*1] C. Zhang, S. Bengio, M. Hardt, B. Recht, and O. Vinyals. Understanding deep learning requires rethinking generalization. In ICLR, 2017.
> [*2] E. B. Baum. On the capabilities of multilayer perceptrons. Journal of Complexity, 1988.

---

> > ### Author Response · Authors · 2025-08-05
> > **Invitation for further discussion**
> >
> > We thank the reviewer for their time so far.
> >
> > Given the importance of their assessment, and noting that they expressed openness to reconsidering the score, we respectfully invite them to review our Author Response above.
> >
> > We are happy to discuss any remaining questions or concerns.

---

> ### Comment · Reviewer_rjvw · 2025-08-06
>
> Thank the author for their rebuttal. After reading the rebuttal and the other reviews, I retain my score of 3 for now and I have further concerns.
>
> The main concern is still the practical relevance. As a CL theory researcher, let me be clear: the connection between this work's setting and CL practice is *really* weak in its current form. Literally no one performs vanilla training in CL. If this work had been under review in 2022, the justification to study vanilla continual linear regression would have worked since there had been no previous CL theory paper. However, with Evron et al. (2022) and around ten works later on CL theory with regularization, replay, etc., there must be some stronger practical reason than analytical harshness.
>
> One possible justification might be: in present CL applications in LLM, people do not like to add regularization in LLM training, and that task similarity might be computed by different subspaces of LoRA. Reasons like this are essential for justifying the practical relevance. Another possible justification might be the link between task ordering and curriculum learning. These proposed reasons might not necessarily hold, and I urge the authors to find such reasons and specify how they can be incorporated into the current paper as a minor edit.
>
> Also, on the difference between this work and Evron et al. (2022), I disagree with the authors on that "(a) they (Evron et al.) never mention reordering tasks to mitigate forgetting". I quote their "The rest of our paper" paragraph at the end of their Section 3 in Pg 6: "In Section 4 we consider arbitrary orderings and show when there is provably no forgetting, and when forgetting is arbitrarily high, i.e., catastrophic. We analyze cyclic and random orderings in Sections 5 and 6. For both these orderings, we derive convergence guarantees and prove forgetting cannot be catastrophic." Indeed, cyclic and random orderings are not greedy orderings, but they are still two kinds of orderings being considered for mitigating forgetting. Also, I partially disagree with that "(e) their convergence results (somewhat restrictively) require task repetition": in their Thm 13, the number of iteration $k$ can be $1$, preventing task repetition. Btw, Evron et al. (2022) is actually marked [24] in your paper, not [15].
>
> I would say that the difference to Evron et al. (2022) can be better described as greedy vs. random ordering. This is why I see the contribution as incremental, and this again affects the relevance of this paper: I am not aware of any paper in CL practice that considers greedy vs random ordering of tasks, so why should people care about this theory paper? Note again that the contribution to the Kaczmarz community is not claimed as the main contribution of the paper.
>
> My other concerns are addressed. I am still open to reconsidering my score if the authors have good explanations for these two questions, but at this stage, I am leaning further toward rejection with greater confidence.

---

> > ### Author Response · Authors · 2025-08-07
> >
> > We thank the reviewer for the effort they devoted to this knowledgeable discussion. We now understand the reviewer’s two remaining concerns better and can address them more properly.
> >
> > As motivation, we note that the interactions between task similarity, orderings, and forgetting remain not well understood—both in practice and in theory.
> > **Even basic, practical questions at the intersection of continual and curriculum learning lack thorough answers**, despite their potential to greatly influence training efficiency.
> > For example, in continual NLP or RL, *how should tasks or environments be ordered?
> > What role does task similarity play?*
> >
> > ---
> > ## Why study vanilla CL?
> > > “no one performs vanilla training in CL … there must be stronger practical reason than analytical harshness”
> >
> > - Vanilla training is the **most natural** approach to learning with gradient algorithms.
> > In contrast, other continual methods (e.g., regularization, replay) are *necessities* introduced to tackle forgetting in CL settings, but **depart** from standard training practices in modern DL.
> > A non-forgetful CL scheme that remains as close as possible to vanilla practices could be considered a *central* goal for the field.
> > - Thus, if task reordering *alone*, even under vanilla training, can improve the efficiency of continual training, it **offers a highly valuable practical insight**.
> >
> > - **In real-world settings,** continual methods may be impractical—e.g., replay may be infeasible when access to prior data is restricted by availability, retention, or privacy limitations; whereas regularization can interfere with optimization (due to the plasticity–stability tradeoff) and introduces design choices (e.g., regularization strength) that are often opaque or costly.
> > - Exploring orderings under replay and regularization is a valuable *future* direction, particularly to test whether they yield different findings from ours; such comparisons could inform similarity-guided orderings tailored to these schemes.
> >
> > ## Comparison to Evron+ [24] and random orderings
> > > “cyclic and random orderings are still being considered for mitigating forgetting”
> >
> > - Indeed, Evron+ [24] proved that random orderings mitigate forgetting under vanilla training.
> > Lesort+ [44] then demonstrated this empirically in deep models trained with vanilla SGD.
> > - However, by focusing on *random* orderings, **Evron+ [24] effectively removed task similarity from the equation**.
> > The reviewer is correct in pointing out their results on similarity in *cyclic* orderings. However, those apply either to a special case of $T=2$ tasks or to the *worst cases* of general $T\geq 3$ settings, which—again—offer no insight into the role of task similarity (within a cycle).
> > - In contrast, **task similarity plays a central role in our work**.
> > We study similarity-guided task orderings through the lens of greedy orderings.
> > Following the common wisdom on Kaczmarz, *we expected similarity-guided orderings to outperform random ones*.
> > - While this is indeed observed in our experiments and recent CL work on deeper models [10, 47], *our goal was to understand how consistently this advantage holds and to what extent* (e.g., via convergence rates).
> > - Specifically:
> >     - Lemma 4.2 shows a rate of $O(1/k)$ in nearly-determined greedy orderings—matching Thm 13 of [24] for random ones.
> >     - In contrast, Sec 5.1 shows that generally, single-pass *greedy orderings can exhibit failure modes that are impossible under random orderings*.
> >     - We show that, unlike in random orderings, repetition in greedy ones often improves performance empirically. Analytically, repetition in greedy orderings mitigates forgetting by achieving a (now improved) bound of $O(1/k^{1 /3})$, eliminating the failure mode.
> > - Finally, our findings have led us to a novel hybrid ordering that combines the benefits of both greedy and random orderings.
> >
> > ## Existing tools for other CL approaches
> > > “in 2022 there had been no CL theory papers. However, with Evron+ (2022) and around ten works later with regularization, replay…”
> >
> > We find the theoretical foundations for other CL approaches not mature enough to support an in-depth analysis of similarity-guided task orderings.
> > Evidently, convergence rates have rarely been established for these methods.
> > Moreover, most of these papers had to either focus on settings with two tasks only [45, 46] or having commutable covariance matrices across tasks [85, 46]—conditions that are *inadequate for studying phenomena driven by task similarity*.
> >
> > Thus, we argue that the vanilla scheme is better suited for a foundational study of task ordering and similarity; once understood, extensions to more complex approaches become more feasible.
> >
> > ---
> > **Finally,** we hope we clarified our analytical choices and the significance of our work.
> > This important discussion can be incorporated easily across the introduction, setting, and discussion sections, only requiring a slightly more precise framing and expanded discussion in a few places.

---

> > > ### Comment · Reviewer_rjvw · 2025-08-09
> > >
> > > Thank you for your detailed explanation. I think these arguments make sense, and they kind of address my concern. Generally speaking, I would be happy to see and to encourage CL theory works. I am raising the score from 3 to 4.
> > >
> > > That said, it will be quite some work to integrate these arguments into the paper. Regardless of the final result, please ensure to carefully incorporate them into the intro and discussion part of your paper.

---

### Official Review · Reviewer_nmNC · 2025-07-03

**Clarity:** 3
**Significance:** 2
**Originality:** 2
**Rating:** 4
**Confidence:** 2

**Summary:**

This paper explores task ordering in continual learning, offering theoretical insights into how task sequences with dissimilar adjacent tasks can enhance performance. The authors specifically examine a greedy strategy designed to maximize the difference between tasks in continual linear regression, discussing both its benefits and potential drawbacks. They also propose a hybrid approach that combines greedy and random task orderings to address the identified limitations.

**Questions:**

Please refer to the Strengths And Weaknesses.

**Ethical Concerns:**

["NO or VERY MINOR ethics concerns only"]

**Final Justification:**

The authors’ response addresses several of my concerns. I agree that the paper’s chief value lies in its theoretical insights into task ordering, which deepen our understanding of continual learning. That said, a stronger demonstration of realistic relevance would enhance the impact of this work.

**Limitations:**

yes

**Paper Formatting Concerns:**

No major formatting issues were found.

**Quality:**

3

**Strengths And Weaknesses:**

Strengths:

1. This paper is well-motivated. Studying task ordering is crucial for understanding the intrinsic nature of continual learning.

2. This paper is clearly written and easy to follow. The geometric intuitions and the theoretical analysis of greedy task ordering in continual linear regression provides insight.

Weaknesses:

1. I am currently confused about the significance of this paper's contribution. As discussed by the authors in the related work section, the importance of task ordering in continual learning has already been demonstrated both empirically and theoretically in previous studies. In particular, the work by Li et al.[47] seems to have addressed this problem under more complex settings, providing theoretical analysis from the perspective of a global task graph and conducting experiments on more realistic image classification tasks. In contrast, this paper focuses on understanding greedy task ordering in the context of continual linear regression. Could the authors further explain the necessity of this research and how it differs from prior work?

2. While I appreciate the theoretical insights presented in this paper, their practical relevance is limited due to the highly idealized experimental setup, which focuses solely on linear regression tasks with data generated from Gaussian distributions. It would significantly enhance the paper's impact to explore how the proposed greedy or hybrid strategies perform in more realistic, real-world scenarios.

3. I am curious about the relationship between greedy task ordering and globally optimal task ordering. As the authors mentioned in the paper, finding a globally optimal task ordering is equivalent to solving a Maximum-Weight Hamiltonian Path problem, which is known to be computationally expensive. However, in practical settings, the number of tasks T is often relatively small, making it feasible to compute the globally optimal ordering or, at the very least, find a solution that outperforms the greedy approach. In such scenarios, how does the performance of greedy task ordering compare to that of a globally optimal (or near-optimal) ordering? Additionally, can the failure cases of greedy task ordering be interpreted or explained through the lens of global optimality?

---

> ### Author Rebuttal · Authors · 2025-07-31
>
> We thank the reviewer for their positive review. We are pleased that they found our work clear and well motivated, studying an important problem in the field of continual learning.
>
> We would like to address the reviewer’s concerns:
> 1. **Significance of contribution in light of prior work.**
> The reviewer asked that we elaborate on the differences from prior work, and particularly to the most closely related work of Li and Hiratani [47]—indeed an important work in the field, which we cite and discuss in many places in our paper (e.g., lines 37-39, 259-260).
> However, we believe that our *theoretical* setup is more revealing than theirs; and that the contributions in our work are independently important and significant to the field, and actually improve the understanding of their findings as well.
>     - A few inherent similarities and differences between their analytical setup and ours:
>         - **Same model:** Both papers study **linear** models, where each task is learned to convergence.
>         - **Our “data model” is more revealing:**
>     Our paper assumes **joint realizability**, i.e., $\exists \mathbf{w}\_{\star} $ such that $\mathbf{y}\_m=\mathbf{X}\_m \mathbf{w}\_\star, \forall m$, but allows **any arbitrary** feature matrices.
> On the other hand, their analysis employs Gaussian feature matrices with a low dimensional latent representation and simplistic cross-task pairwise similarity assumptions, captured by the covariance matrices $\mathbf{C}\^{\mathrm{in}}, \mathbf{C}\^{\mathrm{out}}$.
> This is crucial when analyzing the effects of task similarity on forgetting, since higher order and rank-dependent effects on similarity can’t rise under their simplistic model.
> For instance, analyzing arbitrary feature matrices was essential for us to uncover complex phenomena such as the adversarial construction of Theorem 5.1.
>     - **Different tested aspects and novel findings:**
> We discuss novel phenomena, not previously explored in the context of continual learning (CL), and specifically, not in Li and Hiratani [47]:
>         - We test the effects of repetition in similarity-guided orderings—while Li and Hiratani [47] focus solely on single-pass orderings (see their Limitations section). Consequently, we formally prove a clear separation in worst-case behavior depending on the presence or absence of repetition.
>         - Our geometric interpretation (e.g., in Figure 7) allows a more intuitive discussion of their task typicality.
>         - We formalized greedy orderings in the context of CL, linking to other areas of research such as Kaczmarz and projection methods.
>         - We derive convergence rates for the forgetting under greedy orderings. Importantly, after hard work on the matter, we achieved **a stronger (dimensionality-independent) bound of $\mathcal{O}(1/\sqrt[3]{k})$** for the loss of greedy ordering with repetition, instead of the $\mathcal{O}(1/\log (k))$ bound which was previously presented in Proposition 5.2.
>         - Our elegant linear regression setup is sufficient to reveal a surprising failure mode of greedy orderings (Theorem 5.1), ‘warning’ future practitioners that dissimilarity-guided orderings do *not* invariably perform better.
>         - Drawing inspiration from other fields, we introduce a practical hybrid approach to overcome the potential failure modes we uncovered.
>
>     To address the reviewer’s concern, we will elaborate in the revised manuscript on how our work extends and complements the findings of Li and Hiratani [47].
>
> 2. **Practical relevance of the experimental setup.**
> The goal of our paper (discussed in lines 31-41) is to deepen the *theoretical* understanding of the practical benefits of similarity-guided orderings that were *already* demonstrated in prior empirical work [e.g., 10, 34, 43, 47, 48, 54, 55, 61, 68, 69]. We propose more rigorous, analytical tools and perspectives that provide theoretical justification to such orderings, while thoroughly discussing several aspects of them (e.g., rates of convergence, failure modes, effects of repetitions).
> Hence, the goal of our experiments is *not* to demonstrate the empirical superiority of similarity-guided orderings, but rather to support our analysis, facilitate discussions, and *develop intuition into our analytical findings*—which are the main contributions of our work.
> For instance, the simplified experimental setting provides insight on the effects of dimensionality and rank on the performance of each algorithm, not available via other settings (Appendix B).
>
>
> 3. **Greedy vs optimal task ordering.**
> We thank the reviewer for this interesting question which sparked discussions among the authors.
> The reviewer is right that, when the similarity metric between tasks is *prespecified* and fixed, finding the sequence that maximizes the total pairwise dissimilarity reduces to finding a **maximum‑weight Hamiltonian path**, which can be solved for a small number of tasks or approximated.
> However, (i) we do *not* assume a prespecified metric, and (b) even the optimal path w.r.t. such a metric is not guaranteed to minimize the loss or the distance to the joint solution $\mathbf{w}\_{\star}$!
> In contrast, our Maximum Distance rule (Definition 3.1) greedily maximizes the decrement in the distance to the joint solution (Eq. 3). One would imagine this “decrement” can be a valuable metric for a Hamiltonian path solver—however, it is *impossible* to even use it to define a static metric since it is an *online* quantity depending on the current iterate/node (and indirectly, on the entire path to it). Consequently, it is no longer a Hamiltonian path problem but rather a sequential decision problem whose brute‑force solution requires *simulating the entire training* for all $T!$ orderings.
> It is only in the nearly‑determined case $r=d-1$ of Section 4.2 that the problem maps to finding a Hamiltonian path on a static graph—and then we show that our greedy selection indeed approximates the optimal path (Lemma 4.1).
> Characterizing the globally optimal ordering and deriving its convergence rate remain promising open questions. We will mention this in the revised manuscript.
>
> We hope the important discussion above has helped make the reviewer more confident in their positive assessment. We would be happy to further address any remaining questions.

---

> > ### Author Response · Authors · 2025-08-06
> >
> > We thank the reviewer again for their thorough review, which helped us improve the paper.
> >
> > As the discussion period is coming to an end, we wanted to check whether our rebuttal successfully addressed the reviewer’s concerns and questions.
> >
> > We are happy to clarify any remaining issues.

---

> > ### Comment · Reviewer_nmNC · 2025-08-08
> >
> > Thank you for the thorough clarifications. I have carefully read the authors’ response, which addresses several of my earlier concerns—especially those regarding the novelty of the theoretical framework. I agree that the paper’s primary value lies in its theoretical insights into task ordering, which are helpful for understanding the essence of continual learning. While I still believe that additional experiments in more realistic settings could further enhance the practical relevance, the work’s analytical contributions are informative.

---

> > > ### Author Response · Authors · 2025-08-09
> > >
> > > We sincerely thank the reviewer for their time and effort. We are pleased that they recognize the value of our theoretical insights for the continual learning community.
> > >
> > > It appears that this rebuttal process has helped address some of the reviewer’s concerns. If the reviewer deems it appropriate, we would greatly appreciate a re-assessment of their confidence score.

---

### Decision · Program_Chairs · 2025-09-17

**Decision:**

Accept (poster)

**Comment:**

This paper considers continual learning for (jointly) realizable linear regression, focusing on the greedy task ordering that maximizes dissimilarity between consecutive tasks. The paper makes some advances in our understanding of how promoting task dissimilarity between adjacent tasks can help convergence, what the failure modes of such a greedy ordering are, and how allowing repetition can cure the failure mode. Most reviewers appreciated that the theoretical contributions are solid, and I believe the authors did a good job of articulating what kind of practical insights can be gained from this theory. Even after the submission, the authors made efforts to improve the $O(1/\log k)$ bound to a stronger dimensionality-independent bound of $O(1/k^{1/3})$, and also found another set of failure examples that work with only 3 dimensions.

The biggest weakness of this paper, as commonly pointed out by the reviewers, is that it studies a simple setting and a vanilla algorithm that may appear detached from practical continual learning. While I find this criticism reasonable, I also believe that the theory presented in the paper provides useful insights and makes a solid contribution to the continual learning theory community. Hence, I recommend acceptance.

For the camera-ready version:
* I strongly recommend adding some experiments that more closely resemble real-world conditions, involving real-world data and/or simple neural networks. Such experiments would be a valuable addition to the paper, supporting the theoretical analysis.
* Please carefully incorporate the discussion with the reviewers (why study vanilla CL, practical insights, comparison with Evron+[24], etc.) into the paper. As Reviewer rjvw pointed out, this will require substantial work, but I trust the authors to make a thorough revision.